# Optimal Nuisance Function Tuning for Estimating a Doubly Robust Functional under Proportional Asymptotics

**Sean McGrath**
Yale University
sean.mcgrath@yale.edu

**Debarghya Mukherjee**
Boston University
mdeb@bu.edu

**Rajarshi Mukherjee**
Harvard University
ram521@mail.harvard.edu

**Zixiao Jolene Wang**
Harvard University
zixiaowang@fas.harvard.edu

## Abstract

In this paper, we explore the asymptotically optimal tuning parameter choice in ridge regression for estimating nuisance functions of a statistical functional that has recently gained prominence in conditional independence testing and causal inference. Given a sample of size $n$, we study estimators of the Expected Conditional Covariance (ECC) between variables $Y$ and $A$ given a high-dimensional covariate $X \in \mathbb{R}^p$. Under linear regression models for $Y$ and $A$ on $X$ and the proportional asymptotic regime $p/n \to c \in (0, \infty)$, we evaluate three existing ECC estimators and two sample splitting strategies for estimating the required nuisance functions. Since no consistent estimator of the nuisance functions exists in the proportional asymptotic regime without imposing further structure on the problem, we first derive debiased versions of the ECC estimators that utilize the ridge regression nuisance function estimators. We show that our bias correction strategy yields $\sqrt{n}$-consistent estimators of the ECC across different sample splitting strategies and estimator choices. We then derive the asymptotic variances of these debiased estimators to illustrate the nuanced interplay between the sample splitting strategy, estimator choice, and tuning parameters of the nuisance function estimators for optimally estimating the ECC. Our analysis reveals that prediction-optimal tuning parameters (i.e., those that optimally estimate the nuisance functions) may not lead to the lowest asymptotic variance of the ECC estimator – thereby demonstrating the need to be careful in selecting tuning parameters based on the final goal of inference. Finally, we verify our theoretical results through extensive numerical experiments.

## 1 Introduction

Over the past decade, powerful prediction tools from the machine learning arsenal have been used to construct robust estimators of statistical functionals such as treatment effects and conditional dependence measures. Specifically, the seminal work of Double Machine Learning (DML) develops methods for estimating functionals that require fitting two nuisance functions (e.g., outcome regression and propensity scores for causal effect estimation), which use novel debiasing techniques that impart robustness to the fit of the nuisance functions [1]. DML allows researchers to perform sample splitting where one sample is used to optimally estimate the nuisance functions and the other sample is used to estimate the statistical functional efficiently based on the fitted nuisance functions. Importantly, one can be agnostic of the statistical functional that is the target of inference while fitting

39th Conference on Neural Information Processing Systems (NeurIPS 2025).

nuisance functions. Hence, the tuning of learning algorithms for such function estimation can be operationalized from a prediction optimality perspective.

The validity of DML relies on suitable rates of estimation of the estimated nuisance functions – which need to be verified on a case-by-case basis, depending on the choice of learning method employed for fitting. However, modern complex observational studies can result in situations in which one might not be able to estimate nuisance functions consistently, let alone at a sufficiently fast rate. One regime where this occurs, termed as the inconsistency regime by Celentano and Wainwright [2], is the proportional asymptotic setting, i.e., when the number of variables in the study is proportional to the sample size. In this regime, the DML theory does not apply owing to the inconsistency of any nuisance function estimator. Therefore, an alternative estimation strategy is required.

A number of works have explored the use of certain sample splitting and nuisance function tuning (e.g., undersmoothing) strategies to construct optimal estimators of the functional of interest [3–15]. Indeed, recent literature with nonparametric nuisance function models has shown that even a naive plug-in principle-based approach can be asymptotically minimax optimal when the nuisance functions are tuned carefully [14, 15]. Moreover, a fascinating interplay has been observed between the choice of the estimation technique for the functional of interest and whether different splits are used to estimate the nuisance functions [14]. This phenomenon has been studied in nonparametric problems under Hölder type models for nuisance functions that allow for consistent estimation.

This paper aims to understand optimal tuning functions in the inconsistency regime through a doubly robust functional lens. Specifically, we consider the Expected Conditional Covariance (ECC) between a continuous outcome and treatment variable given baseline confounders – an object that has been central in the development of model agnostic inference [16, 17, 12, 18, 15], learning weighted causal effect estimation to deal with positivity violations [19, 20], conditional independence testing [21], and understanding genetic correlations [22]. Under linear models for the nuisance functions and proportional asymptotics, the overarching goal of this paper is to perform sharp asymptotic analyses and thereby paint a complete picture of the interplay of sample splitting, the type of estimator used, and the choice of optimal tuning of the nuisance functions estimated through ridge-regularized regression.

## 1.1 Our Contributions

We take a first step towards a comprehensive understanding of how the tuning of nuisance function estimators and the choice of data-splitting strategy affect the asymptotic variance of estimators of the ECC. We present a rigorous theoretical analysis of the asymptotic behavior of three different estimators of the ECC: a plug-in based estimator, the doubly robust, i.e., DML estimator, and a variant of the DML estimator introduced in [7]. We focus on the ridge-regularized estimators for the nuisance functions and leave the investigation of other estimators to future work. Since the type of data-splitting plays a pivotal role in the proportional asymptotic regime, we consider two distinct data-splitting strategies: (i) **two-split approach**: where the data is divided into two parts—one used to estimate both nuisance functions and the other to estimate the ECC; and (ii) **three-split approach**: where the nuisance functions are estimated using two separate parts, and the third part is used to estimate the ECC. Below, we summarize our key contributions:

- Since no consistent estimators of the nuisance functions exist in our setting with proportional asymptotics, we show that each of the ECC estimators is asymptotically biased in each data-splitting strategy. We then derive bias-corrected versions of these estimators by inverting the asymptotic bias.

- We show that each of the bias-corrected estimators achieve the optimal $\sqrt{n}$-rate for estimating the ECC and derive expressions for their asymptotic variances. Unlike settings amenable to DML, the asymptotic variance of these estimators is not immune to the effects of nuisance function estimation. Therefore, the tuning parameters can be selected to minimize this asymptotic variance of the ECC. Surprisingly, the optimal tuning parameters for inference *differ from those that are optimal for prediction*, that is, the values that optimally estimate the nuisance functions. This highlights a key difference from the standard DML framework: one *should not estimate* the nuisance functions by minimizing prediction error (which is the gold standard in fixed-dimensional settings), but instead by minimizing the asymptotic variance of the functional of interest.

- We perform extensive numerical experiments that verify and complement our theoretical results. We identify the optimal choices of tuning parameters for minimizing the asymptotic variance of the ECC and compare those to prediction-optimal tuning parameters. We then provide practical guidance on which approach a practitioner should prefer under different circumstances.

## 2 Problem Setup

To set notation, consider a tuple of random variables $(A, Y, X)$, where $A \in \mathbb{R}, Y \in \mathbb{R}$ and $X \in \mathbb{R}^p$. We aim to estimate the ECC of $A$ and $Y$ given $X$, i.e., our parameter of interest is:

$$\theta_0 = \mathbb{E}[\mathsf{cov}(Y, A \mid X)] = \mathbb{E}[AY] - \mathbb{E}[\mathbb{E}[Y \mid X]\mathbb{E}[A \mid X]].$$

As mentioned in the Introduction, the estimation of ECC has a long-standing presence in the causal inference literature among other applications. ECC also serves as a foundational stepping stone toward understanding a broader class of doubly robust functionals [19]. It is easy to show that $\theta_0$ is a doubly robust with respect to two nuisance functions, $\mathbb{E}[A \mid X]$ and $\mathbb{E}[Y \mid X]$, i.e., one can consistently estimate $\theta_0$ as long as at least one of these two nuisance functions is consistently estimated. Despite its simple structure, the ECC functional encapsulates many of the key features necessary for a thorough understanding of doubly robust estimators. (e.g., minimal smoothness or convergence rate required for the nuisance estimators to achieve $\sqrt{n}$-consistency for $\theta_0$, feasibility of adaptive inference, etc.). While there is a relatively clear understanding of the statistical properties of $\theta_0$ when $X$ is finite-dimensional (or in ultra high-dimensional regimes under suitable sparsity assumptions), our knowledge of its behavior in the proportional asymptotics setting remains limited [23]. This motivates us to take a first step toward analyzing the ECC functional under proportional asymptotics, which we believe will serve as a foundation for understanding the broader class of doubly robust functionals in this asymptotic regime.

### 2.1 Estimation Strategies

In this section, we present three different estimation strategies of $\theta_0$. To formalize our analysis and investigate the behavior of $\theta_0$ under the proportional asymptotic regime, we now describe the data-generating process in detail. We assume to observe $n$ i.i.d. observations $\{(X_i, Y_i, A_i)\}_{1 \le i \le n}$. We assume that the dimensionality of $X$ grows proportionally with the sample size, i.e., $p/n = c \in (0, \infty)$. In this paper, we assume $(Y, A)$ follows a bivariate normal distribution conditional on $X$, i.e.,

$$\begin{aligned} A &= X^\top \alpha_0 + \epsilon, & \begin{pmatrix} \epsilon \\ \mu \end{pmatrix} &\sim \mathcal{N}\left(\begin{pmatrix} 0 \\ 0 \end{pmatrix}, \begin{pmatrix} 1 & \rho \\ \rho & 1 \end{pmatrix}\right) \perp\!\!\!\perp X. \\ Y &= X^\top \beta_0 + \mu, & & \end{aligned} \tag{2.1}$$

We assume that coordinates of $X \in \mathbb{R}^p$ are i.i.d. subgaussian random variables, with mean 0, variance 1, and uniformly bounded (over $p$) subgaussian norm. It is immediate from our data-generating process that the nuisance functions can be expressed as $\mathbb{E}[A \mid X] = X^\top \alpha_0$ and $\mathbb{E}[Y \mid X] = X^\top \beta_0$. The key challenge in estimating $\theta_0$ under the proportional asymptotic regime lies in accurately estimating the nuisance parameters $\alpha_0$ and $\beta_0$. In the proportional asymptotic regime, particularly when $p > n$, a common strategy is to use convex regularized estimators. For simplicity and clarity of exposition, we focus in this paper on ridge regression. The analysis of other estimators is an important direction for future work. Given tuning parameters $\lambda_1, \lambda_2 > 0$, the ridge regression estimators of $\alpha_0$ and $\beta_0$ are defined as:

$$\begin{aligned} \hat{\alpha}(\lambda_1) &= \arg\min_\alpha \left\{ \tfrac{1}{n}\|\mathbf{a} - \mathbf{X}\alpha\|_2^2 + \lambda_1\|\alpha\|_2^2 \right\} = (\hat{\Sigma} + \lambda_1 \mathbf{I}_p)^{-1}\tfrac{\mathbf{X}^\top \mathbf{a}}{n} \\ \hat{\beta}(\lambda_2) &= \arg\min_\beta \left\{ \tfrac{1}{n}\|\mathbf{y} - \mathbf{X}\beta\|_2^2 + \lambda_2\|\beta\|_2^2 \right\} = (\hat{\Sigma} + \lambda_2 \mathbf{I}_p)^{-1}\tfrac{\mathbf{X}^\top \mathbf{y}}{n} \end{aligned} \tag{2.2}$$

where $\mathbf{a} = (A_1, \ldots, A_n)^\top, \mathbf{y} = (Y_1, \ldots, Y_n)^\top$ and $\mathbf{X} \in \mathbb{R}^{n \times p}$ is the matrix of $(X_1, \ldots, X_n)^\top$ and $\hat{\Sigma} = (\mathbf{X}^\top \mathbf{X})/n$ is the sample covariance matrix of $X$.

To construct our first estimator, observe that we can express $\theta_0$ as

$$\theta_0 = \mathbb{E}[AY] - \mathbb{E}[\mathbb{E}[Y \mid X]\mathbb{E}[A \mid X]] = \mathbb{E}[AY] - \alpha_0^\top \mathbb{E}[XX^\top]\beta_0 = \mathbb{E}[AY] - \alpha_0^\top \beta_0$$

As both $A$ and $Y$ are scalar random variables, one may estimate $\mathbb{E}[AY]$ by the sample average $(\sum_i A_i Y_i)/n$, which is $\sqrt{n}$-consistent and asymptotically normal (CAN), provided that

$\mathrm{var}(AY) < \infty$ (which can be ensured if $\|\alpha_0\|_2$ and $\|\beta_0\|_2$ are assumed to uniformly (over $p$) bounded, see Appendix for more details). However, the key difficulty lies in estimating $\alpha_0^\top \beta_0$ under the proportional asymptotic regime. The *integral-based plug-in estimator* of $\theta_0$ involves estimating $\alpha_0^\top \beta_0$ by $\hat\alpha(\lambda_1)^\top \hat\beta(\lambda_2)$, i.e.,

$$\hat\theta^{\mathrm{INT}} = \tfrac{1}{n} \sum_{i=1}^n A_i Y_i - \hat\alpha(\lambda_1)^\top \hat\beta(\lambda_2)\,. \tag{2.3}$$

Our second estimator is based on the work of Newey and Robins [7]. Owing to the representation $\theta_0 = \mathbb{E}[Y(A - \mathbb{E}[A|X])] = \mathbb{E}[Y(A - X^\top \alpha(\lambda_1))]$, we can consider the following *Newey-Robins plug-in estimator* of $\theta_0$

$$\hat\theta^{\mathrm{NR}} = \tfrac{1}{n} \sum_{i=1}^n Y_i(A_i - X_i^\top \hat\alpha(\lambda_1))\,. \tag{2.4}$$

Observe that one may construct a similar estimator of $\theta_0$ based on exchanging the roles of $A$ and $Y$, i.e., a plug-in estimator based on $\theta_0 = \mathbb{E}[A(Y - \mathbb{E}[Y|X])]$. It is straightforward to see that both estimators share essentially the same asymptotic properties. Therefore, we focus on the former.

Our third estimation strategy is inspired by the one-step influence function-based correction strategy to construct a semi-parametrically efficient estimator. In our case, this estimator is given by

$$\hat\theta^{\mathrm{DR}} = \tfrac{1}{n} \sum_{i=1}^n (A_i - X_i^\top \hat\alpha(\lambda_1))(Y_i - X_i^\top \hat\beta(\lambda_2))\,. \tag{2.5}$$

Typically, in a fixed-dimensional setup (or in an ultra-high-dimensional setup), it is possible to estimate the nuisance functions $\mathbb{E}[Y \mid X]$ and $\mathbb{E}[A \mid X]$ at a rate faster than $n^{1/4}$ under some appropriate smoothness/sparsity assumption. This leads to the traditional DML wisdom: *one should first split the data into two parts: use the first part to estimate the nuisance components, namely $\mathbb{E}[Y \mid X]$ and $\mathbb{E}[A \mid X]$, and then plug these estimates to construct $\hat\theta^{\mathrm{DR}}$ using the second part of the data. Finally, cross-fitting should be used to reduce asymptotic variance.* Following this wisdom, it is not hard to show that the estimator is $\sqrt{n}$-Consistent and Asymptotically Normal (CAN) as well as a semiparametrically efficient estimator of $\theta_0$. However, this traditional wisdom is not readily applicable to the proportional asymptotic regime for two main reasons. First, although the mean functions in our setting are infinitely smooth (since they are linear), we typically do not assume any form of sparsity. As a result, it is not possible to estimate $\mathbb{E}[Y \mid X]$ or $\mathbb{E}[A \mid X]$ at a rate faster than $n^{-1/4}$. In other words, the product of the estimation errors is not asymptotically negligible; in fact, we will show that this product is itself $\sqrt{n}$-CAN and contributes non-trivially to the asymptotic variance. Second, as demonstrated in Jiang et al. [23], cross-fitting is not straightforward in this regime: estimates obtained by rotating the data are generally not asymptotically independent, and the resulting correlations between folds introduce additional contributions to the asymptotic variance. In fact, the precise characterization of the minimum achievable variance when $c = p/n > 1$ remains an open problem.

## 2.2 Sample Splitting

The three estimators introduced in the previous subsection all involve the estimation of two nuisance parameters, $(\alpha_0, \beta_0)$. As a result, one can consider two distinct sample splitting strategies. The first is the **two-split approach**, where both $\alpha_0$ and $\beta_0$ are estimated using one half of the sample ($\mathcal{D}_1$), and the resulting estimates are then combined with the other half of the sample ($\mathcal{D}_2$) to construct $\hat\theta^{\mathrm{INT}}$, $\hat\theta^{\mathrm{NR}}$, and $\hat\theta^{\mathrm{DR}}$. The second is the **three-split approach**, where the data is divided into three parts: one ($\mathcal{D}_1$) used to estimate $\alpha_0$, another ($\mathcal{D}_2$) to estimate $\beta_0$, and the third ($\mathcal{D}_3$) to construct the final estimator. To make this distinction explicit, we adopt a notational convention: for instance, $\hat\theta^{\mathrm{INT}}_{\mathrm{2sp}}$ and $\hat\theta^{\mathrm{INT}}_{\mathrm{3sp}}$ denote the versions of $\hat\theta^{\mathrm{INT}}$ constructed using the two-split and three-split strategies, respectively. Same convention are also followed for $\hat\theta^{\mathrm{NR}}$ and $\hat\theta^{\mathrm{DR}}$. However, these two sample splitting strategies result in different effective sample sizes being used for constructing the final estimator. For notational convenience, we consistently use $n$ to denote the number of samples involved in the construction of the final estimator. This means we implicitly assume access to $2n$ total samples when employing the two-split approach, and $3n$ samples for the three-split approach.

## 3 Main results

### 3.1 Bias Correction

In the previous section, we introduced three distinct strategies for estimating the target parameter $\theta_0$, and all of them rely on estimating the nuisance components $\alpha_0$ and $\beta_0$ (directly or indirectly), which can be learned either from a common auxiliary sample (two-split approach) or from two separate subsamples (three-split approach). However, all six resulting estimators suffer from a non-negligible asymptotic bias, which must be carefully removed to even achieve consistency for $\theta_0$. In the following sections, we will motivate and develop the bias correction strategy for each of the six estimators.

#### 3.1.1 Integral-Based Estimator

To illustrate the origin of this bias, and to motivate our debiasing strategy, let us consider the three-split version of $\hat{\theta}^{\text{INT}}$ (Equation (2.3)), i.e., we estimate $\alpha_0$ and $\beta_0$ from two separate subsample, then use a third subsample to estimate $\hat{\theta}^{\text{INT}}$. Since $\mathbb{E}[AY] = \theta_0 + \alpha_0^\top \beta_0$, it follows immediately that:

$$\mathbb{E}[\hat{\theta}^{\text{INT}}] = \theta_0 + \alpha_0^\top \beta_0 - \mathbb{E}\left[\hat{\alpha}(\lambda_1)^\top \hat{\beta}(\lambda_2)\right] . \tag{3.1}$$

As $\hat{\alpha}(\lambda_1)$ and $\hat{\beta}(\lambda_2)$ are estimated from a separate sub-sample, they are independent, and $X$ is independent of $(\epsilon, \mu)$. Using this, it is easy to see that:

$$\mathbb{E}\left[\hat{\alpha}(\lambda_1)^\top \hat{\beta}(\lambda_2)\right] = \mathbb{E}\left[\alpha_0^\top \hat{\Sigma}_1(\hat{\Sigma}_1 + \lambda_1 \mathbf{I}_p)^{-1}(\hat{\Sigma}_2 + \lambda_2 \mathbf{I}_p)^{-1}\hat{\Sigma}_2 \beta_0\right]$$

If $p/n \to c \in (0, \infty)$, one can show using tools from random matrix theory that (see Appendix for details):

$$\mathbb{E}\left[\alpha_0^\top \hat{\Sigma}_1(\hat{\Sigma}_1 + \lambda_1 \mathbf{I}_p)^{-1}(\hat{\Sigma}_2 + \lambda_2 \mathbf{I}_p)^{-1}\hat{\Sigma}_2 \beta_0\right] = \alpha_0^\top \beta_0 \left(\int \frac{x \, dF_{\text{MP}}(x)}{x + \lambda_1}\right) \left(\int \frac{x \, dF_{\text{MP}}(x)}{x + \lambda_2}\right) + o(1)$$

$$\triangleq \alpha_0^\top \beta_0 \, g_{\text{3sp}}^{\text{INT}}(\lambda_1, \lambda_2) + o(1) .$$

where $F_{\text{MP}}$ is the famous Marchenko-Pastur distribution (Section 3 of [24]), which arises as the limiting spectral distribution (LSD) of the eigenvalues of the sample covariance matrix ($\hat{\Sigma}_1$ or $\hat{\Sigma}_2$). The above sketch indicates that $\mathbb{E}[\hat{\alpha}(\lambda_1)^\top \hat{\beta}(\lambda_2)] \approx (\alpha_0^\top \beta_0) g_{\text{3sp}}^{\text{INT}}(\lambda_1, \lambda_2)$, i.e. the ridge-regression based plug-in estimates approximate $(\alpha_0^\top \beta_0)$ up to a scalar function $g_{\text{3sp}}^{\text{INT}}(\lambda_1, \lambda_2)$. This scalar function, however, is deterministic, as it does not depend on any unknown parameters. Therefore, it naturally motivates a simple debiasing procedure by *inverting the asymptotic bias*:

$$\mathbb{E}\left[\hat{\alpha}(\lambda_1)^\top \hat{\beta}(\lambda_2)\right] \approx (\alpha_0^\top \beta_0) \, g_{\text{3sp}}^{\text{INT}}(\lambda_1, \lambda_2) \implies \mathbb{E}\left[(\hat{\alpha}(\lambda_1)^\top \hat{\beta}(\lambda_2))/g_{\text{3sp}}^{\text{INT}}(\lambda_1, \lambda_2)\right] \approx \alpha_0^\top \beta_0 .$$

Consequently, we can re-define a debiased version of $\hat{\theta}^{\text{INT}}$ (using three-split) as:

$$\hat{\theta}_{\text{3sp}}^{\text{INT,db}} = \frac{1}{n} \sum_{i \in \mathcal{D}_3} A_i Y_i - \frac{\hat{\alpha}(\lambda_1)^\top \hat{\beta}(\lambda_2)}{g_{\text{3sp}}^{\text{INT}}(\lambda_1, \lambda_2)} . \tag{3.2}$$

In Theorem 3.1, we establish that the above estimator is not only asymptotically unbiased but also achieves the optimal $\sqrt{n}$-rate for estimating $\theta_0$. However, in the case of the two-split approach, where $\alpha_0$ and $\beta_0$ are estimated from the same subsample, the bias correction step becomes slightly more involved, due to the correlation between the associated estimation errors. As evident from Equation (2.2), the error in estimating $\hat{\alpha}(\lambda_1)$ depends on $\epsilon$, while that in $\hat{\beta}(\lambda_2)$ depends on $\mu$, which are themselves correlated. As a result, an additional bias term arises, given by:

$$\mathbb{E}[\hat{\alpha}(\lambda_1)^\top \hat{\beta}(\lambda_2)] = \mathbb{E}[\alpha_0^\top \hat{\Sigma}_1(\hat{\Sigma}_1 + \lambda_1 \mathbf{I}_p)^{-1}(\hat{\Sigma}_1 + \lambda_2 \mathbf{I}_p)^{-1}\hat{\Sigma}_1 \beta_0]$$
$$+ \frac{1}{n^2}\mathbb{E}[\epsilon^\top \mathbf{X}(\hat{\Sigma}_1 + \lambda_1 \mathbf{I}_p)^{-1}(\hat{\Sigma}_1 + \lambda_2 \mathbf{I}_p)^{-1}\mathbf{X}^\top \mu] \tag{3.3}$$

Again, some tools from random matrix theory yield (details are provided in the Appendix):

$$\mathbb{E}[\hat{\alpha}(\lambda_1)^\top \hat{\beta}(\lambda_2)] = (\alpha_0^\top \beta_0) \int \frac{x^2 \, dF_{\text{MP}}(x)}{(x + \lambda_1)(x + \lambda_2)} + \theta_0 \int \frac{cx \, dF_{\text{MP}}(x)}{(x + \lambda_1)(x + \lambda_2)} + o(1)$$
$$\triangleq (\alpha_0^\top \beta_0) \, g_{1,\text{2sp}}^{\text{INT}}(\lambda_1, \lambda_2) + \theta_0 \, g_{2,\text{2sp}}^{\text{INT}}(\lambda_1, \lambda_2) .$$

Interestingly, now the bias of $\hat{\alpha}(\lambda_1)^\top \hat{\beta}(\lambda_2)$ not only involves $\alpha_0^\top \beta_0$, but also involves $\rho = \theta_0$. Therefore, we cannot simply divide $\hat{\alpha}(\lambda_1)^\top \hat{\beta}(\lambda_2)$ by some function of the tuning parameters to

remove the bias, as we also have to take care of the second part. Following the same inversion trick as in the three-split case, we now have:

$$\mathbb{E}[\hat{\alpha}(\lambda_1)^\top \hat{\beta}(\lambda_2)/g_{1,2\text{sp}}^{\text{INT}}(\lambda_1, \lambda_2)] \approx \alpha_0^\top \beta_0 + \theta_0 \left(g_{2,2\text{sp}}^{\text{INT}}(\lambda_1, \lambda_2)/g_{1,2\text{sp}}^{\text{INT}}(\lambda_1, \lambda_2)\right).$$

And consequently:

$$\mathbb{E}\left[\frac{1}{n}\sum_{i \in \mathcal{D}_2} A_i Y_i - \frac{\hat{\alpha}(\lambda_1)^\top \hat{\beta}(\lambda_2)}{g_{1,2\text{sp}}^{\text{INT}}(\lambda_1, \lambda_2)}\right] \approx \theta_0 \left(1 - \frac{g_{2,2\text{sp}}^{\text{INT}}(\lambda_1, \lambda_2)}{g_{1,2\text{sp}}^{\text{INT}}(\lambda_1, \lambda_2)}\right).$$

Now, the above equation suggests using the inversion trick one more time (as both $g_{1,2\text{sp}}^{\text{INT}}$ and $g_{2,2\text{sp}}^{\text{INT}}$ are deterministic) to obtain the following debiased version of $\hat{\theta}^{\text{INT}}$:

$$\hat{\theta}_{2sp}^{\text{INT,db}} = \left(1 - \frac{g_{2,2\text{sp}}^{\text{INT}}(\lambda_1, \lambda_2)}{g_{1,2\text{sp}}^{\text{INT}}(\lambda_1, \lambda_2)}\right)^{-1} \left[\frac{1}{n}\sum_{i \in \mathcal{D}_2} A_i Y_i - \frac{\hat{\alpha}(\lambda_1)^\top \hat{\beta}(\lambda_2)}{g_{1,2\text{sp}}^{\text{INT}}(\lambda_1, \lambda_2)}\right]. \tag{3.4}$$

We next present one of our main results, establishing that these debiased versions of the integral-based estimator are indeed $\sqrt{n}$-consistent.

**Theorem 3.1.** *Consider the debiased version of the integral-based estimator $\hat{\theta}^{\text{INT,db}}$ either obtained by a two-split approach (Equation (3.4)) or a three-split approach (Equation (3.2)). Assume $X_{ij}$ are iid subgaussian random variables with mean 0, variance 1, with uniformly (over $p$) bounded subgaussian norm. Furthermore, assume that $(\epsilon, \nu) \perp\!\!\!\perp X$ and follows a bivariate normal distribution with mean 0, variance 1, and correlation $\rho$, and there exists $C_\alpha, C_\beta > 0$ such that $\|\alpha_0\|_2 \leq C_\alpha$ and $\|\beta_0\|_2 \leq C_\beta$ for all $p \in \mathbb{N}$. Then, we have:*

$$\sqrt{n}(\hat{\theta}^{\text{INT,db}} - \theta_0) = O_p(1) \quad \text{both for two and three split versions}.$$

### 3.1.2 Newey-Robins (NR) Estimator

In this subsection, we extend our bias-correction technique to the estimator $\hat{\theta}^{\text{NR}}$ (Equation (2.4)). To understand the nature of the bias, we once again consider the three-split version of $\hat{\theta}^{\text{NR}}$. The expected value of this estimator is given by:

$$\mathbb{E}\left[\hat{\theta}_{3\text{sp}}^{\text{NR}}\right] = \theta_0 + \alpha_0^\top \beta_0 - \mathbb{E}[\hat{\alpha}(\lambda_1)^\top \beta_0] = \theta_0 + \alpha_0^\top \beta_0 - \mathbb{E}[\beta_0^\top (\hat{\Sigma}_1 + \lambda_1 \mathbf{I}_p)^{-1} \hat{\Sigma}_1 \alpha_0]$$

$$= \theta_0 + \alpha_0^\top \beta_0 \left(1 - \int \frac{x}{x + \lambda_1} dF_{\text{MP}}(x)\right) + o(1)$$

$$\triangleq \theta_0 + \alpha_0^\top \beta_0 \, g_{3\text{sp}}^{\text{NR}}(\lambda_1, \lambda_2) + o(1),$$

where, as before, the last line follows using some techniques from random matrix theory, which can be found in the Appendix. To eliminate this additional bias (i.e., the second term in the equation above), we adopt the same inversion technique introduced in the previous subsection. As noted earlier, when $\hat{\alpha}(\lambda_1)$ and $\hat{\beta}(\lambda_2)$ are estimated from two independent subsamples, we have $\mathbb{E}\left[\hat{\alpha}(\lambda_1)^\top \hat{\beta}(\lambda_2)/g_{3\text{sp}}^{\text{INT}}(\lambda_1, \lambda_2)\right] \approx \alpha_0^\top \beta_0$. We leverage this observation to correct the bias as follows:

$$\hat{\theta}_{3\text{sp}}^{\text{NR,db}} = \left\{\frac{1}{n}\sum_{i \in \mathcal{D}_3} Y_i(A_i - X_i^\top \hat{\alpha}(\lambda_1))\right\} - \hat{\alpha}(\lambda_1)^\top \hat{\beta}(\lambda_2) \frac{g_{3\text{sp}}^{\text{NR}}(\lambda_1, \lambda_2)}{g_{3\text{sp}}^{\text{INT}}(\lambda_1, \lambda_2)}. \tag{3.5}$$

For the two-split approach, the bias correction is a bit more involved as of $\hat{\theta}^{\text{INT}}$, due to the correlation between the estimation error of $\alpha_0$ and $\beta_0$. However, a similar approach as before yields the following debiased estimator:

$$\hat{\theta}_{2\text{sp}}^{\text{NR,db}} = \left(1 - \frac{g_{2,2\text{sp}}^{\text{INT}}(\boldsymbol{\lambda}) g_{3\text{sp}}^{\text{NR}}(\boldsymbol{\lambda})}{g_{1,2\text{sp}}^{\text{INT}}(\boldsymbol{\lambda})}\right)^{-1} \left[\frac{1}{n}\sum_{i \in \mathcal{D}_2} Y_i(A_i - X_i^\top \hat{\alpha}(\lambda_1)) - \hat{\alpha}(\lambda_1)^\top \hat{\beta}(\lambda_2)\frac{g_{2,2\text{sp}}^{\text{INT}}(\boldsymbol{\lambda}) g_{3\text{sp}}^{\text{NR}}(\boldsymbol{\lambda})}{g_{1,2\text{sp}}^{\text{INT}}(\boldsymbol{\lambda})}\right]$$
$$\tag{3.6}$$

with $\boldsymbol{\lambda} = (\lambda_1, \lambda_2)$. The following theorem establishes that these bias-corrected estimators are also $\sqrt{n}$-consistent:

**Theorem 3.2.** *Consider the debiased version of the Newey-Robins estimator $\hat{\theta}^{\text{NR,db}}$ either obtained by a two-split approach (Equation (3.6)) or three-split approach (Equation (3.5)). Under the same assumptions as in Theorem 3.1, we have:*

$$\sqrt{n}(\hat{\theta}^{\text{NR,db}} - \theta_0) = O_p(1) \quad \text{both for two and three split versions}.$$

### 3.1.3 Doubly Robust Estimator

Last but not least, we turn to the bias correction procedure for the doubly robust estimator of $\theta_0$, denoted by $\hat{\theta}^{\mathrm{DR}}$ and defined in Equation (2.5). As discussed in Section 2, this estimator has a well-established history in the literature, particularly from the perspective of double machine learning. A straightforward computation of the expectation yields:

$$\mathbb{E}[\hat{\theta}^{\mathrm{DR}}] = \theta_0 + \mathbb{E}[(\alpha_0 - \hat{\alpha}(\lambda_1))(\beta_0 - \hat{\beta}(\lambda_2))]. \tag{3.7}$$

Therefore, the primary source of the bias is the second term of the above equation. Under the three-split setup, one can show that:

$$\mathbb{E}[(\alpha_0 - \hat{\alpha}(\lambda_1))^\top (\beta_0 - \hat{\beta}(\lambda_2))] = \alpha_0^\top \beta_0 \left( \int \frac{\lambda_1}{x+\lambda_1}\, dF_{\mathrm{MP}}(x) \right) \left( \int \frac{\lambda_2}{x+\lambda_2}\, dF_{\mathrm{MP}}(x) \right) + o(1)$$
$$\triangleq (\alpha_0^\top \beta_0)\, g_{3\mathrm{sp}}^{\mathrm{DR}}(\lambda_1, \lambda_2) + o(1).$$

The calculation proceeds similarly to that for $\hat{\theta}^{\mathrm{INT}}$ or $\hat{\theta}^{\mathrm{NR}}$, which we skip for brevity; the key step is to derive the asymptotic bias of $\hat{\alpha}(\lambda_1)^\top \hat{\beta}(\lambda_2)$. This characterization of bias leads to the following bias-corrected estimator:

$$\hat{\theta}_{3\mathrm{sp}}^{\mathrm{DR,db}} = \hat{\theta}^{\mathrm{DR}} - \hat{\alpha}(\lambda_1)^\top \hat{\beta}(\lambda_2) \frac{g_{3\mathrm{sp}}^{\mathrm{DR}}(\lambda_1,\lambda_2)}{g_{3\mathrm{sp}}^{\mathrm{INT}}(\lambda_1,\lambda_2)}. \tag{3.8}$$

However, as explained for the previous two estimators, the bias takes a more complicated form for the two-sample split. Similar calculation using the tools from deterministic equivalence in the random matrix theory yields:

$$\mathbb{E}[\hat{\theta}^{\mathrm{DR}}] = \theta_0 + \mathbb{E}[(\alpha_0 - \hat{\alpha}(\lambda_1))^\top (\beta_0 - \hat{\beta}(\lambda_2))]$$
$$= \theta_0 + (\alpha_0^\top \beta_0) \int \frac{\lambda_1 \lambda_2\, dF_{\mathrm{MP}}(x)}{(x+\lambda_1)(x+\lambda_2)} + \rho \int \frac{cx\, dF_{\mathrm{MP}}(x)}{(x+\lambda_1)(x+\lambda_2)} + o(1)$$
$$= \theta_0 \left( 1 + \int \frac{cx\, dF_{\mathrm{MP}}(x)}{(x+\lambda_1)(x+\lambda_2)} \right) + (\alpha_0^\top \beta_0) \int \frac{\lambda_1 \lambda_2\, dF_{\mathrm{MP}}(x)}{(x+\lambda_1)(x+\lambda_2)} + o(1)$$
$$\triangleq \theta_0 \left( 1 + g_{2,2\mathrm{sp}}^{\mathrm{INT}}(\lambda_1, \lambda_2) \right) + (\alpha_0^\top \beta_0)\, g_{2,2\mathrm{sp}}^{\mathrm{DR}}(\lambda_1, \lambda_2) + o(1).$$

The details can be found in the Appendix, which we do not provide here for the sake of space. Based on this expression, we propose the following bias-corrected estimator:

$$\hat{\theta}_{2\mathrm{sp}}^{\mathrm{DR,db}} = \left( 1 + g_{2,2\mathrm{sp}}^{\mathrm{INT}}(\boldsymbol{\lambda}) \left( 1 - \frac{g_{2,2\mathrm{sp}}^{\mathrm{DR}}(\boldsymbol{\lambda})}{g_{1,2\mathrm{sp}}^{\mathrm{INT}}(\boldsymbol{\lambda})} \right) \right)^{-1} \left[ \hat{\theta}^{\mathrm{DR}} - \frac{\hat{\alpha}(\lambda_1)^\top \hat{\beta}(\lambda_2) g_{2,2\mathrm{sp}}^{\mathrm{DR}}(\boldsymbol{\lambda})}{g_{1,2\mathrm{sp}}^{\mathrm{INT}}(\boldsymbol{\lambda})} \right]. \tag{3.9}$$

The following theorem establishes that these bias-corrected estimators are also $\sqrt{n}$-consistent:

**Theorem 3.3.** *Consider the debiased version of the doubly robust estimator $\hat{\theta}^{\mathrm{DR,db}}$ either obtained by a two-split approach (Equation (3.9)) or a three-split approach (Equation (3.8)). Under the same assumptions as in Theorem 3.1, we have:*

$$\sqrt{n}(\hat{\theta}^{\mathrm{DR,db}} - \theta_0) = O_p(1) \quad \text{both for two and three split versions}.$$

### 3.2 Limiting Variance

In the previous subsection, we established that the bias-corrected versions of all three estimators, namely $\hat{\theta}^{\mathrm{INT}}$ are $\sqrt{n}$-consistent for any choice of $\lambda_1, \lambda_2 > 0$. This naturally raises a question for practitioners: *which estimator should one use in practice?* The answer, of course, depends on the downstream objective. Since our goal is to perform inference on $\theta_0$, the ideal choice would be the estimator that achieves the *minimum asymptotic variance*. Therefore, it is imperative to develop a precise understanding of the asymptotic variance associated with each estimator, how this variance is influenced by the choice of estimator, and the sample splitting strategy employed. However, quantifying the asymptotic variance of this debiased estimator in the proportional asymptotic regime remains a challenging task and requires much finer analysis. To the best of our knowledge, only a few papers (e.g., [2, 25, 23]) have addressed related questions; although, they are limited to the setting where $c = p/n \in (0, 1)$. In this paper, we take a first step toward characterizing the asymptotic variance of our debiased estimators in the regime where $c = p/n \in (0, \infty)$, i.e., we allow $p > n$. To facilitate this analysis, we assume that the covariates $X_{ij}$ are independent and follow $\mathcal{N}(0, 1)$.

Gaussianity enables the use of powerful tools, particularly those involving the analysis of Haar-distributed eigenvectors and the independence between eigenvalues and eigenvectors, both of which play a crucial role in simplifying our analysis. While we believe that our results extend beyond the Gaussian setting, we leave such generalizations as interesting avenues for future research. The following theorem provides the exact asymptotic variance of the six debiased estimators introduced previously:

**Theorem 3.4.** *Consider the same setup as in Theorem 3.1, along with the assumption that $X_{ij} \overset{\text{i.i.d.}}{\sim} \mathcal{N}(0,1)$. Furthermore, assume that there exists $(u, v, \varrho)$ such that:*

$$\lim_{n\uparrow\infty} \|\alpha_0\|_2^2 = u, \quad \lim_{n\uparrow\infty} \|\beta_0\|_2^2 = v, \quad \lim_{n\uparrow\infty} \alpha_0^\top \beta_0 = \varrho.$$

*Then there exists six functions $\{V_{2\text{sp}}^{\text{INT}}, V_{3\text{sp}}^{\text{INT}}, V_{2\text{sp}}^{\text{NR}}, V_{3\text{sp}}^{\text{NR}}, V_{2\text{sp}}^{\text{DR}}, V_{3\text{sp}}^{\text{DR}}\}$ such that:*

$$\lim_{n\uparrow\infty} n \times \mathsf{var}\left(\hat{\theta}_v^{A,db}\right) = V_v^A(\lambda_1, \lambda_2)$$

*for $A \in \{\text{INT}, \text{NR}, \text{DR}\}$ and $v \in \{2\text{sp}, 3\text{sp}\}$. The expression of $V_{2\text{sp}}^{\text{INT}}$ and $V_{3\text{sp}}^{\text{INT}}$ are provided in Equation (3.11) and Equation (3.12) respectively, and the expression for the rest of the variances can be found in the Appendix.*

The above theorem provides us with the precise limiting variance of six estimators. Ideally, one should choose the estimation which has minimum variance over all possible limiting variances, i.e. $\hat{\theta}^{\text{OPT}}$ should be $\hat{\theta}_{v^*}^{A^*,db}$, where $A^*$ and $v^*$ should be chosen as:

$$(A^*, v^*) = \arg\min_{\substack{A \in \{\text{INT,NR,DR}\} \\ v \in \{2\text{sp},3\text{sp}\}}} \min_{\lambda_1, \lambda_2 > 0} V_v^A(\lambda_1, \lambda_2). \tag{3.10}$$

The reason we do not include the exact expressions for all the asymptotic variances is primarily due to space constraints; the formulas are quite involved and lengthy. Below, we present the expressions for $V_{2sp}^{\text{INT}}$ and $V_{3sp}^{\text{INT}}$ to illustrate the complexity and structure of these quantities:

$$
\begin{aligned}
V_{2sp}^{\text{INT}} = \left(1 - \frac{g_{2,2\text{sp}}^{\text{INT}}(\boldsymbol{\lambda})}{g_{1,2\text{sp}}^{\text{INT}}(\boldsymbol{\lambda})}\right)^{-2} & \left(\mathsf{var}(AY) + \frac{u^2+v^2+2\varrho+c(1+\rho^2)}{(g_{1,2\text{sp}}^{\text{INT}}(\boldsymbol{\lambda}))^2} \int \frac{x^2 \, dF_{\text{MP}}(x)}{(x+\lambda_1)^2(x+\lambda_2)^2} \right. \\
& \left. + \frac{u^2v^2+\varrho^2}{c\,(g_{1,2\text{sp}}^{\text{INT}}(\boldsymbol{\lambda}))^2} \mathsf{var}_{F_{\text{MP}}}\left(\frac{X^2}{(X+\lambda_1)(X+\lambda_2)}\right)\right).
\end{aligned}
\tag{3.11}
$$

$$
\begin{aligned}
V_{3sp}^{\text{INT}} = \mathsf{var}(AY) &+ \frac{1}{(g_{3\text{sp}}^{\text{INT}}(\lambda_1,\lambda_2))^2}\left[\frac{\varrho^2+u^2v^2}{c}\mathsf{var}_{X\sim F_{\text{MP}}}\left(\frac{X}{X+\lambda_2}\right) + u^2\int \frac{x\,dF_{\text{MP}}(x)}{(x+\lambda_2)^2}\right]\left(\int \frac{x\,dF_{\text{MP}}(x)}{x+\lambda_1}\right)^2 \\
&+ \frac{1}{c}\mathsf{var}_{X\sim F_{\text{MP}}}\left(\frac{X}{X+\lambda_1}\right)\left\{\varrho^2\left(\int \frac{x\,dF_{\text{MP}}(x)}{x+\lambda_2}\right)^2 + u^2\left[v^2\left(\int \frac{x^2\,dF_{\text{MP}}(x)}{(x+\lambda_2)^2}\right) + c\left(\int \frac{x\,dF_{\text{MP}}(x)}{(x+\lambda_2)^2}\right)\right]\right\} \\
&+ \left\{v^2\left(\int \frac{x^2\,dF_{\text{MP}}(x)}{(x+\lambda_2)^2}\right) + c\left(\int \frac{x\,dF_{\text{MP}}(x)}{(x+\lambda_2)^2}\right)\right\}\left(\int \frac{x\,dF_{\text{MP}}(x)}{(x+\lambda_1)^2}\right).
\end{aligned}
\tag{3.12}
$$

It is worth noting that the optimal choice of $(\lambda_1, \lambda_2)$ (which is the minimizer of Equation (3.10)) differs significantly from the prediction-optimal choice of $(\lambda_1, \lambda_2)$ (derived in the Appendix), as illustrated by our simulation in Section 4. This departs from the traditional DML framework, where the nuisance parameters are typically estimated using prediction-optimal methods. However, in the proportional asymptotic regime, such prediction-optimal choices do not necessarily yield the best performance for inference. As a result, it is more appropriate to directly minimize the asymptotic variance rather than predict optimal choices to achieve a more accurate inference/narrower confidence interval.

In practice, the asymptotic variances can be estimated by using resampling techniques. We outline a parametric bootstrap approach to estimate the asymptotic variances in the Appendix, which we evaluate in our simulations.

# 4 Simulations

We assessed the performance of the ECC estimators through simulations under the different sample splitting strategies. We considered the size of the splits to be $500$, making the total sample size $N = 1000$ for the two sample splitting case and $N = 1500$ for the three sample splitting case. For

each sample splitting case, we set $p = 250$ (i.e., $c = 0.25$) and $p = 1000$ (i.e., $c = 2$). In each case, we generated $10,000$ independent data sets, each consisting of iid copies of $(A, X, Y)$. Specifically, we consider a random vector $X$ of dimension $p$, generated as $X \sim \mathcal{N}(0, I_p)$. Then, we generated $A$ and $Y$ with $\rho = 0.5$ by Equation (2.1). We applied the integral-based estimator, the Newey-Robins estimator, and the doubly robust estimator, where we set $\lambda_1 = \lambda_2 = \lambda$ and set the bias correction constants by using Monte Carlo with $10,000$ iterations. Each estimator was applied across 100 distinct values of $\lambda$, spanning a grid from 0.05 to 10.

We investigated the impact of the tuning parameter $\lambda$ on the asymptotic variance of the debiased estimators. Results for $p = 1000$ (i.e., $c = 2$) are presented in Figure 1, while results for the $p = 250$ setting (i.e., $c = 0.5$) are provided in the Appendix. Interestingly, we demonstrate that prediction-optimal choices of the tuning parameters fail to yield statistically optimal estimates of ECC, and one should rather aim at optimizing the variance of the ECC directly and carefully to account for the type of sample splitting and estimator chosen. We note that in the two-split scenario, the variance blows up at values around $\lambda = 1.48$ due to the bias-correction constant diverging. However, this does not affect our primary conclusions, as our goal is to select the value of $\lambda$ that minimizes the asymptotic variance.

Due to space constraints, we present additional details of the simulations and our other analyses in the Appendix. In brief, we found that the debiased estimators had negligible bias (unlike the original estimators) and that the parametric bootstrap approach often performed well for estimating the asymptotic variance.

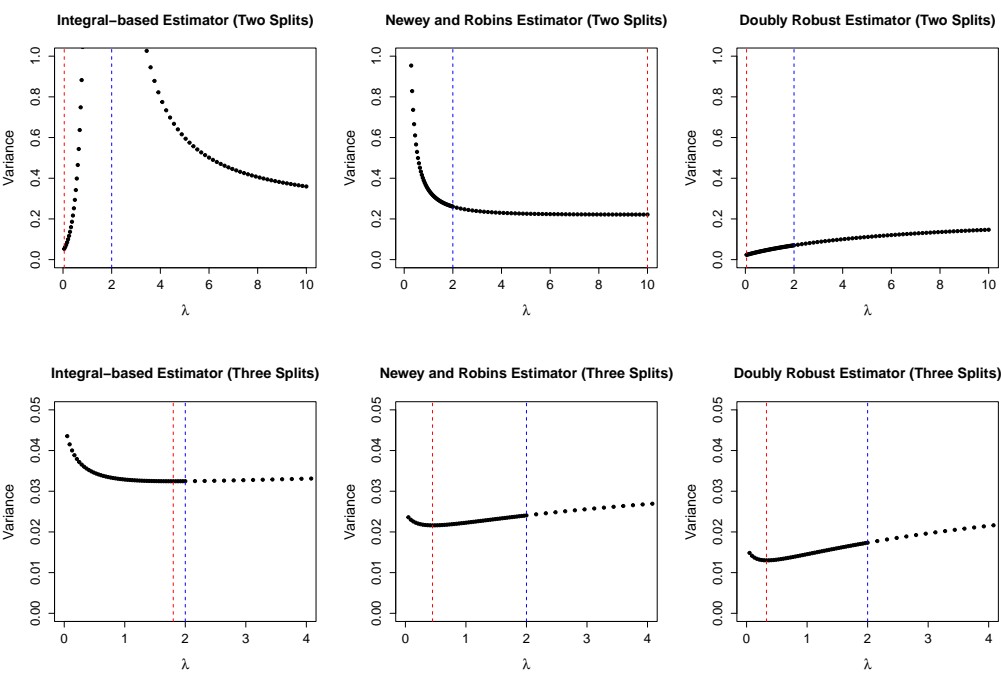

Figure 1: Variances of debiased estimators as functions of tuning parameter $\lambda$. Top row: Two-split ($N = 1000$ total, split into two subsamples of 500 each). Bottom row: Three-split ($N = 1500$ total, split into three subsamples of 500 each). Red line indicates the optimal $\lambda$ for estimator variance; Blue line is for prediction-optimal $\lambda$. The figures shown here are zoomed-in views; complete versions are included in the Appendix.

## 5 Conclusion

In this paper, we have explored the intricate interplay among sample splitting, estimator choice, and optimal tuning of nuisance functions for estimating the expected conditional covariance (ECC), a popular doubly robust functional in the literature on causal inference and missing data. The

philosophy we espouse in this regard can be best understood within the DML framework, where typically prediction-optimal nuisance functions are used for downstream efficient estimation of a low-dimensional functional that depends on them. In this paper, we explore the subtleties of this discourse when no consistent nuisance functions exist. In particular, for ECC, this challenging regime arises when the dimension of the problem is proportional to the sample size, and our results uncover a precise asymptotic picture of the interplay between sample splitting, choice of estimator, and optimal tuning of the regularized nuisance function in this regard under a stylized linear model for the nuisance functions. Our analysis reveals an interesting phenomenon: prediction-optimal tuning parameters are not necessarily the best choices for estimating ECC, in terms of its asymptotic variance (and consequently the length of the confidence interval), which differs from standard DML wisdom.

While this is the first work to analyze ECC in the proportional over-parameterized regime, our framework invites several natural extensions. For example, our analysis assumes the Gaussianity of either the errors or the covariates for variance calculation – an assumption that remains to be explored through the lens of universality principles. It is, however, worth noting that, unlike the operational domain of the DML proposal, we consider a regime where asymptotic optimality for ECC has not yet been established, and the literature has been active in constructing estimators that navigate specific contours of the problem difficulty, such as debiasing regularized estimators of the nuisance functions. Indeed, the estimators studied here are not the only ones that achieve $\sqrt{n}$-consistency in this regime; one can completely bypass nuisance function estimation using a method-of-moments approach [26]. One can also envision appealing to more sophisticated debiasing techniques when generic convex regularization, such as Lasso, is employed to estimate the nuisance functions [25] and derive results analogous to those presented here. In this regime, where an efficiency theory is unavailable, the choice of estimator that minimizes asymptotic variance is a priori unclear. We believe that the template of exploration presented here might serve as a framework for such future endeavors. Specifically, the availability of a large class of sample-split-sensitive, optimally tuned estimators of double-robust functionals might shed light on eventual efficiency bounds for estimating the functional itself. A case study of ECC presented here hopefully serves as a compass for future conjectures in this journey.

## Acknowledgments and Disclosure of Funding

Rajarshi Mukherjee was supported by NSF CAREER 2338760. Debarghya Mukherjee was supported by NSF DMS-2515787. The content is solely the responsibility of the authors and does not necessarily represent the official views of the NSF.

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

# A  Proof of Theorem 3.1

## A.1  Proof of Three-Split

Recall that in the three-split strategy, we assume to have $3n$ observations: We use the first $n$ observations ($\mathcal{D}_1$) to estimate $\hat{\alpha}(\lambda_1)$, second set of $n$ observations ($\mathcal{D}_2$) to estimate $\hat{\beta}(\lambda_2)$, and the last set of $n$ observations ($\mathcal{D}_3$) to construct the final estimator. For the three-sample splitting, the debiased version of the integral-based estimator takes the following form:

$$\hat{\theta}_{3\text{sp}}^{\text{db,INT}} = \frac{1}{n}\sum_{i \in \mathcal{D}_3} Y_i A_i - \frac{\hat{\alpha}(\lambda_1)^\top \hat{\beta}(\lambda_2)}{g_{3sp}^{\text{INT}}(\lambda_1,\lambda_2)}\,. \tag{A.1}$$

To establish the $\sqrt{n}$-consistency of $\hat{\theta}_{3\text{sp}}^{\text{db,INT}}$, we start with the definition of the ridge-regularized estimators:

$$\hat{\alpha}(\lambda_1) = \arg\min_\alpha \left\{\tfrac{1}{n}\|\mathbf{a}_1 - \mathbf{X}_1\alpha\|_2^2 + \lambda_1\|\alpha\|_2^2\right\} = (\hat{\Sigma}_1 + \lambda_1\mathbf{I}_p)^{-1}\frac{\mathbf{X}_1^\top \mathbf{a}_1}{n}$$

$$= (\hat{\Sigma}_1 + \lambda_1\mathbf{I}_p)^{-1}\hat{\Sigma}_1\alpha_0 + (\hat{\Sigma}_1 + \lambda_1\mathbf{I}_p)^{-1}\frac{\mathbf{X}_1^\top \boldsymbol{\epsilon}_1}{n}$$

$$= \alpha_0 - \lambda_1(\hat{\Sigma}_1 + \lambda_1\mathbf{I}_p)^{-1}\alpha_0 + (\hat{\Sigma}_1 + \lambda_1\mathbf{I}_p)^{-1}\frac{\mathbf{X}_1^\top \boldsymbol{\epsilon}_1}{n}$$

$$\hat{\beta}(\lambda_2) = \arg\min_\beta \left\{\frac{1}{n}\|\mathbf{y}_2 - \mathbf{X}_2\beta\|_2^2 + \lambda_2\|\beta\|_2^2\right\} = (\hat{\Sigma}_2 + \lambda_2\mathbf{I}_p)^{-1}\frac{\mathbf{X}_2^\top \mathbf{y}_2}{n}$$

$$= (\hat{\Sigma}_2 + \lambda_2\mathbf{I}_p)^{-1}\hat{\Sigma}_2\beta_0 + (\hat{\Sigma}_2 + \lambda_2\mathbf{I}_p)^{-1}\frac{\mathbf{X}_2^\top \boldsymbol{\mu}_2}{n}$$

$$= \beta_0 - \lambda_2(\hat{\Sigma}_2 + \lambda_2\mathbf{I}_p)^{-1}\beta_0 + (\hat{\Sigma}_2 + \lambda_2\mathbf{I}_p)^{-1}\frac{\mathbf{X}_2^\top \boldsymbol{\mu}_2}{n}$$

Hence, we have:

$$\hat{\alpha}(\lambda_1)^\top \hat{\beta}(\lambda_2) = \alpha_0^\top \beta_0 - \alpha_0^\top \lambda_2(\hat{\Sigma}_2 + \lambda_2\mathbf{I})^{-1}\beta_0 + \alpha_0^\top(\hat{\Sigma}_2 + \lambda_2\mathbf{I})^{-1}\frac{\mathbf{X}_2^\top \boldsymbol{\mu}_2}{n}$$

$$- \lambda_1\alpha_0^\top(\hat{\Sigma}_1 + \lambda_1\mathbf{I})^{-1}\beta_0 + \lambda_1\lambda_2\alpha_0^\top(\hat{\Sigma}_1 + \lambda_1\mathbf{I})^{-1}(\hat{\Sigma}_2 + \lambda_2\mathbf{I})^{-1}\beta_0$$

$$- \lambda_1\alpha_0^\top(\hat{\Sigma}_1 + \lambda_1\mathbf{I})^{-1}(\hat{\Sigma}_2 + \lambda_2\mathbf{I})^{-1}\frac{\mathbf{X}_2^\top \boldsymbol{\mu}_2}{n}$$

$$+ \beta_0^\top(\hat{\Sigma}_1 + \lambda_1\mathbf{I})^{-1}\frac{\mathbf{X}_1^\top \boldsymbol{\epsilon}_1}{n} - \lambda_2\beta_0^\top(\hat{\Sigma}_2 + \lambda_2\mathbf{I})^{-1}(\hat{\Sigma}_1 + \lambda_1\mathbf{I})^{-1}\frac{\mathbf{X}_1^\top \boldsymbol{\epsilon}_1}{n}$$

$$+ \frac{\boldsymbol{\epsilon}_1^\top \mathbf{X}_1}{n}(\hat{\Sigma}_1 + \lambda_1\mathbf{I})^{-1}(\hat{\Sigma}_2 + \lambda_2\mathbf{I})^{-1}\frac{\mathbf{X}_2^\top \boldsymbol{\mu}_2}{n}$$

$$= \sum_{j=1}^{9} T_j\,.$$

Combining the parameter $\alpha_0^\top \beta_0$, and the terms that contribute to its estimation bias, namely $T_1, T_2, T_4, T_5$ (observe that all the other terms have mean 0), let us define an aggregated term $\mathscr{B}_n$ as:

$$\mathscr{B}_n = \alpha_0^\top \left(\mathbf{I} - \lambda_1(\hat{\Sigma}_1 + \lambda_1\mathbf{I})^{-1}\right)\left(\mathbf{I} - \lambda_2(\hat{\Sigma}_2 + \lambda_2\mathbf{I})^{-1}\right)\beta_0$$

Using this new notation, we have:

$$\hat{\theta}_{3\text{sp}}^{\text{INT,db}} - \theta_0 = \frac{1}{n}\sum_{i \in \mathcal{D}_3} A_i Y_i - \mathbb{E}[AY] - \left(\frac{\hat{\alpha}(\lambda_1)^\top \hat{\beta}(\lambda_2)}{g_{3\text{sp}}(\lambda_1,\lambda_2)} - \alpha_0^\top \beta_0\right)$$

$$= \frac{1}{n}\sum_{i \in \mathcal{D}_3} A_i Y_i - \mathbb{E}[AY] - \left(\frac{\mathscr{B}_n}{g_{3\text{sp}}^{\text{INT}}(\lambda_1,\lambda_2)} - \alpha_0^\top \beta_0 + \frac{\sum_{j \notin \{1,2,4,5\}} T_j}{g_{3\text{sp}}^{\text{INT}}(\lambda_1,\lambda_2)}\right)$$

As a consequence,

$$\sqrt{n}(\hat{\theta}_{3\text{sp}}^{\text{INT,db}} - \theta_0)$$

$$= \frac{1}{\sqrt{n}} \left( \sum_{i \in \mathcal{D}_3} A_i Y_i - \mathbb{E}[AY] \right) - \sqrt{n} \left( \frac{\mathscr{B}_n}{g_{3\mathrm{sp}}^{\mathrm{INT}}(\lambda_1, \lambda_2)} - \alpha_0^\top \beta_0 \right) - \sqrt{n} \frac{\sum_{j \notin \{1,2,4,5\}} T_j}{g_{3\mathrm{sp}}^{\mathrm{INT}}(\lambda_1, \lambda_2)} . \quad \text{(A.2)}$$

That the first term is $O_p(1)$ (in fact, asymptotically normal) follows from CLT as $\mathrm{var}(AY) < \infty$. We now show that:

$$\left| \mathscr{B}_n - (\alpha_0^\top \beta_0) \, g_{3\mathrm{sp}}^{\mathrm{INT}}(\lambda_1, \lambda_2) \right| = O_p\left( \frac{1}{\sqrt{n}} \right) . \quad \text{(A.3)}$$

Towards that end, we apply Theorem 1.1 of [27]. Conditioning on $\mathcal{D}_2$, and defining $\omega := \left( \mathbf{I} - \lambda_2(\hat{\Sigma}_2 + \lambda_2\mathbf{I})^{-1} \right) \beta_0$, we have:

$$\mathbb{E}\left[ \sqrt{n} \left| \mathscr{B}_n - (1 - \lambda_1 m_{\mathrm{MP}}(-\lambda_1)) \alpha_0^\top \omega \right| \mid \mathcal{D}_2 \right] \le \|\alpha_0\|_2 \|\omega_2\| \frac{C}{\psi(\lambda_1)} , \quad \text{(A.4)}$$

for some function $\psi(\cdot)$ and constant $C > 0$ (see the proof of Theorem 1.1 of [27] for details). That $\|\alpha_0\| \le C_\alpha$ is by our modelling assumption (see Theorem 3.1). For the other term, observe that:

$$\begin{aligned}
\|\omega\|_2^2 &= \left\| \left( \mathbf{I} - \lambda_2(\hat{\Sigma}_2 + \lambda_2\mathbf{I})^{-1} \right) \beta_0 \right\|_2^2 = \beta_0^\top \left( \mathbf{I} - \lambda_2(\hat{\Sigma}_2 + \lambda_2\mathbf{I})^{-1} \right)^2 \beta_0 \\
&= \sum_{j=1}^{p} (\hat{v}_j^\top \beta_0)^2 \left( 1 - \frac{\lambda_2}{(\hat{\lambda}_j + \lambda_2)} \right)^2 \\
&= \sum_{j=1}^{p} (\hat{v}_j^\top \beta_0)^2 \left( \frac{\hat{\lambda}_j}{\hat{\lambda}_j + \lambda_2} \right)^2 \\
&\le \sum_{j=1}^{p} (\hat{v}_j^\top \beta_0)^2 = \|\beta_0\|_2^2 \le C_\beta^2 .
\end{aligned}$$

Therefore, using this bound and taking expectation on both sides of Equation (A.4), we have:

$$\mathbb{E}\left[ \sqrt{n} \left| \mathscr{B}_n - (1 - \lambda_1 m_{\mathrm{MP}}(-\lambda_1)) \alpha_0^\top \omega \right| \right] \le \frac{C C_\alpha C_\beta}{\psi(\lambda_1)} . \quad \text{(A.5)}$$

Now, another application of Theorem 1.1 of [27] yields the following for some constant $C' > 0$:

$$\mathbb{E}\left[ \sqrt{n} \left| \alpha_0^\top \omega - (1 - \lambda_2 m_{\mathrm{MP}}(-\lambda_2)) \alpha_0^\top \beta_0 \right| \right] \le \frac{C' C_\alpha C_\beta}{\psi(\lambda_2)} . \quad \text{(A.6)}$$

Combining Equation (A.5) and (A.6), we conclude Equation (A.3). Therefore, it only remains to establish the tightness of the last term of Equation (A.2). As a starter, consider $\sqrt{n}T_3$:

$$\sqrt{n}T_3 = \alpha_0^\top (\hat{\Sigma}_2 + \lambda_2\mathbf{I})^{-1} \frac{\mathbf{X}_2^\top \boldsymbol{\mu}_2}{\sqrt{n}} .$$

As $\mu \sim \mathcal{N}(0, 1)$ is of $X$, we have:

$$\sqrt{n} \, T_3 \mid \mathbf{X}_2 \sim \mathcal{N}\left( 0, \alpha_0^\top (\hat{\Sigma}_2 + \lambda_2\mathbf{I})^{-1} \hat{\Sigma}_2 (\hat{\Sigma}_2 + \lambda_2\mathbf{I})^{-1} \alpha_0 \right) .$$

Furthermore, observe that

$$\begin{aligned}
\alpha_0^\top (\hat{\Sigma}_2 + \lambda_2\mathbf{I})^{-1} \hat{\Sigma}_2 (\hat{\Sigma}_2 + \lambda_2\mathbf{I})^{-1} \alpha_0 &= \sum_{j=1}^{p} (\hat{v}_j^\top \alpha_0)^2 \frac{\hat{\lambda}_j}{(\hat{\lambda}_j + \lambda_2)^2} \\
&\le C(\lambda_2) \sum_{j=1}^{p} (\hat{v}_j^\top \alpha_0)^2 \\
&= C(\lambda_2) \|\alpha_0\|_2^2 \le C(\lambda_1, \lambda_2) C_\alpha^2 .
\end{aligned}$$

Here, $C(\lambda_2)$ is the maximum value of $f(x) = x/(x + \lambda_2)^2$. This implies $T_3 = O_p(n^{-1/2})$. Now, all that remains to show is that $\sqrt{n} \, T_9$ is tight. From the expression of $T_9$, we have:

$$\sqrt{n} \, T_9 = \frac{\epsilon_1^\top \mathbf{X}_1}{\sqrt{n}} (\hat{\Sigma}_1 + \lambda_1\mathbf{I})^{-1} (\hat{\Sigma}_2 + \lambda_2\mathbf{I})^{-1} \frac{\mathbf{X}_2^\top \boldsymbol{\mu}_2}{n}$$

Now conditional of $\mathcal{D}_2$ and $\mathbf{X}_1$, $\sqrt{n}\,T_9$ follows a normal distribution:

$$\sqrt{n}\,T_9 \mid (\mathcal{D}_2, \mathbf{X}_1) \sim \mathcal{N}\left(0, \frac{\boldsymbol{\mu}_2^\top \mathbf{X}_2}{n}(\hat{\Sigma}_2 + \lambda_2 \mathbf{I})^{-1}(\hat{\Sigma}_1 + \lambda_1 \mathbf{I})^{-1}\hat{\Sigma}_1(\hat{\Sigma}_1 + \lambda_1 \mathbf{I})^{-1}(\hat{\Sigma}_2 + \lambda_2 \mathbf{I})^{-1}\frac{\mathbf{X}_2^\top \boldsymbol{\mu}_2}{n}\right)$$

Therefore, all we need to show is that the variance has a uniform upper bound with probability going to 1. Towards that end, we use the fact $\|\hat{\Sigma}\|_{\mathrm{op}} \leq 2(1 + \sqrt{c})^2$ with probability going to 1 ([24]). First of all, we have:

$$\frac{\boldsymbol{\mu}_2^\top \mathbf{X}_2}{n}(\hat{\Sigma}_2 + \lambda_2 \mathbf{I})^{-1}(\hat{\Sigma}_1 + \lambda_1 \mathbf{I})^{-1}\hat{\Sigma}_1(\hat{\Sigma}_1 + \lambda_1 \mathbf{I})^{-1}(\hat{\Sigma}_2 + \lambda_2 \mathbf{I})^{-1}\frac{\mathbf{X}_2^\top \boldsymbol{\mu}_2}{n}$$

$$\leq \frac{2(1+\sqrt{c})^2}{\lambda_1^2 \lambda_2^2}\frac{\boldsymbol{\mu}_2^\top \mathbf{X}_2 \mathbf{X}_2^\top \boldsymbol{\mu}}{n^2}$$

From the independence of $\mu$ and $X$ and the fact that $\mu \sim \mathcal{N}(0,1)$, we have:

$$\mathbb{E}\left[\frac{\boldsymbol{\mu}_2^\top \mathbf{X}_2 \mathbf{X}_2^\top \boldsymbol{\mu}}{n^2}\right] = \frac{1}{n}\mathbb{E}\left[\mathrm{tr}(\hat{\Sigma}_1)\right] = c\mathbb{E}\left[\int x\, d\hat{\pi}_{n,2}(x)\right] = c\int x\, dF_{\mathrm{MP}}(x) + O(n^{-1})\,.$$

Therefore, the variance is uniformly upper bounded with probability approaching 1, which, in turn, implies that $T_9 = O_p(n^{-1/2})$. This concludes the proof.

## A.2 Proof of Two-Split

We first consider the case when $\lambda_1 \neq \lambda_2$. We start with the definition of $\hat{\theta}_{2\mathrm{sp}}^{\mathrm{INT,db}}$, as defined in Equation (3.4):

$$\hat{\theta}_{2\mathrm{sp}}^{\mathrm{INT,db}} - \theta_0 = \left(1 - \frac{g_{2,2\mathrm{sp}}^{\mathrm{INT}}(\lambda_1, \lambda_2)}{g_{1,2\mathrm{sp}}^{\mathrm{INT}}(\lambda_1, \lambda_2)}\right)^{-1}\left[\frac{1}{n}\sum_{i \in \mathcal{D}_2}A_i Y_i - \frac{\hat{\alpha}(\lambda_1)^\top \hat{\beta}(\lambda_2)}{g_{1,2\mathrm{sp}}^{\mathrm{INT}}(\lambda_1, \lambda_2)}\right]$$

$$= \left(1 - \frac{g_{2,2\mathrm{sp}}^{\mathrm{INT}}(\lambda_1, \lambda_2)}{g_{1,2\mathrm{sp}}^{\mathrm{INT}}(\lambda_1, \lambda_2)}\right)^{-1}\left[\frac{1}{n}\sum_{i \in \mathcal{D}_2}A_i Y_i - \mathbb{E}[AY] + \mathbb{E}[AY] - \frac{\hat{\alpha}(\lambda_1)^\top \hat{\beta}(\lambda_2)}{g_{1,2\mathrm{sp}}^{\mathrm{INT}}(\lambda_1, \lambda_2)}\right] - \theta_0$$

That $(\sum_i A_i Y_i)/n - \mathbb{E}[AY]$ is $O_p(n^{-1/2})$, follows from CLT. Furthermore, we have $\mathbb{E}[AY] = \alpha_0^\top \beta_0 + \theta_0$. Therefore, all we need to show

$$\left(1 - \frac{g_{2,2\mathrm{sp}}^{\mathrm{INT}}(\lambda_1, \lambda_2)}{g_{1,2\mathrm{sp}}^{\mathrm{INT}}(\lambda_1, \lambda_2)}\right)^{-1}\left[\alpha_0^\top \beta_0 + \theta_0 - \frac{\hat{\alpha}(\lambda_1)^\top \hat{\beta}(\lambda_2)}{g_{1,2\mathrm{sp}}^{\mathrm{INT}}(\lambda_1, \lambda_2)}\right] - \theta_0 = O_p(n^{-1/2})\,,$$

which is equivalent to show the following:

$$\hat{\alpha}(\lambda_1)^\top \hat{\beta}(\lambda_2) = (\alpha_0^\top \beta_0)\, g_{1,2\mathrm{sp}}^{\mathrm{INT}}(\lambda_1, \lambda_2) + \theta_0\, g_{2,2\mathrm{sp}}^{\mathrm{INT}}(\lambda_1, \lambda_2) + O_p(n^{-1/2})\,.$$

For the rest of the proof, we denote by $R(\lambda) = (\hat{\Sigma}_1 + \lambda \mathbf{I})^{-1}$ for notational simplicity. We use the following identity in our proof:

$$\frac{1}{\lambda_2 - \lambda_1}(R(\lambda_1) - R(\lambda_2)) = R(\lambda_1)R(\lambda_2) \tag{A.7}$$

which follows from the fact $A^{-1} - B^{-1} = A^{-1}(B - A)B^{-1}$. We start with the definition of $\hat{\alpha}(\lambda_1)^\top \hat{\beta}(\lambda_2)$:

$$\hat{\alpha}(\lambda_1)^\top \hat{\beta}(\lambda_2) - (\alpha_0^\top \beta_0)\, g_{1,2\mathrm{sp}}^{\mathrm{INT}}(\lambda_1, \lambda_2) - \theta_0\, g_{2,2\mathrm{sp}}^{\mathrm{INT}}(\lambda_1, \lambda_2)$$

$$= \underbrace{\alpha_0^\top (\mathbf{I} - \lambda_1 R(\lambda_1))(\mathbf{I} - \lambda_2 R(\lambda_2))\beta_0 - (\alpha_0^\top \beta_0)\, g_{1,2\mathrm{sp}}^{\mathrm{INT}}(\lambda_1, \lambda_2)}_{:=T_1}$$

$$+ \underbrace{\alpha_0^\top \hat{\Sigma}_1 R(\lambda_1)R(\lambda_2)\frac{\mathbf{X}_1^\top \boldsymbol{\mu}_1}{n}}_{:=T_2} + \underbrace{\beta_0^\top \hat{\Sigma}_1 R(\lambda_1)R(\lambda_2)\frac{\mathbf{X}_1^\top \boldsymbol{\epsilon}_1}{n}}_{:=T_3}$$

$$+ \underbrace{\frac{\boldsymbol{\epsilon}^\top \mathbf{X}_1}{n} R(\lambda_1) R(\lambda_2) \frac{\mathbf{X}_1^\top \boldsymbol{\mu}_1}{n} - \theta_0 \, g_{2,2\text{sp}}^{\text{INT}}(\lambda_1, \lambda_2)}_{:= T_4} .$$

Our goal is to show each $T_i$ is $O_p(n^{-1/2})$. We start with $T_2$: from the normality of $\mu$ and its independence with $X$, we have:

$$\sqrt{n} \, T_2 \mid \mathbf{X}_1 \sim \mathcal{N}\left(0, \alpha_0^\top \hat{\Sigma}_1 R(\lambda_1) R(\lambda_2) \hat{\Sigma}_1 R(\lambda_2) R(\lambda_1) \hat{\Sigma}_1 \alpha_0\right)$$

Now to show that variance is uniformly upper bounded with probability going to 1, denote by $\hat{\Sigma}_1 = \hat{\mathbf{V}} \hat{\Lambda} \hat{\mathbf{V}}^\top$, the eigendecomposition of $\hat{\Sigma}_1$. Then we have:

$$\alpha_0^\top \hat{\Sigma}_1 R(\lambda_1) R(\lambda_2) \hat{\Sigma}_1 R(\lambda_2) R(\lambda_1) \hat{\Sigma}_1 \alpha_0 = \sum_{j=1}^{p} (\hat{v}_j^\top \alpha_0)^2 \frac{\hat{\lambda}_j^3}{(\hat{\lambda}_j + \lambda_1)^2 (\hat{\lambda}_j + \lambda_2)^2}$$

$$\leq C(\lambda_1, \lambda_2) \sum_{j=1}^{p} (\hat{v}_j^\top \alpha_0)^2$$

$$= C(\lambda_1, \lambda_2) \|\alpha_0\|_2^2 \leq C(\lambda_1, \lambda_2) C_\alpha .$$

Here $C(a, b)$ is the maximum of the function $f(x) = x^3/((x+a)^2(x+b)^2)$ over $[0, \infty)$. This shows that $T_2 = O_p(n^{-1/2})$. The proof of $T_3 = O_p(n^{-1/2})$ is similar, hence skipped. For $T_4$, first observe that

$$n \times \text{var}\left(\frac{\boldsymbol{\epsilon}^\top \mathbf{X}_1}{n} R(\lambda_1) R(\lambda_2) \frac{\mathbf{X}_1^\top \boldsymbol{\mu}_1}{n} \mid \mathbf{X}_1\right) = \frac{(1+\theta_0)^2}{n} \text{tr}\left(R(\lambda_1) R(\lambda_2) \hat{\Sigma}_1 R(\lambda_1) R(\lambda_2) \hat{\Sigma}_1\right)$$

$$= (1+\theta_0)^2 \int \frac{x^2}{(x+\lambda_1)^2 (x+\lambda_2)^2} \, d\hat{\pi}_{n,1}(x)$$

$$\leq (1+\theta_0)^2 C'(\lambda_1, \lambda_2)$$

where $C'(a, b)$ is the maximum of the function $f(x) = x^2/((x+a)^2(x+b)^2)$ over $[0, \infty)$. As the upper bound does not depend on $\mathbf{X}_1$, we have $T_4 - \mathbb{E}[T_4] = O_p(n^{-1/2})$. Now, let us consider $\mathbb{E}[T_4]$:

$$\mathbb{E}[T_4] = \frac{\theta_0}{n} \mathbb{E}\left[\text{tr}\left(R(\lambda_1) R(\lambda_2) \hat{\Sigma}_1\right)\right] - \theta_0 \, g_{2,2\text{sp}}^{\text{INT}}(\lambda_1, \lambda_2)$$

$$= c\theta_0 \, \mathbb{E}\left[\int \frac{x \, d\hat{\pi}_{n,1}(x)}{(x+\lambda_1)(x+\lambda_2)} - \int \frac{x \, dF_{\text{MP}}(x)}{(x+\lambda_1)(x+\lambda_2)}\right]$$

$$= O(n^{-1}) \quad [\text{Theorem 1.1 of [28]}] .$$

Therefore, $T_4 = O_p(n^{-1/2})$. Finally, we need to show $T_1 = O_p(n^{-1/2})$.

$$\alpha_0^\top (\mathbf{I} - \lambda_1 R(\lambda_1))(\mathbf{I} - \lambda_2 R(\lambda_2)) \beta_0 - (\alpha_0^\top \beta_0) \, g_{1,2\text{sp}}^{\text{INT}}(\lambda_1, \lambda_2)$$

$$= \alpha_0^\top \beta_0 (1 - g_{1,2\text{sp}}^{\text{INT}}(\lambda_1, \lambda_2)) - \lambda_1 \alpha_0^\top R(\lambda_1) \beta_0 - \lambda_2 \alpha_0^\top R(\lambda_2) \beta_0 + \lambda_1 \lambda_2 \alpha_0^\top R(\lambda_1) R(\lambda_2) \beta_0$$

$$= \alpha_0^\top \beta_0 (1 - g_{1,2\text{sp}}^{\text{INT}}(\lambda_1, \lambda_2)) - \lambda_1 \alpha_0^\top R(\lambda_1) \beta_0 - \lambda_2 \alpha_0^\top R(\lambda_2) \beta_0$$

$$\quad + \frac{\lambda_1 \lambda_2}{\lambda_2 - \lambda_1} \alpha_0^\top (R(\lambda_1) - R(\lambda_2)) \beta_0$$

$$= \alpha_0^\top \beta_0 (1 - g_{1,2\text{sp}}^{\text{INT}}(\lambda_1, \lambda_2)) - \left(\lambda_1 - \frac{\lambda_1 \lambda_2}{\lambda_2 - \lambda_1}\right) \alpha_0^\top R(\lambda_1) \beta_0 - \left(\lambda_2 + \frac{\lambda_1 \lambda_2}{\lambda_2 - \lambda_1}\right) \alpha_0^\top R(\lambda_2) \beta_0$$

$$= \alpha_0^\top \beta_0 (1 - g_{1,2\text{sp}}^{\text{INT}}(\lambda_1, \lambda_2)) + \frac{\lambda_1^2}{\lambda_1 - \lambda_2} \alpha_0^\top R(\lambda_1) \beta_0 - \frac{\lambda_2^2}{\lambda_1 - \lambda_2} \alpha_0^\top R(\lambda_2) \beta_0$$

From Theorem 1.1 of [27], we have:

$$\alpha_0^\top R(\lambda_1) \beta_0 = \alpha_0^\top \beta_0 \int \frac{x \, dF_{\text{MP}}(x)}{x + \lambda_1} + O_p(n^{-1/2}),$$

$$\alpha_0^\top R(\lambda_2) \beta_0 = \alpha_0^\top \beta_0 \int \frac{x \, dF_{\text{MP}}(x)}{x + \lambda_2} + O_p(n^{-1/2}).$$

This implies:

$$\alpha_0^\top (\mathbf{I} - \lambda_1 R(\lambda_1))(\mathbf{I} - \lambda_2 R(\lambda_2))\beta_0 - (\alpha_0^\top \beta_0)\, g_{1,2\mathrm{sp}}^{\mathrm{INT}}(\lambda_1, \lambda_2)$$

$$= \alpha_0^\top \beta_0 \underbrace{\left\{ (1 - g_{1,2\mathrm{sp}}^{\mathrm{INT}}(\lambda_1, \lambda_2)) + \frac{\lambda_1^2}{\lambda_1 - \lambda_2} \int \frac{x\, dF_{\mathrm{MP}}(x)}{x + \lambda_1} - \frac{\lambda_2^2}{\lambda_1 - \lambda_2} \int \frac{x\, dF_{\mathrm{MP}}(x)}{x + \lambda_2} \right\}}_{=0} + O_p(n^{-1/2}).$$

where the last equality follows from the definition of $g_{1,2\mathrm{sp}}^{\mathrm{INT}}(\lambda_1, \lambda_2)$. This completes the proof.

Now, consider the case $\lambda_1 = \lambda_2 = \lambda$. The entire proof almost remains identical to that of $\lambda_1 \neq \lambda_2$, except that we cannot use Equation (A.7) to establish that

$$\alpha_0^\top R^2(\lambda)\beta_0 - \alpha_0^\top \beta_0 \int \frac{dF_{\mathrm{MP}}(x)}{(x + \lambda)^2} = O_p(n^{-1/2}).$$

However, observing that $R^2(\lambda) = \partial_z(\hat{\Sigma}_1 - z\mathbf{I})^{-1}|_{z=\lambda}$, one can follow the exact same argument as used to prove Theorem 1.1 of [27] and arrive at the conclusion. We skip it here for brevity.

# B  Proof of Theorem 3.2

## B.1  Proof of Three-Split

The proof shares certain similarity with that of the proof of Theorem 3.1. Recall the definition of $\hat{\theta}_{3\mathrm{sp}}^{\mathrm{NR,db}}$:

$$\hat{\theta}_{3\mathrm{sp}}^{\mathrm{NR,db}} = \left\{ \frac{1}{n} \sum_{i \in \mathcal{D}_3} Y_i(A_i - X_i^\top \hat{\alpha}(\lambda_1)) \right\} - \hat{\alpha}(\lambda_1)^\top \hat{\beta}(\lambda_2) \frac{g_{3\mathrm{sp}}^{\mathrm{NR}}(\lambda_1, \lambda_2)}{g_{3\mathrm{sp}}^{\mathrm{INT}}(\lambda_1, \lambda_2)}$$

Expanding the definition of $Y_i$ and $A_i$ in terms of $X_i$ and $(\epsilon_i, \mu_i)$, we have:

$$\hat{\theta}_{3\mathrm{sp}}^{\mathrm{NR,db}} - \theta_0 = \underbrace{(\alpha_0 - \hat{\alpha}(\lambda_1))^\top \hat{\Sigma}_3 \beta_0 - \hat{\alpha}(\lambda_1)^\top \hat{\beta}(\lambda_2) \frac{g_{3\mathrm{sp}}^{\mathrm{NR}}(\lambda_1, \lambda_2)}{g_{3\mathrm{sp}}^{\mathrm{INT}}(\lambda_1, \lambda_2)}}_{:=T_1} + \underbrace{\frac{1}{n} \sum_i \epsilon_i X_i^\top \beta_0}_{:=T_2}$$

$$+ \underbrace{\frac{1}{n} \sum_{i \in \mathcal{D}_3} \mu_i X_i^\top(\alpha_0 - \hat{\alpha}(\lambda_1))}_{:=T_3} + \underbrace{\frac{1}{n} \sum_{i \in \mathcal{D}_3} \mu_i \epsilon_i - \theta_0}_{:=T_4}$$

It is immediate $T_4 = O_p(n^{-1/2})$ by CLT. For $T_2$, observe that:

$$\mathrm{var}(\epsilon\, X^\top \beta_0) = \|\beta_0\|_2^2 \leq C_\beta^2.$$

Now, for $T_3$, observe that,

$$n \times \mathrm{var}(T_3) = n \times \mathbb{E}\left[ \mathrm{var}\left( \frac{1}{n} \sum_{i \in \mathcal{D}_3} \mu_i X_i^\top(\alpha_0 - \hat{\alpha}(\lambda_1)) \mid \mathcal{D}_1 \right) \right]$$

$$= \mathbb{E}\left[ \mathrm{var}\left( \mu\, X^\top(\alpha_0 - \hat{\alpha}(\lambda_1)) \mid \mathcal{D}_1 \right) \right]$$

$$= \mathbb{E}\left[ \|\alpha_0 - \hat{\alpha}(\lambda_1)\|_2^2 \right]$$

$$= \mathbb{E}\left[ \|\lambda_1\, R(\lambda_1)\alpha_0\|_2^2 + \left\| R(\lambda_1)\frac{\mathbf{X}_1^\top \epsilon_1}{n} \right\|_2^2 \right]$$

$$\leq C_\alpha^2 + \frac{1}{n}\mathbb{E}\left[ \mathrm{tr}\left( R^2(\lambda_1)\hat{\Sigma}_1 \right) \right]$$

$$\leq C_\alpha^2 + c\mathbb{E}\left[ \int \frac{x}{(x + \lambda_1)^2}\, d\hat{\pi}_{n,1}(x) \right] \leq C_\alpha^2 + c\, C(\lambda_1).$$

Here $C(\lambda_1)$ is the maximum value of the function $f(x) = x/(x+\lambda_1)^2$. This, along with the fact that $\mathbb{E}[T_3] = 0$, implies $T_3 = O_p(n^{-1/2})$. We now focus on $T_1$. Towards that end, first observe that, conditional on $\mathcal{D}_1$:

$$
\begin{aligned}
(\alpha_0 - \hat{\alpha}(\lambda_1))^\top \hat{\Sigma}_3 \beta_0 &= \frac{1}{n} \sum_{i \in \mathcal{D}_3} \left(X_i^\top \beta_0\right) \left(X_i^\top (\alpha_0 - \hat{\alpha}(\lambda_1))\right) \\
&= \mathbb{E}\left[\left(X_i^\top \beta_0\right)\left(X_i^\top(\alpha_0 - \hat{\alpha}(\lambda_1))\right) \mid \mathcal{D}_1\right] + O_p(n^{-1/2}) \\
&= (\alpha_0 - \hat{\alpha}(\lambda_1))^\top \beta_0 + O_p(n^{-1/2}).
\end{aligned}
\tag{B.1}
$$

Here, the remainder in the second equation is $O_p(n^{-1/2})$ because

$$
\mathsf{var}\left(\left(X_i^\top \beta_0\right)\left(X_i^\top(\alpha_0 - \hat{\alpha}(\lambda_1))\right) \mid \mathcal{D}_1\right) \leq C\|\beta_0\|_2^2\|\alpha_0 - \hat{\alpha}(\lambda_1)\|_2^2
$$

which is uniformly upper bounded with probability going to 1. Therefore, to conclude $T_1 = O_p(n^{-1/2})$, all we need to show is:

$$
\left|(\alpha_0 - \hat{\alpha}(\lambda_1))^\top \beta_0 - \hat{\alpha}(\lambda_1)^\top \hat{\beta}(\lambda_2) \frac{g_{3\mathrm{sp}}^{\mathrm{NR}}(\lambda_1, \lambda_2)}{g_{3\mathrm{sp}}^{\mathrm{INT}}(\lambda_1, \lambda_2)}\right| = O_p(n^{-1/2}).
\tag{B.2}
$$

We start with the following decomposition:

$$
\left|(\alpha_0 - \hat{\alpha}(\lambda_1))^\top \beta_0 - \hat{\alpha}(\lambda_1)^\top \hat{\beta}(\lambda_2) \frac{g_{3\mathrm{sp}}^{\mathrm{NR}}(\lambda_1, \lambda_2)}{g_{3\mathrm{sp}}^{\mathrm{INT}}(\lambda_1, \lambda_2)}\right|
$$

$$
\leq \left|(\alpha_0 - \hat{\alpha}(\lambda_1))^\top \beta_0 - (\alpha_0^\top \beta_0)\, g_{3\mathrm{sp}}^{\mathrm{NR}}(\lambda_1, \lambda_2)\right| + g_{3\mathrm{sp}}^{\mathrm{NR}}(\lambda_1, \lambda_2)\left|(\alpha_0^\top \beta_0) - \frac{\hat{\alpha}(\lambda_1)^\top \hat{\beta}(\lambda_2)}{g_{3\mathrm{sp}}^{\mathrm{INT}}(\lambda_1, \lambda_2)}\right|
$$

That the second summand is $O_p(n^{-1/2})$ is already established in the proof of Theorem 3.1. Therefore, we need to show:

$$
\left|(\alpha_0 - \hat{\alpha}(\lambda_1))^\top \beta_0 - (\alpha_0^\top \beta_0)\, g_{3\mathrm{sp}}^{\mathrm{NR}}(\lambda_1, \lambda_2)\right| = O_p(n^{-1/2}).
$$

Towards that end we use the definition $\hat{\alpha}(\lambda_1)$ and Theorem 1.1 of [27]. From the definition of $\hat{\alpha}(\lambda_1)$, we have:

$$
(\alpha_0 - \hat{\alpha}(\lambda_1))^\top \beta_0 = \left\{\lambda_1(\hat{\Sigma}_1 + \lambda_1 \mathbf{I})^{-1}\alpha_0 + (\hat{\Sigma}_1 + \lambda_1 \mathbf{I})^{-1}\frac{\mathbf{X}_1^\top \boldsymbol{\epsilon}_1}{n}\right\}^\top \beta_0
$$

Hence:

$$
(\alpha_0 - \hat{\alpha}(\lambda_1))^\top \beta_0 - (\alpha_0^\top \beta_0)\, g_{3\mathrm{sp}}^{\mathrm{NR}}(\lambda_1, \lambda_2)
$$

$$
= \lambda_1 \alpha_0^\top (\hat{\Sigma}_1 + \lambda_1 \mathbf{I})^{-1}\beta_0 - (\alpha_0^\top \beta_0)\, g_{3\mathrm{sp}}^{\mathrm{NR}}(\lambda_1, \lambda_2) + \beta_0^\top (\hat{\Sigma}_1 + \lambda_1 \mathbf{I})^{-1}\frac{\mathbf{X}_1^\top \boldsymbol{\epsilon}_1}{n}
$$

Now, from the normality of $\epsilon$, we have:

$$
\beta_0^\top (\hat{\Sigma}_1 + \lambda_1 \mathbf{I})^{-1}\frac{\mathbf{X}_1^\top \boldsymbol{\epsilon}_1}{\sqrt{n}} \mid \mathbf{X}_1 \sim \mathcal{N}\left(0, \beta_0^\top (\hat{\Sigma}_1 + \lambda_1 \mathbf{I})^{-1}\hat{\Sigma}_1(\hat{\Sigma}_1 + \lambda_1 \mathbf{I})^{-1}\beta_0\right)
$$

Now, using a similar calculation as before,

$$
\beta_0^\top (\hat{\Sigma}_1 + \lambda_1 \mathbf{I})^{-1}\hat{\Sigma}_1(\hat{\Sigma}_1 + \lambda_1 \mathbf{I})^{-1}\beta_0 \leq C_\beta^2 C(\lambda_1),
$$

where, as before, $C(\lambda_1)$ is the maximum value of the function $f(x) = x/(x+\lambda_1)^2$. For that final part, we have from Theorem 1.1 of [27]:

$$
\mathbb{E}\left[\sqrt{n}\left|\lambda_1 \alpha_0^\top (\hat{\Sigma}_1 + \lambda_1 \mathbf{I})^{-1}\beta_0 - (\alpha_0^\top \beta_0)\, g_{3\mathrm{sp}}^{\mathrm{NR}}(\lambda_1, \lambda_2)\right|\right] \leq \frac{CC_\alpha C_\beta}{\psi(\lambda_1)}.
$$

This completes the proof.

## B.2 Proof of Two-Split

In this subsection, we establish the $\sqrt{n}$-consistency of $\hat{\theta}_{2\mathrm{sp}}^{\mathrm{NR,db}}$. From the definition (Equation (3.6)) we have:

$$
\hat{\theta}_{2\mathrm{sp}}^{\mathrm{NR,db}} - \theta_0
$$

$$
= \left(1 - \frac{g_{2,2\mathrm{sp}}^{\mathrm{INT}}(\boldsymbol{\lambda}) g_{3\mathrm{sp}}^{\mathrm{NR}}(\boldsymbol{\lambda})}{g_{1,2\mathrm{sp}}^{\mathrm{INT}}(\boldsymbol{\lambda})}\right)^{-1} \left[\frac{1}{n}\sum_{i\in\mathcal{D}_2} Y_i(A_i - X_i^\top \hat{\alpha}(\lambda_1)) - \hat{\alpha}(\lambda_1)^\top \hat{\beta}(\lambda_2)\frac{g_{3\mathrm{sp}}^{\mathrm{NR}}(\boldsymbol{\lambda})}{g_{1,2\mathrm{sp}}^{\mathrm{INT}}(\boldsymbol{\lambda})}\right] - \theta_0
$$

$$
= \left(1 - \frac{g_{2,2\mathrm{sp}}^{\mathrm{INT}}(\boldsymbol{\lambda}) g_{3\mathrm{sp}}^{\mathrm{NR}}(\boldsymbol{\lambda})}{g_{1,2\mathrm{sp}}^{\mathrm{INT}}(\boldsymbol{\lambda})}\right)^{-1} \left[\frac{1}{n}\sum_{i\in\mathcal{D}_2} Y_i(A_i - X_i^\top \hat{\alpha}(\lambda_1)) - \mathbb{E}[Y(A - X^\top \hat{\alpha}(\lambda_1))] +\right.
$$

$$
\left. + \mathbb{E}[Y(A - X^\top \hat{\alpha}(\lambda_1))] - \hat{\alpha}(\lambda_1)^\top \hat{\beta}(\lambda_2)\frac{g_{3\mathrm{sp}}^{\mathrm{NR}}(\boldsymbol{\lambda})}{g_{1,2\mathrm{sp}}^{\mathrm{INT}}(\boldsymbol{\lambda})}\right]
$$

$$
= \left(1 - \frac{g_{2,2\mathrm{sp}}^{\mathrm{INT}}(\boldsymbol{\lambda}) g_{3\mathrm{sp}}^{\mathrm{NR}}(\boldsymbol{\lambda})}{g_{1,2\mathrm{sp}}^{\mathrm{INT}}(\boldsymbol{\lambda})}\right)^{-1} \left[\frac{1}{n}\sum_{i\in\mathcal{D}_2} Y_i(A_i - X_i^\top \hat{\alpha}(\lambda_1)) - \mathbb{E}[Y(A - X^\top \hat{\alpha}(\lambda_1))]\right.
$$

$$
\left. + \theta_0 + (\alpha_0 - \hat{\alpha}(\lambda_1))^\top \beta_0 - (\alpha_0^\top \beta_0)g_{3\mathrm{sp}}^{\mathrm{NR}}(\boldsymbol{\lambda}) + (\alpha_0^\top \beta_0)g_{3\mathrm{sp}}^{\mathrm{NR}}(\boldsymbol{\lambda}) - \hat{\alpha}(\lambda_1)^\top \hat{\beta}(\lambda_2)\frac{g_{3\mathrm{sp}}^{\mathrm{NR}}(\boldsymbol{\lambda})}{g_{1,2\mathrm{sp}}^{\mathrm{INT}}(\boldsymbol{\lambda})}\right]
$$

Furthermore, in the proof of the $\sqrt{n}$-consistency of $\hat{\theta}_{2\mathrm{sp}}^{\mathrm{INT,db}}$, we have already established that:

$$
\hat{\alpha}(\lambda_1)^\top \hat{\beta}(\lambda_2) = (\alpha_0^\top \beta_0)\, g_{1,2\mathrm{sp}}^{\mathrm{INT}}(\boldsymbol{\lambda}) + \theta_0\, g_{2,2\mathrm{sp}}^{\mathrm{INT}}(\boldsymbol{\lambda}) + O_p(n^{-1/2}),
$$

which implies

$$
\hat{\alpha}(\lambda_1)^\top \hat{\beta}(\lambda_2)\frac{g_{3\mathrm{sp}}^{\mathrm{NR}}(\boldsymbol{\lambda})}{g_{1,2\mathrm{sp}}^{\mathrm{INT}}(\boldsymbol{\lambda})} = (\alpha_0^\top \beta_0)g_{3\mathrm{sp}}^{\mathrm{NR}}(\boldsymbol{\lambda}) + \theta_0\,\frac{g_{2,2\mathrm{sp}}^{\mathrm{INT}}(\boldsymbol{\lambda}) g_{3\mathrm{sp}}^{\mathrm{NR}}(\boldsymbol{\lambda})}{g_{1,2\mathrm{sp}}^{\mathrm{INT}}(\boldsymbol{\lambda})} + O_p(n^{-1/2}).
$$

This implies that all we need to show is:

$$
(\alpha_0 - \hat{\alpha}(\lambda_1))^\top \beta_0 - (\alpha_0^\top \beta_0)g_{3\mathrm{sp}}^{\mathrm{NR}}(\boldsymbol{\lambda}) = O_p(n^{-1/2}).
$$

Towards that end, first note that:

$$
(\alpha_0 - \hat{\alpha}(\lambda_1))^\top \beta_0 - (\alpha_0^\top \beta_0)g_{3\mathrm{sp}}^{\mathrm{NR}}(\boldsymbol{\lambda})
$$

$$
= \underbrace{\lambda_1 \beta_0^\top R(\lambda_1)\alpha_0 - (\alpha_0^\top \beta_0)g_{3\mathrm{sp}}^{\mathrm{NR}}(\boldsymbol{\lambda})}_{:=T_1} - \underbrace{\beta_0^\top R(\lambda_1)\frac{\mathbf{X}_1^\top \epsilon_1}{n}}_{:=T_2}
$$

That $T_1 = O_p(n^{-1/2})$ follows from an application of Theorem 1.1 of [27]. For $T_2$, we again use normality of $\epsilon$ and its independence with $X$.

$$
\sqrt{n}\, T_2 \mid \mathbf{X}_1 \sim \mathcal{N}\left(0, \beta_0^\top R(\lambda_1)\hat{\Sigma}_1 R(\lambda_1)\beta_0\right)
$$

That the variance is upper bounded follows by a similar argument used to show that $T_2$ is $O_p(n^{-1/2})$ in the proof of the $\sqrt{n}$-consistency of $\hat{\theta}_{2\mathrm{sp}}^{\mathrm{INT,db}}$. This completes the proof for $\lambda_1 \neq \lambda_2$. The argument for $\lambda_1 = \lambda_2$ is similar to that used in the proof of Theorem 3.1 (two-split case) and hence skipped.

## C    Proof of Theorem 3.3

### C.1    Proof of Three-Split

We start with the following decomposition of $\hat{\theta}^{\mathrm{DR,db}}$:

$$
\hat{\theta}_{3\mathrm{sp}}^{\mathrm{DR,db}} - \theta_0
$$

$$= \frac{1}{n} \sum_{i \in \mathcal{D}_3} (Y_i - X_i^\top \hat{\beta}(\lambda_2))(A_i - X_i^\top \hat{\alpha}(\lambda_1)) - \hat{\alpha}(\lambda_1)^\top \hat{\beta}(\lambda_2) \frac{g_{3\text{sp}}^{\text{DR}}(\lambda_1, \lambda_2)}{g_{3\text{sp}}^{\text{INT}}(\lambda_1, \lambda_2)} - \theta_0$$

$$= \underbrace{(\beta_0 - \hat{\beta}(\lambda_2))^\top \hat{\Sigma}_3 (\alpha_0 - \hat{\alpha}(\lambda_1)) - \hat{\alpha}(\lambda_1)^\top \hat{\beta}(\lambda_2) \frac{g_{3\text{sp}}^{\text{DR}}(\lambda_1, \lambda_2)}{g_{3\text{sp}}^{\text{INT}}(\lambda_1, \lambda_2)}}_{:=T_1} + \underbrace{(\beta_0 - \hat{\beta}(\lambda_2))^\top \frac{1}{n} \sum_{i \in \mathcal{D}_3} X_i \epsilon_i}_{:=T_2}$$

$$+ \underbrace{(\alpha_0 - \hat{\alpha}(\lambda_1))^\top \frac{1}{n} \sum_{i \in \mathcal{D}_3} X_i \mu_i}_{:=T_3} + \underbrace{\frac{1}{n} \sum_{i \in \mathcal{D}_3} \mu_i \epsilon_i - \theta_0}_{:=T_4}$$

That $T_2, T_3, T_4$ are $O_p(n^{-1/2})$ follows from the exact same argument as in the proof of Theorem 3.2 (three-split case). Therefore, we focus on $T_1$. Similar argument as in Equation (B.1) yields:

$$(\beta_0 - \hat{\beta}(\lambda_2))^\top \hat{\Sigma}_3 (\alpha_0 - \hat{\alpha}(\lambda_1)) = (\beta_0 - \hat{\beta}(\lambda_2))^\top (\alpha_0 - \hat{\alpha}(\lambda_1)) + O_p(n^{-1/2}). \tag{C.1}$$

Furthermore, from the definition of $\hat{\alpha}(\lambda_1)$ and $\hat{\beta}(\lambda_2)$, we have:

$$\begin{aligned}
(\beta_0 - \hat{\beta}(\lambda_2))^\top (\alpha_0 - \hat{\alpha}(\lambda_1)) &= \lambda_1 \lambda_2 \alpha_0^\top (\hat{\Sigma}_1 + \lambda_1 \mathbf{I})^{-1} (\hat{\Sigma}_2 + \lambda_2 \mathbf{I})^{-1} \beta_0 \\
&\quad - \lambda_1 \alpha_0^\top (\hat{\Sigma}_1 + \lambda_1 \mathbf{I})^{-1} (\hat{\Sigma}_2 + \lambda_2 \mathbf{I})^{-1} \frac{\mathbf{X}_2^\top \boldsymbol{\mu}_2}{n} \\
&\quad - \lambda_2 \beta_0^\top (\hat{\Sigma}_2 + \lambda_2 \mathbf{I})^{-1} (\hat{\Sigma}_1 + \lambda_1 \mathbf{I})^{-1} \frac{\mathbf{X}_1^\top \boldsymbol{\epsilon}_1}{n} \\
&\quad + \frac{\boldsymbol{\epsilon}^\top \mathbf{X}_1}{n} (\hat{\Sigma}_1 + \lambda_1 \mathbf{I})^{-1} (\hat{\Sigma}_2 + \lambda_2 \mathbf{I})^{-1} \frac{\mathbf{X}_2^\top \boldsymbol{\mu}_2}{n} \\
&:= T_{11} + T_{12} + T_{13} + T_{14}. \tag{C.2}
\end{aligned}$$

First, we show that $T_{12}, T_{13}$ and $T_{14}$ are $O_p(n^{-1/2})$. Towards that, first observe that:

$$\sqrt{n}\, T_{12} \mid \mathcal{D}_1, \mathbf{X}_2 \sim \lambda_1 \mathcal{N}\left(0, \alpha_0^\top (\hat{\Sigma}_1 + \lambda_1 \mathbf{I})^{-1} (\hat{\Sigma}_2 + \lambda_2 \mathbf{I})^{-1} \hat{\Sigma}_2 (\hat{\Sigma}_2 + \lambda_2 \mathbf{I})^{-1} (\hat{\Sigma}_1 + \lambda_1 \mathbf{I})^{-1} \alpha_0\right)$$

To show that the variance is uniformly upper bounded, first observe that:

$$\alpha_0^\top (\hat{\Sigma}_1 + \lambda_1 \mathbf{I})^{-1} (\hat{\Sigma}_2 + \lambda_2 \mathbf{I})^{-1} \hat{\Sigma}_2 (\hat{\Sigma}_2 + \lambda_2 \mathbf{I})^{-1} (\hat{\Sigma}_1 + \lambda_1 \mathbf{I})^{-1} \alpha_0 \le \frac{C_\alpha^2}{\lambda_1^2 \lambda_2^2} \|\hat{\Sigma}_2\|_{\text{op}}.$$

Furthermore, we know $\|\hat{\Sigma}_2\|_{\text{op}} \le 2(1 + \sqrt{c})^2$ with probability going to 1 ([24]). Therefore, with probability going to 1:

$$\alpha_0^\top (\hat{\Sigma}_1 + \lambda_1 \mathbf{I})^{-1} (\hat{\Sigma}_2 + \lambda_2 \mathbf{I})^{-1} \hat{\Sigma}_2 (\hat{\Sigma}_2 + \lambda_2 \mathbf{I})^{-1} (\hat{\Sigma}_1 + \lambda_1 \mathbf{I})^{-1} \alpha_0 \le \frac{2C_\alpha^2}{\lambda_1^2 \lambda_2^2} (1 + \sqrt{c})^2.$$

which establishes that $T_{12} = O_p(n^{-1/2})$. The proof of $T_{13}$ is the same and hence skipped for brevity. Observing that $T_{14}$ is exactly same as $T_9$ defined in the proof of Theorem 3.1 (three-split part), we conclude that $T_{14}$ is $O_p(n^{-1/2})$. Therefore, combining Equation (C.1) and (C.3), all we need to show

$$\left| \lambda_1 \lambda_2 \alpha_0^\top (\hat{\Sigma}_1 + \lambda_1 \mathbf{I})^{-1} (\hat{\Sigma}_2 + \lambda_2 \mathbf{I})^{-1} \beta_0 - \hat{\alpha}(\lambda_1)^\top \hat{\beta}(\lambda_2) \frac{g_{3\text{sp}}^{\text{DR}}(\lambda_1, \lambda_2)}{g_{3\text{sp}}^{\text{INT}}(\lambda_1, \lambda_2)} \right| = O_p(n^{-1/2}). \tag{C.3}$$

A similar calculation, as used to show $\mathscr{B}_n - \alpha_0^\top \beta_0\, g_{3\text{sp}}^{\text{INT}}(\lambda_1, \lambda_2) = O_p(n^{-1/2})$ (Theorem 3.1), we have:

$$\left| \lambda_1 \lambda_2 \alpha_0^\top (\hat{\Sigma}_1 + \lambda_1 \mathbf{I})^{-1} (\hat{\Sigma}_2 + \lambda_2 \mathbf{I})^{-1} \beta_0 - (\alpha_0^\top \beta_0)\, \lambda_1 \lambda_2 m_{\text{MP}}(-\lambda_1) m_{\text{MP}}(-\lambda_2) \right| = O_p(n^{-1/2}).$$

Recall that, by our definition of $g_{3\text{sp}}^{\text{DR}}(\lambda_1, \lambda_2)$ (Section 3), we have:

$$g_{3\text{sp}}^{\text{DR}}(\lambda_1, \lambda_2) = \lambda_1 \lambda_2 m_{\text{MP}}(-\lambda_1) m_{\text{MP}}(-\lambda_2).$$

Therefore, we have established:

$$\left| \lambda_1 \lambda_2 \alpha_0^\top (\hat{\Sigma}_1 + \lambda_1 \mathbf{I})^{-1} (\hat{\Sigma}_2 + \lambda_2 \mathbf{I})^{-1} \beta_0 - (\alpha_0^\top \beta_0) \, g_{3\mathrm{sp}}^{\mathrm{DR}}(\lambda_1, \lambda_2) \right| = O_p(n^{-1/2}) \,.$$

Furthermore, we have already established in Theorem 3.1 that:

$$\left| \alpha_0^\top \beta_0 - \frac{\hat{\alpha}(\lambda_1)^\top \hat{\beta}(\lambda_2)}{g_{3\mathrm{sp}}^{\mathrm{INT}}(\lambda_1, \lambda_2)} \right| = O_p(n^{-1/2}) \,.$$

Combining this with the above equation concludes Equation (C.3), which concludes the proof.

## C.2 Proof of Two-Split

We start with the definition, as mentioned in Equation (3.9):

$$\hat{\theta}_{2\mathrm{sp}}^{\mathrm{DR,db}} - \theta_0$$
$$= H(\boldsymbol{\lambda}) \left[ \frac{1}{n} \sum_{i \in \mathcal{D}_2} (Y_i - X_i^\top \hat{\beta}(\lambda_2))(A_i - X_i^\top \hat{\alpha}(\lambda_1)) - \frac{\hat{\alpha}(\lambda_1)^\top \hat{\beta}(\lambda_2) g_{2,2\mathrm{sp}}^{\mathrm{DR}}(\boldsymbol{\lambda})}{g_{1,2\mathrm{sp}}^{\mathrm{INT}}(\boldsymbol{\lambda})} \right] - \theta_0 \,,$$

where

$$H(\boldsymbol{\lambda}) = \left( 1 + g_{2,2\mathrm{sp}}^{\mathrm{INT}}(\boldsymbol{\lambda}) \left( 1 - \frac{g_{2,2\mathrm{sp}}^{\mathrm{DR}}(\boldsymbol{\lambda})}{g_{1,2\mathrm{sp}}^{\mathrm{INT}}(\boldsymbol{\lambda})} \right) \right)^{-1} .$$

First, observe that:

$$\mathbb{E}\left[ (Y - X^\top \hat{\beta}(\lambda_2))(A - X^\top \hat{\alpha}(\lambda_1)) \mid \mathcal{D}_1 \right] = (\alpha_0 - \hat{\alpha}(\lambda_1))^\top (\beta_0 - \hat{\beta}(\lambda_2)) + \theta_0 \,.$$

Using this, we can expand the expression as:

$$\hat{\theta}_{2\mathrm{sp}}^{\mathrm{DR,db}} - \theta_0$$
$$= H(\boldsymbol{\lambda}) \left[ \frac{1}{n} \sum_{i \in \mathcal{D}_2} (Y_i - X_i^\top \hat{\beta}(\lambda_2))(A_i - X_i^\top \hat{\alpha}(\lambda_1)) - \mathbb{E}\left[ (Y - X^\top \hat{\beta}(\lambda_2))(A - X^\top \hat{\alpha}(\lambda_1)) \mid \mathcal{D}_1 \right] \right.$$
$$\left. + \left\{ (\alpha_0 - \hat{\alpha}(\lambda_1))^\top (\beta_0 - \hat{\beta}(\lambda_2)) + \theta_0 \right\} - \frac{\hat{\alpha}(\lambda_1)^\top \hat{\beta}(\lambda_2) g_{2,2\mathrm{sp}}^{\mathrm{DR}}(\boldsymbol{\lambda})}{g_{1,2\mathrm{sp}}^{\mathrm{INT}}(\boldsymbol{\lambda})} \right] - \theta_0$$

The first term inside the square bracket is $O_p(n^{-1/2})$ by CLT. Therefore, all we need to show is:

$$H(\boldsymbol{\lambda}) \left[ \left\{ (\alpha_0 - \hat{\alpha}(\lambda_1))^\top (\beta_0 - \hat{\beta}(\lambda_2)) + \theta_0 \right\} - \frac{\hat{\alpha}(\lambda_1)^\top \hat{\beta}(\lambda_2) g_{2,2\mathrm{sp}}^{\mathrm{DR}}(\boldsymbol{\lambda})}{g_{1,2\mathrm{sp}}^{\mathrm{INT}}(\boldsymbol{\lambda})} \right] - \theta_0 = O_p(n^{-1/2}) \,.$$

The term insider the curly bracket can be expanded as follows:

$$(\alpha_0 - \hat{\alpha}(\lambda_1))^\top (\beta_0 - \hat{\beta}(\lambda_2)) = \lambda_1 \lambda_2 \alpha_0^\top R(\lambda_1) R(\lambda_2) \beta_0 - \lambda_1 \alpha_0^\top R(\lambda_1) R(\lambda_2) \frac{\mathbf{X}_1^\top \mu_1}{n}$$
$$- \lambda_2 \beta_0^\top R(\lambda_2) R(\lambda_1) \frac{\mathbf{X}_1^\top \epsilon}{n} + \frac{\epsilon^\top \mathbf{X}_1}{n} R(\lambda_1) R(\lambda_2) \frac{\mathbf{X}_1^\top \mu_1}{n}$$

That the second and third term are $O_p(n^{-1/2})$ follows from the same argument used to show that $T_2$ and $T_3$ are $O_p(n^{-1/2})$ in the proof of $\sqrt{n}$-consistency of $\hat{\theta}_{2\mathrm{sp}}^{\mathrm{INT,db}}$. Furthermore, using as same technique used to proved that $T_4 = O_p(n^{-1/2})$ in the said proof, we can conclude:

$$\frac{\epsilon^\top \mathbf{X}_1}{n} R(\lambda_1) R(\lambda_2) \frac{\mathbf{X}_1^\top \mu_1}{n} = \theta_0 g_{2,2\mathrm{sp}}^{\mathrm{INT}}(\boldsymbol{\lambda}) + O_p(n^{-1/2})$$

and the same technique used to prove that $T_1 = O_p(n^{-1/2})$ in that proof can be used to show:

$$\lambda_1 \lambda_2 \alpha_0^\top R(\lambda_1) R(\lambda_2) \beta_0 = (\alpha_0^\top \beta_0) \, g_{2,2\mathrm{sp}}^{\mathrm{DR}}(\boldsymbol{\lambda}) + O_p(n^{-1/2}) \,.$$

Therefore, we conclude:

$$\left\{(\alpha_0 - \hat{\alpha}(\lambda_1))^\top (\beta_0 - \hat{\beta}(\lambda_2)) + \theta_0\right\} = (\alpha_0^\top \beta_0)\, g_{2,2\mathrm{sp}}^{\mathrm{DR}}(\boldsymbol{\lambda}) + \theta_0(1 + g_{2,2\mathrm{sp}}^{\mathrm{INT}}(\boldsymbol{\lambda})) + O_p(n^{-1/2})\,.$$

As a consequence, all we need to show is:

$$H(\boldsymbol{\lambda})\left[(\alpha_0^\top \beta_0)\, g_{2,2\mathrm{sp}}^{\mathrm{DR}}(\boldsymbol{\lambda}) + \theta_0(1 + g_{2,2\mathrm{sp}}^{\mathrm{INT}}(\boldsymbol{\lambda})) - \frac{\hat{\alpha}(\lambda_1)^\top \hat{\beta}(\lambda_2) g_{2,2\mathrm{sp}}^{\mathrm{DR}}(\boldsymbol{\lambda})}{g_{1,2\mathrm{sp}}^{\mathrm{INT}}(\boldsymbol{\lambda})}\right] - \theta_0 = O_p(n^{-1/2})\,.$$

Furthermore, it is also proved in the proof of $\sqrt{n}$-consistency of $\hat{\theta}_{2\mathrm{sp}}^{\mathrm{INT,db}}$ that:

$$\hat{\alpha}(\lambda_1)^\top \hat{\beta}(\lambda_2) = (\alpha_0^\top \beta_0)\, g_{1,2\mathrm{sp}}^{\mathrm{INT}}(\boldsymbol{\lambda}) + \theta_0\, g_{2,2\mathrm{sp}}^{\mathrm{INT}}(\boldsymbol{\lambda}) + O_p(n^{-1/2})\,.$$

Hence,

$$\frac{\hat{\alpha}(\lambda_1)^\top \hat{\beta}(\lambda_2) g_{2,2\mathrm{sp}}^{\mathrm{DR}}(\boldsymbol{\lambda})}{g_{1,2\mathrm{sp}}^{\mathrm{INT}}(\boldsymbol{\lambda})} = (\alpha_0^\top \beta_0) g_{2,2\mathrm{sp}}^{\mathrm{DR}}(\boldsymbol{\lambda}) + \theta_0 \frac{g_{2,2\mathrm{sp}}^{\mathrm{INT}}(\boldsymbol{\lambda}) g_{2,2\mathrm{sp}}^{\mathrm{DR}}(\boldsymbol{\lambda})}{g_{1,2\mathrm{sp}}^{\mathrm{INT}}(\boldsymbol{\lambda})} + O_p(n^{-1/2})\,.$$

This, along with the definition of $H(\boldsymbol{\lambda})$, concludes the proof. The argument for $\lambda_1 = \lambda_2$ is similar to that used in the proof of Theorem 3.1 (two-split case) and hence skipped.

## D    Proof of Theorem 3.4

### D.1    Limiting Variance for $\hat{\theta}_{3\mathrm{sp}}^{\mathrm{INT,db}}$

We start with the definition of $\hat{\theta}_{3\mathrm{sp}}^{\mathrm{INT,db}}$:

$$\hat{\theta}_{3\mathrm{sp}}^{\mathrm{INT,db}} = \frac{1}{n}\sum_{i \in \mathcal{D}_3} A_i Y_i - \frac{\hat{\alpha}(\lambda_1)^\top \hat{\beta}(\lambda_2)}{g_{3\mathrm{sp}}^{\mathrm{INT}}(\lambda_1, \lambda_2)}$$

From the independence of $\mathcal{D}_1, \mathcal{D}_2$ and $\mathcal{D}_3$, we have:

$$n \times \mathsf{var}\left(\hat{\theta}^{\mathrm{INT,db}}\right) = \mathsf{var}(AY) + \frac{1}{(g_{3\mathrm{sp}}^{\mathrm{INT}}(\lambda_1, \lambda_2))^2}\, n \times \mathsf{var}\left(\hat{\alpha}(\lambda_1)^\top \hat{\beta}(\lambda_2)\right)\,.$$

As a consequence, our goal is to derive the limit of $n \times \mathsf{var}\left(\hat{\alpha}(\lambda_1)^\top \hat{\beta}(\lambda_2)\right)$ when they are estimated from different subsamples. Let's start with another expansion (a bit simpler version) of $\hat{\alpha}(\lambda_1)^\top \hat{\beta}(\lambda_2)$:

$$\hat{\alpha}(\lambda_1)^\top \hat{\beta}(\lambda_2)$$
$$= \left\{\hat{\Sigma}_1 \alpha_0 + \frac{\mathbf{X}_1^\top \boldsymbol{\epsilon}_1}{n}\right\}^\top \left(\hat{\Sigma}_1 + \lambda_1 \mathbf{I}\right)^{-1} \left(\hat{\Sigma}_2 + \lambda_2 \mathbf{I}\right)^{-1} \left\{\hat{\Sigma}_2 \beta_0 + \frac{\mathbf{X}_2^\top \boldsymbol{\mu}_2}{n}\right\}$$
$$= \alpha_0^\top \hat{\Sigma}_1 \left(\hat{\Sigma}_1 + \lambda_1 \mathbf{I}\right)^{-1} \left(\hat{\Sigma}_2 + \lambda_2 \mathbf{I}\right)^{-1} \hat{\Sigma}_2 \beta_0 + \alpha_0^\top \hat{\Sigma}_1 \left(\hat{\Sigma}_1 + \lambda_1 \mathbf{I}\right)^{-1} \left(\hat{\Sigma}_2 + \lambda_2 \mathbf{I}\right)^{-1} \frac{\mathbf{X}_2^\top \boldsymbol{\mu}_2}{n}$$
$$+ \beta_0^\top \hat{\Sigma}_2 \left(\hat{\Sigma}_2 + \lambda_2 \mathbf{I}\right)^{-1} \left(\hat{\Sigma}_1 + \lambda_1 \mathbf{I}\right)^{-1} \frac{\mathbf{X}_1^\top \boldsymbol{\epsilon}_1}{n} + \frac{\boldsymbol{\epsilon}_1^\top \mathbf{X}_1}{n} \left(\hat{\Sigma}_1 + \lambda_1 \mathbf{I}\right)^{-1} \left(\hat{\Sigma}_2 + \lambda_2 \mathbf{I}\right)^{-1} \frac{\mathbf{X}_2^\top \boldsymbol{\mu}_2}{n}$$
$$\triangleq T_1 + T_2 + T_3 + T_4\,. \tag{D.1}$$

We will compute the variance from the first principle, i.e.

$$\mathsf{var}\left(\hat{\alpha}(\lambda_1)^\top \hat{\beta}(\lambda_2)\right) = \mathbb{E}\left[\left(\hat{\alpha}(\lambda_1)^\top \hat{\beta}(\lambda_2)\right)^2\right] - \left(\mathbb{E}\left[\hat{\alpha}(\lambda_1)^\top \hat{\beta}(\lambda_2)\right]\right)^2\,.$$

First, let us take care of the easy term, i.e., the expectation. As $(\mu, \eta) \perp\!\!\!\perp X$ and $\boldsymbol{\epsilon}_1$ and $\boldsymbol{\mu}_2$ are also independent (as they are from different subsamples), we have:

$$\mathbb{E}\left[\hat{\alpha}(\lambda_1)^\top \hat{\beta}(\lambda_2)\right] = \mathbb{E}\left[\alpha_0^\top \hat{\Sigma}_1 \left(\hat{\Sigma}_1 + \lambda_1 \mathbf{I}\right)^{-1} \left(\hat{\Sigma}_2 + \lambda_2 \mathbf{I}\right)^{-1} \hat{\Sigma}_2 \beta_0\right]$$

To compute the expectation of the right-hand side, let us first compute the conditional expectation given $\mathcal{D}_2$. For notational simplicity, define $\omega_2 = \left(\hat{\Sigma}_2 + \lambda_2\mathbf{I}\right)^{-1}\hat{\Sigma}_2\beta_0$.

$$\mathbb{E}\left[\hat{\alpha}(\lambda_1)^{\top}\hat{\beta}(\lambda_2) \mid \mathcal{D}_2\right] = \mathbb{E}\left[\alpha_0^{\top}\hat{\Sigma}_1\left(\hat{\Sigma}_1 + \lambda_1\mathbf{I}\right)^{-1}\omega_2 \mid \mathcal{D}_2\right] = \alpha_0^{\top}\omega_2\mathbb{E}\left[\int\frac{x}{x+\lambda_1}\,d\hat{\pi}_{n,1}(x)\right].$$

where the last equality follows from the same calculation as in the proof of Claim 1 of Lemma E.1 and the independence of the eigenvectors and eigenvalues of $\hat{\Sigma}_1$. Here, $\hat{\pi}_{n,1}$ is the ESD of $\hat{\Sigma}_1$. Now taking expectation with respect to $\mathcal{D}_2$, we have:

$$\mathbb{E}\left[\alpha_0^{\top}\omega_2\right] = \mathbb{E}\left[\alpha_0^{\top}\left(\hat{\Sigma}_2 + \lambda_2\mathbf{I}\right)^{-1}\hat{\Sigma}_2\beta_0\right] = \alpha_0^{\top}\beta_0\mathbb{E}\left[\int\frac{x}{x+\lambda_2}\,d\hat{\pi}_{n,2}(x)\right],$$

where $\hat{\pi}_{n,2}$ is ESD of $\hat{\Sigma}_2$. Therefore, we conclude:

$$\mathbb{E}\left[\hat{\alpha}(\lambda_1)^{\top}\hat{\beta}(\lambda_2)\right] = \alpha_0^{\top}\beta_0\,\mathbb{E}\left[\int\frac{x}{x+\lambda_1}\,d\hat{\pi}_{n,1}(x)\right]\mathbb{E}\left[\int\frac{x}{x+\lambda_2}\,d\hat{\pi}_{n,2}(x)\right]$$

$$= \alpha_0^{\top}\beta_0\,\mathbb{E}\left[\int\frac{x}{x+\lambda_1}\,d\hat{\pi}_{n,1}(x)\int\frac{x}{x+\lambda_2}\,d\hat{\pi}_{n,2}(x)\right]. \qquad (\text{D.2})$$

Now, we focus on the second moment. Again, as before, conditioning on $\mathcal{D}_2$, we have:

$$\mathbb{E}\left[\left(\hat{\alpha}(\lambda_1)^{\top}\hat{\beta}(\lambda_2)\right)^2 \mid \mathcal{D}_2\right] = \hat{\beta}(\lambda_2)^{\top}\mathbb{E}\left[\hat{\alpha}(\lambda_1)\hat{\alpha}(\lambda_1)^{\top}\right]\hat{\beta}(\lambda_2).$$

From the definition of $\hat{\alpha}(\lambda_1)$, we have:

$$\hat{\beta}(\lambda_2)^{\top}\mathbb{E}\left[\hat{\alpha}(\lambda_1)\hat{\alpha}(\lambda_1)^{\top}\right]\hat{\beta}(\lambda_2)$$

$$= \hat{\beta}(\lambda_2)^{\top}\mathbb{E}\left[\left(\hat{\Sigma}_1 + \lambda_1\mathbf{I}\right)^{-1}\hat{\Sigma}_1\alpha_0\alpha_0^{\top}\hat{\Sigma}_1\left(\hat{\Sigma}_1 + \lambda_1\mathbf{I}\right)^{-1}\right]\hat{\beta}(\lambda_2)$$

$$+ \hat{\beta}(\lambda_2)^{\top}\mathbb{E}\left[\left(\hat{\Sigma}_1 + \lambda_1\mathbf{I}\right)^{-1}\hat{\Sigma}_1\alpha_0\frac{\epsilon_1^{\top}\mathbf{X}_1}{n}\left(\hat{\Sigma}_1 + \lambda_1\mathbf{I}\right)^{-1}\right]\hat{\beta}(\lambda_2)$$

$$+ \hat{\beta}(\lambda_2)^{\top}\mathbb{E}\left[\left(\hat{\Sigma}_1 + \lambda_1\mathbf{I}\right)^{-1}\frac{\mathbf{X}_1^{\top}\epsilon_1}{n}\alpha_0^{\top}\hat{\Sigma}_1\left(\hat{\Sigma}_1 + \lambda_1\mathbf{I}\right)^{-1}\right]\hat{\beta}(\lambda_2)$$

$$+ \hat{\beta}(\lambda_2)^{\top}\mathbb{E}\left[\left(\hat{\Sigma}_1 + \lambda_1\mathbf{I}\right)^{-1}\frac{\mathbf{X}_1^{\top}\epsilon_1}{n}\frac{\epsilon_1^{\top}\mathbf{X}_1}{n}\left(\hat{\Sigma}_1 + \lambda_1\mathbf{I}\right)^{-1}\right]\hat{\beta}(\lambda_2)$$

The expectation of the second and the third term will be 0, as $\epsilon_1 \perp\!\!\!\perp X_1$. Therefore,

$$\hat{\beta}(\lambda_2)^{\top}\mathbb{E}\left[\hat{\alpha}(\lambda_1)\hat{\alpha}(\lambda_1)^{\top}\right]\hat{\beta}(\lambda_2)$$

$$= \mathbb{E}\left[\left(\hat{\beta}(\lambda_2)^{\top}\left(\hat{\Sigma}_1 + \lambda_1\mathbf{I}\right)^{-1}\hat{\Sigma}_1\alpha_0\right)^2\right] + \hat{\beta}(\lambda_2)^{\top}\mathbb{E}\left[\left(\hat{\Sigma}_1 + \lambda_1\mathbf{I}\right)^{-1}\frac{\mathbf{X}_1^{\top}\mathbf{X}_1}{n^2}\left(\hat{\Sigma}_1 + \lambda_1\mathbf{I}\right)^{-1}\right]\hat{\beta}(\lambda_2)$$

$$= \mathbb{E}\left[\left(\hat{\beta}(\lambda_2)^{\top}\left(\hat{\Sigma}_1 + \lambda_1\mathbf{I}\right)^{-1}\hat{\Sigma}_1\alpha_0\right)^2\right] + \frac{1}{n}\hat{\beta}(\lambda_2)^{\top}\mathbb{E}\left[\left(\hat{\Sigma}_1 + \lambda_1\mathbf{I}\right)^{-1}\hat{\Sigma}_1\left(\hat{\Sigma}_1 + \lambda_1\mathbf{I}\right)^{-1}\right]\hat{\beta}(\lambda_2)$$

Again, by the same calculation as in Claim 1 of Lemma E.1, we have:

$$\hat{\beta}(\lambda_2)^{\top}\mathbb{E}\left[\left(\hat{\Sigma}_1 + \lambda_1\mathbf{I}\right)^{-1}\hat{\Sigma}_1\left(\hat{\Sigma}_1 + \lambda_1\mathbf{I}\right)^{-1}\right]\hat{\beta}(\lambda_2) = \|\hat{\beta}(\lambda_2)\|_2^2\,\mathbb{E}\left[\int\frac{x}{(x+\lambda_1)^2}\,d\hat{\pi}_{n,1}(x)\right]$$

$$(\text{D.3})$$

For the other term, we have, by the same calculation as of Equation (E.2) in the proof of Lemma E.1 (with $\mathbf{u}_n = \alpha_0, \mathbf{w}_n = \hat{\beta}(\lambda_2)$ and $f(x) = x/(x+\lambda)$:

$$\mathbb{E}\left[\left(\hat{\beta}(\lambda_2)^{\top}\left(\hat{\Sigma}_1 + \lambda_1\mathbf{I}\right)^{-1}\hat{\Sigma}_1\alpha_0\right)^2\right] \qquad (\text{D.4})$$

$$= \frac{1}{(p+2)} \left[ \|\alpha_0\|_2^2 \|\hat{\beta}(\lambda_2)\|_2^2 + 2(\alpha_0^\top \hat{\beta}(\lambda_2))^2 \right] \mathbb{E}\left[ \int \frac{x^2}{(x+\lambda_1)^2} \, d\hat{\pi}_{n,1}(x) \right]$$

$$+ \frac{p}{p-1} \left[ \frac{p}{p+2} (\alpha_0^\top \hat{\beta}(\lambda_2))^2 - \frac{\|\alpha_0\|_2^2 \|\hat{\beta}(\lambda_2)\|_2^2}{(p+2)} \right] \mathbb{E}\left[ \left( \int \frac{x}{(x+\lambda_1)} \, d\hat{\pi}_{n,1}(x) \right)^2 \right]$$

$$- \frac{1}{p-1} \left[ \frac{p}{p+2} (\alpha_0^\top \hat{\beta}(\lambda_2))^2 - \frac{\|\alpha_0\|_2^2 \|\hat{\beta}(\lambda_2)\|_2^2}{(p+2)} \right] \mathbb{E}\left[ \int \frac{x^2}{(x+\lambda_1)^2} \, d\hat{\pi}_{n,1}(x) \right]$$

$$= \frac{p}{(p-1)(p+2)} \left[ \left( 1 - \frac{2}{p} \right) (\alpha_0^\top \hat{\beta}(\lambda_2))^2 + \|\alpha_0\|_2^2 \|\hat{\beta}(\lambda_2)\|_2^2 \right] \mathbb{E}\left[ \int \frac{x^2}{(x+\lambda_1)^2} \, d\hat{\pi}_{n,1}(x) \right]$$

$$+ \frac{p^2}{(p-1)(p+2)} \left[ (\alpha_0^\top \hat{\beta}(\lambda_2))^2 - \frac{\|\alpha_0\|_2^2 \|\hat{\beta}(\lambda_2)\|_2^2}{p} \right] \mathbb{E}\left[ \left( \int \frac{x}{(x+\lambda_1)} \, d\hat{\pi}_{n,1}(x) \right)^2 \right] \quad \text{(D.5)}$$

Combining Equation (D.3) and (D.4), we obtain:

---

$$\mathbb{E}\left[ \left( \hat{\alpha}(\lambda_1)^\top \hat{\beta}(\lambda_2) \right)^2 \mid \mathcal{D}_2 \right] \quad \text{(D.6)}$$

$$= \frac{p}{(p-1)(p+2)} \left[ \left( 1 - \frac{2}{p} \right) (\alpha_0^\top \hat{\beta}(\lambda_2))^2 + \|\alpha_0\|_2^2 \|\hat{\beta}(\lambda_2)\|_2^2 \right] \mathbb{E}\left[ \int \frac{x^2}{(x+\lambda_1)^2} \, d\hat{\pi}_{n,1}(x) \right]$$

$$+ \frac{p^2}{(p-1)(p+2)} \left[ (\alpha_0^\top \hat{\beta}(\lambda_2))^2 - \frac{\|\alpha_0\|_2^2 \|\hat{\beta}(\lambda_2)\|_2^2}{p} \right] \mathbb{E}\left[ \left( \int \frac{x}{(x+\lambda_1)} \, d\hat{\pi}_{n,1}(x) \right)^2 \right]$$

$$+ \frac{1}{n} \|\hat{\beta}(\lambda_2)\|_2^2 \, \mathbb{E}\left[ \int \frac{x}{(x+\lambda_1)^2} \, d\hat{\pi}_{n,1}(x) \right]$$

$$= (\alpha_0^\top \hat{\beta}(\lambda_2))^2 \mathbb{E}\left[ \left( \int \frac{x}{x+\lambda_1} \, d\hat{\pi}_{n,1}(x) \right)^2 \right]$$

$$+ \frac{p}{(p-1)(p+2)} \mathbb{E}\left[ \mathsf{var}_{X \sim \hat{\pi}_{n,1}} \left( \frac{X}{X+\lambda_1} \right) \right] \left\{ (\alpha_0^\top \hat{\beta}(\lambda_2))^2 \left( 1 - \frac{2}{p} \right) + \|\alpha_0\|_2^2 \|\hat{\beta}(\lambda_2)\|_2^2 \right\}$$

$$+ \frac{1}{n} \|\hat{\beta}(\lambda_2)\|_2^2 \, \mathbb{E}\left[ \int \frac{x}{(x+\lambda_1)^2} \, d\hat{\pi}_{n,1}(x) \right] \quad \text{(D.7)}$$

---

It is immediate from Equation (D.6) that, for the unconditional expectation, we need to compute the following expectations with respect to $\mathcal{D}_2$, $\mathbb{E}[(\alpha_0^\top \hat{\beta}(\lambda_2))^2]$ and $\mathbb{E}[\|\hat{\beta}(\lambda_2)\|_2^2]$. The calculation is similar to before, with slight modifications. First, let us compute $\mathbb{E}[(\alpha_0^\top \hat{\beta}(\lambda_2))^2]$. From the definition of $\hat{\beta}(\lambda_2)$, we have:

$$\mathbb{E}\left[ \left( \alpha_0^\top \hat{\beta}(\lambda_2) \right)^2 \right] = \mathbb{E}\left[ \left( \alpha_0^\top \left( \hat{\Sigma}_2 + \lambda_2 \mathbf{I} \right)^{-1} \hat{\Sigma}_2 \beta_0 + \alpha_0^\top \left( \hat{\Sigma}_2 + \lambda_2 \mathbf{I} \right)^{-1} \frac{\mathbf{X}_2^\top \boldsymbol{\mu}_2}{n} \right)^2 \right]$$

$$= \mathbb{E}\left[ \left( \alpha_0^\top \left( \hat{\Sigma}_2 + \lambda_2 \mathbf{I} \right)^{-1} \hat{\Sigma}_2 \beta_0 \right)^2 \right] + \mathbb{E}\left[ \alpha_0^\top \left( \hat{\Sigma}_2 + \lambda_2 \mathbf{I} \right)^{-1} \frac{\mathbf{X}_2^\top \boldsymbol{\mu}_2 \boldsymbol{\mu}_2^\top \mathbf{X}_2}{n^2} \left( \hat{\Sigma}_2 + \lambda_2 \mathbf{I} \right)^{-1} \alpha_0 \right]$$

$$+ 2\mathbb{E}\left[ \alpha_0^\top \left( \hat{\Sigma}_2 + \lambda_2 \mathbf{I} \right)^{-1} \hat{\Sigma}_2 \beta_0 \alpha_0^\top \left( \hat{\Sigma}_2 + \lambda_2 \mathbf{I} \right)^{-1} \frac{\mathbf{X}_2^\top \boldsymbol{\mu}_2}{n} \right]$$

$$= \mathbb{E}\left[ \left( \alpha_0^\top \left( \hat{\Sigma}_2 + \lambda_2 \mathbf{I} \right)^{-1} \hat{\Sigma}_2 \beta_0 \right)^2 \right] + \frac{1}{n} \mathbb{E}\left[ \alpha_0^\top \left( \hat{\Sigma}_2 + \lambda_2 \mathbf{I} \right)^{-1} \hat{\Sigma}_2 \left( \hat{\Sigma}_2 + \lambda_2 \mathbf{I} \right)^{-1} \alpha_0 \right]$$

$$= \frac{1}{(p+2)} \left[ \|\alpha_0\|_2^2 \|\beta_0\|_2^2 + 2(\alpha_0^\top \beta_0)^2 \right] \mathbb{E}\left[ \int \frac{x^2}{(x+\lambda_2)^2} \, d\hat{\pi}_{n,2}(x) \right]$$

$$
+ \frac{p}{p-1} \left[ \frac{p}{p+2} (\alpha_0^\top \beta_0)^2 - \frac{\|\alpha_0\|_2^2 \|\beta_0\|_2^2}{(p+2)} \right] \mathbb{E} \left[ \left( \int \frac{x}{(x+\lambda_2)} \, d\hat{\pi}_{n,2}(x) \right)^2 \right]
$$

$$
- \frac{1}{p-1} \left[ \frac{p}{p+2} (\alpha_0^\top \beta_0)^2 - \frac{\|\alpha_0\|_2^2 \|\beta_0\|_2^2}{(p+2)} \right] \mathbb{E} \left[ \int \frac{x^2}{(x+\lambda_2)^2} \, d\hat{\pi}_{n,2}(x) \right]
$$

$$
+ \frac{1}{n} \|\alpha_0\|_2^2 \, \mathbb{E} \left[ \int \frac{x}{(x+\lambda_2)^2} \, d\hat{\pi}_{n,2}(x) \right]
$$

$$
= \frac{p}{(p-1)(p+2)} \left[ \left( 1 - \frac{2}{p} \right) (\alpha_0^\top \beta_0)^2 + \|\alpha_0\|_2^2 \|\beta_0\|_2^2 \right] \mathbb{E} \left[ \int \frac{x^2}{(x+\lambda_2)^2} \, d\hat{\pi}_{n,2}(x) \right]
$$

$$
+ \frac{p^2}{(p-1)(p+2)} \left[ (\alpha_0^\top \beta_0)^2 - \frac{\|\alpha_0\|_2^2 \|\beta_0\|_2^2}{p} \right] \mathbb{E} \left[ \left( \int \frac{x}{(x+\lambda_2)} \, d\hat{\pi}_{n,2}(x) \right)^2 \right]
$$

$$
+ \frac{1}{n} \|\alpha_0\|_2^2 \, \mathbb{E} \left[ \int \frac{x}{(x+\lambda_2)^2} \, d\hat{\pi}_{n,2}(x) \right]
$$

The following simplification of the above expression would be required later:

$$
\mathbb{E} \left[ \left( \alpha_0^\top \hat{\beta}(\lambda_2) \right)^2 \right] = (\alpha_0^\top \beta_0)^2 \mathbb{E} \left[ \left( \int \frac{x}{x+\lambda_2} \, d\hat{\pi}_{n,2}(x) \right)^2 \right]
$$

$$
+ \frac{p}{(p-1)(p+2)} \mathbb{E} \left[ \mathsf{var}_{X \sim \hat{\pi}_{n,2}} \left( \frac{X}{X+\lambda_2} \right) \right] \left\{ (\alpha_0^\top \beta_0)^2 \left( 1 - \frac{2}{p} \right) + \|\alpha_0\|_2^2 \|\beta_0\|_2^2 \right\}
$$

$$
+ \frac{1}{n} \|\alpha_0\|_2^2 \, \mathbb{E} \left[ \int \frac{x}{(x+\lambda_2)^2} \, d\hat{\pi}_{n,2}(x) \right]
$$

$$
\triangleq (\alpha_0^\top \beta_0)^2 \mathbb{E} \left[ \left( \int \frac{x}{x+\lambda_2} \, d\hat{\pi}_{n,2}(x) \right)^2 \right] + R_{1,p} . \tag{D.8}
$$

For the expected squared norm of $\hat{\beta}(\lambda_2)$:

$$
\mathbb{E} \left[ \| \hat{\beta}(\lambda_2) \|_2^2 \right] = \mathbb{E} \left[ \left\| \left( \hat{\Sigma}_2 + \lambda_2 \mathbf{I} \right)^{-1} \hat{\Sigma}_2 \beta_0 + \left( \hat{\Sigma}_2 + \lambda_2 \mathbf{I} \right)^{-1} \frac{\mathbf{X}_2^\top \boldsymbol{\mu}_2}{n} \right\|_2^2 \right]
$$

$$
= \beta_0^\top \mathbb{E} \left[ \hat{\Sigma}_2 \left( \hat{\Sigma}_2 + \lambda \mathbf{I} \right)^{-2} \hat{\Sigma}_2 \right] \beta_0 + \mathbb{E} \left[ \frac{\boldsymbol{\mu}_2^\top \mathbf{X}_2}{n} \left( \hat{\Sigma}_2 + \lambda \mathbf{I} \right)^{-2} \frac{\mathbf{X}_2^\top \boldsymbol{\mu}_2}{n} \right]
$$

$$
+ 2 \beta_0^\top \mathbb{E} \left[ \hat{\Sigma}_2 \left( \hat{\Sigma}_2 + \lambda \mathbf{I} \right)^{-2} \frac{\mathbf{X}_2^\top \boldsymbol{\mu}_2}{n} \right] \nearrow^{0}
$$

$$
= \beta_0^\top \mathbb{E} \left[ \sum_{j=1}^{p} \hat{v}_{j,2} \hat{v}_{j,2}^\top \frac{\hat{\lambda}_{j,2}^2}{(\hat{\lambda}_{j,2} + \lambda)^2} \right] \beta_0 + \mathbb{E} \left[ \mathsf{tr} \left( \frac{\mathbf{X}_2}{n} \left( \hat{\Sigma}_2 + \lambda \mathbf{I} \right)^{-2} \frac{\mathbf{X}_2^\top}{n} \right) \right]
$$

$$
= \mathbb{E} \left[ \sum_{j=1}^{p} (\beta_0^\top \hat{v}_{j,2})^2 \frac{\hat{\lambda}_{j,2}^2}{(\hat{\lambda}_{j,2} + \lambda)^2} \right] + \frac{1}{n} \mathbb{E} \left[ \mathsf{tr} \left( \left( \hat{\Sigma}_2 + \lambda \mathbf{I} \right)^{-2} \hat{\Sigma} \right) \right]
$$

$$
= \|\beta_0\|_2^2 \, \mathbb{E} \left[ \int \frac{x^2}{(x+\lambda_2)^2} \, d\hat{\pi}_{n,2}(x) \right] + c \, \mathbb{E} \left[ \int \frac{x}{(x+\lambda_2)^2} \, d\hat{\pi}_{n,2}(x) \right] . \tag{D.9}
$$

Finally, we put all the pieces together to compute the variance. First, taking expectation with respect to $\mathcal{D}_2$ on the both sides of Equation (D.6), we have:

$$
\mathbb{E} \left[ \left( \hat{\alpha}(\lambda_1)^\top \hat{\beta}(\lambda_2) \right)^2 \right]
$$

$$
= \mathbb{E} \left[ (\alpha_0^\top \hat{\beta}(\lambda_2))^2 \right] \mathbb{E} \left[ \left( \int \frac{x}{x+\lambda_1} \, d\hat{\pi}_{n,1}(x) \right)^2 \right]
$$

$$+ \frac{p}{(p-1)(p+2)} \mathbb{E}\left[\text{var}_{X \sim \hat{\pi}_{n,1}}\left(\frac{X}{X+\lambda_1}\right)\right] \left\{ \mathbb{E}\left[(\alpha_0^\top \hat{\beta}(\lambda_2))^2\right]\left(1 - \frac{2}{p}\right) + \|\alpha_0\|_2^2 \mathbb{E}\left[\|\hat{\beta}(\lambda_2)\|_2^2\right]\right\}$$

$$+ \frac{1}{n}\mathbb{E}\left[\|\hat{\beta}(\lambda_2)\|_2^2\right]\,\mathbb{E}\left[\int \frac{x}{(x+\lambda_1)^2}\,d\hat{\pi}_{n,1}(x)\right]$$

$$= (\alpha_0^\top \beta_0)^2 \mathbb{E}\left[\left(\int \frac{x}{x+\lambda_2}\,d\hat{\pi}_{n,2}(x)\right)^2\right]\mathbb{E}\left[\left(\int \frac{x}{x+\lambda_1}\,d\hat{\pi}_{n,1}(x)\right)^2\right] + R_{1,p}\mathbb{E}\left[\left(\int \frac{x}{x+\lambda_1}\,d\hat{\pi}_{n,1}(x)\right)^2\right]$$

$$+ \frac{p}{(p-1)(p+2)} \mathbb{E}\left[\text{var}_{X \sim \hat{\pi}_{n,1}}\left(\frac{X}{X+\lambda_1}\right)\right] \left\{ \mathbb{E}\left[(\alpha_0^\top \hat{\beta}(\lambda_2))^2\right]\left(1 - \frac{2}{p}\right) + \|\alpha_0\|_2^2 \mathbb{E}\left[\|\hat{\beta}(\lambda_2)\|_2^2\right]\right\}$$

$$+ \frac{1}{n}\mathbb{E}\left[\|\hat{\beta}(\lambda_2)\|_2^2\right]\,\mathbb{E}\left[\int \frac{x}{(x+\lambda_1)^2}\,d\hat{\pi}_{n,1}(x)\right]$$

$$= (\alpha_0^\top \beta_0)^2 \mathbb{E}\left[\left(\int \frac{x}{x+\lambda_1}\,d\hat{\pi}_{n,1}(x) \int \frac{x}{x+\lambda_2}\,d\hat{\pi}_{n,2}(x)\right)^2\right] + R_{1,p}\mathbb{E}\left[\left(\int \frac{x}{x+\lambda_1}\,d\hat{\pi}_{n,1}(x)\right)^2\right]$$

$$+ \frac{p}{(p-1)(p+2)} \mathbb{E}\left[\text{var}_{X \sim \hat{\pi}_{n,1}}\left(\frac{X}{X+\lambda_1}\right)\right] \left\{ \mathbb{E}\left[(\alpha_0^\top \hat{\beta}(\lambda_2))^2\right]\left(1 - \frac{2}{p}\right) + \|\alpha_0\|_2^2 \mathbb{E}\left[\|\hat{\beta}(\lambda_2)\|_2^2\right]\right\}$$

$$+ \frac{1}{n}\mathbb{E}\left[\|\hat{\beta}(\lambda_2)\|_2^2\right]\,\mathbb{E}\left[\int \frac{x}{(x+\lambda_1)^2}\,d\hat{\pi}_{n,1}(x)\right]$$

This, along with Equation (D.2), yields:

$$\text{var}\left(\hat{\alpha}(\lambda_1)^\top \hat{\beta}(\lambda_2)\right)$$

$$= (\alpha_0^\top \beta_0)^2 \text{var}\left(\int \frac{x}{x+\lambda_1}\,d\hat{\pi}_{n,1}(x) \int \frac{x}{x+\lambda_2}\,d\hat{\pi}_{n,2}(x)\right) + R_{1,p}\mathbb{E}\left[\left(\int \frac{x}{x+\lambda_1}\,d\hat{\pi}_{n,1}(x)\right)^2\right]$$

$$+ \frac{p}{(p-1)(p+2)} \mathbb{E}\left[\text{var}_{X \sim \hat{\pi}_{n,1}}\left(\frac{X}{X+\lambda_1}\right)\right] \left\{ \mathbb{E}\left[(\alpha_0^\top \hat{\beta}(\lambda_2))^2\right]\left(1 - \frac{2}{p}\right) + \|\alpha_0\|_2^2 \mathbb{E}\left[\|\hat{\beta}(\lambda_2)\|_2^2\right]\right\}$$

$$+ \frac{1}{n}\mathbb{E}\left[\|\hat{\beta}(\lambda_2)\|_2^2\right]\,\mathbb{E}\left[\int \frac{x}{(x+\lambda_1)^2}\,d\hat{\pi}_{n,1}(x)\right]$$

$$\triangleq T_1 + T_2 + T_3 + T_4\,.$$

To compute $\lim_{n \uparrow \infty} \text{var}\left(\hat{\alpha}(\lambda_1)^\top \hat{\beta}(\lambda_2)\right)$, we compute the limit of each $T_i$ separately. From Theorem 1.1 of [28], we have,

$$\lim_{n \uparrow \infty} n \times T_1 = \varrho^2 \lim_{n \uparrow \infty} n \times \text{var}\left(\int \frac{x}{x+\lambda_1}\,d\hat{\pi}_{n,1}(x) \int \frac{x}{x+\lambda_2}\,d\hat{\pi}_{n,2}(x)\right) = 0\,.$$

From the definition of $R_{1,p}$ in Equation (D.8), we have:

$$\lim_{n \uparrow \infty} n \times R_{1p} = \frac{1}{c}\text{var}_{X \sim F_{\text{MP}}}\left(\frac{X}{X+\lambda_2}\right)\left\{\varrho^2 + u^2 v^2\right\} + u^2 \int \frac{x\,dF_{\text{MP}}(x)}{(x+\lambda_2)^2}\,.$$

Therefore,

$$\lim_{n \uparrow \infty} n \times T_2 = \left[\frac{\varrho^2 + u^2 v^2}{c}\text{var}_{X \sim F_{\text{MP}}}\left(\frac{X}{X+\lambda_2}\right) + u^2 \int \frac{x\,dF_{\text{MP}}(x)}{(x+\lambda_2)^2}\right]\left(\int \frac{x\,dF_{\text{MP}}(x)}{x+\lambda_1}\right)^2\,.$$

Furthermore, from Equation (D.9), we have:

$$\lim_{n \uparrow \infty} \mathbb{E}[\|\hat{\beta}(\lambda_2)\|_2^2] = v^2 \left(\int \frac{x^2\,dF_{\text{MP}}(x)}{(x+\lambda_2)^2}\right) + c\left(\int \frac{x\,dF_{\text{MP}}(x)}{(x+\lambda_2)^2}\right)$$

which implies,

$$\lim_{n \uparrow \infty} n \times T_4 = \left\{v^2\left(\int \frac{x^2\,dF_{\text{MP}}(x)}{(x+\lambda_2)^2}\right) + c\left(\int \frac{x\,dF_{\text{MP}}(x)}{(x+\lambda_2)^2}\right)\right\}\left(\int \frac{x\,dF_{\text{MP}}(x)}{(x+\lambda_1)^2}\right)\,.$$

Finally, for $T_3$, first observe that,

$$\lim_{n\uparrow\infty} \mathbb{E}\left[\left(\alpha_0^\top \hat{\beta}(\lambda_2)\right)^2\right] = \varrho^2 \left(\int \frac{x\, dF_{\mathrm{MP}}(x)}{x+\lambda_2}\right)^2.$$

This implies,

$$\lim_{n\uparrow\infty} n \times T_3 = \frac{1}{c}\mathsf{var}_{X\sim F_{\mathrm{MP}}}\left(\frac{X}{X+\lambda_1}\right)\left\{\varrho^2 \left(\int \frac{x\, dF_{\mathrm{MP}}(x)}{x+\lambda_2}\right)^2 + \right.$$
$$\left. u^2\left[v^2\left(\int \frac{x^2\, dF_{\mathrm{MP}}(x)}{(x+\lambda_2)^2}\right) + c\left(\int \frac{x\, dF_{\mathrm{MP}}(x)}{(x+\lambda_2)^2}\right)\right]\right\}.$$

So overall, the limit is:

$$\lim_{n\uparrow\infty} n \times \mathsf{var}\left(\hat{\alpha}(\lambda_1)^\top \hat{\beta}(\lambda_2)\right)$$
$$= \left(\frac{\varrho^2 + u^2 v^2}{c}\mathsf{var}_{X\sim F_{\mathrm{MP}}}\left(\frac{X}{X+\lambda_2}\right) + u^2 \int \frac{x\, dF_{\mathrm{MP}}(x)}{(x+\lambda_2)^2}\right)\left(\int \frac{x\, dF_{\mathrm{MP}}(x)}{x+\lambda_1}\right)^2$$
$$+ \frac{1}{c}\mathsf{var}_{X\sim F_{\mathrm{MP}}}\left(\frac{X}{X+\lambda_1}\right)\left\{\varrho^2 \left(\int \frac{x\, dF_{\mathrm{MP}}(x)}{x+\lambda_2}\right)^2 + \right.$$
$$\left. u^2\left[v^2\left(\int \frac{x^2\, dF_{\mathrm{MP}}(x)}{(x+\lambda_2)^2}\right) + c\left(\int \frac{x\, dF_{\mathrm{MP}}(x)}{(x+\lambda_2)^2}\right)\right]\right\}$$
$$+ \left\{v^2\left(\int \frac{x^2\, dF_{\mathrm{MP}}(x)}{(x+\lambda_2)^2}\right) + c\left(\int \frac{x\, dF_{\mathrm{MP}}(x)}{(x+\lambda_2)^2}\right)\right\}\left(\int \frac{x\, dF_{\mathrm{MP}}(x)}{(x+\lambda_1)^2}\right).$$

Therefore, the limiting variance of $\hat{\theta}_{3\mathrm{sp}}^{\mathrm{INT,db}}$ is:

$$\lim_{n\uparrow\infty} n \times \hat{\mathsf{var}}\left(\theta_{3\mathrm{sp}}^{\mathrm{INT,db}}\right)$$
$$= \mathsf{var}(AY) + \frac{1}{(g_{3\mathrm{sp}}^{\mathrm{INT}}(\lambda_1,\lambda_2))^2}\left[\left(\frac{\varrho^2 + u^2 v^2}{c}\mathsf{var}_{X\sim F_{\mathrm{MP}}}\left(\frac{X}{X+\lambda_2}\right) + u^2 \int \frac{x\, dF_{\mathrm{MP}}(x)}{(x+\lambda_2)^2}\right)\left(\int \frac{x\, dF_{\mathrm{MP}}(x)}{x+\lambda_1}\right)^2\right.$$
$$+ \frac{1}{c}\mathsf{var}_{X\sim F_{\mathrm{MP}}}\left(\frac{X}{X+\lambda_1}\right)\left\{\varrho^2 \left(\int \frac{x\, dF_{\mathrm{MP}}(x)}{x+\lambda_2}\right)^2 + \right.$$
$$\left. u^2\left[v^2\left(\int \frac{x^2\, dF_{\mathrm{MP}}(x)}{(x+\lambda_2)^2}\right) + c\left(\int \frac{x\, dF_{\mathrm{MP}}(x)}{(x+\lambda_2)^2}\right)\right]\right\}$$
$$\left. + \left\{v^2\left(\int \frac{x^2\, dF_{\mathrm{MP}}(x)}{(x+\lambda_2)^2}\right) + c\left(\int \frac{x\, dF_{\mathrm{MP}}(x)}{(x+\lambda_2)^2}\right)\right\}\left(\int \frac{x\, dF_{\mathrm{MP}}(x)}{(x+\lambda_1)^2}\right)\right]$$

This completes the proof.

## D.2 Limiting Variance for $\hat{\theta}_{2\mathrm{sp}}^{\mathrm{INT,db}}$

Recall that in the case of $\hat{\theta}_{2\mathrm{sp}}^{\mathrm{INT,db}}$, we estimate $\alpha_0$ and $\beta_0$ from $\mathcal{D}_1$. The estimator takes the form:

$$\hat{\theta}_{2\mathrm{sp}}^{\mathrm{INT,db}} = \left(1 - \frac{g_{2,2\mathrm{sp}}^{\mathrm{INT}}(\lambda_1,\lambda_2)}{g_{1,2\mathrm{sp}}^{\mathrm{INT}}(\lambda_1,\lambda_2)}\right)^{-1}\left[\frac{1}{n}\sum_{i\in\mathcal{D}_2} A_i Y_i - \frac{\hat{\alpha}(\lambda_1)^\top \hat{\beta}(\lambda_2)}{g_{1,2\mathrm{sp}}^{\mathrm{INT}}(\lambda_1,\lambda_2)}\right].$$

Therefore,

$$n \times \mathsf{var}\left(\hat{\theta}_{2\mathrm{sp}}^{\mathrm{INT,db}}\right) = \left(1 - \frac{g_{2,2\mathrm{sp}}^{\mathrm{INT}}(\lambda_1,\lambda_2)}{g_{1,2\mathrm{sp}}^{\mathrm{INT}}(\lambda_1,\lambda_2)}\right)^{-2}\left[\mathsf{var}(AY) + \frac{n \times \mathsf{var}\left(\hat{\alpha}(\lambda_1)^\top \hat{\beta}(\lambda_2)\right)}{\left(g_{1,2\mathrm{sp}}^{\mathrm{INT}}(\lambda_1,\lambda_2)\right)^2}\right]$$

As a consequence, like the three-split case, we need to derive the limit of $n \times \text{var}\left(\hat{\alpha}(\lambda_1)^\top \hat{\beta}(\lambda_2)\right)$, but the only difference is that they are now estimated from the same subsample. We break the proof into two propositions.

**Proposition D.1.** *The expected value of the conditional variance of $\hat{\alpha}^\top(\lambda_1)\hat{\beta}(\lambda_2)$ given $\mathbf{X}_1$ satisfies:*

$$
\lim_{n \uparrow \infty} n \times \mathbb{E}\left[\text{var}\left(\hat{\alpha}^\top(\lambda_1)\hat{\beta}(\lambda_2) \mid \mathbf{X}_1\right)\right] = \left(u^2 + v^2 + 2\varrho + c(1+\rho^2)\right) \int \frac{x^2 \, dF_{\text{MP}}(x)}{(x+\lambda_1)^2(x+\lambda_2)^2}.
$$

*Proof.* From the definition of the conditional variance, we have:

$$
\text{var}\left(\hat{\alpha}^\top(\lambda_1)\hat{\beta}(\lambda_2) \mid \mathbf{X}_1\right) = \mathbb{E}\left[\left(\hat{\alpha}^\top(\lambda_1)\hat{\beta}(\lambda_2)\right)^2 \mid \mathbf{X}_1\right] - \left(\mathbb{E}\left[\hat{\alpha}^\top(\lambda_1)\hat{\beta}(\lambda_2) \mid \mathbf{X}_1\right]\right)^2
$$

Let us first start with conditional expectation. For notational simplicity, define $R(\lambda) = (\hat{\Sigma}_1 + \lambda \mathbf{I})^{-1}$. From the definition of $\hat{\alpha}^\top(\lambda_1)\hat{\beta}(\lambda_2)$, we have:

$$
\begin{aligned}
\hat{\alpha}^\top(\lambda_1)\hat{\beta}(\lambda_2) &= \left\{\hat{\Sigma}_1 \alpha_0 + \frac{\mathbf{X}_1^\top \boldsymbol{\epsilon}}{n}\right\}^\top R(\lambda_1)R(\lambda_2)\left\{\hat{\Sigma}_1 \beta_0 + \frac{\mathbf{X}_1^\top \boldsymbol{\mu}}{n}\right\} \\
&= \alpha_0^\top \hat{\Sigma}_1 R(\lambda_1)R(\lambda_2)\hat{\Sigma}_1 \beta_0 + \alpha_0^\top \hat{\Sigma}_1 R(\lambda_1)R(\lambda_2)\frac{\mathbf{X}_1^\top \boldsymbol{\mu}}{n} \\
&\quad + \beta_0^\top \hat{\Sigma}_1 R(\lambda_1)R(\lambda_2)\frac{\mathbf{X}_1^\top \boldsymbol{\epsilon}}{n} + \frac{\boldsymbol{\epsilon}^\top \mathbf{X}_1}{n}R(\lambda_1)R(\lambda_2)\frac{\mathbf{X}_1^\top \boldsymbol{\mu}}{n} \\
&\triangleq T_1 + T_2 + T_3 + T_4 \,. \tag{D.10}
\end{aligned}
$$

As we are interested in the conditional variance given $\mathbf{X}_1$, $T_1$, being a function of $\mathbf{X}_1$, will not contribute. Therefore,

$$
\begin{aligned}
\text{var}\left(\hat{\alpha}^\top(\lambda_1)\hat{\beta}(\lambda_2) \mid \mathbf{X}_1\right) &= \text{var}(T_2 + T_3 + T_4 \mid \mathbf{X}_1) \\
&= \mathbb{E}\left[(T_2 + T_3 + T_4)^2 \mid \mathbf{X}_1\right] - \left(\mathbb{E}[(T_2 + T_3 + T_4) \mid \mathbf{X}_1]\right)^2
\end{aligned}
$$

As $\epsilon$ and $\mu$ are centered Normal random variable and independent of $X$, we have:

$$
\mathbb{E}[T_2 \mid \mathbf{X}_1] = \mathbb{E}[T_3 \mid \mathbf{X}_1] = 0 \,.
$$

Therefore,

$$
\begin{aligned}
\mathbb{E}\left[(T_2 + T_3 + T_4) \mid \mathbf{X}_1\right] &= \mathbb{E}[T_4 \mid \mathbf{X}_1] \\
&= \mathbb{E}\left[\frac{\boldsymbol{\epsilon}^\top \mathbf{X}}{n}R(\lambda_1)R(\lambda_2)\frac{\mathbf{X}^\top \boldsymbol{\mu}}{n} \mid \mathbf{X}_1\right] = \frac{\rho}{n}\text{tr}\left(R(\lambda_1)R(\lambda_2)\hat{\Sigma}_1\right).
\end{aligned}
$$
(D.11)

Next, we consider $\mathbb{E}[(T_2 + T_3 + T_4)^2 \mid X]$. Expanding the square, we have:

$$
\begin{aligned}
\mathbb{E}[(T_2 + T_3 + T_4)^2 \mid X] &= \mathbb{E}[T_2^2 \mid \mathbf{X}] + \mathbb{E}[T_3^2 \mid \mathbf{X}] + \mathbb{E}[T_4^2 \mid \mathbf{X}] \\
&\quad + 2\left(\mathbb{E}[T_2 T_3 \mid \mathbf{X}] + \mathbb{E}[T_2 T_4 \mid \mathbf{X}] + \mathbb{E}[T_3 T_4 \mid \mathbf{X}]\right)
\end{aligned}
$$

From the normality of $(\epsilon, \mu)$ and their independence with $X$, we have the following conditional expectation of the six terms above:

$$
\begin{cases}
\mathbb{E}[T_2^2 \mid \mathbf{X}] = \dfrac{\alpha_0^\top \hat{\Sigma}_1 R(\lambda_1) R(\lambda_2) \hat{\Sigma}_1 R(\lambda_1) R(\lambda_2) \alpha_0}{n} \\[2ex]
\mathbb{E}[T_3^2 \mid \mathbf{X}] = \dfrac{\beta_0^\top \hat{\Sigma}_1 R(\lambda_1) R(\lambda_2) \hat{\Sigma}_1 R(\lambda_1) R(\lambda_2) \beta_0}{n} \\[2ex]
\mathbb{E}[T_4^2 \mid \mathbf{X}] = (1 + \rho^2) \dfrac{1}{n^2} \mathsf{tr}\left( \hat{\Sigma} R(\lambda_1) R(\lambda_2) \hat{\Sigma}_1 R(\lambda_1) R(\lambda_2) \right) + \left( \dfrac{\rho}{n} \mathsf{tr}\left( \hat{\Sigma}_1 R(\lambda_1) R(\lambda_2) \right) \right)^2 \\[2ex]
\mathbb{E}[T_2 T_3 \mid \mathbf{X}] = \dfrac{\alpha_0^\top \hat{\Sigma}_1 R(\lambda_1) R(\lambda_2) \hat{\Sigma}_1 R(\lambda_1) R(\lambda_2) \beta_0}{n} \\[2ex]
\mathbb{E}[T_2 T_4 \mid \mathbf{X}] = \dfrac{\mathbb{E}[\epsilon \mu^2]}{n^3} \sum_{i=1}^{n} \alpha_0^\top \hat{\Sigma}_1 R(\lambda_1) R(\lambda_2) X_i X_i^\top R(\lambda_1) R(\lambda_2) X_i = 0 \\[2ex]
\mathbb{E}[T_3 T_4 \mid \mathbf{X}] = \dfrac{\mathbb{E}[\epsilon^2 \mu]}{n^3} \sum_{i=1}^{n} \beta_0^\top \hat{\Sigma}_1 R(\lambda_1) R(\lambda_2) X_i X_i^\top R(\lambda_1) R(\lambda_2) X_i = 0
\end{cases}
$$
(D.12)

Combining Equation (D.11) and (D.12), we conclude that:

$$
\begin{aligned}
&\mathsf{var}\left( \hat{\alpha}^\top(\lambda_1) \hat{\beta}(\lambda_2) \mid \mathbf{X} \right) \\
&= \frac{\alpha_0^\top \hat{\Sigma}_1 R(\lambda_1) R(\lambda_2) \hat{\Sigma}_1 R(\lambda_1) R(\lambda_2) \alpha_0}{n} + \frac{\beta_0^\top \hat{\Sigma}_1 R(\lambda_1) R(\lambda_2) \hat{\Sigma}_1 R(\lambda_1) R(\lambda_2) \beta_0}{n} \\
&\quad + 2 \frac{\alpha_0^\top \hat{\Sigma}_1 R(\lambda_1) R(\lambda_2) \hat{\Sigma}_1 R(\lambda_1) R(\lambda_2) \beta_0}{n} + \frac{(1 + \rho^2)}{n^2} \mathsf{tr}\left( \hat{\Sigma} R(\lambda_1) R(\lambda_2) \hat{\Sigma}_1 R(\lambda_1) R(\lambda_2) \right) \\
&\quad + \cancel{\left( \frac{\rho}{n} \mathsf{tr}\left( \hat{\Sigma} R(\lambda_1) R(\lambda_2) \right) \right)^2} - \cancel{\left( \frac{\rho}{n} \mathsf{tr}\left( R(\lambda_1) R(\lambda_2) \hat{\Sigma} \right) \right)^2} \\
&= \frac{\alpha_0^\top \hat{\Sigma}_1 R(\lambda_1) R(\lambda_2) \hat{\Sigma}_1 R(\lambda_1) R(\lambda_2) \alpha_0}{n} + \frac{\beta_0^\top \hat{\Sigma}_1 R(\lambda_1) R(\lambda_2) \hat{\Sigma}_1 R(\lambda_1) R(\lambda_2) \beta_0}{n} \\
&\quad + 2 \frac{\alpha_0^\top \hat{\Sigma}_1 R(\lambda_1) R(\lambda_2) \hat{\Sigma}_1 R(\lambda_1) R(\lambda_2) \beta_0}{n} + \frac{(1 + \rho^2)}{n^2} \mathsf{tr}\left( \hat{\Sigma} R(\lambda_1) R(\lambda_2) \hat{\Sigma}_1 R(\lambda_1) R(\lambda_2) \right)
\end{aligned}
$$
(D.13)

Therefore, we need to compute the expected value of the four terms on the RHS of the above equation. Towards that end, we first compute the expectation of a general term of the form:

$$
\mathbb{E}\left[ \frac{\mathbf{u}^\top \hat{\Sigma}_1 R(\lambda_1) R(\lambda_2) \hat{\Sigma}_1 R(\lambda_1) R(\lambda_2) \mathbf{v}}{n} \right]
$$

This would cover the first three terms of the RHS of Equation (D.13). Suppose $\hat{\Sigma}_1 = \sum_{j=1}^{p} \hat{v}_j \hat{v}_j^\top \hat{\lambda}_j$ denote the eigendecomposition of $\hat{\Sigma}_1$. Then,

$$
\mathbb{E}\left[ \frac{\mathbf{u}^\top \hat{\Sigma}_1 R(\lambda_1) R(\lambda_2) \hat{\Sigma}_1 R(\lambda_1) R(\lambda_2) \mathbf{v}}{n} \right] = \frac{1}{n} \sum_{j=1}^{p} \mathbb{E}\left[ (\mathbf{u}^\top \hat{v}_j)(\mathbf{v}^\top \hat{v}_j) \frac{\hat{\lambda}_j^2}{(\hat{\lambda}_j + \lambda_1)^2 (\hat{\lambda}_j + \lambda_2)^2} \right]
$$
(D.14)

From the rotational symmetry of the Normal distribution, we know that the eigenvectors and eigenvalues of $\hat{\Sigma}$ are independent. Moreoever each eigenvector $\hat{v}_j$ is uniformly distributed on $\mathbb{S}^{p-1}$. Using this fact, we obtain:

$$
\begin{aligned}
\mathbb{E}\left[ (\mathbf{u}^\top \hat{v}_j)(\mathbf{v}^\top \hat{v}_j) \frac{\hat{\lambda}_j^2}{(\hat{\lambda}_j + \lambda)^4} \right] &= \mathbb{E}[(\mathbf{u}^\top \hat{v}_j)(\mathbf{v}^\top \hat{v}_j)] \mathbb{E}\left[ \frac{\hat{\lambda}_j^2}{(\hat{\lambda}_j + \lambda_1)^2 (\hat{\lambda}_j + \lambda_2)^2} \right] \\
&= \frac{\mathbf{u}^\top \mathbf{v}}{p} \mathbb{E}\left[ \frac{\hat{\lambda}_j^2}{(\hat{\lambda}_j + \lambda_1)^2 (\hat{\lambda}_j + \lambda_2)^2} \right]
\end{aligned}
$$

In the last equality, we have used the fact that for a random variable $\tau$, uniformly distributed on $\mathbb{S}^{p-1}$, we have $\mathbb{E}[(\mathbf{u}^\top \tau)(\mathbf{v}^\top \tau)] = (\mathbf{u}^\top \mathbf{v})/p$. Therefore, going back to Equation (D.14), we have:

$$\mathbb{E}\left[\frac{\mathbf{u}^\top \hat{\Sigma}_1 R(\lambda_1) R(\lambda_2) \hat{\Sigma}_1 R(\lambda_1) R(\lambda_2) \mathbf{v}}{n}\right] = \frac{\mathbf{u}^\top \mathbf{v}}{n} \mathbb{E}\left[\frac{1}{p}\sum_{j=1}^{p} \frac{\hat{\lambda}_j^2}{(\hat{\lambda}_j + \lambda_1)^2(\hat{\lambda}_j + \lambda_2)^2}\right]$$

$$= \frac{\mathbf{u}^\top \mathbf{v}}{n} \mathbb{E}\left[\int \frac{x^2 \, d\hat{\pi}_n(x)}{(x + \lambda_1)^2 (x + \lambda_2)^2}\right] \qquad \text{(D.15)}$$

We now consider the expectation of the fourth term of the RHS of Equation (D.13). Using the eigen decomposition of $\hat{\Sigma}$ we have:

$$\frac{(1+\rho^2)}{n^2}\mathbb{E}\left[\operatorname{tr}\left(\hat{\Sigma}R(\lambda_1)R(\lambda_2)\hat{\Sigma}_1 R(\lambda_1)R(\lambda_2)\right)\right] = \frac{(1+\rho^2)}{n^2}\mathbb{E}\left[\sum_{j=1}^{p}\frac{\hat{\lambda}_j^2}{(\hat{\lambda}_j + \lambda_1)^2(\hat{\lambda}_j + \lambda_2)^2}\right]$$

$$= \frac{c(1+\rho^2)}{n}\mathbb{E}\left[\frac{1}{p}\sum_{j=1}^{p}\frac{\hat{\lambda}_j^2}{(\hat{\lambda}_j + \lambda_1)^2(\hat{\lambda}_j + \lambda_2)^2}\right]$$

$$= \frac{c(1+\rho^2)}{n}\mathbb{E}\left[\int \frac{x^2 \, d\hat{\pi}_n(x)}{(x + \lambda_1)^2(x + \lambda_2)^2}\right]. \qquad \text{(D.16)}$$

Finally, combining Equation (D.13), (D.15), and (D.16), we conclude that:

$$n\,\mathbb{E}\left[\operatorname{var}\left(\hat{\alpha}^\top(\lambda_1)\hat{\beta}(\lambda_2) \mid \mathbf{X}\right)\right] = \left(\|\alpha_0 + \beta_0\|_2^2 + c(1 + \rho^2)\right)\mathbb{E}\left[\int \frac{x^2 \, d\hat{\pi}_n(x)}{(x + \lambda_1)^2(x + \lambda_2)^2}\right]. \qquad \text{(D.17)}$$

Now, using DCT and the fact that ESD of $\hat{\Sigma}_1$ follows Marchenko-Pastur law, we conclude the proof of the Proposition. $\qquad \square$

**Proposition D.2.** *The expected value of the conditional variance of $\hat{\alpha}^\top(\lambda_1)\hat{\beta}(\lambda_2)$ given $\mathbf{X}$ satisfies:*

$$\lim_{n\uparrow\infty} n \times \operatorname{var}\left(\mathbb{E}\left[\hat{\alpha}^\top(\lambda_1)\hat{\beta}(\lambda_2) \mid \mathbf{X}\right]\right) = \frac{1}{c}\left[u^2 v^2 + \varrho^2\right] \operatorname{var}_{X \sim F_{\mathrm{MP}}}\left(\frac{X^2}{(X + \lambda_1)(X + \lambda_2)}\right).$$

*Proof.* It is immediate from Equation (D.10) that

$$\operatorname{var}\left(\mathbb{E}\left[\hat{\alpha}^\top(\lambda_1)\hat{\beta}(\lambda_2) \mid \mathbf{X}\right]\right) = \operatorname{var}\left(\alpha_0^\top \hat{\Sigma}_1 R(\lambda_1) R(\lambda_2) \hat{\Sigma}_1 \beta_0 + \mathbb{E}[T_4 \mid \mathbf{X}]\right)$$

$$= \operatorname{var}\left(\alpha_0^\top \hat{\Sigma}_1 R(\lambda_1) R(\lambda_2) \hat{\Sigma}_1 \beta_0 + \frac{\rho}{n}\operatorname{tr}\left(R(\lambda_1) R(\lambda_2) \hat{\Sigma}\right)\right).$$

Next, we compute $\operatorname{var}\left(\alpha_0^\top \hat{\Sigma}_1 R(\lambda_1) R(\lambda_2) \hat{\Sigma}_1 \beta_0\right)$ and then argue that the variance of the second term is of negligible order. Towards that end, an application of Lemma E.1 yields:

$$\lim_{n\uparrow\infty} n \times \operatorname{var}\left(\alpha_0^\top \hat{\Sigma}_1 R(\lambda_1) R(\lambda_2) \hat{\Sigma}_1 \beta_0\right) = \frac{u^2 v^2 + \varrho^2}{c}\operatorname{var}_{X \sim F_{\mathrm{MP}}}\left(\frac{X^2}{(X + \lambda_1)(X + \lambda_2)}\right).$$

Therefore, to conclude the proof of the Proposition, we need to show that

$$\lim_{n\uparrow\infty} n \times \operatorname{var}\left(\frac{\rho}{n}\operatorname{tr}\left(R(\lambda_1) R(\lambda_2) \hat{\Sigma}\right)\right) = 0.$$

Towards that goal, we will again use Theorem 1.1 of [28].

$$\lim_{n\uparrow\infty} n \times \operatorname{var}\left(\frac{\rho}{n}\operatorname{tr}\left(R(\lambda_1) R(\lambda_2) \hat{\Sigma}\right)\right) = \lim_{n\uparrow\infty} n \times \operatorname{var}\left(c\rho \int \frac{x \, d\hat{\pi}_n(x)}{(x + \lambda_1)(x + \lambda_2)}\right)$$

$$= c^2 \rho^2 \lim_{n\uparrow\infty} n \times \operatorname{var}\left(\int \frac{x \, d\hat{\pi}_n(x)}{(x + \lambda_1)(x + \lambda_2)}\right) = 0$$

Here, the last inequality follows from Theorem 1.1 of [28]. This concludes the proof of the Proposition. $\qquad \square$

Finally, combining the results of Proposition D.1 and D.2, we conclude the proof of the limiting variance of $\hat{\theta}_{2\mathrm{sp}}^{\mathrm{INT,db}}$.

## D.3 Limiting Variance of $\hat{\theta}_{2\text{sp}}^{\text{DR,db}}$

Recall that $\hat{\theta}_{2\text{sp}}^{\text{DR,db}}$ has the following form:

$$
\begin{aligned}
\hat{\theta}_{2\text{sp}}^{\text{DR,db}} &= H_1(\boldsymbol{\lambda}) \left[ \hat{\theta}^{\text{DR}} - \hat{\alpha}(\lambda_1)^\top \hat{\beta}(\lambda_2) H_2(\boldsymbol{\lambda}) \right] \\
&= H_1(\boldsymbol{\lambda}) \left[ \frac{1}{n} \sum_{i \in \mathcal{D}_2} (Y_i - X_i^\top \hat{\beta}(\lambda_2))(A_i - X_i^\top \hat{\alpha}(\lambda_1)) - \hat{\alpha}(\lambda_1)^\top \hat{\beta}(\lambda_2) H_2(\boldsymbol{\lambda}) \right] \\
&= H_1(\boldsymbol{\lambda}) \left[ (\alpha_0 - \hat{\alpha}(\lambda_1)) \hat{\Sigma}_2 (\beta_0 - \hat{\beta}(\lambda_2)) + \frac{1}{n} \sum_{i \in \mathcal{D}_2} (X_i^\top (\beta_0 - \hat{\beta}(\lambda_2))) \epsilon_i \right. \\
&\qquad \left. + \frac{1}{n} \sum_{i \in \mathcal{D}_2} (X_i^\top (\alpha_0 - \hat{\alpha}(\lambda_1))) \mu_i + \frac{1}{n} \sum_{i \in \mathcal{D}_2} \mu_i \epsilon_i - \hat{\alpha}(\lambda_1)^\top \hat{\beta}(\lambda_2) H_2(\boldsymbol{\lambda}) \right]
\end{aligned}
$$

We next use the tower property conditional on $\mathcal{D}_1$. The conditional expectation is:

$$
\mathbb{E}\left[ \hat{\theta}_{2\text{sp}}^{\text{DR,db}} \mid \mathcal{D}_1 \right] = H_1(\boldsymbol{\lambda}) \left[ (\alpha_0 - \hat{\alpha}(\lambda_1))^\top (\beta_0 - \hat{\beta}(\lambda_2)) + \theta_0 - \hat{\alpha}(\lambda_1)^\top \hat{\beta}(\lambda_2) H_2(\boldsymbol{\lambda}) \right]
$$

Taking the variance with respect $\mathcal{D}_1$, we have:

$$
\begin{aligned}
&\mathsf{var}\left( \mathbb{E}\left[ \hat{\theta}_{2\text{sp}}^{\text{DR,db}} \mid \mathcal{D}_1 \right] \right) \\
&= H_1^2(\boldsymbol{\lambda}) \mathsf{var}\left( (\alpha_0 - \hat{\alpha}(\lambda_1))^\top (\beta_0 - \hat{\beta}(\lambda_2)) - \hat{\alpha}(\lambda_1)^\top \hat{\beta}(\lambda_2) H_2(\boldsymbol{\lambda}) \right) \\
&= H_1^2(\boldsymbol{\lambda}) \mathsf{var}\left( \hat{\alpha}(\lambda_1)^\top \hat{\beta}(\lambda_2)(1 - H_2(\boldsymbol{\lambda})) - \hat{\alpha}(\lambda_1)^\top \beta_0 - \alpha_0^\top \hat{\beta}(\lambda_2) \right) \\
&= H_1^2(\boldsymbol{\lambda}) \left[ (1 - H_2(\boldsymbol{\lambda}))^2 \mathsf{var}(\hat{\alpha}(\lambda_1)^\top \hat{\beta}(\lambda_2)) + \mathsf{var}(\hat{\alpha}(\lambda_1)^\top \beta_0) + \mathsf{var}(\alpha_0^\top \hat{\beta}(\lambda_2)) \right. \\
&\quad - 2(1 - H_2(\boldsymbol{\lambda})) \mathsf{cov}\left( \hat{\alpha}(\lambda_1)^\top \hat{\beta}(\lambda_2), \, \hat{\alpha}(\lambda_1)^\top \beta_0 \right) - 2(1 - H_2(\boldsymbol{\lambda})) \mathsf{cov}\left( \hat{\alpha}(\lambda_1)^\top \hat{\beta}(\lambda_2), \, \alpha_0^\top \hat{\beta}(\lambda_2) \right) \\
&\quad \left. + 2\mathsf{cov}(\hat{\alpha}(\lambda_1)^\top \beta_0, \alpha_0^\top \hat{\beta}(\lambda_2)) \right] .
\end{aligned}
$$

We now derive the limit of each of the summands. For the first term, we have from Proposition D.1 and D.2:

$$
\begin{aligned}
\lim_{n \uparrow \infty} n \times \mathsf{var}(\hat{\alpha}(\lambda_1)^\top \hat{\beta}(\lambda_2)) &= \left( u^2 + v^2 + 2\varrho + c(1 + \rho^2) \right) \int \frac{x^2}{(x + \lambda_1)^2 (x + \lambda_2)^2} \, dF_{\text{MP}}(x) \\
&\quad + \frac{1}{c} \left[ u^2 v^2 + \varrho^2 \right] \mathsf{var}_{X \sim F_{\text{MP}}} \left( \frac{X^2}{(X + \lambda_1)(X + \lambda_2)} \right) .
\end{aligned}
$$

For the second summand, we have:

$$
\begin{aligned}
\mathsf{var}(\hat{\alpha}(\lambda_1)^\top \beta_0) &= \mathsf{var}\left( \left( R(\lambda_1)\hat{\Sigma}_1 \alpha_0 + R(\lambda_1) \frac{\mathbf{X}_1^\top \boldsymbol{\epsilon}_1}{n} \right)^\top \beta_0 \right) \\
&= \mathsf{var}(\alpha_0^\top \hat{\Sigma}_1 R(\lambda_1) \beta_0) + \mathsf{var}\left( \beta_0^\top R(\lambda_1) \frac{\mathbf{X}_1^\top \boldsymbol{\epsilon}_1}{n} \right) \\
&= \mathsf{var}\left( \alpha_0^\top \hat{\Sigma}_1 R(\lambda_1) \beta_0 \right) + \frac{1}{n} \mathbb{E}\left[ \beta_0^\top R(\lambda_1) \hat{\Sigma}_1 R(\lambda_1) \beta_0 \right] \\
&= \mathsf{var}\left( \alpha_0^\top \hat{\Sigma}_1 R(\lambda_1) \beta_0 \right) + \|\beta_0\|_2^2 \, \mathbb{E}\left[ \int \frac{x \, d\hat{\pi}_{n,1}(x)}{(x + \lambda_1)^2} \right]
\end{aligned}
$$

From Lemma E.1, we have:

$$
\lim_{n \uparrow \infty} n \times \mathsf{var}\left( \alpha_0^\top \hat{\Sigma}_1 R(\lambda_1) \beta_0 \right) = \frac{1}{c} (\varrho^2 + u^2 v^2) \, \mathsf{var}_{X \sim F_{\text{MP}}} \left( \frac{X}{X + \lambda_1} \right)
$$

Therefore,

$$\lim_{n\uparrow\infty} n \times \mathsf{var}(\hat\alpha(\lambda_1)^\top\beta_0) = \frac{1}{c}(\varrho^2 + u^2v^2)\,\mathsf{var}_{X\sim F_{\mathrm{MP}}}\left(\frac{X}{X+\lambda_1}\right) + v^2\int\frac{x\,dF_{\mathrm{MP}}(x)}{(x+\lambda_1)^2}\,.$$

A similar calculation yields:

$$\lim_{n\uparrow\infty} n \times \mathsf{var}(\alpha_0^\top\hat\beta(\lambda_2)) = \frac{1}{c}(\varrho^2 + u^2v^2)\,\mathsf{var}_{X\sim F_{\mathrm{MP}}}\left(\frac{X}{X+\lambda_2}\right) + u^2\int\frac{x\,dF_{\mathrm{MP}}(x)}{(x+\lambda_2)^2}\,.$$

Next, we compute the limits of covariance terms:

$$\mathsf{cov}\left(\hat\alpha(\lambda_1)^\top\hat\beta(\lambda_2),\ \hat\alpha(\lambda_1)^\top\beta_0\right)$$

$$= \mathsf{cov}\left(\alpha_0^\top\hat\Sigma_1 R(\lambda_1)R(\lambda_2)\hat\Sigma_1\beta_0,\ \alpha_0^\top\hat\Sigma_1 R(\lambda_1)\beta_0\right) + \mathsf{cov}\left(\alpha_0^\top\hat\Sigma_1 R(\lambda_1)R(\lambda_2)\frac{\mathbf{X}_1^\top\boldsymbol\mu_1}{n}, \beta_0^\top R(\lambda_1)\frac{\mathbf{X}_1^\top\boldsymbol\epsilon_1}{n}\right)$$

$$+ \mathsf{cov}\left(\beta_0^\top R(\lambda_1)\frac{\mathbf{X}_1^\top\boldsymbol\epsilon_1}{n},\ \beta_0^\top\hat\Sigma_1 R(\lambda_1)R(\lambda_2)\frac{\mathbf{X}_1^\top\boldsymbol\epsilon_1}{n}\right) + \mathsf{cov}\left(\frac{\boldsymbol\mu_1^\top\mathbf{X}_1}{n}R(\lambda_1)R(\lambda_2)\frac{\mathbf{X}_1^\top\boldsymbol\epsilon_1}{n}, \alpha_0^\top\hat\Sigma_1 R(\lambda_1)\beta_0\right)\,.$$

For the first covariance term, an application of Lemma E.3 yields:

$$\mathsf{cov}\left(\alpha_0^\top\hat\Sigma_1 R(\lambda_1)R(\lambda_2)\hat\Sigma_1\beta_0,\ \alpha_0^\top\hat\Sigma_1 R(\lambda_1)\beta_0\right)$$

$$\longrightarrow \frac{1}{c}(u^2v^2 + \varrho^2)\,\mathsf{cov}_{X\sim F_{\mathrm{MP}}}\left(\frac{X^2}{(X+\lambda_1)(X+\lambda_2)},\ \frac{X}{X+\lambda_1}\right)\,.$$

For the second term, we have:

$$\mathsf{cov}\left(\alpha_0^\top\hat\Sigma_1 R(\lambda_1)R(\lambda_2)\frac{\mathbf{X}_1^\top\boldsymbol\mu_1}{n}, \beta_0^\top R(\lambda_1)\frac{\mathbf{X}_1^\top\boldsymbol\epsilon_1}{n}\right)$$

$$= \mathbb{E}\left[\left(\alpha_0^\top\hat\Sigma_1 R(\lambda_1)R(\lambda_2)\frac{\mathbf{X}_1^\top\boldsymbol\mu_1}{n}\right)\left(\beta_0^\top R(\lambda_1)\frac{\mathbf{X}_1^\top\boldsymbol\epsilon_1}{n}\right)\right]$$

$$= \frac{\theta_0}{n}\mathbb{E}\left[\beta_0^\top R(\lambda_1)\hat\Sigma_1 R(\lambda_1)R(\lambda_2)\hat\Sigma_1\alpha_0\right]$$

$$= \frac{\theta_0}{n}\alpha_0^\top\beta_0\mathbb{E}\left[\int\frac{x^2\,d\hat\pi_{n,1}(x)}{(x+\lambda_1)^2(x+\lambda_2)}\right]$$

Hence,

$$\lim_{n\uparrow\infty} n \times \mathsf{cov}\left(\alpha_0^\top\hat\Sigma_1 R(\lambda_1)R(\lambda_2)\frac{\mathbf{X}_1^\top\boldsymbol\mu_1}{n}, \beta_0^\top R(\lambda_1)\frac{\mathbf{X}_1^\top\boldsymbol\epsilon_1}{n}\right) \longrightarrow \theta_0\varrho\int\frac{x^2\,dF_{\mathrm{MP}}(x)}{(x+\lambda_1)^2(x+\lambda_2)}\,.$$

For the third term,

$$\mathsf{cov}\left(\beta_0^\top R(\lambda_1)\frac{\mathbf{X}_1^\top\boldsymbol\epsilon_1}{n},\ \beta_0^\top\hat\Sigma_1 R(\lambda_1)R(\lambda_2)\frac{\mathbf{X}_1^\top\boldsymbol\epsilon_1}{n}\right) = \mathbb{E}\left[\frac{\boldsymbol\epsilon^\top\mathbf{X}_1}{n}R(\lambda_1)\beta_0\beta_0^\top\hat\Sigma_1 R(\lambda_1)R(\lambda_2)\frac{\mathbf{X}_1^\top\boldsymbol\epsilon_1}{n}\right]$$

$$= \frac{1}{n}\mathbb{E}\left[\beta_0^\top\hat\Sigma_1 R(\lambda_1)R(\lambda_2)\hat\Sigma_1 R(\lambda_1)\beta_0\right]$$

$$= \frac{\|\beta_0\|_2^2}{n}\mathbb{E}\left[\int\frac{x^2\,d\hat\pi_{n,1}(x)}{(x+\lambda_1)^2(x+\lambda_2)}\right]$$

Therefore,

$$\lim_{n\uparrow\infty} n \times \mathsf{cov}\left(\beta_0^\top R(\lambda_1)\frac{\mathbf{X}_1^\top\boldsymbol\epsilon_1}{n},\ \beta_0^\top\hat\Sigma_1 R(\lambda_1)R(\lambda_2)\frac{\mathbf{X}_1^\top\boldsymbol\epsilon_1}{n}\right) \longrightarrow v^2\int\frac{x^2\,dF_{\mathrm{MP}}(x)}{(x+\lambda_1)^2(x+\lambda_2)}\,.$$

For the fourth term:

$$\mathsf{cov}\left(\frac{\boldsymbol\mu_1^\top\mathbf{X}_1}{n}R(\lambda_1)R(\lambda_2)\frac{\mathbf{X}_1^\top\boldsymbol\epsilon_1}{n}, \alpha_0^\top\hat\Sigma_1 R(\lambda_1)\beta_0\right)$$

$$= \mathbb{E}\left[\left(\frac{\boldsymbol\mu_1^\top\mathbf{X}_1}{n}R(\lambda_1)R(\lambda_2)\frac{\mathbf{X}_1^\top\boldsymbol\epsilon_1}{n}\right)\left(\alpha_0^\top\hat\Sigma_1 R(\lambda_1)\beta_0\right)\right]$$

$$- \mathbb{E}\left[\frac{\boldsymbol{\mu}_1^\top \mathbf{X}_1}{n} R(\lambda_1) R(\lambda_2) \frac{\mathbf{X}_1^\top \boldsymbol{\epsilon}_1}{n}\right] \mathbb{E}\left[\alpha_0^\top \hat{\Sigma}_1 R(\lambda_1) \beta_0\right]$$

$$= \frac{1}{n} \mathbb{E}\left[\text{tr}\left(R(\lambda_1) R(\lambda_2) \hat{\Sigma}_1\right) \alpha_0^\top \hat{\Sigma}_1 R(\lambda_1) \beta_0\right] - \frac{1}{n} \mathbb{E}\left[\text{tr}\left(R(\lambda_1) R(\lambda_2) \hat{\Sigma}_1\right)\right] \mathbb{E}\left[\alpha_0^\top \hat{\Sigma}_1 R(\lambda_1) \beta_0\right]$$

$$= c\mathbb{E}\left[\left(\int \frac{x\, d\hat{\pi}_{n,1}(x)}{(x+\lambda_1)(x+\lambda_2)}\right) \alpha_0^\top \hat{\Sigma}_1 R(\lambda_1) \beta_0\right] - c\mathbb{E}\left[\int \frac{x\, d\hat{\pi}_{n,1}(x)}{(x+\lambda_1)(x+\lambda_2)}\right] \mathbb{E}\left[\alpha_0^\top \hat{\Sigma}_1 R(\lambda_1) \beta_0\right]$$

$$= c\,(\alpha_0^\top \beta_0)\left\{\mathbb{E}\left[\int \frac{x\, d\hat{\pi}_{n,1}(x)}{(x+\lambda_1)(x+\lambda_2)} \int \frac{x\, d\hat{\pi}_{n,1}(x)}{(x+\lambda_1)}\right] - \mathbb{E}\left[\int \frac{x\, d\hat{\pi}_{n,1}(x)}{(x+\lambda_1)(x+\lambda_2)}\right] \mathbb{E}\left[\int \frac{x\, d\hat{\pi}_{n,1}(x)}{(x+\lambda_1)}\right]\right\}$$

$$\longrightarrow 0\,.$$

The last conclusion follows from Theorem 1.1 of [28]. Therefore, we conclude:

$$\lim_{n\uparrow\infty} n \times \text{cov}\left(\hat{\alpha}(\lambda_1)^\top \hat{\beta}(\lambda_2),\ \hat{\alpha}(\lambda_1)^\top \beta_0\right)$$

$$= \frac{1}{c}(u^2 v^2 + \varrho^2)\,\text{cov}_{X\sim F_{\text{MP}}}\left(\frac{X^2}{(X+\lambda_1)(X+\lambda_2)},\ \frac{X}{X+\lambda_1}\right)$$

$$+ \theta_0 \varrho \int \frac{x^2\, dF_{\text{MP}}(x)}{(x+\lambda_1)^2(x+\lambda_2)} + v^2 \int \frac{x^2\, dF_{\text{MP}}(x)}{(x+\lambda_1)^2(x+\lambda_2)}\,.$$

Now, we analyze the second covariance term:

$$\text{cov}\left(\hat{\alpha}(\lambda_1)^\top \hat{\beta}(\lambda_2),\ \alpha_0^\top \hat{\beta}(\lambda_2)\right)$$

$$= \text{cov}\left(\alpha_0^\top \hat{\Sigma}_1 R(\lambda_1) R(\lambda_2) \hat{\Sigma}_1 \beta_0,\ \alpha_0^\top R(\lambda_2) \hat{\Sigma}_1 \beta_0\right) + \text{cov}\left(\alpha_0^\top \hat{\Sigma}_1 R(\lambda_1) R(\lambda_2) \frac{\mathbf{X}_1^\top \boldsymbol{\mu}_1}{n},\ \alpha_0^\top R(\lambda_2) \frac{\mathbf{X}_1^\top \boldsymbol{\mu}_1}{n}\right)$$

$$+ \text{cov}\left(\alpha_0^\top R(\lambda_2) \frac{\mathbf{X}_1^\top \boldsymbol{\mu}_1}{n},\ \beta_0^\top \hat{\Sigma}_1 R(\lambda_1) R(\lambda_2) \frac{\mathbf{X}_1^\top \boldsymbol{\epsilon}_1}{n}\right) + \text{cov}\left(\frac{\boldsymbol{\mu}_1^\top \mathbf{X}_1}{n} R(\lambda_1) R(\lambda_2) \frac{\mathbf{X}_1^\top \boldsymbol{\epsilon}_1}{n},\ \alpha_0^\top R(\lambda_2) \hat{\Sigma}_1 \beta_0\right)\,.$$

Same calculation as for the previous term yields:

$$\lim_{n\uparrow\infty} n \times \text{cov}\left(\hat{\alpha}(\lambda_1)^\top \hat{\beta}(\lambda_2),\ \alpha_0^\top \hat{\beta}(\lambda_2)\right)$$

$$= \frac{1}{c}(u^2 v^2 + \varrho^2)\,\text{cov}_{X\sim F_{\text{MP}}}\left(\frac{X^2}{(X+\lambda_1)(X+\lambda_2)},\ \frac{X}{X+\lambda_2}\right)$$

$$+ \theta_0 \varrho \int \frac{x^2\, dF_{\text{MP}}(x)}{(x+\lambda_1)(x+\lambda_2)^2} + u^2 \int \frac{x^2\, dF_{\text{MP}}(x)}{(x+\lambda_1)(x+\lambda_2)^2}\,.$$

Finally, for the last covariance term:

$$\text{cov}(\hat{\alpha}(\lambda_1)^\top \beta_0, \alpha_0^\top \hat{\beta}(\lambda_2))$$

$$= \text{cov}\left(\alpha_0^\top \hat{\Sigma}_1 R(\lambda_1) \beta_0,\ \alpha_0^\top R(\lambda_2) \hat{\Sigma}_1 \beta_0\right) + \text{cov}\left(\alpha_0^\top R(\lambda_2) \frac{\mathbf{X}_1^\top \boldsymbol{\mu}_1}{n},\ \beta_0^\top R(\lambda_1) \frac{\mathbf{X}_1^\top \boldsymbol{\epsilon}_1}{n}\right)$$

Same calculation as before yields:

$$\lim_{n\uparrow\infty} n \times \text{cov}(\hat{\alpha}(\lambda_1)^\top \beta_0, \alpha_0^\top \hat{\beta}(\lambda_2))$$

$$= \frac{1}{c}(u^2 v^2 + \varrho^2)\,\text{cov}_{X\sim F_{\text{MP}}}\left(\frac{X}{X+\lambda_1},\ \frac{X}{X+\lambda_2}\right) + \varrho \int \frac{x\, dF_{\text{MP}}(x)}{(x+\lambda_1)(x+\lambda_2)}\,.$$

Combining the limits, we obtain:

$$\boxed{\begin{aligned}
&\lim_{n\uparrow\infty} n \times \text{var}\left(\mathbb{E}\left[\hat{\theta}_{2\text{sp}}^{\text{DR,db}} \mid \mathcal{D}_1\right]\right) \\
&= H_1^2(\boldsymbol{\lambda})(1 - H_2(\boldsymbol{\lambda}))^2 \left[(u^2 + v^2 + 2\varrho + c(1+\rho^2)) \int \frac{x^2}{(x+\lambda_1)^2(x+\lambda_2)^2}\, dF_{\text{MP}}(x)\right. \\
&\qquad \left. + \frac{1}{c}\left[u^2 v^2 + \varrho^2\right] \text{var}_{X\sim F_{\text{MP}}}\left(\frac{X^2}{(X+\lambda_1)(X+\lambda_2)}\right)\right]
\end{aligned}}$$

$$
\begin{aligned}
&+ H_1^2(\boldsymbol{\lambda})\left(\frac{1}{c}(\varrho^2 + u^2 v^2)\, \mathsf{var}_{X \sim F_{\mathrm{MP}}}\left(\frac{X}{X + \lambda_1}\right) + v^2 \int \frac{x\, dF_{\mathrm{MP}}(x)}{(x + \lambda_1)^2}\right) \\
&+ H_1^2(\boldsymbol{\lambda})\left(\frac{1}{c}(\varrho^2 + u^2 v^2)\mathsf{var}_{X \sim F_{\mathrm{MP}}}\left(\frac{X}{X + \lambda_2}\right) + u^2 \int \frac{x\, dF_{\mathrm{MP}}(x)}{(x + \lambda_2)^2}\right) \\
&- 2H_1^2(\boldsymbol{\lambda})(1 - H_2(\boldsymbol{\lambda}))\left[\frac{1}{c}(u^2 v^2 + \varrho^2)\, \mathsf{cov}_{X \sim F_{\mathrm{MP}}}\left(\frac{X^2}{(X + \lambda_1)(X + \lambda_2)}, \frac{X}{X + \lambda_1}\right)\right. \\
&\qquad\qquad \left. + \theta_0 \varrho \int \frac{x^2\, dF_{\mathrm{MP}}(x)}{(x + \lambda_1)^2(x + \lambda_2)} + v^2 \int \frac{x^2\, dF_{\mathrm{MP}}(x)}{(x + \lambda_1)^2(x + \lambda_2)}\right] \\
&- 2H_1^2(\boldsymbol{\lambda})(1 - H_2(\boldsymbol{\lambda}))\left[\frac{1}{c}(u^2 v^2 + \varrho^2)\, \mathsf{cov}_{X \sim F_{\mathrm{MP}}}\left(\frac{X^2}{(X + \lambda_1)(X + \lambda_2)}, \frac{X}{X + \lambda_2}\right)\right. \\
&\qquad\qquad \left. + \theta_0 \varrho \int \frac{x^2\, dF_{\mathrm{MP}}(x)}{(x + \lambda_1)(x + \lambda_2)^2} + u^2 \int \frac{x^2\, dF_{\mathrm{MP}}(x)}{(x + \lambda_1)(x + \lambda_2)^2}\right] \\
&+ 2H_1^2\left[\frac{1}{c}(u^2 v^2 + \varrho^2)\, \mathsf{cov}_{X \sim F_{\mathrm{MP}}}\left(\frac{X}{X + \lambda_1}, \frac{X}{X + \lambda_2}\right) + \varrho \int \frac{x\, dF_{\mathrm{MP}}(x)}{(x + \lambda_1)(x + \lambda_2)}\right]
\end{aligned}
\tag{D.18}
$$

Next, we derive the limit of the expectation of the conditional variance of $\hat{\theta}_{2\mathrm{sp}}^{\mathrm{DR,db}}$. The conditional variance given $\mathcal{D}_1$ is:

$$
\begin{aligned}
&\mathsf{var}\left(\hat{\theta}_{2\mathrm{sp}}^{\mathrm{DR,db}} \mid \mathcal{D}_1\right) \\
&= \frac{H_1^2(\boldsymbol{\lambda})}{n}\mathsf{var}\left((Y - X^\top \hat{\beta}(\lambda_2))(A - X^\top \hat{\alpha}(\lambda_1)) \mid \mathcal{D}_1\right) \\
&= \frac{H_1^2(\boldsymbol{\lambda})}{n}\left(\mathsf{var}\left((\alpha_0 - \hat{\alpha}(\lambda_1))^\top XX^\top(\beta_0 - \hat{\beta}(\lambda_2))\right) + \mathsf{var}\left(X^\top(\beta_0 - \hat{\beta}(\lambda_2))\epsilon\right)\right. \\
&\quad + \mathsf{var}\left(X^\top(\alpha_0 - \hat{\alpha}(\lambda_1))\mu\right) + \mathsf{var}(\mu\epsilon) \\
&\quad \left. + 2\mathsf{cov}\left(X^\top(\alpha_0 - \hat{\alpha}(\lambda_1))\mu,\ X^\top(\beta_0 - \hat{\beta}(\lambda_2))\epsilon\right)\right) \\
&= \frac{H_1^2(\boldsymbol{\lambda})}{n}\left(\|\alpha_0 - \hat{\alpha}(\lambda_1)\|_2^2\|\beta_0 - \hat{\beta}(\lambda_2)\|_2^2 + ((\alpha_0 - \hat{\alpha}(\lambda_1))^\top(\beta_0 - \hat{\beta}(\lambda_2)))^2 + \|\alpha_0 - \hat{\alpha}(\lambda_1)\|_2^2\right. \\
&\quad \left. + \|\beta_0 - \hat{\beta}(\lambda_2)\|_2^2 + 2\rho(\alpha_0 - \hat{\alpha}(\lambda_1))^\top(\beta_0 - \hat{\beta}(\lambda_2)) + 1 + \rho^2\right) \\
&\triangleq \frac{H_1^2(\boldsymbol{\lambda})}{n}\left(T_1 + T_2 + T_3 + T_4 + 2\rho\, T_5 + 1 + \rho^2\right).
\end{aligned}
$$

In the second equality, rest of the covariance terms will be 0 because of independence between $X$ and $(\epsilon, \nu)$ and the fact that $\mathbb{E}[\mu\epsilon^2] = \mathbb{E}[\mu^2\epsilon] = 0$. Now we take the expectation of $T_i$ for $1 \le i \le 5$. For notational simplicity, define $R(\lambda) = (\hat{\Sigma}_1 + \lambda\mathbf{I})^{-1}$. We start with the expectation of $T_1$:

$$
\begin{aligned}
\mathbb{E}[T_1] &= \mathbb{E}\left[\left\|-\lambda_1 R(\lambda_1)\alpha_0 + R(\lambda_1)\frac{\mathbf{X}_1^\top \boldsymbol{\epsilon}_1}{n}\right\|_2^2 \left\|-\lambda_2 R(\lambda_2)\beta_0 + R(\lambda_2)\frac{\mathbf{X}_1^\top \boldsymbol{\mu}_1}{n}\right\|_2^2\right] \\
&= \mathbb{E}\left[\left\{\lambda_1^2\|R(\lambda_1)\alpha_0\|_2^2 + \left\|R(\lambda_1)\frac{\mathbf{X}_1^\top \boldsymbol{\epsilon}_1}{n}\right\|_2^2 - 2\lambda_1 \alpha_0^\top R^2(\lambda_1)\frac{\mathbf{X}_1^\top \boldsymbol{\epsilon}_1}{n}\right\}\right. \\
&\qquad \left. \times \left\{\lambda_2^2\|R(\lambda_2)\beta_0\|_2^2 + \left\|R(\lambda_2)\frac{\mathbf{X}_1^\top \boldsymbol{\mu}_1}{n}\right\|_2^2 - 2\lambda_2 \beta_0^\top R^2(\lambda_2)\frac{\mathbf{X}_1^\top \boldsymbol{\mu}_1}{n}\right\}\right] \\
&= \mathbb{E}\left[\lambda_1^2 \lambda_2^2\|R(\lambda_1)\alpha_0\|_2^2\|R(\lambda_2)\beta_0\|_2^2 + \lambda_1^2\|R(\lambda_1)\alpha_0\|_2^2\left\|R(\lambda_2)\frac{\mathbf{X}_1^\top \boldsymbol{\mu}_1}{n}\right\|_2^2\right. \\
&\qquad \left. + \lambda_2^2\|R(\lambda_2)\beta_0\|_2^2\left\|R(\lambda_1)\frac{\mathbf{X}_1^\top \boldsymbol{\epsilon}_1}{n}\right\|_2^2 + \left\|R(\lambda_1)\frac{\mathbf{X}_1^\top \boldsymbol{\epsilon}_1}{n}\right\|_2^2\left\|R(\lambda_2)\frac{\mathbf{X}_1^\top \boldsymbol{\mu}_1}{n}\right\|_2^2\right.
\end{aligned}
$$

$$+ 4\lambda_1\lambda_2\alpha_0^\top R^2(\lambda_1)\frac{\mathbf{X}_1^\top \boldsymbol{\epsilon}_1}{n}\frac{\boldsymbol{\mu}_1^\top \mathbf{X}_1}{n} R^2(\lambda_2)\beta_0\Big] \quad \text{[Rest have mean 0]}$$

$$\triangleq \lambda_1^2\lambda_2^2 T_{11} + \lambda_1^2 T_{12} + \lambda_2^2 T_{13} + T_{14} + 4\lambda_1\lambda_2 T_{15}\,.$$

Now, we find the limit of each of the above summands:

$$T_{11} = \mathbb{E}\left[\|R(\lambda_1)\alpha_0\|_2^2\|R(\lambda_2)\beta_0\|_2^2\right]$$

$$\longrightarrow u^2 v^2 \int \frac{1}{(x+\lambda_1)^2}\, dF_{\mathrm{MP}}(x) \int \frac{1}{(x+\lambda_2)^2}\, dF_{\mathrm{MP}}(x)\,.$$

$$T_{12} = \mathbb{E}\left[\|R(\lambda_1)\alpha_0\|_2^2 \left\|R(\lambda_2)\frac{\mathbf{X}_1^\top \boldsymbol{\mu}_1}{n}\right\|_2^2\right] = \frac{1}{n}\mathbb{E}\left[\|R(\lambda_1)\alpha_0\|_2^2 \mathrm{tr}\left(R^2(\lambda_2)\hat{\Sigma}_1\right)\right]$$

$$= c\,\mathbb{E}\left[\alpha_0^\top R^2(\lambda_1)\alpha_0\left(\frac{1}{p}\sum_{j=1}^p\frac{\hat{\lambda}_j}{(\hat{\lambda}_j+\lambda_2)^2}\right)\right]$$

$$= c\,\|\alpha_0\|_2^2\,\mathbb{E}\left[\left(\frac{1}{p}\sum_{J=1}^p\frac{1}{(\hat{\lambda}_j+\lambda_1)^2}\right)\left(\frac{1}{p}\sum_{j=1}^p\frac{\hat{\lambda}_j}{(\hat{\lambda}_j+\lambda_2)^2}\right)\right]$$

$$\longrightarrow cu^2 \int \frac{dF_{\mathrm{MP}}(x)}{(x+\lambda_1)^2} \int \frac{x\,dF_{\mathrm{MP}}(x)}{(x+\lambda_2)^2}\,.$$

A similar calculation yields:

$$T_{13} \longrightarrow cv^2 \int \frac{1}{(x+\lambda_2)^2}\, dF_{\mathrm{MP}}(x) \int \frac{x}{(x+\lambda_1)^2}\, dF_{\mathrm{MP}}(x)\,.$$

For $T_{14}$, we have:

$$T_{14} = \mathbb{E}\left[\left\|R(\lambda_1)\frac{\mathbf{X}_1^\top \boldsymbol{\epsilon}_1}{n}\right\|_2^2 \left\|R(\lambda_2)\frac{\mathbf{X}_1^\top \boldsymbol{\mu}_1}{n}\right\|_2^2\right]$$

$$= \mathbb{E}\left[\left(\boldsymbol{\epsilon}_1^\top\frac{\mathbf{X}_1 R^2(\lambda_1)\mathbf{X}_1^\top}{n^2}\boldsymbol{\epsilon}_1\right)\left(\boldsymbol{\mu}_1^\top\frac{\mathbf{X}_1 R^2(\lambda_2)\mathbf{X}_1^\top}{n^2}\boldsymbol{\mu}_1\right)\right]$$

$$= \left(\frac{1}{n}\mathrm{tr}\left(R^2(\lambda_1)\hat{\Sigma}_1\right)\right)\left(\frac{1}{n}\mathrm{tr}\left(R^2(\lambda_2)\hat{\Sigma}_1\right)\right) + \frac{1}{n^2}\mathrm{tr}\left(R^2(\lambda_1)\hat{\Sigma}_1 R^2(\lambda_2)\hat{\Sigma}_1\right)$$

$$\longrightarrow c^2 \int \frac{x}{(x+\lambda_1)^2}\, dF_{\mathrm{MP}}(x) \int \frac{x}{(x+\lambda_2)^2}\, dF_{\mathrm{MP}}(x)\,.$$

Finally for $T_{15}$:

$$T_{15} = \mathbb{E}\left[\alpha_0^\top R^2(\lambda_1)\frac{\mathbf{X}_1^\top \boldsymbol{\epsilon}_1}{n}\frac{\boldsymbol{\mu}_1^\top \mathbf{X}_1}{n} R^2(\lambda_2)\beta_0\right] = \theta_0\,\frac{1}{n}\mathbb{E}\left[\alpha_0^\top R^2(\lambda_1)\hat{\Sigma}R^2(\lambda_2)\beta_0\right]$$

$$= \theta_0\,\frac{\alpha_0^\top\beta_0}{h}\mathbb{E}\left[\frac{1}{p}\sum_{j=1}^p\frac{\hat{\lambda}_j}{(\hat{\lambda}_j+\lambda_1)^2(\hat{\lambda}_j+\lambda_2)^2}\right]$$

$$\longrightarrow 0\,.$$

Combining the limits we conclude:

$$\mathbb{E}[T_1] \longrightarrow u^2 v^2 \int \frac{\lambda_1^2\,dF_{\mathrm{MP}}(x)}{(x+\lambda_1)^2} \int \frac{\lambda_2^2\,dF_{\mathrm{MP}}(x)}{(x+\lambda_2)^2} + cu^2 \int \frac{\lambda_1^2\,dF_{\mathrm{MP}}(x)}{(x+\lambda_1)^2} \int \frac{x\,dF_{\mathrm{MP}}(x)}{(x+\lambda_2)^2}$$

$$+ cv^2 \int \frac{\lambda_2^2\,dF_{\mathrm{MP}}(x)}{(x+\lambda_2)^2} \int \frac{x\,dF_{\mathrm{MP}}(x)}{(x+\lambda_1)^2} + c^2 \int \frac{x\,dF_{\mathrm{MP}}(x)}{(x+\lambda_1)^2} \int \frac{x\,dF_{\mathrm{MP}}(x)}{(x+\lambda_2)^2}$$

Now we focus on $T_2$:

$$\mathbb{E}[T_2] = \mathbb{E}\left[((\alpha_0 - \hat{\alpha}(\lambda_1))^\top(\beta_0 - \hat{\beta}(\lambda_2)))^2]\right]$$

$$= \mathbb{E}\left[\left[\lambda_1\lambda_2\alpha_0^\top R(\lambda_1)R(\lambda_2)\beta_0 - \lambda_1\alpha_0^\top R(\lambda_1)R(\lambda_2)\frac{\mathbf{X}_1^\top\boldsymbol{\mu}_1}{n}\right.\right.$$

$$\left.\left. + -\lambda_2\beta_0^\top R(\lambda_1)R(\lambda_2)\frac{\mathbf{X}_1^\top\boldsymbol{\epsilon}_1}{n} + \frac{\boldsymbol{\epsilon}^\top\mathbf{X}_1}{n}R(\lambda_1)R(\lambda_2)\frac{\mathbf{X}_1^\top\boldsymbol{\mu}_1}{n}\right]^2\right]$$

$$= \mathbb{E}\left[\lambda_1^2\lambda_2^2(\alpha_0^\top R(\lambda_1)R(\lambda_2)\beta_0)^2 + \lambda_1^2\left(\alpha_0^\top R(\lambda_1)R(\lambda_2)\frac{\mathbf{X}_1^\top\boldsymbol{\mu}_1}{n}\right)^2\right.$$

$$+ \lambda_2^2\left(\beta_0^\top R(\lambda_1)R(\lambda_2)\frac{\mathbf{X}_1^\top\boldsymbol{\epsilon}_1}{n}\right)^2 + \left(\frac{\boldsymbol{\epsilon}^\top\mathbf{X}_1}{n}R(\lambda_1)R(\lambda_2)\frac{\mathbf{X}_1^\top\boldsymbol{\mu}_1}{n}\right)^2$$

$$+ 2\lambda_1\lambda_2\alpha_0^\top R(\lambda_1)R(\lambda_2)\beta_0\frac{\boldsymbol{\epsilon}^\top\mathbf{X}_1}{n}R(\lambda_1)R(\lambda_2)\frac{\mathbf{X}_1^\top\boldsymbol{\mu}_1}{n}$$

$$\left. + 2\lambda_1\lambda_2\left(\alpha_0^\top R(\lambda_1)R(\lambda_2)\frac{\mathbf{X}_1^\top\boldsymbol{\mu}_1}{n}\right)\left(\beta_0^\top R(\lambda_1)R(\lambda_2)\frac{\mathbf{X}_1^\top\boldsymbol{\epsilon}_1}{n}\right)\right]$$

$$\triangleq \lambda_1^2\lambda_2^2 T_{21} + \lambda_1^2 T_{22} + \lambda_2^2 T_{23} + T_{24} + 2\lambda_1\lambda_2(T_{25} + T_{26}).$$

We start with $T_{21}$:

$$\mathbb{E}[T_{21}]$$

$$= \mathbb{E}[(\alpha_0^\top R(\lambda_1)R(\lambda_2)\beta_0)^2]$$

$$= \mathbb{E}\left[\left(\sum_{j=1}^p \frac{(\hat{v}_j^\top\alpha_0)(\hat{v}_j^\top\beta_0)}{(\hat{\lambda}_j + \lambda_1)(\hat{\lambda}_j + \lambda_2)}\right)^2\right]$$

$$= \frac{1}{(p+2)}\left[\|\alpha_0\|_2^2\|\beta_0\|_2^2 + 2(\alpha_0^\top\beta_0)^2\right]\mathbb{E}\left[\int f^2(x)\,d\hat{\pi}_{n,1}(x)\right]$$

$$+ \frac{p}{p-1}\left[\frac{p}{p+2}(\alpha_0^\top\beta_0)^2 - \frac{\|\alpha_0\|_2^2\|\beta_0\|_2^2}{(p+2)}\right]\mathbb{E}\left[\left(\int f(x)\,d\hat{\pi}_{n,1}(x)\right)^2\right]$$

$$- \frac{1}{p-1}\left[\frac{p}{p+2}(\alpha_0^\top\beta_0)^2 - \frac{\|\alpha_0\|_2^2\|\beta_0\|_2^2}{(p+2)}\right]\mathbb{E}\left[\int f^2(x)\,d\hat{\pi}_{n,1}(x)\right] \quad \text{(Equation (E.2))}$$

where $f(x) = 1/(x+\lambda_1)(x+\lambda_2)$. Therefore,

$$\lim_{n\uparrow\infty}\mathbb{E}[T_{21}] = \varrho^2\left(\int \frac{dF_{\text{MP}}(x)}{(x+\lambda_1)(x+\lambda_2)}\right)^2$$

For $T_{22}$:

$$\mathbb{E}[T_{22}] = \mathbb{E}\left[\left(\alpha_0^\top R(\lambda_1)R(\lambda_2)\frac{\mathbf{X}_1^\top\boldsymbol{\mu}_1}{n}\right)^2\right] = \frac{1}{n}\mathbb{E}\left[\alpha_0^\top R(\lambda_1)R(\lambda_2)\hat{\Sigma}_1 R(\lambda_1)R(\lambda_2)\alpha_0\right]$$

$$= c\,\|\alpha_0\|_2^2\mathbb{E}\left[\int \frac{x\,d\hat{\pi}_{n,1}(x)}{(x+\lambda_1)^2(x+\lambda_2)^2}\right]$$

$$\longrightarrow c\,u^2\int \frac{x\,dF_{\text{MP}}(x)}{(x+\lambda_1)^2(x+\lambda_2)^2}.$$

Similarly, for $T_{23}$:

$$\mathbb{E}[T_{23}] \longrightarrow c\,v^2\int \frac{x\,dF_{\text{MP}}(x)}{(x+\lambda_1)^2(x+\lambda_2)^2}.$$

For $T_{24}$, we have:

$$\mathbb{E}[T_{24}] = \mathbb{E}\left[\left(\frac{\boldsymbol{\epsilon}^\top\mathbf{X}_1}{n}R(\lambda_1)R(\lambda_2)\frac{\mathbf{X}_1^\top\boldsymbol{\mu}_1}{n}\right)^2\right]$$

$$= \theta_0^2 \mathbb{E}\left[\left(\frac{1}{n}\mathsf{tr}\left(R(\lambda_1)R(\lambda_2)\hat{\Sigma}_1\right)\right)^2\right]$$

$$+ (1 + \theta_0^2)\mathbb{E}\left[\frac{1}{n^2}\mathsf{tr}\left(R(\lambda_1)R(\lambda_2)\hat{\Sigma}_1 R(\lambda_1)R(\lambda_2)\hat{\Sigma}_1\right)\right]$$

$$= \theta_0^2 \mathbb{E}\left[\left(c\int \frac{x\,d\hat{\pi}_{n,1}(x)}{(x+\lambda_1)(x+\lambda_2)}\right)^2\right] + \frac{(1+\theta_0^2)}{n}\, c\, \mathbb{E}\left[\int \frac{x^2\,d\hat{\pi}_{n,1}(x)}{(x+\lambda_1)^2(x+\lambda_2)^2}\right]$$

$$\longrightarrow c^2\theta_0^2 \left(\int \frac{x\,d\hat{F}_{\mathrm{MP}}(x)}{(x+\lambda_1)(x+\lambda_2)}\right)^2.$$

Now, for $T_{25}$:

$$\mathbb{E}[T_{25}] = \mathbb{E}\left[\left(\alpha_0^\top R(\lambda_1)R(\lambda_2)\frac{\mathbf{X}_1^\top \boldsymbol{\mu}_1}{n}\right)\left(\beta_0^\top R(\lambda_1)R(\lambda_2)\frac{\mathbf{X}_1^\top \boldsymbol{\epsilon}_1}{n}\right)\right]$$

$$= \mathbb{E}\left[\frac{\boldsymbol{\epsilon}_1^\top \mathbf{X}_1}{n}R(\lambda_1)R(\lambda_2)\beta_0\alpha_0^\top R(\lambda_1)R(\lambda_2)\frac{\mathbf{X}_1^\top \boldsymbol{\mu}_1}{n}\right]$$

$$= \frac{1}{n}\mathbb{E}\left[\mathsf{tr}\left(\beta_0\alpha_0^\top R(\lambda_1)R(\lambda_2)\hat{\Sigma}_1 R(\lambda_1)R(\lambda_2)\right)\right]$$

$$= \frac{1}{n}\mathbb{E}\left[\alpha_0^\top R(\lambda_1)R(\lambda_2)\hat{\Sigma}_1 R(\lambda_1)R(\lambda_2)\beta_0\right]$$

$$= c\,(\alpha_0^\top \beta_0)\,\mathbb{E}\left[\int \frac{x\,d\hat{\pi}_{n,1}(x)}{(x+\lambda_1)^2(x+\lambda_2)^2}\right]$$

$$\longrightarrow c\,\varrho\int \frac{x\,dF_{\mathrm{MP}}(x)}{(x+\lambda_1)^2(x+\lambda_2)^2}$$

Finally, for $T_{26}$:

$$\mathbb{E}[T_{26}] = \mathbb{E}\left[\left(\alpha_0^\top R(\lambda_1)R(\lambda_2)\frac{\mathbf{X}_1^\top \boldsymbol{\mu}_1}{n}\right)\left(\beta_0^\top R(\lambda_1)R(\lambda_2)\frac{\mathbf{X}_1^\top \boldsymbol{\epsilon}_1}{n}\right)\right]$$

$$= \mathbb{E}\left[\frac{\boldsymbol{\epsilon}_1^\top \mathbf{X}_1}{n}R(\lambda_1)R(\lambda_2)\beta_0\alpha_0^\top R(\lambda_1)R(\lambda_2)\frac{\mathbf{X}_1^\top \boldsymbol{\mu}_1}{n}\right]$$

$$\longrightarrow c\,\varrho\int \frac{x\,dF_{\mathrm{MP}}(x)}{(x+\lambda_1)^2(x+\lambda_2)^2}.$$

Combining all the terms, we conclude:

$$\lim_{n\uparrow\infty}\mathbb{E}[T_2] = \varrho^2\left(\int \frac{\lambda_1^2\lambda_2^2\,dF_{\mathrm{MP}}(x)}{(x+\lambda_1)(x+\lambda_2)}\right)^2 + \int \frac{(cu^2\lambda_1^2 + cv^2\lambda_2^2)\,x\,dF_{\mathrm{MP}}(x)}{(x+\lambda_1)^2(x+\lambda_2)^2} \qquad (\text{D.19})$$

$$+ c^2\theta_0^2\left(\int \frac{x\,d\hat{F}_{\mathrm{MP}}(x)}{(x+\lambda_1)(x+\lambda_2)}\right)^2 + \int \frac{4\lambda_1\lambda_2 c\varrho\,x\,dF_{\mathrm{MP}}(x)}{(x+\lambda_1)^2(x+\lambda_2)^2} \qquad (\text{D.20})$$

For $T_3$ we have:

$$\mathbb{E}[T_3] = \mathbb{E}\left[\|\alpha_0 - \hat{\alpha}(\lambda_1)\|_2^2\right] = \mathbb{E}\left[\left\|-\lambda_1 R(\lambda_1)\alpha_0 + R(\lambda_1)\frac{\mathbf{X}_1^\top \boldsymbol{\epsilon}_1}{n}\right\|^2\right]$$

$$= \lambda_1^2\mathbb{E}\left[\alpha_0^\top R^2(\lambda_1)\alpha_0\right] + \mathbb{E}\left[\frac{\boldsymbol{\epsilon}^\top \mathbf{X}_1}{n}R^2(\lambda_1)\frac{\mathbf{X}_1^\top \boldsymbol{\epsilon}_1}{n}\right]$$

$$= \lambda_1^2\|\alpha_0\|_2^2\mathbb{E}\left[\int \frac{d\hat{\pi}_{n,1}(x)}{(x+\lambda_1)^2}\right] + \frac{1}{n}\mathsf{tr}\left(R^2(\lambda_1)\hat{\Sigma}_1\right)$$

$$= \lambda_1^2\|\alpha_0\|_2^2\mathbb{E}\left[\int \frac{d\hat{\pi}_{n,1}(x)}{(x+\lambda_1)^2}\right] + c\,\mathbb{E}\left[\int \frac{x\,d\hat{\pi}_{n,1}(x)}{(x+\lambda_1)^2}\right]$$

$$\longrightarrow u^2 \int \frac{\lambda_1^2\, dF_{\mathrm{MP}}(x)}{(x+\lambda_1)^2} + c \int \frac{x\, dF_{\mathrm{MP}}(x)}{(x+\lambda_1)^2} \,.$$

A similar calculation yields:

$$\mathbb{E}[T_4] \longrightarrow v^2 \int \frac{\lambda_2^2\, dF_{\mathrm{MP}}(x)}{(x+\lambda_2)^2} + c \int \frac{x\, dF_{\mathrm{MP}}(x)}{(x+\lambda_2)^2} \,.$$

Finally, for $T_5$, we have:

$$\mathbb{E}[T_5] = \mathbb{E}\left[(\alpha_0 - \hat{\alpha}(\lambda_1))^\top (\beta_0 - \hat{\beta}(\lambda_2))\right]$$

$$= \mathbb{E}\left[\left(-\lambda_1 R(\lambda_1)\alpha_0 + R(\lambda_1)\frac{\mathbf{X}_1^\top \boldsymbol{\epsilon}_1}{n}\right)^\top \left(-\lambda_2 R(\lambda_2)\beta_0 + R(\lambda_2)\frac{\mathbf{X}_1^\top \boldsymbol{\mu}_1}{n}\right)\right]$$

$$= \lambda_1\lambda_2 \mathbb{E}\left[\alpha_0^\top R(\lambda_1)R(\lambda_2)\beta_0\right] + \mathbb{E}\left[\frac{\boldsymbol{\epsilon}^\top \mathbf{X}_1}{n} R(\lambda_1)R(\lambda_2)\frac{\mathbf{X}_1^\top \boldsymbol{\mu}_1}{n}\right]$$

$$= \lambda_1\lambda_2 \alpha_0^\top \beta_0 \mathbb{E}\left[\int \frac{d\hat{\pi}_{n,1}(x)}{(x+\lambda_1)(x+\lambda_2)}\right] + \frac{1}{n}\mathrm{tr}\left(R(\lambda_1)R(\lambda_2)\hat{\Sigma}_1\right)$$

$$= \lambda_1\lambda_2 \alpha_0^\top \beta_0 \mathbb{E}\left[\int \frac{d\hat{\pi}_{n,1}(x)}{(x+\lambda_1)(x+\lambda_2)}\right] + c\,\mathbb{E}\left[\int \frac{x\, d\hat{\pi}_{n,1}(x)}{(x+\lambda_1)(x+\lambda_2)}\right]$$

$$\longrightarrow \lambda_1\lambda_2 \varrho \int \frac{d\hat{F}_{\mathrm{MP}}(x)}{(x+\lambda_1)(x+\lambda_2)} + c \int \frac{x\, dF_{\mathrm{MP}}(x)}{(x+\lambda_1)(x+\lambda_2)} \,.$$

Combining the limits of $(T_1, \ldots, T_5)$ we conclude that:

$$\lim_{n\uparrow\infty} n \times \mathbb{E}\left[\mathrm{var}\left(\hat{\theta}_{2\mathrm{sp}}^{\mathrm{DR,db}} \mid \mathcal{D}_1\right)\right]$$

$$= H_1^2(\boldsymbol{\lambda})\left[u^2 v^2 \int \frac{\lambda_1^2\, dF_{\mathrm{MP}}(x)}{(x+\lambda_1)^2}\int \frac{\lambda_2^2\, dF_{\mathrm{MP}}(x)}{(x+\lambda_2)^2} + cu^2 \int \frac{\lambda_1^2\, dF_{\mathrm{MP}}(x)}{(x+\lambda_1)^2}\int \frac{x\, dF_{\mathrm{MP}}(x)}{(x+\lambda_2)^2}\right.$$

$$+ cv^2 \int \frac{\lambda_2^2\, dF_{\mathrm{MP}}(x)}{(x+\lambda_2)^2}\int \frac{x\, dF_{\mathrm{MP}}(x)}{(x+\lambda_1)^2} + c^2 \int \frac{x\, dF_{\mathrm{MP}}(x)}{(x+\lambda_1)^2}\int \frac{x\, dF_{\mathrm{MP}}(x)}{(x+\lambda_2)^2}$$

$$+ \varrho^2 \left(\int \frac{\lambda_1^2\lambda_2^2\, dF_{\mathrm{MP}}(x)}{(x+\lambda_1)(x+\lambda_2)}\right)^2 + \int \frac{(cu^2\lambda_1^2 + cv^2\lambda_2^2)\, x\, dF_{\mathrm{MP}}(x)}{(x+\lambda_1)^2(x+\lambda_2)^2}$$

$$+ c^2\theta_0^2 \left(\int \frac{x\, d\hat{F}_{\mathrm{MP}}(x)}{(x+\lambda_1)(x+\lambda_2)}\right)^2 + \int \frac{4\lambda_1\lambda_2 c\varrho\, x\, dF_{\mathrm{MP}}(x)}{(x+\lambda_1)^2(x+\lambda_2)^2}$$

$$+ u^2 \int \frac{\lambda_1^2\, dF_{\mathrm{MP}}(x)}{(x+\lambda_1)^2} + c \int \frac{x\, dF_{\mathrm{MP}}(x)}{(x+\lambda_1)^2} + v^2 \int \frac{\lambda_2^2\, dF_{\mathrm{MP}}(x)}{(x+\lambda_2)^2} + c \int \frac{x\, dF_{\mathrm{MP}}(x)}{(x+\lambda_2)^2}$$

$$\left.+ 2\theta_0\lambda_1\lambda_2\varrho \int \frac{d\hat{F}_{\mathrm{MP}}(x)}{(x+\lambda_1)(x+\lambda_2)} + c \int \frac{x\, dF_{\mathrm{MP}}(x)}{(x+\lambda_1)(x+\lambda_2)} + 1 + \theta_0^2\right] \,. \tag{D.21}$$

Adding the limits in Equation (D.18) and (D.21), we conclude the proof.

## D.4 Limiting Variance of $\hat{\theta}_{2\mathrm{sp}}^{\mathrm{NR,db}}$

Let us start with the definition of $\hat{\theta}_{2\mathrm{sp}}^{\mathrm{NR,db}}$:

$$\hat{\theta}_{2\mathrm{sp}}^{\mathrm{NR,db}} = H_1(\boldsymbol{\lambda})\left[\frac{1}{n}\sum_{i\in\mathcal{D}_2} Y_i(A_i - X_i^\top \hat{\alpha}(\lambda_1)) - \hat{\alpha}(\lambda_1)^\top \hat{\beta}(\lambda_2)H_2(\boldsymbol{\lambda})\right]$$

where the definition of functions $H_1(\cdot), H_2(\cdot)$ is immediate from Equation (3.6). The term $H_1(\boldsymbol{\lambda})$ will be squared. As for the doubly-robust estimator, we start with the tower property of variance, condition on $\mathcal{D}_1$:

$$\mathrm{var}\left(\hat{\theta}_{2\mathrm{sp}}^{\mathrm{NR,db}}\right) = \mathrm{var}\left(\mathbb{E}\left[\hat{\theta}_{2\mathrm{sp}}^{\mathrm{NR,db}} \mid \mathcal{D}_1\right]\right) + \mathbb{E}\left[\mathrm{var}\left(\hat{\theta}_{2\mathrm{sp}}^{\mathrm{NR,db}} \mid \mathcal{D}_1\right)\right]$$

For the conditional expectation, we have:

$$\mathbb{E}\left[\hat{\theta}_{2\mathrm{sp}}^{\mathrm{NR,db}} \mid \mathcal{D}_1\right] = H_1(\boldsymbol{\lambda})\left[(\alpha_0 - \hat{\alpha}(\lambda_1))^\top \beta_0 + \theta_0 - \hat{\alpha}(\lambda_1)^\top \hat{\beta}(\lambda_2) H_2(\boldsymbol{\lambda})\right]$$

Hence,

$$\mathrm{var}\left(\mathbb{E}\left[\hat{\theta}_{2\mathrm{sp}}^{\mathrm{NR,db}} \mid \mathcal{D}_1\right]\right) = H_1^2(\boldsymbol{\lambda})\mathrm{var}\left(\hat{\alpha}(\lambda_1)^\top \beta_0 + \hat{\alpha}(\lambda_1)^\top \hat{\beta}(\lambda_2) H_2(\boldsymbol{\lambda})\right)$$

$$= H_1^2(\boldsymbol{\lambda})\left[\mathrm{var}(\hat{\alpha}(\lambda_1)^\top \beta_0) + H_2^2(\boldsymbol{\lambda})\mathrm{var}(\hat{\alpha}(\lambda_1)^\top \hat{\beta}(\lambda_2)) + \right.$$

$$\left. 2H_2(\boldsymbol{\lambda})\mathrm{cov}(\hat{\alpha}(\lambda_1)^\top \beta_0, \hat{\alpha}(\lambda_1)^\top \hat{\beta}(\lambda_2))\right]$$

In the proof of the limiting variance of $\hat{\theta}_{2\mathrm{sp}}^{\mathrm{INT,db}}$, we have already established that:

$$\lim_{n\uparrow\infty} n \times \mathrm{var}\left(\hat{\alpha}(\lambda_1)^\top \hat{\beta}(\lambda_2))\right) = (u^2 + v^2 + 2\varrho + c(1+\rho^2))\int \frac{x^2}{(x+\lambda_1)^2(x+\lambda_2)^2}\,dF_{\mathrm{MP}}(x)$$

$$+ \frac{1}{c}\left[u^2 v^2 + \varrho^2\right]\mathrm{var}_{X\sim F_{\mathrm{MP}}}\left(\frac{X^2}{(X+\lambda_1)(X+\lambda_2)}\right) \qquad \text{(D.22)}$$

From the proof of the limiting variance of $\hat{\theta}_{2\mathrm{sp}}^{\mathrm{NR,db}}$, we have:

$$\lim_{n\uparrow\infty} n \times \mathrm{var}(\hat{\alpha}(\lambda_1)^\top \beta_0) = \frac{1}{c}(\varrho^2 + u^2 v^2)\,\mathrm{var}_{X\sim F_{\mathrm{MP}}}\left(\frac{X}{X+\lambda_1}\right) + v^2\int \frac{x\,dF_{\mathrm{MP}}(x)}{(x+\lambda_1)^2}$$

$$\lim_{n\uparrow\infty} n \times \mathrm{cov}(\hat{\alpha}(\lambda_1)^\top \beta_0, \hat{\alpha}(\lambda_1)^\top \hat{\beta}(\lambda_2)) = \frac{1}{c}(u^2 v^2 + \varrho^2)\,\mathrm{cov}_{X\sim F_{\mathrm{MP}}}\left(\frac{X^2}{(X+\lambda_1)(X+\lambda_2)}, \frac{X}{X+\lambda_1}\right)$$

$$+ \theta_0\varrho\int \frac{x^2\,dF_{\mathrm{MP}}(x)}{(x+\lambda_1)^2(x+\lambda_2)} + v^2\int \frac{x^2\,dF_{\mathrm{MP}}(x)}{(x+\lambda_1)^2(x+\lambda_2)}\,.$$

Therefore, we obtain:

$$\lim_{n\uparrow\infty} \mathrm{var}\left(\mathbb{E}\left[\hat{\theta}_{2\mathrm{sp}}^{\mathrm{NR,db}} \mid \mathcal{D}_1\right]\right)$$

$$= H_1^2(\boldsymbol{\lambda})\left[\frac{1}{c}(\varrho^2 + u^2 v^2)\,\mathrm{var}_{X\sim F_{\mathrm{MP}}}\left(\frac{X}{X+\lambda_1}\right) + v^2\int \frac{x\,dF_{\mathrm{MP}}(x)}{(x+\lambda_1)^2}\right.$$

$$+ H_2^2(\boldsymbol{\lambda})\left((u^2 + v^2 + 2\varrho + c(1+\rho^2))\int \frac{x^2}{(x+\lambda_1)^2(x+\lambda_2)^2}\,dF_{\mathrm{MP}}(x)\right.$$

$$+ \frac{1}{c}\left[u^2 v^2 + \varrho^2\right]\mathrm{var}_{X\sim F_{\mathrm{MP}}}\left(\frac{X^2}{(X+\lambda_1)(X+\lambda_2)}\right)\bigg)$$

$$+ 2H_2(\boldsymbol{\lambda})\left(\frac{1}{c}(u^2 v^2 + \varrho^2)\,\mathrm{cov}_{X\sim F_{\mathrm{MP}}}\left(\frac{X^2}{(X+\lambda_1)(X+\lambda_2)}, \frac{X}{X+\lambda_1}\right)\right.$$

$$\left.\left. + \theta_0\varrho\int \frac{x^2\,dF_{\mathrm{MP}}(x)}{(x+\lambda_1)^2(x+\lambda_2)} + v^2\int \frac{x^2\,dF_{\mathrm{MP}}(x)}{(x+\lambda_1)^2(x+\lambda_2)}\right)\right] \qquad \text{(D.23)}$$

Next, we consider the expectation of the conditional variance. The conditional variance of $\hat{\theta}_{2\mathrm{sp}}^{\mathrm{NR,db}}$ given $\mathcal{D}_1$ is as follows:

$$\mathrm{var}\left(\hat{\theta}_{2\mathrm{sp}}^{\mathrm{NR,db}} \mid \mathcal{D}_1\right) = \frac{H_1^2(\boldsymbol{\lambda})}{n}\mathrm{var}\left(Y(A - X^\top \hat{\alpha}(\lambda_1))\right)$$

$$= \frac{H_1^2(\boldsymbol{\lambda})}{n}\mathrm{var}\left((X^\top \beta_0 + \mu)(X^\top(\alpha_0 - \hat{\alpha}(\lambda_1)) + \epsilon)\right)$$

$$= \frac{H_1^2(\boldsymbol{\lambda})}{n}\left[\mathrm{var}(\beta_0^\top XX^\top(\alpha_0 - \hat{\alpha}(\lambda_1))) + \mathrm{var}\left(\epsilon\,X^\top \beta_0\right) + \mathrm{var}\left(\mu\,X^\top(\alpha_0 - \hat{\alpha}(\lambda_1))\right)\right.$$

$$\left. + \mathrm{var}(\mu\epsilon) + 2\mathrm{cov}\left(\epsilon\,X^\top \beta_0, \mu\,X^\top(\alpha_0 - \hat{\alpha}(\lambda_1))\right)\right]$$

$$= \frac{H_1^2(\boldsymbol{\lambda})}{n} \left( \|\alpha_0 - \hat{\alpha}(\lambda_1)\|_2^2 \|\beta_0\|_2^2 + ((\alpha_0 - \hat{\alpha}(\lambda_1))^\top \beta_0)^2 + \|\alpha_0 - \hat{\alpha}(\lambda_1)\|_2^2 \right.$$
$$\left. + \|\beta_0\|_2^2 + 2\rho(\alpha_0 - \hat{\alpha}(\lambda_1))^\top \beta_0 + 1 + \theta_0^2 \right)$$

We have already established:

$$\lim_{n\uparrow\infty} \mathbb{E}[\|\alpha_0 - \hat{\alpha}(\lambda_1)\|_2^2] = u^2 \int \frac{\lambda_1^2\, dF_{\mathrm{MP}}(x)}{(x+\lambda_1)^2} + c \int \frac{x\, dF_{\mathrm{MP}}(x)}{(x+\lambda_1)^2},$$

$$\lim_{n\uparrow\infty} \mathbb{E}[(\alpha_0 - \hat{\alpha}(\lambda_1))^\top \beta_0] = \lambda_1 \varrho \int \frac{dF_{\mathrm{MP}}(x)}{x+\lambda_1},$$

$$\lim_{n\uparrow\infty} \mathbb{E}[((\alpha_0 - \hat{\alpha}(\lambda_1))^\top \beta_0)^2] = \lambda_1^2 \varrho^2 \left( \int \frac{dF_{\mathrm{MP}}(x)}{(x+\lambda_1)} \right)^2.$$

Combining these we have:

$$\lim_{n\uparrow\infty} n \times \mathbb{E}\left[ \mathsf{var}\left( \hat{\theta}_{2\mathrm{sp}}^{\mathrm{NR,db}} \mid \mathcal{D}_1 \right) \right] = H_1^2(\boldsymbol{\lambda}) \left\{ v^2 \left( u^2 \int \frac{\lambda_1^2\, dF_{\mathrm{MP}}(x)}{(x+\lambda_1)^2} + c \int \frac{x\, dF_{\mathrm{MP}}(x)}{(x+\lambda_1)^2} \right) \right.$$
$$+ \lambda_1^2 \varrho^2 \left( \int \frac{dF_{\mathrm{MP}}(x)}{(x+\lambda_1)} \right)^2$$
$$+ v^2 + u^2 \int \frac{\lambda_1^2\, dF_{\mathrm{MP}}(x)}{(x+\lambda_1)^2} + c \int \frac{x\, dF_{\mathrm{MP}}(x)}{(x+\lambda_1)^2}$$
$$\left. + 2\theta_0 \lambda_1 \varrho \int \frac{dF_{\mathrm{MP}}(x)}{x+\lambda_1} + 1 + \theta_0^2 \right\} \tag{D.24}$$

Finally, adding the expressions in Equation (D.23) and (D.24), we conclude the proof.

## D.5 Limiting Variance of $\hat{\theta}_{3\mathrm{sp}}^{\mathrm{DR,db}}$

We now establish the limiting variance of $\hat{\theta}_{3\mathrm{sp}}^{\mathrm{DR,db}}$. Recall the definition of $\hat{\theta}_{3\mathrm{sp}}^{\mathrm{DR,db}}$, as defined in Equation (3.8):

$$\hat{\theta}_{3\mathrm{sp}}^{\mathrm{DR,db}} = \frac{1}{n} \sum_{i \in \mathcal{D}_3} (Y_i - X_i^\top \hat{\beta}(\lambda_2))(A_i - X_i^\top \hat{\alpha}(\lambda_1)) - \hat{\alpha}(\lambda_1)^\top \hat{\beta}(\lambda_2) H_1(\boldsymbol{\lambda}),$$

where the definition of $H_1(\boldsymbol{\lambda})$ is immediate from Equation (3.8). As before, we will use the tower property, but now we will condition on $\mathcal{D}_{1:2} = (\mathcal{D}_1, \mathcal{D}_2)$. The conditional expectation is:

$$\mathbb{E}\left[ \hat{\theta}_{3\mathrm{sp}}^{\mathrm{DR,db}} \big| \mathcal{D}_{1:2} \right] = (\alpha_0 - \hat{\alpha}(\lambda_1))^\top (\beta_0 - \hat{\beta}(\lambda_2)) + \theta_0 - \hat{\alpha}(\lambda_1)^\top \hat{\beta}(\lambda_2) H_1(\boldsymbol{\lambda})$$
$$= \hat{\alpha}(\lambda_1)^\top \hat{\beta}(\lambda_2)(1 - H_1(\boldsymbol{\lambda})) - \alpha_0^\top \hat{\beta}(\lambda_2) - \hat{\alpha}(\lambda_1)^\top \beta_0 + \alpha_0^\top \beta_0 + \theta_0$$

Therefore, we have:

$$\mathsf{var}\left( \mathbb{E}\left[ \hat{\theta}_{3\mathrm{sp}}^{\mathrm{DR,db}} | \mathcal{D}_{1:2} \right] \right) = (1 - H_1(\boldsymbol{\lambda}))^2 \mathsf{var}\left( \hat{\alpha}(\lambda_1)^\top \hat{\beta}(\lambda_2) \right) + \mathsf{var}(\alpha_0^\top \hat{\beta}(\lambda_2)) + \mathsf{var}(\hat{\alpha}(\lambda_1)^\top \beta_0)$$
$$- 2(1 - H_1(\boldsymbol{\lambda})) \mathsf{cov}(\hat{\alpha}(\lambda_1)^\top \hat{\beta}(\lambda_2), \alpha_0^\top \hat{\beta}(\lambda_2))$$
$$- 2(1 - H_1(\boldsymbol{\lambda})) \mathsf{cov}(\hat{\alpha}(\lambda_1)^\top \hat{\beta}(\lambda_2), \hat{\alpha}(\lambda_1)^\top \beta_0).$$

We first use the independence between $\hat{\alpha}(\lambda_1)$ and $\hat{\beta}(\lambda_2)$ to simplify the covariance terms. For the first covariance term,

$$\mathsf{cov}(\hat{\alpha}(\lambda_1)^\top \hat{\beta}(\lambda_2), \alpha_0^\top \hat{\beta}(\lambda_2)) = \mathbb{E}\left[ \hat{\alpha}(\lambda_1)^\top \hat{\beta}(\lambda_2) \alpha_0^\top \hat{\beta}(\lambda_2) \right] - \mathbb{E}[\hat{\alpha}(\lambda_1)^\top \hat{\beta}(\lambda_2)] \mathbb{E}[\alpha_0^\top \hat{\beta}(\lambda_2)]$$
$$= \mathbb{E}\left[ \mathbb{E}[\hat{\alpha}(\lambda_1)]^\top \hat{\beta}(\lambda_2) \alpha_0^\top \hat{\beta}(\lambda_2) \right] - \mathbb{E}[\hat{\alpha}(\lambda_1)]^\top \mathbb{E}[\hat{\beta}(\lambda_2)] \alpha_0^\top \mathbb{E}[\hat{\beta}(\lambda_2)]$$
$$= \mathbb{E}[\alpha_0^\top \mathbb{E}[\hat{\Sigma}_1 R_1(\lambda_1)] \hat{\beta}(\lambda_2) \alpha_0^\top \hat{\beta}(\lambda_2)]$$

$$- \mathbb{E}\left[\alpha_0^\top \mathbb{E}[\hat{\Sigma}_1 R_1(\lambda_1)]\hat{\beta}(\lambda_2)\right]\mathbb{E}[\alpha_0^\top \hat{\beta}(\lambda_2)]$$

$$= \mathbb{E}\left[\int \frac{x\, d\hat{\pi}_{n,1}(x)}{x+\lambda_1}\right]\mathsf{var}\left(\alpha_0^\top \hat{\beta}(\lambda_2)\right) := \mu_{n,1}\mathsf{var}\left(\alpha_0^\top \hat{\beta}(\lambda_2)\right)$$

Similarly,

$$\mathsf{cov}(\hat{\alpha}(\lambda_1)^\top \hat{\beta}(\lambda_2), \hat{\alpha}(\lambda_1)^\top \beta_0) = \mathbb{E}\left[\int \frac{x\, d\hat{\pi}_{n,2}(x)}{x+\lambda_2}\right]\mathsf{var}\left(\hat{\alpha}(\lambda_1)^\top \beta_0\right) := \mu_{n,2}\mathsf{var}\left(\hat{\alpha}(\lambda_1)^\top \beta_0\right).$$

Plugging these in the expression of the conditional variance, we have:

$$\mathsf{var}\left(\mathbb{E}\left[\hat{\theta}_{3\mathrm{sp}}^{\mathrm{DR,db}}|\mathcal{D}_{1:2}\right]\right) = (1-H_1(\boldsymbol{\lambda}))^2\mathsf{var}\left(\hat{\alpha}(\lambda_1)^\top \hat{\beta}(\lambda_2)\right)$$

$$+ (1-2(1-H_1(\boldsymbol{\lambda}))\mu_{n,1})\,\mathsf{var}(\alpha_0^\top \hat{\beta}(\lambda_2))$$

$$+ (1-2(1-H_1(\boldsymbol{\lambda}))\mu_{n,2})\,\mathsf{var}\left(\hat{\alpha}(\lambda_1)^\top \beta_0\right)$$

It is immediate that:

$$\mu_{n,1} \overset{n\uparrow\infty}{\longrightarrow} \int \frac{x\, dF_{\mathrm{MP}}(x)}{x+\lambda_1},$$

$$\mu_{n,2} \overset{n\uparrow\infty}{\longrightarrow} \int \frac{x\, dF_{\mathrm{MP}}(x)}{x+\lambda_2}.$$

We next find the limit of each summand individually, and then add them up. From the proof of the limiting variance of $\hat{\theta}_{3\mathrm{sp}}^{\mathrm{INT,db}}$, we have:

$$\lim_{n\uparrow\infty} n \times \mathsf{var}\left(\hat{\alpha}(\lambda_1)^\top \hat{\beta}(\lambda_2)\right)$$

$$= \left[\frac{1}{c}\mathsf{var}_{X\sim F_{\mathrm{MP}}}\left(\frac{X}{X+\lambda_2}\right)\{\varrho^2 + u^2 v^2\} + u^2 \int \frac{x}{(x+\lambda_2)^2}\, dF_{\mathrm{MP}}(x)\right]\left(\int \frac{x}{x+\lambda_1}\, dF_{\mathrm{MP}}(x)\right)^2$$

$$+ \frac{1}{c}\mathsf{var}_{X\sim F_{\mathrm{MP}}}\left(\frac{X}{X+\lambda_1}\right)\left\{\varrho^2\left(\int \frac{x}{x+\lambda_2}\, dF_{\mathrm{MP}}(x)\right)^2 + \right.$$

$$\left. u^2\left[v^2\left(\int \frac{x^2}{(x+\lambda_2)^2}\, dF_{\mathrm{MP}}(x)\right) + c\left(\int \frac{x}{(x+\lambda_2)^2}\, dF_{\mathrm{MP}}(x)\right)\right]\right\}$$

$$+ \left\{v^2\left(\int \frac{x^2}{(x+\lambda_2)^2}\, dF_{\mathrm{MP}}(x)\right) + c\left(\int \frac{x}{(x+\lambda_2)^2}\, dF_{\mathrm{MP}}(x)\right)\right\}\left(\int \frac{x}{(x+\lambda_1)^2}\, dF_{\mathrm{MP}}(x)\right).$$

From the proof of the limiting variance of $\hat{\theta}_{2\mathrm{sp}}^{\mathrm{DR,db}}$, we have:

$$\lim_{n\uparrow\infty} n \times \mathsf{var}(\hat{\alpha}(\lambda_1)^\top \beta_0) = \frac{1}{c}(\varrho^2 + u^2 v^2)\,\mathsf{var}_{X\sim F_{\mathrm{MP}}}\left(\frac{X}{X+\lambda_1}\right) + v^2 \int \frac{x\, dF_{\mathrm{MP}}(x)}{(x+\lambda_1)^2},$$

$$\lim_{n\uparrow\infty} n \times \mathsf{var}(\alpha_0^\top \hat{\beta}(\lambda_2)) = \frac{1}{c}(\varrho^2 + u^2 v^2)\,\mathsf{var}_{X\sim F_{\mathrm{MP}}}\left(\frac{X}{X+\lambda_2}\right) + u^2 \int \frac{x\, dF_{\mathrm{MP}}(x)}{(x+\lambda_2)^2}.$$

Therefore, combining all the terms, we have:

$$\lim_{n\uparrow\infty}\mathsf{var}\left(\mathbb{E}\left[\hat{\theta}_{3\mathrm{sp}}^{\mathrm{DR,db}}|\mathcal{D}_{1:2}\right]\right)$$

$$= (1-H_1(\boldsymbol{\lambda}))^2\left\{\left[\frac{1}{c}\mathsf{var}_{X\sim F_{\mathrm{MP}}}\left(\frac{X}{X+\lambda_2}\right)\{\varrho^2 + u^2 v^2\}\right.\right.$$

$$\left. + u^2 \int \frac{x}{(x+\lambda_2)^2}\, dF_{\mathrm{MP}}(x)\right]\left(\int \frac{x}{x+\lambda_1}\, dF_{\mathrm{MP}}(x)\right)^2$$

$$+ \frac{1}{c}\mathsf{var}_{X\sim F_{\mathrm{MP}}}\left(\frac{X}{X+\lambda_1}\right)\left\{\varrho^2\left(\int \frac{x}{x+\lambda_2}\, dF_{\mathrm{MP}}(x)\right)^2 + \right.$$

$$\left.\left. u^2\left[v^2\left(\int \frac{x^2}{(x+\lambda_2)^2}\, dF_{\mathrm{MP}}(x)\right) + c\left(\int \frac{x}{(x+\lambda_2)^2}\, dF_{\mathrm{MP}}(x)\right)\right]\right\}\right\}$$

$$\left| \begin{array}{l} + \left\{ v^2 \left( \int \frac{x^2}{(x+\lambda_2)^2}\, dF_{\mathrm{MP}}(x) \right) + c \left( \int \frac{x}{(x+\lambda_2)^2}\, dF_{\mathrm{MP}}(x) \right) \right\} \left( \int \frac{x}{(x+\lambda_1)^2}\, dF_{\mathrm{MP}}(x) \right) \right. \\[2mm] + \left( 1 - 2(1 - H_1(\boldsymbol{\lambda})) \int \frac{x\, dF_{\mathrm{MP}}(x)}{x+\lambda_2} \right) \left\{ \frac{1}{c}(\varrho^2 + u^2 v^2)\, \mathsf{var}_{X \sim F_{\mathrm{MP}}} \left( \frac{X}{X+\lambda_2} \right) + u^2 \int \frac{x\, dF_{\mathrm{MP}}(x)}{(x+\lambda_2)^2} \right\} \\[2mm] \left. + \left( 1 - 2(1 - H_1(\boldsymbol{\lambda})) \int \frac{x\, dF_{\mathrm{MP}}(x)}{x+\lambda_1} \right) \left\{ \frac{1}{c}(\varrho^2 + u^2 v^2)\, \mathsf{var}_{X \sim F_{\mathrm{MP}}} \left( \frac{X}{X+\lambda_2} \right) + u^2 \int \frac{x\, dF_{\mathrm{MP}}(x)}{(x+\lambda_2)^2} \right\} \right|$$

$$\text{(D.25)}$$

Next, we derive the limit of the expectation of the conditional variance. Towards that end,

$$
\begin{aligned}
\mathsf{var}\left( \hat{\theta}_{3\mathrm{sp}}^{\mathrm{DR,db}} | \mathcal{D}_{1:2} \right) &= \frac{1}{n} \mathsf{var}\left( (Y - X^\top \hat{\beta}(\lambda_2))(A - X^\top \hat{\alpha}(\lambda_1)) \right) \\
&= \frac{1}{n} \left\{ \mathsf{var}\left( (\alpha_0 - \hat{\alpha}(\lambda_1))^\top X X^\top (\beta_0 - \hat{\beta}(\lambda_2)) \right) + \mathsf{var}\left( X^\top (\beta_0 - \hat{\beta}(\lambda_2))\epsilon \right) \right. \\
&\quad + \mathsf{var}\left( X^\top (\alpha_0 - \hat{\alpha}(\lambda_1))\mu \right) + \mathsf{var}(\mu\epsilon) \\
&\quad \left. + 2\mathsf{cov}\left( X^\top (\alpha_0 - \hat{\alpha}(\lambda_1))\mu,\ X^\top (\beta_0 - \hat{\beta}(\lambda_2))\epsilon \right) \right\} \\
&= \frac{1}{n} \left( \|\alpha_0 - \hat{\alpha}(\lambda_1)\|_2^2 \|\beta_0 - \hat{\beta}(\lambda_2)\|_2^2 + ((\alpha_0 - \hat{\alpha}(\lambda_1))^\top (\beta_0 - \hat{\beta}(\lambda_2)))^2 \right. \\
&\quad \left. + \|\alpha_0 - \hat{\alpha}(\lambda_1)\|_2^2 + \|\beta_0 - \hat{\beta}(\lambda_2)\|_2^2 + 2\theta_0 (\alpha_0 - \hat{\alpha}(\lambda_1))^\top (\beta_0 - \hat{\beta}(\lambda_2)) + 1 + \theta_0^2 \right)
\end{aligned}
$$

Now, we derive the limit of each summand separately. As already established in the proof of the limiting variance of $\hat{\theta}_{2\mathrm{sp}}^{\mathrm{DR,db}}$, we have:

$$
\lim_{n \uparrow \infty} \mathbb{E}\left[ \|\alpha_0 - \hat{\alpha}(\lambda_1)\|_2^2 \right] = u^2 \int \frac{\lambda_1^2\, dF_{\mathrm{MP}}(x)}{(x+\lambda_1)^2} + c \int \frac{x\, dF_{\mathrm{MP}}(x)}{(x+\lambda_1)^2}
$$

$$
\lim_{n \uparrow \infty} \mathbb{E}\left[ \|\beta_0 - \hat{\beta}(\lambda_2)\|_2^2 \right] = v^2 \int \frac{\lambda_2^2\, dF_{\mathrm{MP}}(x)}{(x+\lambda_2)^2} + c \int \frac{x\, dF_{\mathrm{MP}}(x)}{(x+\lambda_2)^2}.
$$

As $\alpha_0$ and $\beta_0$ are estimated from different subsamples, we have:

$$
\begin{aligned}
\lim_{n \uparrow \infty} \mathbb{E}\left[ \|\alpha_0 - \hat{\alpha}(\lambda_1)\|_2^2 \|\beta_0 - \hat{\beta}(\lambda_2)\|_2^2 \right] &= \left\{ \left( u^2 \int \frac{\lambda_1^2\, dF_{\mathrm{MP}}(x)}{(x+\lambda_1)^2} + c \int \frac{x\, dF_{\mathrm{MP}}(x)}{(x+\lambda_1)^2} \right) \right. \\
&\quad \left. \times \left( v^2 \int \frac{\lambda_2^2\, dF_{\mathrm{MP}}(x)}{(x+\lambda_2)^2} + c \int \frac{x\, dF_{\mathrm{MP}}(x)}{(x+\lambda_2)^2} \right) \right\}
\end{aligned}
$$

$$
\lim_{n \uparrow \infty} \mathbb{E}\left[ (\alpha_0 - \hat{\alpha}(\lambda_1))^\top (\beta_0 - \hat{\beta}(\lambda_2)) \right] = \lambda_1 \lambda_2 \varrho \int \frac{dF_{\mathrm{MP}}(x)}{x+\lambda_1} \int \frac{dF_{\mathrm{MP}}(x)}{x+\lambda_2}
$$

Finally, for the square of the inner product, we have:

$$
\begin{aligned}
&\mathbb{E}\left[ ((\alpha_0 - \hat{\alpha}(\lambda_1))^\top (\beta_0 - \hat{\beta}(\lambda_2)))^2 \right] \\
&= \lambda_2^2 \mathbb{E}\left[ (\alpha_0 - \hat{\alpha}(\lambda_1))^\top R_2(\lambda_2) \beta_0 \beta_0^\top R_2(\lambda_2)(\alpha_0 - \hat{\alpha}(\lambda_1)) \right] \\
&\quad - \lambda_2 \underbrace{\mathbb{E}\left[ (\alpha_0 - \hat{\alpha}(\lambda_1))^\top R_2(\lambda_2) \beta_0 \frac{\boldsymbol{\mu}_2^\top \mathbf{X}_2}{n} R_2(\lambda_2)(\alpha_0 - \hat{\alpha}(\lambda_1)) \right]}_{=0} \\
&\quad - \lambda_2 \underbrace{\mathbb{E}\left[ (\alpha_0 - \hat{\alpha}(\lambda_1))^\top R_2(\lambda_2) \frac{\mathbf{X}_2^\top \mu_2}{n} \beta_0^\top R_2(\lambda_2)(\alpha_0 - \hat{\alpha}(\lambda_1)) \right]}_{=0} \\
&\quad + \mathbb{E}\left[ (\alpha_0 - \hat{\alpha}(\lambda_1))^\top R_2(\lambda_2) \frac{\mathbf{X}_2^\top \mu_2}{n} \frac{\boldsymbol{\mu}_2^\top \mathbf{X}_2}{n} R_2(\lambda_2)(\alpha_0 - \hat{\alpha}(\lambda_1)) \right] \\
&= \lambda_2^2 \mathbb{E}\left[ \left( (\alpha_0 - \hat{\alpha}(\lambda_1))^\top R_2(\lambda_2) \beta_0 \right)^2 \right] \\
&\quad + \frac{1}{n} \mathbb{E}\left[ (\alpha_0 - \hat{\alpha}(\lambda_1))^\top R(\lambda_2) \hat{\Sigma}_2 R(\lambda_2)(\alpha_0 - \hat{\alpha}(\lambda_1)) \right].
\end{aligned}
$$

Let us analyze the above two summands separately. For the second summand, observe that,

$$\frac{1}{n}\mathbb{E}\left[(\alpha_0 - \hat{\alpha}(\lambda_1))^\top R(\lambda_2)\hat{\Sigma}_2 R(\lambda_2)(\alpha_0 - \hat{\alpha}(\lambda_1))\right]$$

$$= \frac{1}{n}\mathbb{E}\left[\|\alpha_0 - \hat{\alpha}(\lambda_1)\|_2^2\right]\mathbb{E}\left[\int \frac{x\, d\hat{\pi}_{n,2}(x)}{(x+\lambda_2)^2}\right]$$

Therefore, this summand is $O(n^{-1})$ and hence asymptotically negligible. For the first summand:

$$\mathbb{E}\left[\left((\alpha_0 - \hat{\alpha}(\lambda_1))^\top R_2(\lambda_2)\beta_0\right)^2\right]$$

$$= \frac{1}{(p+2)}\left[\mathbb{E}[\|\alpha_0 - \hat{\alpha}(\lambda_1)\|_2^2]\|\beta_0\|_2^2 + 2\mathbb{E}[((\alpha_0 - \hat{\alpha}(\lambda_1))^\top \beta_0)^2]\right]\mathbb{E}\left[\int \frac{1}{(x+\lambda_2)^2}\, d\hat{\pi}_n(x)\right]$$

$$+ \frac{p}{p-1}\left[\frac{p}{p+2}\mathbb{E}[((\alpha_0 - \hat{\alpha}(\lambda_1))^\top \beta_0)^2] - \frac{\mathbb{E}[\|\alpha_0 - \hat{\alpha}(\lambda_1)\|_2^2]\|\beta_0\|_2^2}{(p+2)}\right]\mathbb{E}\left[\left(\int \frac{1}{(x+\lambda_2)}\, d\hat{\pi}_n(x)\right)^2\right]$$

$$- \frac{1}{p-1}\left[\frac{p}{p+2}\mathbb{E}[((\alpha_0 - \hat{\alpha}(\lambda_1))^\top \beta_0)^2] - \frac{\mathbb{E}[\|\alpha_0 - \hat{\alpha}(\lambda_1)\|_2^2]\|\beta_0\|_2^2}{(p+2)}\right]\mathbb{E}\left[\int \frac{1}{(x+\lambda_2)^2}\, d\hat{\pi}_n(x)\right]$$

$$= \frac{p^2}{(p-1)(p+2)}\mathbb{E}[((\alpha_0 - \hat{\alpha}(\lambda_1))^\top \beta_0)^2]\mathbb{E}\left[\left(\int \frac{1}{(x+\lambda_2)}\, d\hat{\pi}_n(x)\right)^2\right] + O(p^{-1}).$$

As a consequence,

$$\lim_{n\uparrow\infty}\mathbb{E}\left[\left((\alpha_0 - \hat{\alpha}(\lambda_1))^\top R_2(\lambda_2)\beta_0\right)^2\right] = \left\{\lim_{n\uparrow\infty}\mathbb{E}[((\alpha_0 - \hat{\alpha}(\lambda_1))^\top \beta_0)^2]\right\}\left(\int \frac{dF_{\mathrm{MP}}(x)}{(x+\lambda_2)}\right)^2$$

Now, to find the limit of the right hand side,

$$\mathbb{E}[((\alpha_0 - \hat{\alpha}(\lambda_1))^\top \beta_0)^2]$$

$$= \mathbb{E}\left[\left\{\left(-\lambda_1 R_1(\lambda_1)\alpha_0 + R_1(\lambda_1)\frac{\mathbf{X}_1^\top \epsilon_1}{n}\right)^\top \beta_0\right\}^2\right]$$

$$= \mathbb{E}\left[\left(-\lambda_1 \alpha_0^\top R_1(\lambda_1)\beta_0 + \beta_0^\top R_1(\lambda_1)\frac{\mathbf{X}_1^\top \epsilon_1}{n}\right)^2\right]$$

$$= \lambda_1^2 \mathbb{E}\left[(\alpha_0^\top R_1(\lambda_1)\beta_0)^2\right] + \mathbb{E}\left[\frac{\epsilon_1^\top \mathbf{X}_1}{n}R_1(\lambda_1)\beta_0\beta_0^\top R_1(\lambda_1)\frac{\mathbf{X}_1^\top \epsilon_1}{n}\right]$$

By a similar calculation as before, we have:

$$\mathbb{E}\left[(\alpha_0^\top R_1(\lambda_1)\beta_0)^2\right] = \frac{p^2}{(p-1)(p+2)}(\alpha_0^\top \beta_0)^2\mathbb{E}\left[\left(\int \frac{1}{(x+\lambda_2)}\, d\hat{\pi}_{n,1}(x)\right)^2\right] + O(p^{-1})$$

Therefore,

$$\lim_{n\uparrow\infty}\mathbb{E}\left[(\alpha_0^\top R_1(\lambda_1)\beta_0)^2\right] = \varrho^2 \left(\int \frac{dF_{\mathrm{MP}}(x)}{(x+\lambda_1)}\right)^2$$

Also,

$$\mathbb{E}\left[\frac{\epsilon_1^\top \mathbf{X}_1}{n}R_1(\lambda_1)\beta_0\beta_0^\top R_1(\lambda_1)\frac{\mathbf{X}_1^\top \epsilon_1}{n}\right] = \frac{1}{n}\mathbb{E}\left[\beta_0^\top R_1(\lambda_1)\hat{\Sigma}_1 R_1(\lambda_1)\beta_0\right] = O(n^{-1}).$$

Hence, we have:

$$\lim_{n\uparrow\infty}\mathbb{E}\left[\left((\alpha_0 - \hat{\alpha}(\lambda_1))^\top R_2(\lambda_2)\beta_0\right)^2\right] = \lambda_1^2\varrho^2 \left(\int \frac{dF_{\mathrm{MP}}(x)}{(x+\lambda_1)}\right)^2 \left(\int \frac{dF_{\mathrm{MP}}(x)}{(x+\lambda_2)}\right)^2.$$

This, in turn, implies:

$$\lim_{n\uparrow\infty}\mathbb{E}\left[((\alpha_0 - \hat{\alpha}(\lambda_1))^\top (\beta_0 - \hat{\beta}(\lambda_2)))^2\right] = \lambda_1^2\lambda_2^2\varrho^2 \left(\int \frac{dF_{\mathrm{MP}}(x)}{(x+\lambda_1)}\right)^2 \left(\int \frac{dF_{\mathrm{MP}}(x)}{(x+\lambda_2)}\right)^2.$$

Collecting all terms, we conclude:

$$\lim_{n\uparrow\infty} n \times \mathbb{E}\left[\mathsf{var}\left(\hat{\theta}_{3\mathrm{sp}}^{\mathrm{DR,db}} \mid \mathcal{D}_{1:2}\right)\right] = \left\{\left(u^2 \int \frac{\lambda_1^2\, dF_{\mathrm{MP}}(x)}{(x+\lambda_1)^2} + c \int \frac{x\, dF_{\mathrm{MP}}(x)}{(x+\lambda_1)^2}\right)\right.$$

$$\left.\times \left(v^2 \int \frac{\lambda_2^2\, dF_{\mathrm{MP}}(x)}{(x+\lambda_2)^2} + c \int \frac{x\, dF_{\mathrm{MP}}(x)}{(x+\lambda_2)^2}\right)\right\}$$

$$+ u^2 \int \frac{\lambda_1^2\, dF_{\mathrm{MP}}(x)}{(x+\lambda_1)^2} + c \int \frac{x\, dF_{\mathrm{MP}}(x)}{(x+\lambda_1)^2}$$

$$+ v^2 \int \frac{\lambda_2^2\, dF_{\mathrm{MP}}(x)}{(x+\lambda_2)^2} + c \int \frac{x\, dF_{\mathrm{MP}}(x)}{(x+\lambda_2)^2}$$

$$+ 2\theta_0 \lambda_1 \lambda_2 \varrho \int \frac{dF_{\mathrm{MP}}(x)}{x+\lambda_1} \int \frac{dF_{\mathrm{MP}}(x)}{x+\lambda_2}$$

$$+ \lambda_1^2 \lambda_2^2 \varrho^2 \left(\int \frac{dF_{\mathrm{MP}}(x)}{(x+\lambda_1)}\right)^2 \left(\int \frac{dF_{\mathrm{MP}}(x)}{(x+\lambda_2)}\right)^2$$

$$+ 1 + \theta_0^2\,. \tag{D.26}$$

Combining Equation (D.25) and (D.26) we conclude the proof.

## D.6 Limiting Variance of $\hat{\theta}_{3\mathrm{sp}}^{\mathrm{NR,db}}$

The calculation of limiting variance of $\hat{\theta}_{3\mathrm{sp}}^{\mathrm{NR,db}}$ is similar to that of $\hat{\theta}_{3\mathrm{sp}}^{\mathrm{DR,db}}$. We again use the tower property, conditioning on $\mathcal{D}_{1:2} = (\mathcal{D}_1, \mathcal{D}_2)$. Recall the definition in Equation (3.5):

$$\hat{\theta}_{3\mathrm{sp}}^{\mathrm{NR,db}} = \left\{\frac{1}{n} \sum_{i \in \mathcal{D}_3} Y_i (A_i - X_i^\top \hat{\alpha}(\lambda_1))\right\} - \hat{\alpha}(\lambda_1)^\top \hat{\beta}(\lambda_2) H_1(\boldsymbol{\lambda})$$

where the definition of $H_1(\boldsymbol{\lambda})$ is evident from the definition. The conditional expectation is:

$$\mathbb{E}\left[\hat{\theta}_{3\mathrm{sp}}^{\mathrm{NR,db}} \mid \mathcal{D}_{1:2}\right] = \beta_0^\top (\alpha_0 - \hat{\alpha}(\lambda_1)) + \theta_0 - \hat{\alpha}(\lambda_1)^\top \hat{\beta}(\lambda_2) H_1(\boldsymbol{\lambda})$$

Therefore, we have:

$$\mathsf{var}\left(\mathbb{E}\left[\hat{\theta}_{3\mathrm{sp}}^{\mathrm{NR,db}} | \mathcal{D}_{1:2}\right]\right) = \mathsf{var}\left(\beta_0^\top \hat{\alpha}(\lambda_1) + \hat{\alpha}(\lambda_1)^\top \hat{\beta}(\lambda_2) H_1(\boldsymbol{\lambda})\right)$$

$$= \mathsf{var}(\beta_0^\top \hat{\alpha}(\lambda_1)) + H_1^2(\boldsymbol{\lambda})\mathsf{var}(\hat{\alpha}(\lambda_1)^\top \hat{\beta}(\lambda_2)) + 2H_1(\boldsymbol{\lambda})\mathsf{cov}(\beta_0^\top \hat{\alpha}(\lambda_1),\, \hat{\alpha}(\lambda_1)^\top \hat{\beta}(\lambda_2))\,.$$

We have already established in the proof of the limiting variance of $\hat{\theta}_{3\mathrm{sp}}^{\mathrm{DR,db}}$ that:

$$\lim_{n\uparrow\infty} n \times \mathsf{var}\left(\hat{\alpha}(\lambda_1)^\top \hat{\beta}(\lambda_2)\right)$$

$$= \left[\frac{1}{c}\mathsf{var}_{X\sim F_{\mathrm{MP}}}\left(\frac{X}{X+\lambda_2}\right)\{\varrho^2 + u^2 v^2\} + u^2 \int \frac{x}{(x+\lambda_2)^2}\, dF_{\mathrm{MP}}(x)\right]\left(\int \frac{x}{x+\lambda_1}\, dF_{\mathrm{MP}}(x)\right)^2$$

$$+ \frac{1}{c}\mathsf{var}_{X\sim F_{\mathrm{MP}}}\left(\frac{X}{X+\lambda_1}\right)\left\{\varrho^2 \left(\int \frac{x}{x+\lambda_2}\, dF_{\mathrm{MP}}(x)\right)^2 + \right.$$

$$\left. u^2 \left[v^2 \left(\int \frac{x^2}{(x+\lambda_2)^2}\, dF_{\mathrm{MP}}(x)\right) + c \left(\int \frac{x}{(x+\lambda_2)^2}\, dF_{\mathrm{MP}}(x)\right)\right]\right\}$$

$$+ \left\{v^2 \left(\int \frac{x^2}{(x+\lambda_2)^2}\, dF_{\mathrm{MP}}(x)\right) + c \left(\int \frac{x}{(x+\lambda_2)^2}\, dF_{\mathrm{MP}}(x)\right)\right\}\left(\int \frac{x}{(x+\lambda_1)^2}\, dF_{\mathrm{MP}}(x)\right)\,.$$

Furthermore, we have also established:

$$\lim_{n\uparrow\infty} n \times \mathsf{var}(\hat{\alpha}(\lambda_1)^\top \beta_0) = \frac{1}{c}(\varrho^2 + u^2 v^2)\, \mathsf{var}_{X\sim F_{\mathrm{MP}}}\left(\frac{X}{X+\lambda_1}\right) + v^2 \int \frac{x\, dF_{\mathrm{MP}}(x)}{(x+\lambda_1)^2}\,,$$

and

$$\lim_{n\uparrow\infty} n \times \mathsf{cov}(\hat{\alpha}(\lambda_1)^\top \hat{\beta}(\lambda_2), \hat{\alpha}(\lambda_1)^\top \beta_0)$$

$$= \int \frac{x \, dF_{\mathrm{MP}}(x)}{x + \lambda_2} \left( \frac{1}{c}(\varrho^2 + u^2 v^2) \, \mathsf{var}_{X \sim F_{\mathrm{MP}}} \left( \frac{X}{X + \lambda_1} \right) + v^2 \int \frac{x \, dF_{\mathrm{MP}}(x)}{(x + \lambda_1)^2} \right).$$

Combining the terms, we get:

$$\lim_{n \uparrow \infty} \mathsf{var} \left( \mathbb{E} \left[ \hat{\theta}_{3\mathrm{sp}}^{\mathrm{NR,db}} | \mathcal{D}_{1:2} \right] \right)$$

$$= H_1^2(\boldsymbol{\lambda}) \left\{ \left[ \frac{1}{c} \mathsf{var}_{X \sim F_{\mathrm{MP}}} \left( \frac{X}{X + \lambda_2} \right) \{ \varrho^2 + u^2 v^2 \} \right. \right.$$

$$\left. + u^2 \int \frac{x}{(x + \lambda_2)^2} \, dF_{\mathrm{MP}}(x) \right] \left( \int \frac{x}{x + \lambda_1} \, dF_{\mathrm{MP}}(x) \right)^2$$

$$+ \frac{1}{c} \mathsf{var}_{X \sim F_{\mathrm{MP}}} \left( \frac{X}{X + \lambda_1} \right) \left\{ \varrho^2 \left( \int \frac{x}{x + \lambda_2} \, dF_{\mathrm{MP}}(x) \right)^2 + \right.$$

$$\left. u^2 \left[ v^2 \left( \int \frac{x^2}{(x + \lambda_2)^2} \, dF_{\mathrm{MP}}(x) \right) + c \left( \int \frac{x}{(x + \lambda_2)^2} \, dF_{\mathrm{MP}}(x) \right) \right] \right\}$$

$$+ \left\{ v^2 \left( \int \frac{x^2}{(x + \lambda_2)^2} \, dF_{\mathrm{MP}}(x) \right) + c \left( \int \frac{x}{(x + \lambda_2)^2} \, dF_{\mathrm{MP}}(x) \right) \right\} \left( \int \frac{x}{(x + \lambda_1)^2} \, dF_{\mathrm{MP}}(x) \right) \right\}$$

$$+ \frac{1}{c}(\varrho^2 + u^2 v^2) \, \mathsf{var}_{X \sim F_{\mathrm{MP}}} \left( \frac{X}{X + \lambda_1} \right) + v^2 \int \frac{x \, dF_{\mathrm{MP}}(x)}{(x + \lambda_1)^2}$$

$$+ 2 H_1(\boldsymbol{\lambda}) \int \frac{x \, dF_{\mathrm{MP}}(x)}{x + \lambda_2} \left( \frac{1}{c}(\varrho^2 + u^2 v^2) \, \mathsf{var}_{X \sim F_{\mathrm{MP}}} \left( \frac{X}{X + \lambda_1} \right) + v^2 \int \frac{x \, dF_{\mathrm{MP}}(x)}{(x + \lambda_1)^2} \right). \quad \text{(D.27)}$$

Next, we establish the limit of the expectation of the conditional variance:

$$\mathsf{var} \left( \theta_{3\mathrm{sp}}^{\mathrm{NR,db}} | \mathcal{D}_{1:2} \right) = \frac{1}{n} \mathsf{var} \left( Y(A - X^\top \hat{\alpha}(\lambda_1)) \right)$$

$$= \frac{1}{n} \left\{ \mathsf{var} \left( (\alpha_0 - \hat{\alpha}(\lambda_1))^\top X X^\top \beta_0 \right) + \mathsf{var} \left( X^\top \beta_0 \epsilon \right) \right.$$

$$\left. + \mathsf{var} \left( X^\top (\alpha_0 - \hat{\alpha}(\lambda_1)) \mu \right) + \mathsf{var}(\mu \epsilon) + 2 \mathsf{cov} \left( X^\top (\alpha_0 - \hat{\alpha}(\lambda_1)) \mu, \ X^\top \beta_0 \epsilon \right) \right\}$$

$$= \frac{1}{n} \left( \| \alpha_0 - \hat{\alpha}(\lambda_1) \|_2^2 \| \beta_0 \|_2^2 + ((\alpha_0 - \hat{\alpha}(\lambda_1))^\top \beta_0)^2 \right.$$

$$\left. + \| \alpha_0 - \hat{\alpha}(\lambda_1) \|_2^2 + \| \beta_0 \|_2^2 + 2 \theta_0 (\alpha_0 - \hat{\alpha}(\lambda_1))^\top \beta_0 + 1 + \theta_0^2 \right)$$

In the proof of the limiting variance of $\hat{\theta}_{3\mathrm{sp}}^{\mathrm{DR,db}}$, we have already established that:

$$\lim_{n \uparrow \infty} \mathbb{E} \left[ \| \alpha_0 - \hat{\alpha}(\lambda_1) \|_2^2 \right] = u^2 \int \frac{\lambda_1^2 \, dF_{\mathrm{MP}}(x)}{(x + \lambda_1)^2} + c \int \frac{x \, dF_{\mathrm{MP}}(x)}{(x + \lambda_1)^2}$$

$$\lim_{n \uparrow \infty} \mathbb{E}[((\alpha_0 - \hat{\alpha}(\lambda_1))^\top \beta_0)^2] = \lambda_1^2 \varrho^2 \left( \int \frac{dF_{\mathrm{MP}}(x)}{(x + \lambda_1)} \right)^2$$

$$\lim_{n \uparrow \infty} \mathbb{E}[(\alpha_0 - \hat{\alpha}(\lambda_1))^\top \beta_0] = \lambda_1 \varrho \int \frac{dF_{\mathrm{MP}}(x)}{x + \lambda_1}.$$

Hence, combining them, we conclude:

$$\lim_{n \uparrow \infty} n \times \mathbb{E} \left[ \mathsf{var} \left( \theta_{3\mathrm{sp}}^{\mathrm{NR,db}} | \mathcal{D}_{1:2} \right) \right] = v^2 \left( u^2 \int \frac{\lambda_1^2 \, dF_{\mathrm{MP}}(x)}{(x + \lambda_1)^2} + c \int \frac{x \, dF_{\mathrm{MP}}(x)}{(x + \lambda_1)^2} \right)$$

$$+ \lambda_1^2 \varrho^2 \left( \int \frac{dF_{\mathrm{MP}}(x)}{(x + \lambda_1)} \right)^2$$

$$+ v^2 + u^2 \int \frac{\lambda_1^2 \, dF_{\mathrm{MP}}(x)}{(x + \lambda_1)^2} + c \int \frac{x \, dF_{\mathrm{MP}}(x)}{(x + \lambda_1)^2}$$

$$+ 2\theta_0\lambda_1\varrho \int \frac{dF_{\mathrm{MP}}(x)}{x + \lambda_1} + 1 + \theta_0^2. \qquad (\mathrm{D.28})$$

Combining the limits in Equation (D.27) and (D.28),

we conclude the proof.

# E    Some Auxiliary Lemmas

**Lemma E.1.** *Suppose* $\mathbf{X} \in \mathbb{R}^{n \times p}$ *is a standard Gaussian random matrix, i.e.* $X_{ij} \overset{i.i.d.}{\sim} \mathcal{N}(0, 1)$ *and* $p/n = c$. *Define the corresponding covariance matrix* $\hat{\Sigma} = (\mathbf{X}^\top \mathbf{X})/n$. *Suppose* $\hat{\Sigma}$ *admits the following eigendecomposition:*

$$\hat{\Sigma} = \hat{\mathbf{V}}^\top \hat{\Lambda} \hat{\mathbf{V}} = \sum_{j=1}^{p} \hat{v}_j \hat{v}_j^\top \hat{\lambda}_j,$$

*where* $\hat{\mathbf{V}} \in \mathbb{O}(p)$ *is the eigenmatrix of* $\hat{\Sigma}$ *and* $\{\hat{v}_j\}_{1 \leq j \leq p}$ *are columns of* $\hat{\mathbf{V}}$. *Then for any two sequences of vectors* $\{\mathbf{u}_n\}_{n \in \mathbb{N}}$ *and* $\{\mathbf{w}_n\}_{n \in \mathbb{N}}$, *with*

$$\lim_{n \uparrow \infty} \|\mathbf{u}_n\|_2 = u, \qquad \lim_{n \uparrow \infty} \|\mathbf{w}_n\|_2 = v, \qquad \lim_{n \uparrow \infty} \mathbf{u}_n^\top \mathbf{w}_n = \varrho,$$

*we have for any bounded function* $f$:

$$\lim_{n \uparrow \infty} n \times \mathsf{var} \left( \mathbf{u}_n^\top \mathbf{V}^\top f(\hat{\Lambda}) \mathbf{V} \mathbf{w}_n \right) = \lim_{n \uparrow \infty} n \times \mathsf{var} \left( \sum_{j=1}^{p} (\hat{v}_j^\top \mathbf{u}_n)(\hat{v}_j^\top \mathbf{w}_n) f(\hat{\lambda}_j) \right)$$

$$= \frac{1}{c} \left( \varrho^2 + u^2 v^2 \right) \mathsf{var}_{X \sim F_{\mathrm{MP}}} \left( f(X) \right).$$

*Proof.* The proof hinges on the following three key facts:

1. From rotational invariance of a Gaussian random variable, we know that $\hat{\mathbf{V}}$ is independent of $\hat{\Lambda}$. Furthermore, $\hat{\mathbf{V}}$ follows Haar distribution on $\mathbb{O}(p)$, the set of all $p \times p$ orthonormal matrices.

2. For any $1 \leq i \leq p$, $\hat{v}_i$ (the $i^{th}$ eigenvector), is marginally uniformly distributed on the sphere. As a consequence $\hat{v}_i \overset{d}{=} g/\|g\|$, where $g \sim \mathcal{N}(0, \mathbf{I}_p)$.

3. For any $i \neq j$, the marginal distribution of $(\hat{v}_i, \hat{v}_j)$ is same for all $i \neq j$, which follows from Haar measure.

These facts are elementary and well-known. Based on these facts, we first establish a few claims:

**Claim 1:** For any $1 \leq i \leq p$, we have:

$$\mathbb{E} \left[ (\hat{v}_i^\top \mathbf{u}_n)(\hat{v}_i^\top \mathbf{w}_n) \right] = \frac{\mathbf{u}_n^\top \mathbf{w}_n}{p}.$$

The proof of the claim is simple by using a symmetry argument, thanks to the second key fact. As $\hat{v}_i$ has same marginal distribution for all $i$, the value of $\mathbb{E} \left[ (\hat{v}_i^\top \mathbf{u}_n)(\hat{v}_i^\top \mathbf{w}_n) \right]$ should also be the same for all $i$. Denote the common value by $C$. Then,

$$C = \mathbb{E} \left[ (\hat{v}_1^\top \mathbf{u}_n)(\hat{v}_1^\top \mathbf{w}_n) \right] = \cdots = \mathbb{E} \left[ (\hat{v}_p^\top \mathbf{u}_n)(\hat{v}_p^\top \mathbf{w}_n) \right]$$

$$= \frac{1}{p} \sum_{i=1}^{p} \mathbb{E} \left[ (\hat{v}_i^\top \mathbf{u}_n)(\hat{v}_i^\top \mathbf{w}_n) \right]$$

$$= \frac{1}{p} \mathbf{u}_n^\top \mathbb{E} \left[ \sum_{i=1}^{p} \hat{v}_i \hat{v}_i^\top \right] \mathbf{w}_n = \frac{1}{p} \mathbf{u}_n^\top \mathbb{E} \left[ \mathbf{V} \mathbf{V}^\top \right] \mathbf{w}_n = \frac{\mathbf{u}_n^\top \mathbf{w}_n}{p}.$$

The last equality follows from the fact that $\mathbf{V}\mathbf{V}^\top = \mathbf{I}_p$ as $\mathbf{V} \in \mathbb{O}(p)$, which completes the proof of the claim. In our following two claims, we compute the expectation, the second moment, and cross moments of the inner product:

**Claim 2:** For any $1 \leq i \leq p$, we have:

$$\mathbb{E}\left[(\hat{v}_i^\top \mathbf{u}_n)^2(\hat{v}_i^\top \mathbf{w}_n)^2\right] = \frac{1}{p(p+2)}\left[\|\mathbf{u}_n\|_2^2\|\mathbf{w}_n\|_2^2 + 2(\mathbf{u}_n^\top \mathbf{w}_n)^2\right]$$

To prove this claim, we again use the second key fact. Suppose $g \sim \mathcal{N}(0, \mathbf{I}_p)$. Then for any $1 \leq i \leq p$:

$$\mathbb{E}\left[(\hat{v}_i^\top \mathbf{u}_n)^2(\hat{v}_i^\top \mathbf{w}_n)^2\right] = \mathbb{E}\left[\frac{(g^\top \mathbf{u}_n)^2(g^\top \mathbf{w}_n)^2}{\|g\|_2^4}\right]$$

Now as $g/\|g\| \perp\!\!\!\perp \|g\|$, we have:

$$\mathbb{E}\left[(g^\top \mathbf{u}_n)^2(g^\top \mathbf{w}_n)^2\right] = \mathbb{E}\left[\frac{(g^\top \mathbf{u}_n)^2(g^\top \mathbf{w}_n)^2}{\|g\|_2^4}\right]\mathbb{E}[\|g\|_2^4]$$

$$\implies \mathbb{E}\left[\frac{(g^\top \mathbf{u}_n)^2(g^\top \mathbf{w}_n)^2}{\|g\|_2^4}\right] = \frac{\mathbb{E}\left[(g^\top \mathbf{u}_n)^2(g^\top \mathbf{w}_n)^2\right]}{\mathbb{E}[\|g\|_2^4]}.$$

Now as $\|g\|_2^2 \sim \chi_p^2$, we have $\mathbb{E}[\|g\|_2^4] = p(p+2)$. On the other hand, as $g \sim \mathcal{N}(0, \mathbf{I}_p)$, we know

$$\begin{pmatrix} g^\top \mathbf{u}_n \\ g^\top \mathbf{w}_n \end{pmatrix} \sim \mathcal{N}\left(\begin{pmatrix} 0 \\ 0 \end{pmatrix}, \begin{pmatrix} \|\mathbf{u}_n\|_2^2 & \mathbf{u}_n^\top \mathbf{w}_n \\ \mathbf{u}_n^\top \mathbf{w}_n & \|\mathbf{w}_n\|_2^2 \end{pmatrix}\right).$$

Therefore,

$$\mathbb{E}\left[(g^\top \mathbf{u}_n)^2(g^\top \mathbf{w}_n)^2\right] = \|\mathbf{u}_n\|_2^2\|\mathbf{w}_n\|_2^2 + 2(\mathbf{u}_n^\top \mathbf{w}_n)^2.$$

Therefore, we can conclude:

$$\mathbb{E}\left[(\hat{v}_i^\top \mathbf{u}_n)^2(\hat{v}_i^\top \mathbf{w}_n)^2\right] = \mathbb{E}\left[\frac{(g^\top \mathbf{u}_n)^2(g^\top \mathbf{w}_n)^2}{\|g\|_2^4}\right] = \frac{\mathbb{E}\left[(g^\top \mathbf{u}_n)^2(g^\top \mathbf{w}_n)^2\right]}{\mathbb{E}[\|g\|_2^4]}$$

$$= \frac{1}{p(p+2)}\left[\|\mathbf{u}_n\|_2^2\|\mathbf{w}_n\|_2^2 + 2(\mathbf{u}_n^\top \mathbf{w}_n)^2\right].$$

**Claim 3:** For any $1 \leq i \neq j \leq p$, we have:

$$\mathbb{E}\left[(\hat{v}_i^\top \mathbf{u}_n)(\hat{v}_i^\top \mathbf{w}_n)(\hat{v}_j^\top \mathbf{u}_n)(\hat{v}_j^\top \mathbf{w}_n)\right] = \frac{1}{p(p-1)}\left[\frac{p}{p+2}(\mathbf{u}_n^\top \mathbf{w}_n)^2 - \frac{\|\mathbf{u}_n\|_2^2\|\mathbf{w}_n\|_2^2}{(p+2)}\right].$$

Again, we prove this claim with a simple symmetry argument using the third key fact. The main observation is that for any $i \neq j$, the value of the expectation is the same as the marginal distribution of $(\hat{v}_i, \hat{v}_j)$ is the same for any $i \neq j$. Let us denote the common value by $D_1$. Then we have:

$$\sum_{i,j}\mathbb{E}\left[(\hat{v}_i^\top \mathbf{u}_n)(\hat{v}_i^\top \mathbf{w}_n)(\hat{v}_j^\top \mathbf{u}_n)(\hat{v}_j^\top \mathbf{w}_n)\right] = \sum_{i=1}^{p}\mathbb{E}\left[(\hat{v}_i^\top \mathbf{u}_n)^2(\hat{v}_i^\top \mathbf{w}_n)^2\right]$$

$$+ \sum_{i\neq j}\mathbb{E}\left[(\hat{v}_i^\top \mathbf{u}_n)(\hat{v}_i^\top \mathbf{w}_n)(\hat{v}_j^\top \mathbf{u}_n)(\hat{v}_j^\top \mathbf{w}_n)\right]$$

$$= \sum_{i=1}^{p}\mathbb{E}\left[(\hat{v}_i^\top \mathbf{u}_n)^2(\hat{v}_i^\top \mathbf{w}_n)^2\right] + C_1 p(p-1)$$

$$= \frac{1}{(p+2)}\left[\|\mathbf{u}_n\|_2^2\|\mathbf{w}_n\|_2^2 + 2(\mathbf{u}_n^\top \mathbf{w}_n)^2\right] + C_1 p(p-1).$$

Here, the last equality follows from our conclusion in Claim 2. Now, if we consider, the LHS of the above equation, we have:

$$\sum_{i,j}\mathbb{E}\left[(\hat{v}_i^\top \mathbf{u}_n)(\hat{v}_i^\top \mathbf{w}_n)(\hat{v}_j^\top \mathbf{u}_n)(\hat{v}_j^\top \mathbf{w}_n)\right] = \mathbb{E}\left[\sum_{i,j}(\hat{v}_i^\top \mathbf{u}_n)(\hat{v}_i^\top \mathbf{w}_n)(\hat{v}_j^\top \mathbf{u}_n)(\hat{v}_j^\top \mathbf{w}_n)\right]$$

$$= \mathbb{E}\left[\left(\sum_i (\hat{v}_i^\top \mathbf{u}_n)(\hat{v}_i^\top \mathbf{w}_n)\right)^2\right]$$

$$= \mathbb{E}\left[\left(\mathbf{u}_n^\top \left(\sum_{i=1}^p \hat{v}_i \hat{v}_i^\top\right) \mathbf{w}_n\right)^2\right]$$

$$= \mathbb{E}\left[\left(\mathbf{u}_n^\top \hat{\mathbf{V}} \hat{\mathbf{V}}^\top \mathbf{w}_n\right)^2\right] = (\mathbf{u}_n^\top \mathbf{w}_n)^2.$$

Therefore, we have:

$$(\mathbf{u}_n^\top \mathbf{w}_n)^2 = \frac{1}{(p+2)}\left[\|\mathbf{u}_n\|_2^2 \|\mathbf{w}_n\|_2^2 + 2(\mathbf{u}_n^\top \mathbf{w}_n)^2\right] + C_1 p(p-1)$$

$$\implies C_1 = \frac{1}{p(p-1)}\left[(\mathbf{u}_n^\top \mathbf{w}_n)^2 - \frac{1}{(p+2)}\left[\|\mathbf{u}_n\|_2^2 \|\mathbf{w}_n\|_2^2 + 2(\mathbf{u}_n^\top \mathbf{w}_n)^2\right]\right]$$

This completes the proof. Now, we use the conclusions from these three claims to complete the proof of the Lemma. First of all, from the definition of variance, we have:

$$\mathsf{var}\left(\sum_{j=1}^p (\hat{v}_j^\top \mathbf{u}_n)(\hat{v}_j^\top \mathbf{w}_n)f(\hat{\lambda}_j)\right) = \mathbb{E}\left[\left(\sum_{j=1}^p (\hat{v}_j^\top \mathbf{u}_n)(\hat{v}_j^\top \mathbf{w}_n)f(\hat{\lambda}_j)\right)^2\right] - \left(\mathbb{E}\left[\sum_{j=1}^p (\hat{v}_j^\top \mathbf{u}_n)(\hat{v}_j^\top \mathbf{w}_n)f(\hat{\lambda}_j)\right]\right)^2$$

Let us first consider the second term, expectation. From the independence of eigenvalues and eigenvectors of $\hat{\Sigma}$, we have:

$$\mathbb{E}\left[\sum_{j=1}^p (\hat{v}_j^\top \mathbf{u}_n)(\hat{v}_j^\top \mathbf{w}_n)f(\hat{\lambda}_j)\right] = \sum_{j=1}^p \mathbb{E}\left[(\hat{v}_j^\top \mathbf{u}_n)(\hat{v}_j^\top \mathbf{w}_n)\right]\mathbb{E}\left[f(\hat{\lambda}_j)\right]$$

$$= \frac{\mathbf{u}_n^\top \mathbf{w}_n}{p}\sum_{j=1}^p \mathbb{E}\left[f(\hat{\lambda}_j)\right] = \left(\mathbf{u}_n^\top \mathbf{w}_n\right)\mathbb{E}\left[\int f(x)\,d\hat{\pi}_n(x)\right]$$

Here $\hat{\pi}_n$ is the empirical spectral distribution of the eigenvalues of $\hat{\Sigma}$. As we know that $\hat{\pi}_n$ converges to Marchenko-Pasteur (MP) law and $f$ is a bounded function, using dominated convergence theorem, we can conclude:

$$\left(\mathbb{E}\left[\sum_{j=1}^p (\hat{v}_j^\top \mathbf{u}_n)(\hat{v}_j^\top \mathbf{w}_n)f(\hat{\lambda}_j)\right]\right)^2 = (\mathbf{u}_n^\top \mathbf{w}_n)^2 \left(\mathbb{E}\left[\int f(x)\,d\hat{\pi}_n(x)\right]\right)^2 \qquad \text{(E.1)}$$

Now, coming back to the second moment:

$$\mathbb{E}\left[\left(\sum_{j=1}^p (\hat{v}_j^\top \mathbf{u}_n)(\hat{v}_j^\top \mathbf{w}_n)f(\hat{\lambda}_j)\right)^2\right]$$

$$= \mathbb{E}\left[\sum_{ij} (\hat{v}_i^\top \mathbf{u}_n)(\hat{v}_i^\top \mathbf{w}_n)(\hat{v}_j^\top \mathbf{u}_n)(\hat{v}_j^\top \mathbf{w}_n)f(\hat{\lambda}_i)f(\hat{\lambda}_j)\right]$$

$$= \sum_{ij} \mathbb{E}\left[(\hat{v}_i^\top \mathbf{u}_n)(\hat{v}_i^\top \mathbf{w}_n)(\hat{v}_j^\top \mathbf{u}_n)(\hat{v}_j^\top \mathbf{w}_n)\right]\mathbb{E}\left[f(\hat{\lambda}_i)f(\hat{\lambda}_j)\right]$$

$$= \sum_{i=1}^p \mathbb{E}\left[(\hat{v}_i^\top \mathbf{u}_n)^2(\hat{v}_i^\top \mathbf{w}_n)^2\right]\mathbb{E}\left[f^2(\hat{\lambda}_i)\right] + \sum_{i\neq j} \mathbb{E}\left[(\hat{v}_i^\top \mathbf{u}_n)(\hat{v}_i^\top \mathbf{w}_n)(\hat{v}_j^\top \mathbf{u}_n)(\hat{v}_j^\top \mathbf{w}_n)\right]\mathbb{E}\left[f(\hat{\lambda}_i)f(\hat{\lambda}_j)\right]$$

$$= \frac{1}{p(p+2)}\left[\|\mathbf{u}_n\|_2^2 \|\mathbf{w}_n\|_2^2 + 2(\mathbf{u}_n^\top \mathbf{w}_n)^2\right]\sum_{i=1}^p \mathbb{E}\left[f^2(\hat{\lambda}_i)\right]$$

$$+ \frac{1}{p(p-1)} \left[ \frac{p}{p+2}(\mathbf{u}_n^\top \mathbf{w}_n)^2 - \frac{\|\mathbf{u}_n\|_2^2 \|\mathbf{w}_n\|_2^2}{(p+2)} \right] \sum_{i \neq j} \mathbb{E}\left[ f(\hat{\lambda}_i) f(\hat{\lambda}_j) \right]$$

$$= \frac{1}{(p+2)} \left[ \|\mathbf{u}_n\|_2^2 \|\mathbf{w}_n\|_2^2 + 2(\mathbf{u}_n^\top \mathbf{w}_n)^2 \right] \mathbb{E}\left[ \int f^2(x)\, d\hat{\pi}_n(x) \right]$$

$$+ \frac{1}{p(p-1)} \left[ \frac{p}{p+2}(\mathbf{u}_n^\top \mathbf{w}_n)^2 - \frac{\|\mathbf{u}_n\|_2^2 \|\mathbf{w}_n\|_2^2}{(p+2)} \right] \left\{ \mathbb{E}\left[ \left( \sum_{i=1}^{p} f(\hat{\lambda}_i) \right)^2 \right] - \sum_{i=1}^{p} \mathbb{E}\left[ f^2(\hat{\lambda}_i) \right] \right\}$$

$$= \frac{1}{(p+2)} \left[ \|\mathbf{u}_n\|_2^2 \|\mathbf{w}_n\|_2^2 + 2(\mathbf{u}_n^\top \mathbf{w}_n)^2 \right] \mathbb{E}\left[ \int f^2(x)\, d\hat{\pi}_n(x) \right]$$

$$+ \frac{p}{p-1} \left[ \frac{p}{p+2}(\mathbf{u}_n^\top \mathbf{w}_n)^2 - \frac{\|\mathbf{u}_n\|_2^2 \|\mathbf{w}_n\|_2^2}{(p+2)} \right] \mathbb{E}\left[ \left( \int f(x)\, d\hat{\pi}_n(x) \right)^2 \right]$$

$$- \frac{1}{p-1} \left[ \frac{p}{p+2}(\mathbf{u}_n^\top \mathbf{w}_n)^2 - \frac{\|\mathbf{u}_n\|_2^2 \|\mathbf{w}_n\|_2^2}{(p+2)} \right] \mathbb{E}\left[ \int f^2(x)\, d\hat{\pi}_n(x) \right] \tag{E.2}$$

This, along with Equation (E.1), yields:

$$\mathsf{var}\left( \sum_{j=1}^{p} (\hat{v}_j^\top \mathbf{u}_n)(\hat{v}_j^\top \mathbf{w}_n) f(\hat{\lambda}_j) \right)$$

$$= \frac{1}{(p+2)} \left[ \|\mathbf{u}_n\|_2^2 \|\mathbf{w}_n\|_2^2 + 2(\mathbf{u}_n^\top \mathbf{w}_n)^2 \right] \mathbb{E}\left[ \int f^2(x)\, d\hat{\pi}_n(x) \right]$$

$$+ \frac{p}{p-1} \left[ \frac{p}{p+2}(\mathbf{u}_n^\top \mathbf{w}_n)^2 - \frac{\|\mathbf{u}_n\|_2^2 \|\mathbf{w}_n\|_2^2}{(p+2)} \right] \mathbb{E}\left[ \left( \int f(x)\, d\hat{\pi}_n(x) \right)^2 \right]$$

$$- \frac{1}{p-1} \left[ \frac{p}{p+2}(\mathbf{u}_n^\top \mathbf{w}_n)^2 - \frac{\|\mathbf{u}_n\|_2^2 \|\mathbf{w}_n\|_2^2}{(p+2)} \right] \mathbb{E}\left[ \int f^2(x)\, d\hat{\pi}_n(x) \right]$$

$$- (\mathbf{u}_n^\top \mathbf{w}_n)^2 \left( \mathbb{E}\left[ \int f(x)\, d\hat{\pi}_n(x) \right] \right)^2$$

$$= \left[ \frac{p-2}{(p+2)(p-1)}(\mathbf{u}_n^\top \mathbf{w}_n)^2 + \frac{p}{(p+2)(p-1)} \|\mathbf{u}_n\|_2^2 \|\mathbf{w}_n\|_2^2 \right] \mathbb{E}\left[ \int f^2(x)\, d\hat{\pi}_n(x) \right]$$

$$- \left[ \frac{p-2}{(p-1)(p+2)}(\mathbf{u}_n^\top \mathbf{w}_n)^2 + \frac{p}{(p-1)(p+2)} \|\mathbf{u}_n\|_2^2 \|\mathbf{w}_n\|_2^2 \right] \mathbb{E}\left[ \left( \int f(x)\, d\hat{\pi}_n(x) \right)^2 \right]$$

$$+ (\mathbf{u}_n^\top \mathbf{w}_n)^2 \mathsf{var}\left( \int f(x)\, d\hat{\pi}_n(x) \right)$$

$$= \left[ \frac{p-2}{(p+2)(p-1)}(\mathbf{u}_n^\top \mathbf{w}_n)^2 + \frac{p}{(p+2)(p-1)} \|\mathbf{u}_n\|_2^2 \|\mathbf{w}_n\|_2^2 \right] \left\{ \mathbb{E}\left[ \int f^2(x)\, d\hat{\pi}_n(x) \right] - \mathbb{E}\left[ \left( \int f(x)\, d\hat{\pi}_n(x) \right)^2 \right] \right\}$$

$$(\mathbf{u}_n^\top \mathbf{w}_n)^2 \mathsf{var}\left( \int f(x)\, d\hat{\pi}_n(x) \right).$$

Recall that our goal is to compute:

$$\lim_{n \uparrow \infty} n \times \mathsf{var}\left( \sum_{j=1}^{p} (\hat{v}_j^\top \mathbf{u}_n)(\hat{v}_j^\top \mathbf{w}_n) f(\hat{\lambda}_j) \right).$$

As $p/n \longrightarrow c$, we have:

$$\lim_{n \uparrow \infty} n \times \mathsf{var}\left( \sum_{j=1}^{p} (\hat{v}_j^\top \mathbf{u}_n)(\hat{v}_j^\top \mathbf{w}_n) f(\hat{\lambda}_j) \right) = c \lim_{p \uparrow \infty} p \times \mathsf{var}\left( \sum_{j=1}^{p} (\hat{v}_j^\top \mathbf{u}_n)(\hat{v}_j^\top \mathbf{w}_n) f(\hat{\lambda}_j) \right).$$

Consequently, it is immediate to compute the limit with respect to $p$. Towards that goal, first, observe that we have,

$$\lim_{p \uparrow \infty} p \times \text{var}\left(\int f(x)\, d\hat{\pi}_n(x)\right) = 0\,,$$

by Theorem 1.1 of [28], as there the authors have proved that

$$p\left(\int f(x)\, d\hat{\pi}_n(x) - \int f(x)\, dF_{\text{MP}}(x)\right) \stackrel{\mathscr{L}}{\Longrightarrow} \mathcal{N}(0, \sigma_f^2)\,.$$

Therefore, $\text{var}\left(\int f(x)\, d\hat{\pi}_n(x)\right) = O(p^{-2})$ and consequently $p\,\text{var}\left(\int f(x)\, d\hat{\pi}_n(x)\right) \longrightarrow 0$. Furthermore, by DCT, we have:

$$\mathbb{E}\left[\int f^2(x)\, d\hat{\pi}_n(x)\right] - \mathbb{E}\left[\left(\int f(x)\, d\hat{\pi}_n(x)\right)^2\right] \longrightarrow \int f^2(x)\, dF_{\text{MP}}(x) - \left(\int f(x)\, dF_{\text{MP}}(x)\right)$$

$$= \text{var}_{X \sim F_{\text{MP}}}(f(X))\,.$$

Therefore, it is immediate:

$$\lim_{p \uparrow \infty} p \times \text{var}\left(\sum_{j=1}^{p}(\hat{v}_j^\top \mathbf{u}_n)(\hat{v}_j^\top \mathbf{w}_n)f(\hat{\lambda}_j)\right) = \left(\varrho^2 + u^2 v^2\right)\text{var}_{X \sim F_{\text{MP}}}(f(X))\,.$$

and consequently,

$$\lim_{n \uparrow \infty} n \times \text{var}\left(\sum_{j=1}^{p}(\hat{v}_j^\top \mathbf{u}_n)(\hat{v}_j^\top \mathbf{w}_n)f(\hat{\lambda}_j)\right) = \frac{1}{c}\left(\varrho^2 + u^2 v^2\right)\text{var}_{X \sim F_{\text{MP}}}(f(X))\,.$$

This completes the proof. $\qquad\square$

**Lemma E.2.** *Under the same setup as Lemma E.1, we have:*

$$\lim_{n \uparrow \infty} \mathbb{E}\left[\left(\mathbf{u}_n^\top \mathbf{V} f_1(\Lambda)\mathbf{V}^\top \mathbf{u}_n\right)\left(\mathbf{w}_n^\top \mathbf{V} f_2(\Lambda)\mathbf{V}^\top \mathbf{w}_n\right)\right] = u^2 v^2 \int f_1(x)\, dF_{\text{MP}}(x) \int f_2(x)\, dF_{\text{MP}}(x)\,.$$

*Proof.* The proof is essentially similar to that of Claim 2 and Claim 3 of the of the proof of Lemma E.1. First observe that:

$$\mathbb{E}\left[\left(\mathbf{u}_n^\top \hat{\mathbf{V}} f_1(\hat{\Lambda})\hat{\mathbf{V}}^\top \mathbf{u}_n\right)\left(\mathbf{w}_n^\top \hat{\mathbf{V}} f_2(\hat{\Lambda})\hat{\mathbf{V}}^\top \mathbf{w}_n\right)\right]$$

$$= \mathbb{E}\left[\left(\sum_{j=1}^{p}(\hat{v}_j^\top \mathbf{u}_n)^2 f_1(\hat{\lambda}_j)\right)\left(\sum_{j=1}^{p}(\hat{v}_j^\top \mathbf{w}_n)^2 f_2(\hat{\lambda}_j)\right)\right]$$

$$= \sum_{j=1}^{p}\mathbb{E}\left[(\hat{v}_j^\top \mathbf{u}_n)^2(\hat{v}_j^\top \mathbf{w}_n)^2\right]\mathbb{E}\left[f_1(\hat{\lambda}_j)f_2(\hat{\lambda}_j)\right] + \sum_{j \neq j'}\mathbb{E}\left[(\hat{v}_j^\top \mathbf{u}_n)^2(\hat{v}_{j'}^\top \mathbf{w}_n)^2\right]\mathbb{E}\left[f_1(\hat{\lambda}_j)f_2(\hat{\lambda}_{j'})\right]$$

$$= \frac{1}{p+2}\left(\|\mathbf{u}_n\|_2^2\|\mathbf{w}_n\|_2^2 + 2(\mathbf{u}_n^\top \mathbf{w}_n)^2\right)\mathbb{E}\left[\int f_1(x)f_2(x)\, d\hat{\pi}_n(x)\right]$$

$$\quad + \left(\|\mathbf{u}_n\|_2^2\|\mathbf{w}_n\|_2^2 - \frac{1}{p+2}\left(\|\mathbf{u}_n\|_2^2\|\mathbf{w}_n\|_2^2 + 2(\mathbf{u}_n^\top \mathbf{w}_n)^2\right)\right)\frac{1}{p(p-1)}\sum_{j \neq j'}\mathbb{E}\left[f_1(\hat{\lambda}_j)f_2(\hat{\lambda}_{j'})\right]$$

$$= \frac{1}{p+2}\left(\|\mathbf{u}_n\|_2^2\|\mathbf{w}_n\|_2^2 + 2(\mathbf{u}_n^\top \mathbf{w}_n)^2\right)\mathbb{E}\left[\int f_1(x)f_2(x)\, d\hat{\pi}_n(x)\right]$$

$$\quad + \left(\|\mathbf{u}_n\|_2^2\|\mathbf{w}_n\|_2^2 - \frac{1}{p+2}\left(\|\mathbf{u}_n\|_2^2\|\mathbf{w}_n\|_2^2 + 2(\mathbf{u}_n^\top \mathbf{w}_n)^2\right)\right)$$

$$\quad\quad \times \mathbb{E}\left[\frac{p}{p-1}\left(\frac{1}{p}\sum_j f_1(\hat{\lambda}_j)\right)\left(\frac{1}{p}\sum_j f_2(\hat{\lambda}_j)\right) - \frac{1}{p(p-1)}\sum_j f_1(\hat{\lambda}_j)f_2(\hat{\lambda}_j)\right]$$

$$= \frac{1}{p+2} \left( \|\mathbf{u}_n\|_2^2 \|\mathbf{w}_n\|_2^2 + 2(\mathbf{u}_n^\top \mathbf{w}_n)^2 \right) \mathbb{E}\left[ \int f_1(x) f_2(x)\, d\hat{\pi}_n(x) \right]$$

$$+ \left( \|\mathbf{u}_n\|_2^2 \|\mathbf{w}_n\|_2^2 - \frac{1}{p+2} \left( \|\mathbf{u}_n\|_2^2 \|\mathbf{w}_n\|_2^2 + 2(\mathbf{u}_n^\top \mathbf{w}_n)^2 \right) \right)$$

$$\times \mathbb{E}\left[ \frac{p}{p-1} \left( \int f_1(x)\, d\hat{\pi}_n(x) \right) \left( \int f_2(x)\, d\hat{\pi}_n(x) \right) - \frac{1}{(p-1)} \int f_1(x) f_2(x)\, d\hat{\pi}_n(x) \right]$$

Therefore, it is immediate that:

$$\lim_{n\uparrow\infty} \mathbb{E}\left[ \left( \mathbf{u}_n^\top \mathbf{V} f_1(\Lambda) \mathbf{V}^\top \mathbf{u}_n \right) \left( \mathbf{w}_n^\top \mathbf{V} f_2(\Lambda) \mathbf{V}^\top \mathbf{w}_n \right) \right] = u^2 v^2 \int f_1(x)\, dF_{\mathrm{MP}}(x) \int f_2(x)\, dF_{\mathrm{MP}}(x).$$

This concludes the proof. □

**Lemma E.3.** *Under the same setup as Lemma E.1, we have:*

$$\lim_{n\uparrow\infty} \mathrm{cov}\left( \mathbf{u}_n^\top \hat{\mathbf{V}} f_1(\hat{\Lambda}) \hat{\mathbf{V}} \mathbf{w}_n,\ \mathbf{u}_n^\top \hat{\mathbf{V}} f_2(\hat{\Lambda}) \hat{\mathbf{V}} \mathbf{w}_n \right) = \frac{1}{c} (u^2 v^2 + \varrho^2)\, \mathrm{cov}_{X\sim F_{\mathrm{MP}}} \left( f_1(X), f_2(X) \right).$$

*Proof.* The expectation of the product is as follows:

$$\mathbb{E}\left[ \mathbf{u}_n^\top \hat{\mathbf{V}} f_1(\hat{\Lambda}) \hat{\mathbf{V}} \mathbf{w}_n \mathbf{u}_n^\top \hat{\mathbf{V}} f_2(\hat{\Lambda}) \hat{\mathbf{V}} \mathbf{w}_n \right]$$

$$= \mathbb{E}\left[ \left( \sum_{j=1}^p (\hat{v}_j^\top \mathbf{u}_n)(\hat{v}_j^\top \mathbf{w}_n) f_1(\hat{\lambda}_j) \right) \left( \sum_{j=1}^p (\hat{v}_j^\top \mathbf{u}_n)(\hat{v}_j^\top \mathbf{w}_n) f_2(\hat{\lambda}_j) \right) \right]$$

$$= \sum_{j=1}^p \mathbb{E}\left[ (\hat{v}_j^\top \mathbf{u}_n)^2 (\hat{v}_j^\top \mathbf{w}_n)^2 \right] \mathbb{E}\left[ f_1(\hat{\lambda}_j) f_2(\hat{\lambda}_j) \right]$$

$$+ \sum_{j\neq j'} \mathbb{E}\left[ (\hat{v}_j^\top \mathbf{u}_n)(\hat{v}_j^\top \mathbf{w}_n)(\hat{v}_{j'}^\top \mathbf{u}_n)(\hat{v}_{j'}^\top \mathbf{w}_n) \right] \mathbb{E}\left[ f_1(\hat{\lambda}_j) f_2(\hat{\lambda}_{j'}) \right]$$

$$= \frac{1}{p(p+2)} \left[ \|\mathbf{u}_n\|_2^2 \|\mathbf{w}_n\|_2^2 + 2(\mathbf{u}_n^\top \mathbf{w}_n)^2 \right] \sum_j \mathbb{E}\left[ f_1(\hat{\lambda}_j) f_2(\hat{\lambda}_j) \right]$$

$$+ \left[ (\mathbf{u}_n^\top \mathbf{w}_n)^2 - \frac{1}{(p+2)} \left[ \|\mathbf{u}_n\|_2^2 \|\mathbf{w}_n\|_2^2 + 2(\mathbf{u}_n^\top \mathbf{w}_n)^2 \right] \right] \frac{1}{p(p-1)} \sum_{j\neq j'} \mathbb{E}\left[ f_1(\hat{\lambda}_j) f_2(\hat{\lambda}_{j'}) \right]$$

$$= \frac{1}{p+2} \left[ \|\mathbf{u}_n\|_2^2 \|\mathbf{w}_n\|_2^2 + 2(\mathbf{u}_n^\top \mathbf{w}_n)^2 \right] \mathbb{E}\left[ \int f_1(x) f_2(x)\, d\hat{\pi}_{n,1}(x) \right]$$

$$+ \left[ (\mathbf{u}_n^\top \mathbf{w}_n)^2 - \frac{1}{(p+2)} \left[ \|\mathbf{u}_n\|_2^2 \|\mathbf{w}_n\|_2^2 + 2(\mathbf{u}_n^\top \mathbf{w}_n)^2 \right] \right]$$

$$\times \left( \frac{1}{p(p-1)} \left\{ \mathbb{E}\left[ \left( \sum_j f_1(\hat{\lambda}_j) \right) \left( \sum_j f_2(\hat{\lambda}_j) \right) \right] - \mathbb{E}\left[ \sum_j f_1(\hat{\lambda}_j) f_2(\hat{\lambda}_j) x \right] \right\} \right)$$

$$= \frac{1}{p+2} \left[ \|\mathbf{u}_n\|_2^2 \|\mathbf{w}_n\|_2^2 + 2(\mathbf{u}_n^\top \mathbf{w}_n)^2 \right] \mathbb{E}\left[ \int f_1(x) f_2(x)\, d\hat{\pi}_{n,1}(x) \right]$$

$$+ \left[ (\mathbf{u}_n^\top \mathbf{w}_n)^2 - \frac{1}{(p+2)} \left[ \|\mathbf{u}_n\|_2^2 \|\mathbf{w}_n\|_2^2 + 2(\mathbf{u}_n^\top \mathbf{w}_n)^2 \right] \right]$$

$$\times \left( \left( 1 + \frac{1}{p+1} \right) \mathbb{E}\left[ \int f_1(x)\, d\hat{\pi}_{n,1}(x) \int f_2(x)\, d\hat{\pi}_{n,1}(x) \right] - \frac{1}{p-1} \mathbb{E}\left[ \int f_1(x) f_2(x)\, d\hat{\pi}_{n,1}(x) \right] \right)$$

As for the product of the expectation:

$$\mathbb{E}\left[ \mathbf{u}_n^\top \hat{\mathbf{V}} f_1(\hat{\Lambda}) \hat{\mathbf{V}} \mathbf{w}_n \right] \mathbb{E}\left[ \mathbf{u}_n^\top \hat{\mathbf{V}} f_2(\hat{\Lambda}) \hat{\mathbf{V}} \mathbf{w}_n \right]$$

$$= (\mathbf{u}_n^\top \mathbf{w}_n)^2 \mathbb{E}\left[ \int f_1(x)\, d\hat{\pi}_{n,1}(x) \right] \mathbb{E}\left[ \int f_2(x)\, d\hat{\pi}_{n,1}(x) \right]$$

As a consequence, subtracting the product of the expectation from the expectation of product, we obtain:

$$\text{cov}\left(\mathbf{u}_n^\top \hat{\mathbf{V}} f_1(\hat{\Lambda})\hat{\mathbf{V}}\mathbf{w}_n,\ \mathbf{u}_n^\top \hat{\mathbf{V}} f_2(\hat{\Lambda})\hat{\mathbf{V}}\mathbf{w}_n\right)$$

$$= (\mathbf{u}_n^\top \mathbf{w}_n)^2 \mathbb{E}\left[\text{cov}_{\hat{\pi}_{n,1}}(f_1(x)f_2(x))\right] + \mathbb{E}\left[\int f_1(x)f_2(x)\,d\hat{\pi}_{n,1}(x)\right]$$

$$\times \left\{\frac{1}{p+2}\left[\|\mathbf{u}_n\|_2^2\|\mathbf{w}_n\|_2^2 + 2(\mathbf{u}_n^\top \mathbf{w}_n)^2\right] - \frac{1}{p-1}\left[(\mathbf{u}_n^\top \mathbf{w}_n)^2 - \frac{1}{(p+2)}\left[\|\mathbf{u}_n\|_2^2\|\mathbf{w}_n\|_2^2 + 2(\mathbf{u}_n^\top \mathbf{w}_n)^2\right]\right]\right\}$$

$$+ \frac{1}{p+1}(\mathbf{u}_n^\top \mathbf{w}_n)^2 \mathbb{E}\left[\int f_1(x)\,d\hat{\pi}_{n,1}(x)\int f_2(x)\,d\hat{\pi}_{n,1}(x)\right]$$

$$- \frac{p}{(p+1)(p+2)}\left[\|\mathbf{u}_n\|_2^2\|\mathbf{w}_n\|_2^2 + 2(\mathbf{u}_n^\top \mathbf{w}_n)^2\right]\mathbb{E}\left[\int f_1(x)\,d\hat{\pi}_{n,1}(x)\int f_2(x)\,d\hat{\pi}_{n,1}(x)\right]$$

From Theorem 1.1 of [28], we have:

$$\lim_{n\uparrow\infty} n \times \mathbb{E}\left[\text{cov}_{\hat{\pi}_{n,1}}(f_1(x)f_2(x))\right] = 0.$$

Hence, we have:

$$\lim_{n\uparrow\infty} n \times \text{cov}\left(\mathbf{u}_n^\top \hat{\mathbf{V}} f_1(\hat{\Lambda})\hat{\mathbf{V}}\mathbf{w}_n,\ \mathbf{u}_n^\top \hat{\mathbf{V}} f_2(\hat{\Lambda})\hat{\mathbf{V}}\mathbf{w}_n\right)$$

$$= \frac{1}{c}(u^2 v^2 + \varrho^2)\text{cov}_{X\sim F_{\text{MP}}}(f_1(X), f_2(X)).$$

This completes the proof. $\qquad\square$

## F   Prediction-Optimal Tuning Parameters

In this section, we derive the prediction-optimal tuning parameters $\lambda_1, \lambda_2$. Throughout, denote the nuisance functions $p(X) = X^\top \alpha_0$ and $b(X) = X^\top \beta_0$ and the ridge-regression based estimators $\hat{p}_{\lambda_1}(X) = X^\top \hat{\alpha}(\lambda_1)$ and $\hat{b}_{\lambda_2}(X) = X^\top \hat{\beta}(\lambda_2)$.

**Lemma F.1.** *Under the conditions of Theorem 3.4, the mean squared errors of the estimators of the nuisance functions $p$ and $b$ are given by*

$$\mathbb{E}[(\hat{p}_{\lambda_1}(X) - p(X))^2] = \|\alpha_0\|_2^2 \left(1 + \int \frac{x^2}{(x+\lambda_1)^2}\,dF_{\text{MP}}(x) - 2\int \frac{x}{x+\lambda_1}\,dF_{\text{MP}}(x)\right)$$

$$+ c\int \frac{x}{(x+\lambda_1)^2}\,dF_{\text{MP}}(x) + o(1)$$

$$\mathbb{E}[(\hat{b}_{\lambda_2}(X) - b(X))^2] = \|\beta_0\|_2^2 \left(1 + \int \frac{x^2}{(x+\lambda_2)^2}\,dF_{\text{MP}}(x) - 2\int \frac{x}{x+\lambda_2}\,dF_{\text{MP}}(x)\right)$$

$$+ c\int \frac{x}{(x+\lambda_2)^2}\,dF_{\text{MP}}(x) + o(1)$$

**Lemma F.2.** *The limiting mean squared error of the estimator of the nuisance function $p$ given in Lemma F.1 is minimized (with respect to $\lambda_1$) at $\lambda_1 = c/u^2$ where $u = \lim_{n\uparrow\infty}\|\alpha_0\|_2$. Similarly, the limiting mean squared error of the estimator of $b$ is $c/v^2$ where $v = \lim_{n\uparrow\infty}\|\beta_0\|_2$.*

*Proof of Lemma F.1.* It is straightforward to see that

$$\mathbb{E}[(\hat{p}_{\lambda_1}(X) - p(X))^2] = \mathbb{E}[\|\hat{\alpha}(\lambda_1) - \alpha_0\|_2^2].$$

Plugging in $\hat{\alpha}(\lambda_1) = (\hat{\Sigma}_1 + \lambda_1 \mathbf{I}_p)^{-1}\frac{\mathbf{X}^\top \mathbf{a}}{n}$, we have that

$$\mathbb{E}[\|\hat{\alpha}(\lambda_1) - \alpha_0\|_2^2] = \mathbb{E}\left[\left\|(\hat{\Sigma}_1 + \lambda_1 \mathbf{I}_p)^{-1}\frac{\mathbf{X}^\top \mathbf{a}}{n} - \alpha_0\right\|_2^2\right]$$

$$= \mathbb{E}\left[\left\|(\hat{\Sigma} + \lambda_1 \mathbf{I}_p)^{-1}\hat{\Sigma}\alpha_0 + (\hat{\Sigma} + \lambda_1 \mathbf{I}_p)^{-1}\mathbf{X}^\top \boldsymbol{\epsilon}/n - \alpha_0\right\|_2^2\right]$$

$$= \mathbb{E}\left[\alpha_0^\top \hat{\Sigma}(\hat{\Sigma} + \lambda_1 \mathbf{I}_p)^{-2}\hat{\Sigma}\alpha_0\right] - 2\mathbb{E}\left[\alpha_0^\top \hat{\Sigma}(\hat{\Sigma} + \lambda_1 \mathbf{I}_p)^{-1}\alpha_0\right]$$
$$+ \frac{2}{n}\mathbb{E}\left[\alpha_0^\top \hat{\Sigma}(\hat{\Sigma} + \lambda_1 \mathbf{I}_p)^{-1}\mathbf{X}^\top \boldsymbol{\epsilon}\right] + \|\alpha_0\|_2^2 - 2\mathbb{E}\left[\alpha_0^\top \boldsymbol{\epsilon}\right]$$
$$+ \mathbb{E}\left[\frac{1}{n^2}\boldsymbol{\epsilon}^\top \mathbf{X}(\hat{\Sigma} + \lambda_1 \mathbf{I}_p)^{-2}\mathbf{X}^\top \boldsymbol{\epsilon}\right].$$

Observe that $\mathbb{E}\left[\alpha_0^\top \boldsymbol{\epsilon}\right] = 0$ and $\mathbb{E}\left[\alpha_0^\top \hat{\Sigma}(\hat{\Sigma} + \lambda_1 \mathbf{I}_p)^{-1}\mathbf{X}^\top \boldsymbol{\epsilon}\right] = 0$. Recall from Section D.2 that

$$\mathbb{E}\left[\alpha_0^\top \hat{\Sigma}(\hat{\Sigma} + \lambda_1 \mathbf{I}_p)^{-2}\hat{\Sigma}\alpha_0\right] = \|\alpha_0\|_2^2 \left(\int \frac{x^2}{(x+\lambda)^2}\, dF_{\mathrm{MP}}(x) + o(1)\right).$$

In the same manner,

$$\mathbb{E}\left[\alpha_0^\top \hat{\Sigma}(\hat{\Sigma} + \lambda_1 \mathbf{I}_p)^{-1}\alpha_0\right] = \|\alpha_0\|_2^2 \mathbb{E}\left[\int \frac{x}{x+\lambda}\, d\hat{\pi}_n(x)\right]$$
$$= \|\alpha_0\|_2^2 \left[\int \frac{x}{x+\lambda}\, dF_{\mathrm{MP}}(x) + o(1)\right].$$

Finally, it follows from the same steps used in the analysis of $T_4$ in Section D.2 that

$$\mathbb{E}\left[\frac{1}{n^2}\boldsymbol{\epsilon}^\top \mathbf{X}(\hat{\Sigma} + \lambda_1 \mathbf{I}_p)^{-2}\mathbf{X}^\top \boldsymbol{\epsilon}\right] = \frac{1}{n}\mathbb{E}\left[\sum_{j=1}^p \frac{\hat{\lambda}_j}{(\hat{\lambda}_j + \lambda_1)^2}\right]$$
$$= \frac{c}{p}\mathbb{E}\left[\sum_{j=1}^p \frac{\hat{\lambda}_j}{(\hat{\lambda}_j + \lambda_1)^2}\right]$$
$$= c\mathbb{E}\left[\int \frac{x}{(x+\lambda_1)^2}\, d\hat{\pi}_n(x)\right]$$
$$= c\int \frac{x}{(x+\lambda_1)^2}\, dF_{\mathrm{MP}}(x) + o(1),$$

which concludes the proof of the lemma.

$\square$

*Proof of Lemma F.2.* We here prove the limiting prediction risk of $\hat{p}_{\lambda_1}(X)$ is minimized at $\lambda_1^* = c/u^2$, where $u = \lim_{n\uparrow\infty}\|\alpha_0\|_2$. The proof for the minimizer of the limiting prediction risk of $\hat{b}_{\lambda_2}(X)$ is the same and hence skipped. First of all, observe that we can rewrite the prediction risk of $\hat{p}_{\lambda_1}(X)$ as follows:

$$\lim_{n\uparrow\infty}\mathbb{E}[(\hat{p}_{\lambda_1}(X) - p(X))^2] = u^2\left(1 + \int \frac{x^2\, dF_{\mathrm{MP}}(x)}{(x+\lambda_1)^2} - 2\int \frac{x\, dF_{\mathrm{MP}}(x)}{x+\lambda_1}\right) + c\int \frac{x\, dF_{\mathrm{MP}}(x)}{(x+\lambda_1)^2}$$
$$= u^2\left(\int \left(1 - \frac{x}{x+\lambda_1}\right)^2 dF_{\mathrm{MP}}(x)\right) + c\int \frac{x\, dF_{\mathrm{MP}}(x)}{(x+\lambda_1)^2}$$
$$= u^2 \int \frac{\lambda_1^2\, dF_{\mathrm{MP}}(x)}{(x+\lambda_1)^2} + c\int \frac{x\, dF_{\mathrm{MP}}(x)}{(x+\lambda_1)^2}$$
$$= u^2\left[\int \frac{\lambda_1^2 + (c/u^2)x}{(x+\lambda_1)^2}\, dF_{\mathrm{MP}}(x)\right] := u^2 g(\lambda_1).$$

Our goal is to obtain $\lambda_1^*$ such that:

$$\lambda_1^* = \arg\min_{\lambda_1} g(\lambda_1).$$

The derivative of $g$ with respect to $\lambda_1$ is:

$$g'(\lambda_1) = \frac{d}{d\lambda_1}\int \frac{\lambda_1^2 + (c/u^2)x}{(x+\lambda_1)^2}\, dF_{\mathrm{MP}}(x)$$

$$= \int \frac{(x+\lambda_1)^2 2\lambda_1 - (\lambda_1^2 + (c/u^2)x)2(x+\lambda_1)}{(x+\lambda_1)^4}\, dF_{\mathrm{MP}}(x)$$

$$= 2\int \frac{(x+\lambda_1)\lambda_1 - (\lambda_1^2 + (c/u^2)x)}{(x+\lambda_1)^3}\, dF_{\mathrm{MP}}(x)$$

$$= 2\int \frac{\lambda_1 x - (c/u^2)x}{(x+\lambda_1)^3}\, dF_{\mathrm{MP}}(x)$$

$$= 2(\lambda_1 - (c/u^2))\int \frac{x\, dF_{\mathrm{MP}}(x)}{(x+\lambda_1)^3}\, .$$

Therefore, the derivative is 0 when $\lambda_1 = c/u^2$. Furthermore,

$$g''(\lambda_1) = -6(\lambda_1 - (c/u^2))\int \frac{x\, dF_{\mathrm{MP}}(x)}{(x+\lambda_1)^4} + 2\int \frac{x\, dF_{\mathrm{MP}}(x)}{(x+\lambda_1)^3}\, ,$$

which implies,

$$g''(c/u^2) = 2\int \frac{x\, dF_{\mathrm{MP}}(x)}{(x+(c/u^2))^3} > 0\, .$$

Hence $g(\lambda_1)$ is minimized at $\lambda_1^* = c/u^2$. This completes the proof. $\qquad\square$

## G  Parametric Bootstrap

We use a parametric bootstrap approach to estimate the large-sample variance of our debiased estimators. For illustration purpose, we consider $\hat{\theta}_{3\mathrm{sp}}^{\mathrm{db,INT}}$, which is defined as:

$$\hat{\rho} = \hat{\theta}_{3\mathrm{sp}}^{\mathrm{db,INT}} = \frac{1}{n}\sum_{i\in\mathcal{D}_3} Y_i A_i - \frac{\hat{\alpha}(\lambda_1)^\top \hat{\beta}(\lambda_2)}{g_{3sp}^{\mathrm{INT}}(\lambda_1,\lambda_2)}$$

where $g_{3\mathrm{sp}}^{\mathrm{INT}}(\lambda_1,\lambda_2) = \left(\int \frac{x\, dF_{\mathrm{MP}}(x)}{x+\lambda_1}\right)\left(\int \frac{x\, dF_{\mathrm{MP}}(x)}{x+\lambda_2}\right)$ and $F_{\mathrm{MP}}$ denotes the Marchenko–Pastur law with aspect ratio $c = p/n$.

**Bootstrap algorithm (3-split).**  Given $\hat{\rho}$ and the ridge estimators $(\hat{\alpha}(\lambda_1), \hat{\beta}(\lambda_2))$, we define the following:

1. Define the constants $(c_1, d_1, c_2, d_2, c_3)$ as:

$$c_j = \int \frac{x^2}{(x+\lambda_j)2}\, dF_{\mathrm{MP}}(x), \qquad j \in \{1,2\}$$

$$d_j = (p/n)\int \frac{x}{(x+\lambda_j)^2}\, dF_{\mathrm{MP}}(x), \qquad j \in \{1,2\}$$

$$c_3 = \int \frac{x}{x+\lambda_1}\, dF_{\mathrm{MP}}(x) \int \frac{x}{x+\lambda_2}\, dF_{\mathrm{MP}}(x)\, .$$

Then define two *transformed regression* coefficients $\tilde{\alpha}$ and $\tilde{\beta}$ as:

$$\tilde{\alpha} = \frac{\hat{\alpha}(\lambda_1)}{\|\hat{\alpha}(\lambda_1)\|_2}\sqrt{\frac{\|\hat{\alpha}(\lambda_1)\|_2^2 - d_1}{c_1}}$$

$$\tilde{\beta} = \frac{\hat{\alpha}(\lambda_1)}{\|\hat{\alpha}(\lambda_1)\|_2}\left(\frac{(\hat{\alpha}(\lambda_1)^\top \hat{\beta}(\lambda_2))}{c_3}\frac{1}{\sqrt{\frac{\|\hat{\alpha}(\lambda_1)\|_2^2 - d_1}{c_1}}}\right) + \mathbf{z}\sqrt{\frac{\|\hat{\beta}(\lambda_2)\|_2^2 - d_2}{c_2} - \left(\frac{(\hat{\alpha}(\lambda_1)^\top \hat{\beta}(\lambda_2))}{c_3}\frac{1}{\sqrt{\frac{\|\hat{\alpha}(\lambda_1)\|_2^2 - d_1}{c_1}}}\right)^2}$$

where $\mathbf{z}$ is some unit vector perpendicular to $\hat{\alpha}(\lambda_1)$.

2. Generate $X_1^{(b)}, \ldots, X_n^{(b)} \sim \mathcal{N}(0, I_p)$.

3. Generate $(\epsilon_1^{(b)}, \eta_1^{(b)}), \ldots, (\epsilon_n^{(b)}, \eta_n^{(b)}) \sim \mathcal{N}\left(\begin{pmatrix}0\\0\end{pmatrix}, \begin{pmatrix}1 & \hat{\rho}\\ \hat{\rho} & 1\end{pmatrix}\right)$.

4. Set $A_i^{(b)} = X_i^{(b)^\top} \tilde{\alpha} + \epsilon_i^{(b)}$ and $Y_i^{(b)} = X_i^{(b)^\top} \tilde{\beta} + \eta_i^{(b)}$ for $1 \leq i \leq n$.

5. Estimate $\hat{\theta}_{3sp}^{(b),\mathrm{db,INT}}$ using $\{(X_i^{(b)}, Y_i^{(b)}, A_i^{(b)})\}_{1 \leq i \leq n}$.

6. Repeat Step 2-5 multiple (say $B$) times and return:

$$\widehat{\mathrm{var}} = \frac{1}{B} \sum_{b=1}^{B} \left( \hat{\theta}_{3sp}^{(b),\mathrm{db,INT}} \right)^2 - \left( \frac{1}{B} \sum_{b=1}^{B} \hat{\theta}_{3sp}^{(b),\mathrm{db,INT}} \right)^2 .$$

**Intuition:** Before delving into the technical details, we begin by presenting some intuition behind the motivation for introducing the transformed vectors $(\tilde{\alpha}, \tilde{\beta})$. To start, recall that the limiting variance of $\hat{\theta}_{3sp}^{\mathrm{db,INT}}$ depends on seven variables:

$$\lim_{n \uparrow \infty} n \times \mathrm{var} \left( \hat{\theta}_{3sp}^{(b),\mathrm{db,INT}} \right) = \sigma^2(\rho, \varrho, u, v, c, \lambda_1, \lambda_2) .$$

where $u = \lim_{p \uparrow \infty} \|\alpha\|_2^2$, $v = \lim_{n \uparrow \infty} \|\beta\|_2^2$, $\varrho = \lim_{n \uparrow \infty} \alpha^\top \beta$ and $c$ is the limiting value of $(p/n)$. On the other hand, if we increase the bootstrap sample size $B$, then $\widehat{\mathrm{var}}$ converges to the conditional variance of $\hat{\theta}_{3sp}^{\mathrm{db,INT}}$ given the original data, i.e.,

$$\lim_{B \uparrow \infty} \widehat{\mathrm{var}} = \mathrm{var} \left( \hat{\theta}_{3sp}^{(b),\mathrm{db,INT}} \mid \mathcal{D} \right) .$$

Therefore, the bootstrap variance estimation will be consistent if we can show that:

$$\lim_{n \uparrow \infty} \mathrm{var} \left( \hat{\theta}_{3sp}^{(b),\mathrm{db,INT}} \mid \mathcal{D} \right) \overset{\mathrm{a.s.}}{=} \sigma^2(\rho, \varrho, u, v, c, \lambda_1, \lambda_2) . \tag{G.1}$$

However there is a small issue here: we know that neither $\|\hat{\alpha}(\lambda_1)\|_2^2$ nor $\|\hat{\beta}(\lambda_2)\|_2^2$ gives us a consistent estimate of $u$ of $v$. In particular we can show that:

$$\lim_{n \uparrow \infty} \|\hat{\alpha}(\lambda_1)\|_2^2 \overset{a.s.}{=} c_1 u + d_1$$

$$\lim_{n \uparrow \infty} \|\hat{\beta}(\lambda_2)\|_2^2 \overset{a.s.}{=} c_2 u + d_2$$

$$\lim_{n \uparrow \infty} \hat{\alpha}(\lambda_1)^\top \hat{\beta}(\lambda_2) \overset{a.s.}{=} c_3 \varrho .$$

where $(c_1, d_1, c_2, d_2, c_3)$ is same defined in the Step 1 of our bootstrap algorithm. As a consequence, one can show that:

$$\lim_{n \uparrow \infty} \mathrm{var} \left( \hat{\theta}_{3sp}^{(b),\mathrm{db,INT}} \mid \mathcal{D} \right) \overset{\mathrm{a.s.}}{=} \sigma^2(\rho, c_3\varrho, c_1 u + d_1, c_2 v + d_2, c, \lambda_1, \lambda_2)$$

which does not satisfy the consistency, as required in Equation (G.1). Therefore, we need to construct a transformation of $(\hat{\alpha}(\lambda_1), \hat{\beta}(\lambda_2))$ that yields a consistent estimate of $(u, v, \varrho)$. To that end, consider the definition of $(\tilde{\alpha}, \tilde{\beta})$ provided in Step 1 of the bootstrap algorithm. We have:

$$\lim_{n \uparrow \infty} \|\tilde{\alpha}\|_2^2 = \lim_{n \uparrow \infty} \left\| \frac{\hat{\alpha}(\lambda_1)}{\|\hat{\alpha}(\lambda_1)\|_2} \sqrt{\frac{\|\hat{\alpha}(\lambda_1)\|_2^2 - d_1}{c_1}} \right\|_2^2$$

$$= \lim_{n \uparrow \infty} \frac{\|\hat{\alpha}(\lambda_1)\|_2^2 - d_1}{c_1}$$

$$= \frac{c_1 u + d_1 - d_1}{c_1} = u .$$

$$\lim_{n \uparrow \infty} \|\tilde{\beta}\|_2^2 \tag{G.2}$$

$$= \lim_{n \uparrow \infty} \left\| \frac{\hat{\alpha}(\lambda_1)}{\|\hat{\alpha}(\lambda_1)\|_2} \left( \frac{(\hat{\alpha}(\lambda_1)^\top \hat{\beta}(\lambda_2))}{c_3} \frac{1}{\sqrt{\frac{\|\hat{\alpha}(\lambda_1)\|_2^2 - d_1}{c_1}}} \right) + \mathbf{z} \sqrt{\frac{\|\hat{\beta}(\lambda_2)\|_2^2 - d_2}{c_2} - \left( \frac{(\hat{\alpha}(\lambda_1)^\top \hat{\beta}(\lambda_2))}{c_3} \frac{1}{\sqrt{\frac{\|\hat{\alpha}(\lambda_1)\|_2^2 - d_1}{c_1}}} \right)^2} \right\|_2^2$$

$$\tag{G.3}$$

$$= \lim_{n\uparrow\infty} \left[ \left( \frac{(\hat{\alpha}(\lambda_1)^\top \hat{\beta}(\lambda_2))}{c_3} \frac{1}{\sqrt{\frac{\|\hat{\alpha}(\lambda_1)\|_2^2 - d_1}{c_1}}} \right)^2 + \left\{ \frac{\|\hat{\beta}(\lambda_2)\|_2^2 - d_2}{c_2} - \left( \frac{(\hat{\alpha}(\lambda_1)^\top \hat{\beta}(\lambda_2))}{c_3} \frac{1}{\sqrt{\frac{\|\hat{\alpha}(\lambda_1)\|_2^2 - d_1}{c_1}}} \right)^2 \right\} \right] \tag{G.4}$$

$$= \lim_{n\uparrow\infty} \frac{\|\hat{\beta}(\lambda_2)\|_2^2 - d_2}{c_2} \tag{G.5}$$

$$= \frac{c_2 v + d_2 - d_2}{c_2} = v \tag{G.6}$$

Last, but not least,

$$\lim_{n\uparrow\infty} \tilde{\alpha}^\top \tilde{\beta}$$

$$= \lim_{n\uparrow\infty} \left\langle \frac{\hat{\alpha}(\lambda_1)}{\|\hat{\alpha}(\lambda_1)\|_2} \sqrt{\frac{\|\hat{\alpha}(\lambda_1)\|_2^2 - d_1}{c_1}}, \; \frac{\hat{\alpha}(\lambda_1)}{\|\hat{\alpha}(\lambda_1)\|_2} \left( \frac{(\hat{\alpha}(\lambda_1)^\top \hat{\beta}(\lambda_2))}{c_3} \frac{1}{\sqrt{\frac{\|\hat{\alpha}(\lambda_1)\|_2^2 - d_1}{c_1}}} \right) \right\rangle$$

$$= \lim_{n\uparrow\infty} \frac{(\hat{\alpha}(\lambda_1)^\top \hat{\beta}(\lambda_2))}{c_3} = \varrho \,.$$

Therefore, if we use these transformed vectors $(\tilde{\alpha}, \tilde{\beta})$ in our bootstrap algorithm, we expect the bootstrap variance to converge to the right limit.

For the other estimators in the three-split regime, we apply the same three-split bootstrap procedure described above. The only difference lies in the specific form of the estimators themselves (i.e., the analytical expressions for $\hat{\theta}_{3\mathrm{sp}}^{\mathrm{db,NR}}$ and $\hat{\theta}_{3\mathrm{sp}}^{\mathrm{db,DR}}$), while the underlying bootstrap mechanism and data-generating steps remain identical.

$$\lim_{n\uparrow\infty} n \times \mathsf{var}\left( \hat{\theta}_{3sp}^{(b),\mathrm{db,INT/NR/DR}} \right) = \sigma^2(\rho, \varrho, u, v, c, \lambda_1, \lambda_2) \,.$$

Since all three estimators share the same asymptotic structure and depend on the same limiting quantities $(u, v, \varrho)$. Thus, the consistency argument for the bootstrap variance estimation holds in exactly the same way for NR and DR as for the INT estimator.

## G.1 Two-Split Parametric Bootstrap

With two splits, the norms obey the same limits, but the cross term changes. Let

$$\hat{\alpha}(\lambda_1)^\top \hat{\beta}(\lambda_2) \xrightarrow{\text{a.s.}} g_1^{\mathrm{INT}}(\lambda_1, \lambda_2)\,\varrho + g_2^{\mathrm{INT}}(\lambda_1, \lambda_2)\,\rho,$$

where we have $g_{1,2\mathrm{sp}}^{\mathrm{INT}} = \int \frac{x^2\, dF_{\mathrm{MP}}(x)}{(x+\lambda_1)(x+\lambda_2)}, g_{2,2\mathrm{sp}}^{\mathrm{INT}}(\lambda_1, \lambda_2) = \int \frac{cx\, dF_{\mathrm{MP}}(x)}{(x+\lambda_1)(x+\lambda_2)}$

$$g_1^{\mathrm{INT}}(\lambda_1, \lambda_2) = \int \frac{x^2}{(x+\lambda_1)(x+\lambda_2)}\, dF_{\mathrm{MP}}(x), \qquad g_2^{\mathrm{INT}}(\lambda_1, \lambda_2) = \int \frac{c\,x}{(x+\lambda_1)(x+\lambda_2)}\, dF_{\mathrm{MP}}(x).$$

Let $\hat{\mathbf{a}} := \hat{\alpha}(\lambda_1)/\|\hat{\alpha}(\lambda_1)\|_2$ and let $\mathbf{z}$ be any unit vector orthogonal to $\hat{\mathbf{a}}$. Define

$$\tilde{\alpha} := \hat{\mathbf{a}} \sqrt{\frac{\|\hat{\alpha}(\lambda_1)\|_2^2 - d_1}{c_1}}, \tag{G.7}$$

$$\tilde{\beta} := \hat{\mathbf{a}} \left( \frac{\hat{\alpha}(\lambda_1)^\top \hat{\beta}(\lambda_2) - g_2\hat{\rho}}{g_1} \frac{1}{\sqrt{\frac{\|\hat{\alpha}(\lambda_1)\|_2^2 - d_1}{c_1}}} \right) + \mathbf{z} \sqrt{\frac{\|\hat{\beta}(\lambda_2)\|_2^2 - d_2}{c_2} - \left( \frac{\hat{\alpha}(\lambda_1)^\top \hat{\beta}(\lambda_2) - g_2\hat{\rho}}{g_1} \frac{1}{\sqrt{\frac{\|\hat{\alpha}(\lambda_1)\|_2^2 - d_1}{c_1}}} \right)^2 } \,.$$

We rewrite

$$g_1 = g_1^{\mathrm{INT}}(\lambda_1, \lambda_2) \quad g_2 = g_2^{\mathrm{INT}}(\lambda_1, \lambda_2)$$

$$\|\hat{\alpha}(\lambda_1)\|_2^2 \xrightarrow{\text{a.s.}} c_1 u + d_1, \qquad \|\hat{\beta}(\lambda_2)\|_2^2 \xrightarrow{\text{a.s.}} c_2 v + d_2, \qquad \hat{\alpha}(\lambda_1)^\top \hat{\beta}(\lambda_2) \xrightarrow{\text{a.s.}} g_1\varrho + g_2\rho,$$

$$\lim_{n\uparrow\infty} \|\tilde{\alpha}\|_2^2 = \lim_{n\uparrow\infty} \left\| \frac{\hat{\alpha}(\lambda_1)}{\|\hat{\alpha}(\lambda_1)\|_2} \sqrt{\frac{\|\hat{\alpha}(\lambda_1)\|_2^2 - d_1}{c_1}} \right\|_2^2$$

$$= \lim_{n\uparrow\infty} \frac{\|\hat{\alpha}(\lambda_1)\|_2^2 - d_1}{c_1} = \frac{c_1 u + d_1 - d_1}{c_1} = u \,.$$

$$\lim_{n\uparrow\infty} \|\tilde{\beta}\|_2^2 \tag{G.8}$$

$$= \lim_{n\uparrow\infty} \left\| \frac{\hat{\alpha}(\lambda_1)}{\|\hat{\alpha}(\lambda_1)\|_2} \left( \frac{\hat{\alpha}(\lambda_1)^\top \hat{\beta}(\lambda_2) - g_2 \hat{\rho}}{g_1} \frac{1}{\sqrt{\frac{\|\hat{\alpha}(\lambda_1)\|_2^2 - d_1}{c_1}}} \right) + \mathbf{z} \sqrt{\frac{\|\hat{\beta}(\lambda_2)\|_2^2 - d_2}{c_2} - \left( \frac{\hat{\alpha}(\lambda_1)^\top \hat{\beta}(\lambda_2) - g_2 \hat{\rho}}{g_1} \frac{1}{\sqrt{\frac{\|\hat{\alpha}(\lambda_1)\|_2^2 - d_1}{c_1}}} \right)^2} \right\|_2^2 \tag{G.9}$$

$$= \lim_{n\uparrow\infty} \left[ \left( \frac{\hat{\alpha}(\lambda_1)^\top \hat{\beta}(\lambda_2) - g_2 \hat{\rho}}{g_1} \frac{1}{\sqrt{\frac{\|\hat{\alpha}(\lambda_1)\|_2^2 - d_1}{c_1}}} \right)^2 + \left\{ \frac{\|\hat{\beta}(\lambda_2)\|_2^2 - d_2}{c_2} - \left( \frac{\hat{\alpha}(\lambda_1)^\top \hat{\beta}(\lambda_2) - g_2 \hat{\rho}}{g_1} \frac{1}{\sqrt{\frac{\|\hat{\alpha}(\lambda_1)\|_2^2 - d_1}{c_1}}} \right)^2 \right\} \right] \tag{G.10}$$

$$= \lim_{n\uparrow\infty} \frac{\|\hat{\beta}(\lambda_2)\|_2^2 - d_2}{c_2} = \frac{c_2 v + d_2 - d_2}{c_2} = v \,. \tag{G.11}$$

Last, but not least,

$$\lim_{n\uparrow\infty} \tilde{\alpha}^\top \tilde{\beta} = \lim_{n\uparrow\infty} \left\langle \frac{\hat{\alpha}(\lambda_1)}{\|\hat{\alpha}(\lambda_1)\|_2} \sqrt{\frac{\|\hat{\alpha}(\lambda_1)\|_2^2 - d_1}{c_1}}, \; \frac{\hat{\alpha}(\lambda_1)}{\|\hat{\alpha}(\lambda_1)\|_2} \left( \frac{\hat{\alpha}(\lambda_1)^\top \hat{\beta}(\lambda_2) - g_2 \hat{\rho}}{g_1} \frac{1}{\sqrt{\frac{\|\hat{\alpha}(\lambda_1)\|_2^2 - d_1}{c_1}}} \right) \right\rangle$$

$$= \lim_{n\uparrow\infty} \frac{\hat{\alpha}(\lambda_1)^\top \hat{\beta}(\lambda_2) - g_2 \hat{\rho}}{g_1} = \frac{g_1 \varrho + g_2 \rho - g_2 \rho}{g_1} = \varrho \,,$$

Here, the bootstrap procedure proceeds analogously, but with one key distinction: the transformed coefficients now depend explicitly on the estimated correlation parameter $\hat{\rho}$. As a result, the algorithm requires incorporating an additional estimation step for $\hat{\rho}$ before constructing the transformed vectors $(\tilde{\alpha}, \tilde{\beta})$. Nevertheless, as established in Theorem 3.1, $\hat{\rho}$ is a consistent estimator of the true $\rho$, ensuring consistency of the bootstrap. In Section H.4 we present our simulation results, which confirm that the bootstrap variance approximates the Monte Carlo variance well.

## H  Additional Details and Results for the Simulation Study

Here, we provide additional details and results of the simulation study. These results serve the following purposes:

1. We illustrate the asymptotic bias of the integral-based estimator, Newey-Robins estimator, and doubly robust estimator, and we verify that the debiased versions of these estimators are indeed asymptotically unbiased.

2. We illustrate the asymptotic variances of the debiased estimators across a range of values for the nuisance parameters when $c = 0.5$ and $c = 2$. We then compare the optimal tuning parameters (that minimize the variance) to those that minimize the prediction error (i.e., the prediction-optimal tuning parameters), as in the simulations presented in the main text. For $c = 2$, where we do not trim the y-axis (unlike the main text).

3. We verify the derivations in Appendix F regarding the asymptotic mean squared error of the nuisance functions and the prediction-optimal tuning parameters.

4. We illustrate that the parametric bootstrap approach performs well for estimating the asymptotic variance of the debiased estimators.

We considered four scenarios by varying whether two or three sample splits are used and whether $c = 0.5$ or $c = 2$. In each scenario, recall that we generated $10,000$ independent data sets. For

each data set, we generated $N$ iid copies of $(A, X, Y)$ as follows. We let $X$ have dimension $p$ and generated $X$ by $X \sim \text{Normal}(0, \mathbf{I}_p)$. Then, we generated $A$ and $Y$ by

$$\begin{pmatrix} A \\ Y \end{pmatrix} \Big| X \sim \text{Normal}\left( \begin{pmatrix} X^\top \alpha_0 \\ X^\top \beta_0 \end{pmatrix}, \begin{pmatrix} 1 & \rho \\ \rho & 1 \end{pmatrix} \right)$$

where $\rho = 0.5$. In each scenario, the true values of $\alpha_0$ and $\beta_0$ were set by sampling its entries independently from a $\text{Uniform}(0, 1)$ distribution and then re-scaling them so that $\|\alpha_0\|_2 = \|\beta_0\|_2 = 1$.

Throughout our simulations, we consider the case where the nuisance functions are tuned identically, i.e., $\lambda_1 = \lambda_2 = \lambda$. We consider a discrete grid of 100 values for $\lambda$ (ranging from 0.05 to 10). For each of the debiased estimators, we find the value of $\lambda$ resulting in the lowest asymptotic variance for each estimator and compare it to the prediction-optimal $\lambda$ (i.e., $\lambda_1 = \lambda_2 = c$).

The code will be released at the GitHub link: https://github.com/zixiaowang17/Optimal-Nuisance-Function-Tuning.

### H.1 Bias and Variance in Settings with Two Splits

#### H.1.1 Setup

The total sample size of each data set was set to $N = 1000$ and we split the sample into two disjoint subsamples of size $n = 500$, denoted by $\mathcal{D}_1$ and $\mathcal{D}_2$. We have $p = 250$ in the setting with $c = 0.5$ and $p = 1000$ in the setting with $c = 2$. For all estimators, we estimated $\alpha_0$ and $\beta_0$ using $\mathcal{D}_1$ and estimated $\rho$ using $\mathcal{D}_3$. In this case, the non-debiased estimators are given by

$$\hat{\rho}_{2sp}^{\text{INT}} = \frac{1}{n} \sum_{i \in \mathcal{D}_2} A_i Y_i - \hat{\alpha}(\lambda)^\top \hat{\beta}(\lambda)$$

$$\hat{\rho}_{2sp}^{\text{NR}} = \frac{1}{n} \sum_{i \in \mathcal{D}_2} A_i (Y_i - X_i^\top \hat{\beta}(\lambda))$$

$$\hat{\rho}_{2sp}^{\text{DR}} = \frac{1}{n} \sum_{i \in \mathcal{D}_2} (Y_i - X_i^\top \hat{\beta}(\lambda))(A_i - X_i^\top \hat{\alpha}(\lambda))$$

and the debiased estimators are given by

$$\hat{\rho}_{2sp}^{\text{INT,db}} := \left( 1 - \frac{g_{2,2sp}^{\text{INT}}(\lambda, \lambda)}{g_{1,2sp}^{\text{INT}}(\lambda, \lambda)} \right)^{-1} \left[ \frac{1}{n} \sum_{i \in \mathcal{D}_2} A_i Y_i - \frac{\hat{\alpha}(\lambda)^\top \hat{\beta}(\lambda)}{g_{1,2sp}^{\text{INT}}(\lambda, \lambda)} \right]$$

$$\hat{\rho}_{2sp}^{\text{NR,db}} := \left( 1 - \frac{g_{2,2sp}^{\text{INT}}(\lambda, \lambda) g_{3sp}^{\text{NR}}(\lambda, \lambda)}{g_{1,2sp}^{\text{INT}}(\lambda, \lambda)} \right)^{-1} \left[ \hat{\rho}_{2sp}^{\text{NR}} - \hat{\alpha}(\lambda)^\top \hat{\beta}(\lambda) \frac{g_{2,2sp}^{\text{INT}}(\lambda, \lambda) g_{3sp}^{\text{NR}}(\lambda, \lambda)}{g_{1,2sp}^{\text{INT}}(\lambda_1, \lambda_2)} \right]$$

$$\hat{\rho}_{2sp}^{\text{DR,db}} := \left( 1 + g_{2,2sp}^{\text{INT}}(\lambda, \lambda) \left( 1 - \frac{g_{2,2sp}^{\text{DR}}(\lambda, \lambda)}{g_{1,2sp}^{\text{INT}}(\lambda, \lambda)} \right) \right)^{-1} \left[ \hat{\rho}_{2sp}^{\text{DR}} - \frac{\hat{\alpha}(\lambda)^\top \hat{\beta}(\lambda) g_{2,2sp}^{\text{DR}}(\lambda, \lambda)}{g_{1,2sp}^{\text{INT}}(\lambda, \lambda)} \right].$$

The values of the constants were set by performing Monte Carlo integration with 10,000 iterations.

#### H.1.2 Results

We first summarize the results for the setting with $c = 0.5$. The bias of the estimators is given in Figure 2. In line with Theorems 3.1, 3.2, and 3.3, the estimators without bias corrections have a considerable degree of bias and the debiased versions of the estimators have nearly zero bias. The variance of the debiased estimators are given in Figure 3. We observe a clear difference between the value of $\lambda$ for minimizing the variance of the estimator and the prediction-optimal $\lambda$.

The results for the setting with $c = 2$ are summarized in Figures 4 and 5. Broadly, similar conclusions held: the debiased versions of the estimators generally had close to zero bias and there were differences between the optimal versus prediction-optimal $\lambda$ values. As noted in the main text, the variance of the integral-based estimator blows up at around $\lambda = 1.48$, which may explain why the empirical bias of the debiased integral-based estimator was not 0 at that value of $\lambda$.

Interestingly, we observe that the value of $c$ can plays a role in whether the prediction-optimal $\lambda$ values are greater than or less than the optimal $\lambda$. Taking the plug-in estimator as an example, the prediction-optimal value for $\lambda$ is less than the optimal one when $c = 0.5$ but is greater than the optimal one when $c = 2$.

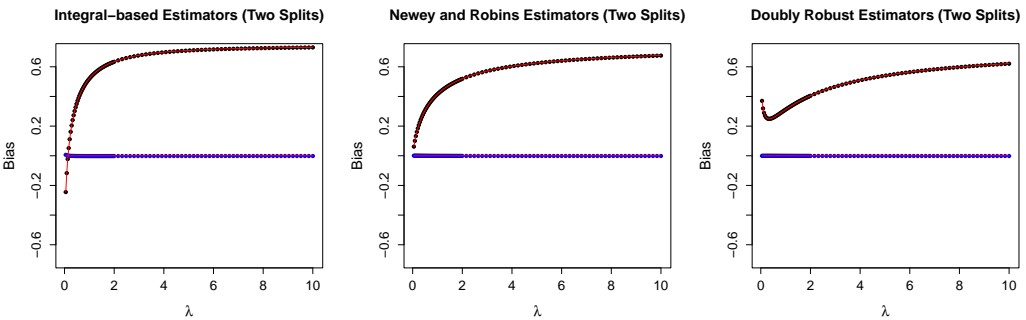

Figure 2: Bias of the non-debiased estimators (black dots) and the debiased versions (blue dots) in the setting with two splits and $c = 0.5$. The red lines illustrate the derived asymptotic bias of the estimators, with the horizontal line at zero.

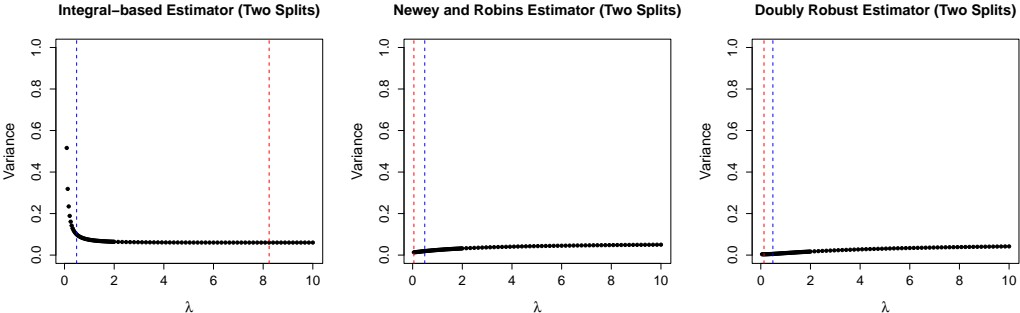

Figure 3: Variance of the debiased estimators in the setting with two splits and $c = 0.5$. The red line indicates the value of $\lambda$ resulting in the smallest asymptotic variance for estimating $\rho$; The blue line is for the prediction-optimal $\lambda$.

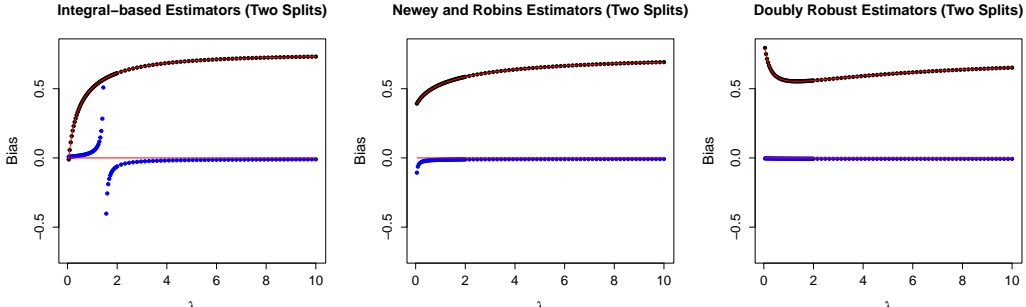

Figure 4: Bias of the non-debiased estimators (black dots) and the debiased versions (blue dots) in the setting with two splits and $c = 2$. The red lines illustrate the derived asymptotic bias of the estimators, with the horizontal line at zero.

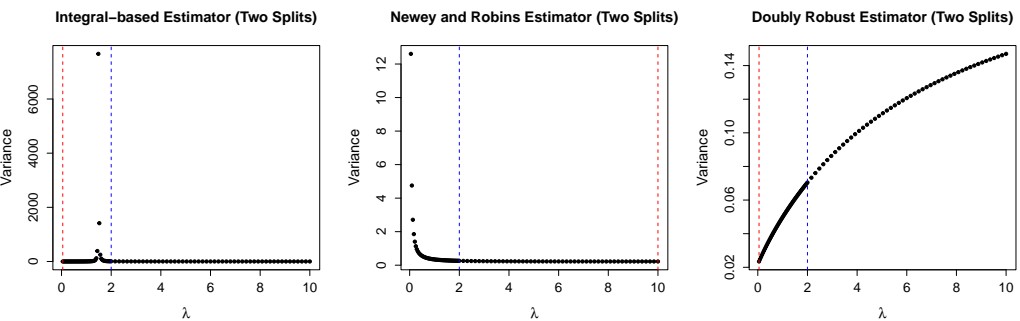

Figure 5: Variance of the debiased estimators in the setting with two splits and $c = 2$. The red line indicates the value of $\lambda$ resulting in the smallest asymptotic variance for estimating $\rho$; The blue line is for the prediction-optimal $\lambda$.

## H.2  Bias and Variance in Settings with Three Splits

### H.2.1  Setup

The total sample size of each data set was set to $N = 1500$ and we split the sample into three disjoint subsamples of size $n = 500$, denoted by $\mathcal{D}_1$, $D_2$ and $\mathcal{D}_3$. Once again, we have $p = 250$ in the setting with $c = 0.5$ and $p = 1000$ in the setting with $c = 2$. For all estimators, we estimated $\alpha_0$ using $\mathcal{D}_1$, $\beta_0$ using $\mathcal{D}_2$, and $\rho$ using $\mathcal{D}_3$. In this case, recall that the non-debiased estimators are given by

$$\hat{\rho}_{3sp}^{\text{INT}} = \frac{1}{n} \sum_{i \in \mathcal{D}_3} A_i Y_i - \hat{\alpha}(\lambda)^\top \hat{\beta}(\lambda)$$

$$\hat{\rho}_{3sp}^{\text{NR}} = \frac{1}{n} \sum_{i \in \mathcal{D}_3} A_i (Y_i - X_i^\top \hat{\beta}(\lambda))$$

$$\hat{\rho}_{3sp}^{\text{DR}} = \frac{1}{n} \sum_{i \in \mathcal{D}_3} (Y_i - X_i^\top \hat{\beta}(\lambda))(A_i - X_i^\top \hat{\alpha}(\lambda))$$

and the debiased estimators are given by

$$\hat{\rho}_{3sp}^{\text{INT,db}} = \frac{1}{n} \sum_{i \in \mathcal{D}_3} A_i Y_i - \frac{\hat{\alpha}(\lambda)^\top \hat{\beta}(\lambda)}{g_{3\text{sp}}^{\text{INT}}(\lambda, \lambda_2)}$$

$$\hat{\rho}_{3sp}^{\mathrm{NR,db}} = \hat{\rho}_{3sp}^{\mathrm{NR}} - \hat{\alpha}(\lambda)^\top \hat{\beta}(\lambda) \frac{g_{3\mathrm{sp}}^{\mathrm{NR}}(\lambda, \lambda)}{g_{3\mathrm{sp}}^{\mathrm{INT}}(\lambda, \lambda)}$$

$$\hat{\rho}_{3sp}^{\mathrm{DR,db}} = \hat{\rho}_{3sp}^{\mathrm{DR}} - \hat{\alpha}(\lambda)^\top \hat{\beta}(\lambda) \frac{g_{3\mathrm{sp}}^{\mathrm{DR}}(\lambda, \lambda)}{g_{3\mathrm{sp}}^{\mathrm{INT}}(\lambda, \lambda)} .$$

The values of the constants were set by performing Monte Carlo integration with 10,000 iterations.

### H.2.2 Results

For the setting with $c = 0.5$, we present the bias results in Figure 6 and variance results in Figure 7. The analogous results for the setting with $c = 2$ are given in Figures 8 and 9. We had similar findings in these simulations compared to the ones with three splits. That is, the debiased estimators indeed have a bias of approximately 0 and there are clear differences between the optimal and the prediction-optimal $\lambda$ values.

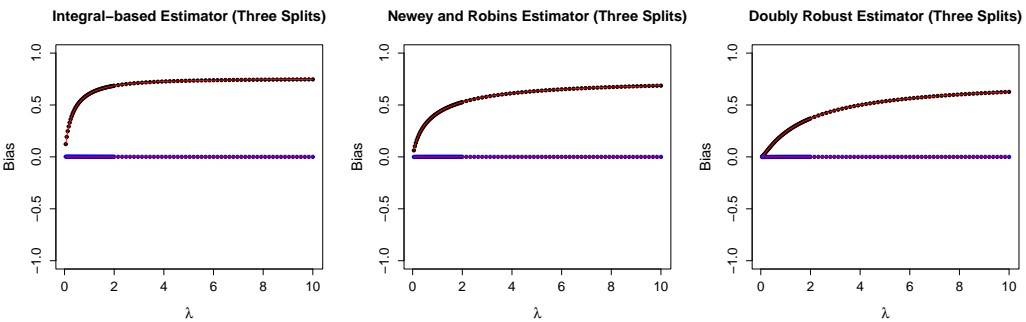

Figure 6: Bias of the non-debiased estimators (black dots) and the debiased versions (blue dots) in the setting with three splits and $c = 0.5$. The red lines illustrate the derived asymptotic bias of the estimators, with the horizontal line at zero.

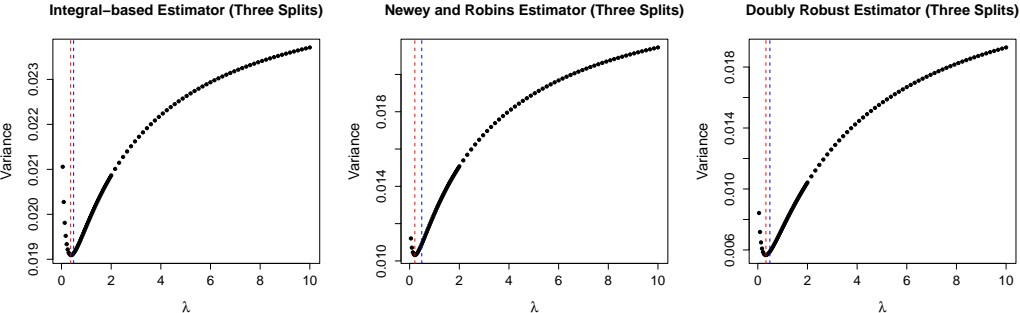

Figure 7: Variance of the debiased estimators in the setting with three splits and $c = 0.5$. The red line indicates the value of $\lambda$ resulting in the smallest asymptotic variance for estimating $\rho$; The blue line is for the prediction-optimal $\lambda$.

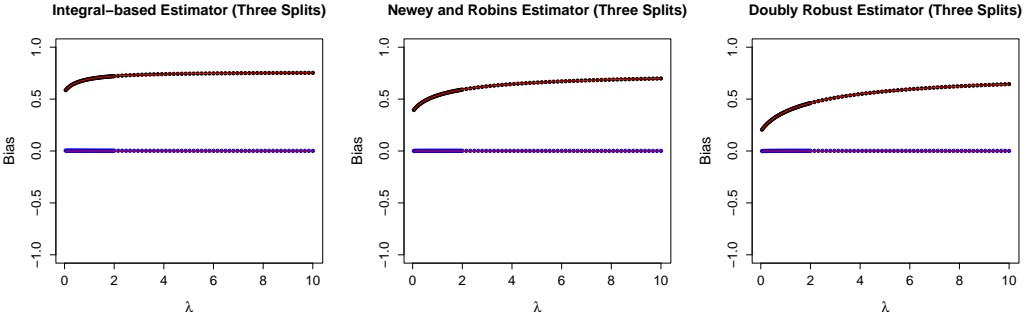

Figure 8: Bias of the non-debiased estimators (black dots) and the debiased versions (blue dots) in the setting with three splits and $c = 2$. The red lines illustrate the derived asymptotic bias of the estimators, with the horizontal line at zero.

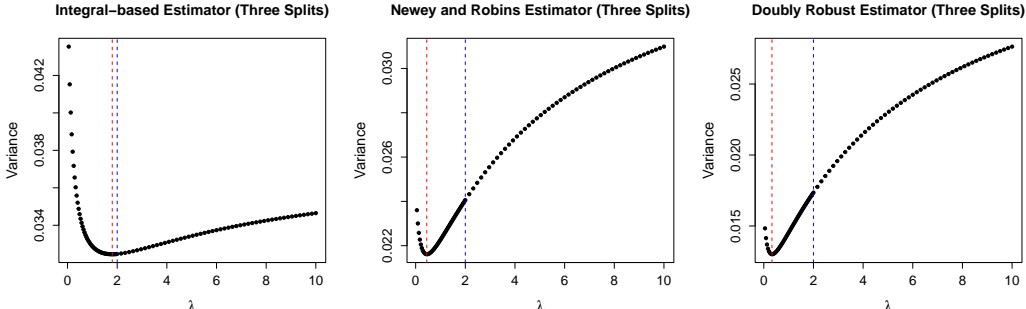

Figure 9: Variance of the debiased estimators in the setting with three splits and $c = 2$. The red line indicates the value of $\lambda$ resulting in the smallest asymptotic variance for estimating $\rho$; The blue line is for the prediction-optimal $\lambda$.

## H.3 Verification of Prediction-Optimal Tuning Parameters

Figure 10 presents the mean squared error (MSE) profiles for estimating the nuisance functions $p(X) = X^\top \alpha_0$ (left panel) and $b(X) = X^\top \beta_0$ (right panel) across a range of regularization parameters $\lambda$ in the setting with $c = 0.5$. In each panel, black points represent the empirical average prediction error across simulations, while the red curve shows the derived value of the limiting MSE (Lemma F.1). The vertical red dashed line indicates the value of $\lambda$ that minimizes the empirical MSE. As shown in Lemma F.2, the value of $\lambda$ that minimizes the MSE is $c/\|\alpha_0\|_2^2 = c/\|\beta_0\|_2^2 = 0.5$ for each nuisance function. We present the results for $c = 2$ in Figure 11, where similar conclusions held.

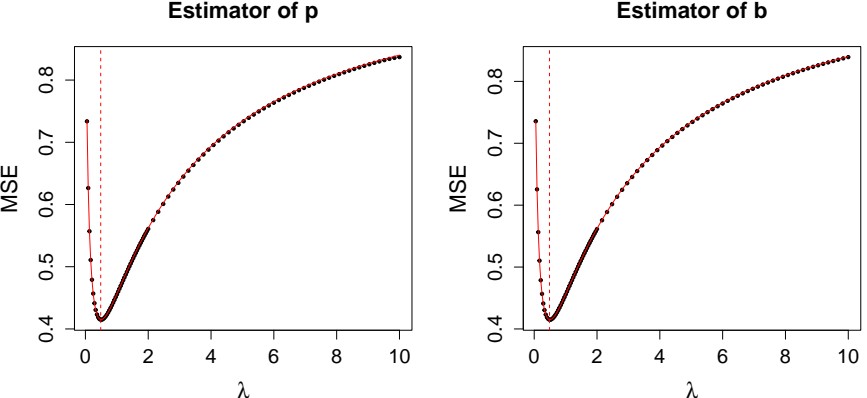

Figure 10: MSE of the nuisance function estimators in the setting with $c = 0.5$ (Left: $\alpha$; Right: $\beta$)

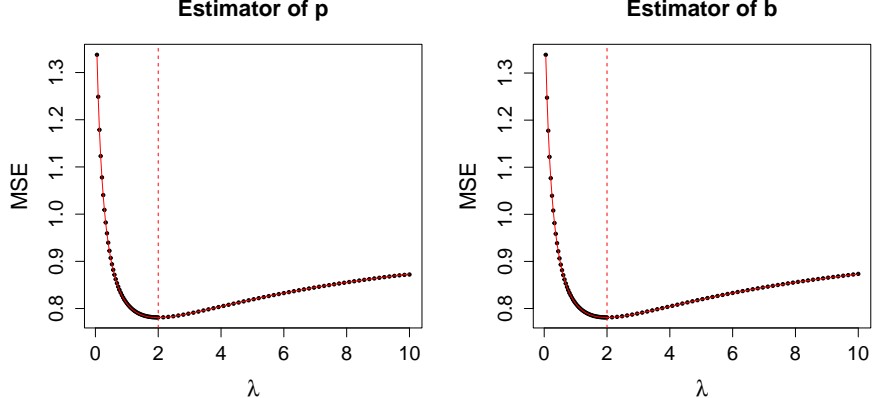

Figure 11: MSE of the nuisance function estimators in the setting with $c = 2$ (Left: $\alpha$; Right: $\beta$)

## H.4 Parametric Bootstrap for Estimating the Asymptotic Variance

Our next set of results evaluate the performance of the parametric bootstrap approach described in Appendix G. In particular, we compared the estimated variance of the debiased ECC estimators (with $10,000$ bootstrap samples) to their true variances, obtained by Monte Carlo integration with $10,000$ samples.

### H.4.1 Settings with Two Splits

Figure 12 compares the estimated variance of the ECC estimators with the true variance in the setting with two splits and $c = 0.5$. Table 1 summarizes the results across $\lambda$ values. We find that the estimated variance was very close to the true variance for each $\lambda$ value.

Table 1: Ratio of the true standard error to the bootstrap estimated standard error of the debiased ECC estimators in the setting with two splits and $c = 0.5$. The entries show summary statistics of the ratios calculated across the 100 different $\lambda$ values. Ratios greater than 1 indicate that the true standard error is greater than the bootstrap estimated standard error.

| Estimator | Min | 1st Qu. | Median | Mean | 3rd Qu. | Max |
|---|---|---|---|---|---|---|
| INT (integral-based) | 0.995 | 0.996 | 0.999 | 1.001 | 1.004 | 1.019 |
| NR (Newey–Robins) | 0.992 | 0.992 | 0.993 | 0.996 | 0.996 | 1.035 |
| IF (doubly robust) | 0.983 | 0.989 | 0.992 | 0.991 | 0.993 | 0.996 |

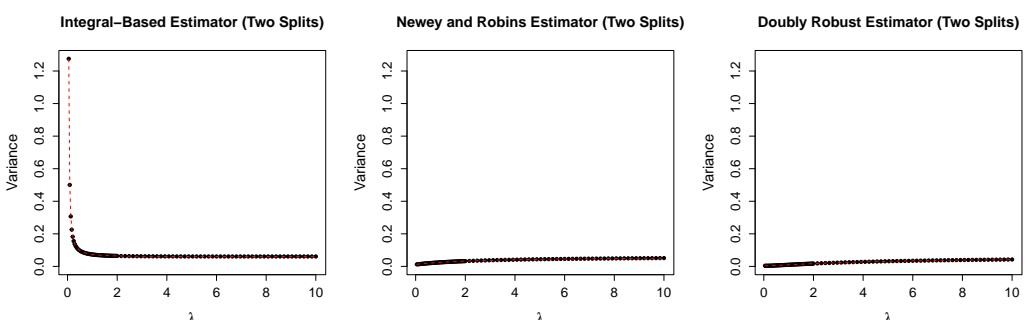

Figure 12: Bootstrap estimated (black dots) versus true (red line) variance of the debiased ECC estimators in the setting with two splits and $c = 0.5$ across the $\lambda$ values.

Analogous results in the $c = 2$ setting are given in Figures 13 and Table 2. As noted in the main text, the variance blows up when $\lambda \approx 1.5$ due to the bias correction constant diverging for the Integral-based estimator. We also noticed a similar phenomenon for the Newey and Robins estimator for $\lambda$ close to 0. We plan to explore these subtitles in detail in the future. Compared to the $c = 0.5$ setting, the differences between the estimated and true variances were larger, especially for the integral-based estimator.

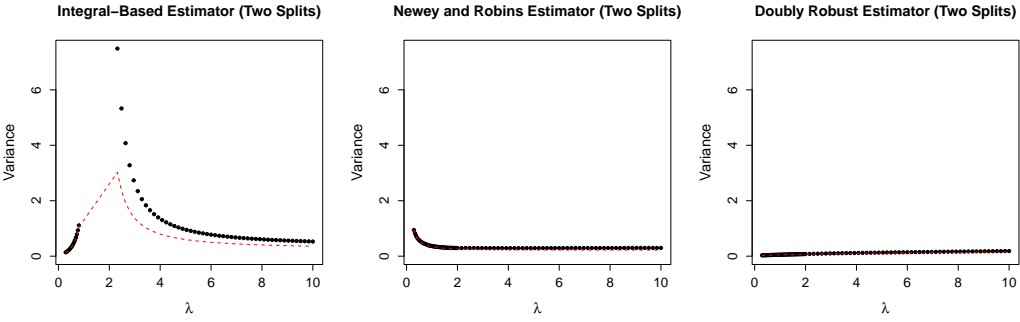

Figure 13: Bootstrap estimated (black dots) versus true (red line) variance of the debiased ECC estimators in the setting with two splits and $c = 2$ across the $\lambda$ values.

Table 2: Ratio of the true standard error to the bootstrap estimated standard error of the debiased ECC estimators in the setting with two splits and $c = 2$. The entries show summary statistics of the ratios calculated across the 100 different $\lambda$ values. Ratios greater than 1 indicate that the true standard error is greater than the bootstrap estimated standard error.

| Estimator | Min | 1st Qu. | Median | Mean | 3rd Qu. | Max |
|---|---|---|---|---|---|---|
| INT (integral-based) | 0.636 | 0.787 | 0.813 | 0.831 | 0.824 | 0.995 |
| NR (Newey-Robins) | 0.862 | 0.872 | 0.913 | 0.920 | 0.965 | 1.005 |
| IF (doubly robust) | 0.894 | 0.917 | 0.936 | 0.941 | 0.966 | 0.995 |

### H.4.2 Settings with Three Splits

Figure 14 and Table 3 summarize the simulation results in the setting with three splits and $c = 0.5$. Analogously, Figure 15 and Table 4 give the results for the setting with $c = 2$.

Similar to the results in the settings with two splits, the estimated variance was close to the true variance.

Table 3: Ratio of the true standard error to the bootstrap estimated standard error of the debiased ECC estimators in the setting with three splits and $c = 0.5$. The entries show summary statistics of the ratios calculated across the 100 different $\lambda$ values. Ratios greater than 1 indicate that the true standard error is greater than the bootstrap estimated standard error.

| Estimator | Min | 1st Qu. | Median | Mean | 3rd Qu. | Max |
|---|---|---|---|---|---|---|
| Integral-based estimator | 1.026 | 1.027 | 1.031 | 1.031 | 1.034 | 1.042 |
| Newey-Robins estimator | 1.014 | 1.016 | 1.024 | 1.023 | 1.029 | 1.031 |
| Doubly robust estimator | 0.990 | 1.015 | 1.016 | 1.016 | 1.018 | 1.019 |

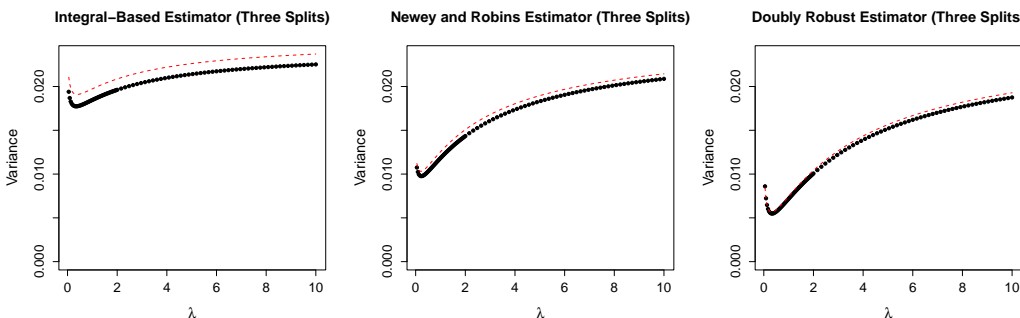

Figure 14: Bootstrap estimated (black dots) versus true (red line) variance of the debiased ECC estimators in the setting with three splits and $c = 0.5$ across the $\lambda$ values.

Table 4: Ratio of the true standard error to the bootstrap estimated standard error of the debiased ECC estimators in the setting with three splits and $c = 2$. The entries show summary statistics of the ratios calculated across the 100 different $\lambda$ values. Ratios greater than 1 indicate that the true standard error is greater than the bootstrap estimated standard error.

| Estimator | Min | 1st Qu. | Median | Mean | 3rd Qu. | Max |
|---|---|---|---|---|---|---|
| INT (integral-based) | 0.934 | 0.935 | 0.939 | 0.941 | 0.944 | 0.978 |
| NR (Newey–Robins) | 0.935 | 0.935 | 0.940 | 0.942 | 0.946 | 0.977 |
| IF (doubly robust) | 0.935 | 0.936 | 0.943 | 0.946 | 0.951 | 0.994 |

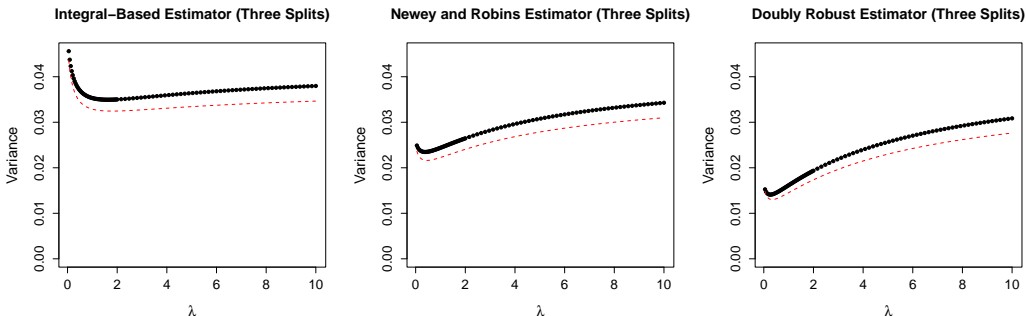

Figure 15: Bootstrap estimated (black dots) versus true (red line) variance of the debiased ECC estimators in the setting with three splits and $c = 2$ across the $\lambda$ values.

