# OpenReview forum: "Optimal Nuisance Function Tuning for Estimating a Doubly Robust Functional under Proportional Asymptotics"
_NeurIPS.cc/2025/Conference — NeurIPS 2025 spotlight_

### Official Review · Reviewer_Fm5s · 2025-07-02

**Clarity:** 4
**Significance:** 4
**Originality:** 3
**Rating:** 6
**Confidence:** 4

**Summary:**

The authors study parameter tuning for estimation of the ECC in linear models under the asymptotic regime $\frac{p}{n} \rightarrow c$ as $n \rightarrow \infty$ (including $p > n$). In this setting, the linear model weights are nuisance parameters for which naive plug-in, Newey--Robins or doubly robust estimators have worse-than-$\frac{1}{\sqrt{n}}$ asymptotic bias. The authors propose debiased versions of these estimators. The authors derive asymptotic variances of their estimators---including estimators utilizing with additional subsampling---as a function of the ridge-regression regularization parameters for the linear weights.

**Questions:**

Overall, I am very happy with this paper. No questions/suggestions significant enough to influence my score come to mind. I am keen to see other reviewers' comments and will follow up with any significant questions during the next stage.

**Ethical Concerns:**

["NO or VERY MINOR ethics concerns only"]

**Final Justification:**

`Reviewer FNJX` raised concerns about the validity of the authors' analysis, which, to my reading, have been addressed by the authors.

All three reviewers (except myself) have downweighted their scores based on concerns about the scope of the authors' investigation due to linear Gaussian assumptions and a focus on estimators of the ECC. The authors rebut that their derivation strategy shows promise for future work in extending their results. However, given the simplicity of the authors' argument for why Gaussian assumptions should be unnecessary, a short empirical investigation with, e.g., sub-Gaussian noise would have strengthened their rebuttal.

The authors reiterate the potential use of their approach in structure discovery without a priori sparsity assumptions, which is an exciting direction with possible applications in assessing the sensitivity of structure discovery approaches to these assumptions.

**Limitations:**

Yes.

**Paper Formatting Concerns:**

None.

**Quality:**

4

**Strengths And Weaknesses:**

The paper was a pleasure to read, with contributions well motivated and technical results clearly spelled out. The authors compare a wide range of approaches and make clear which results are novel and/or surprising. Applications to structure discovery and/or covariate selection are clear.

---

> ### Author Rebuttal · Authors · 2025-07-30
>
> Thank you very much for your kind words! If you have any other comments/concerns during the discussion period, kindly do let us know. We would be happy to answer them.

---

### Official Review · Reviewer_AoFP · 2025-07-03

**Clarity:** 3
**Significance:** 3
**Originality:** 3
**Rating:** 5
**Confidence:** 3

**Summary:**

- DML relies on estimating nuisance functions (eg. Propensity score, outcome regression) accurately. However, in high-dimensional settings, ridge, Lasso, and many other estimators become inconsistent. Besides, in the propotional regime, the invalidity of orthogonalization may be invalid, thus the product of two inconsistent nuisance estimators can introduce substantial bias. Furthermore, cross-fitting of DML becomes problematic: estimates from different folds can be correlated;

- This paper studies how to optimally tune nuisance function estimators in high-dimensional settings of the ECC. They derive bias-corrected versions of three ECC estimators (plug-in, Newey-Robins, and doubly robust) that achieve $/sqrt{n}$-consistency, each applied under two distinct sample-splitting strategies. They also provide exact asymptotic variance formulas for these estimators, showing that the tuning parameters that are optimal for prediction do not necessarily minimize the variance of the final ECC estimator.

- Simulations validate the theoretical findings and provide practical insights into selecting tuning parameters for reliable inference in high-dimensional settings.

**Questions:**

See my questions in the above “Weakness” part and below:

* Section 3.2 and Section 4, no concrete method is provided for how practitioners can implement this in real-data setting when ground truth is unavailable. Can you add empirical study or a practical algorithm or heuristic for tuning under real constrains?
* Can you add a simple simulation example under eg non-Gaussian error, nonlinearity in E[Y|X] and E[A|X]? This would strengthen the empirical case with robustness to non-Gaussian and misspecified settings.
* In the two-split simulations, the authors note that the variance “blow up” around a certain $\lambda$, but this is unexplained? Any insights?

**Ethical Concerns:**

["NO or VERY MINOR ethics concerns only"]

**Final Justification:**

I would like to increase my score to 5.

**Limitations:**

See the limitations also in the above “Weakness” and “Questions” part.

**Paper Formatting Concerns:**

Figures in Section 4 appear compressed and may be difficult to read when printed or viewed on small screens.

**Quality:**

3

**Strengths And Weaknesses:**

**Strengths:**
- The paper is technically strong. It addresses a challenging of inference for doubly robust functionals when nuisance functions cannot be consistently estimated due to high-dimensionality.
- The authors provide rigorous bias correction techniques for three estimators (integral-based, Newey-Robins, and doubly robust), and prove $\sqrt{n}$-consistency and asymptotic normality.
- The asymptotic variance analysis is detailed, revealing a nuanced dependence of estimator performance on tuning parameters and sample splitting schemes.
- The simulation studies illustrating theoretical findings.
- The insights regarding inference-optimal and prediction-optimal can be useful for the causal and robust statistical inference applications.
- The paper builds on concepts like sample splitting and debiasing from classical DML literature, its methodological contributions—especially the derivation of explicit variance formulas under ridge regularization—are novel.

**Weakness:**

- This paper only analyzes ridge regression for estimating nuisance functions $E[A|X]$, and $E[Y|X]$ under the linearity assumptions. While this helps derive the closed-form asymptotic variance but narrows the applicability.
- The results do not generalize to model nonlinearity or non-Gaussian errors. It is unclear whether similar variance or bias correction methods extend to more realistic scenarios.
- The asymptotic results depend on the assumption of Gaussian and sub-Gaussian. It would be great if the author could add a discussion of the robustness of the results to deviations from Gaussianity. Though it can be a future work for the theoretical study, adding empirical verification of performance under violations of this assumption might be helpful to explain the robustness of the results.
- Any minimax lower bound for estimating ECC under proportional asymptotic?
- How sensitive are confidence intervals to variance estimation?
- In theory, it’s elegant to minimize asymptotic variance w.r.t. $\lambda_1$, $\lambda_2$, but these variance expressions are complicated, require knowledge of unknown population parameters (eg. Inner product between nuisance coefficients). However, there’s no practical method is proposed for running parameter sections, may not be practically estimable in finite samples.

---

> ### Author Rebuttal · Authors · 2025-07-31
>
> We greatly appreciate the reviewer’s feedback and suggestions, and would like to address them as follows.
>
> > 1. **Linearity assumption and ridge regression**:
>
> We agree that the linear model assumption may appear restrictive at first glance. However, we would like to emphasize that our analysis includes the challenging proportional asymptotic regime where the number of covariates $p$ is larger than the sample size $n$, and $p/n \to c \in (1, \infty)$. This regime is fundamentally different from the standard high-dimensional setting, where valid inference typically relies on some form of sparsity assumption. Theoretical understanding of the proportional regime is still in its early stages. In fact, results on simple linear regression [1] and AIPW estimators under the linear outcome regression and propensity score model and $p < n$ (ref. [23] of our paper) have only recently been established. From this perspective, focusing on the linear model is not merely a simplifying assumption, but rather a necessary and meaningful first step toward developing a deeper understanding of more complex models in this regime.
>
> A natural next step is to extend the linear regression setting to regression in a general feature space, such as kernel ridge regression. Since our analysis relies on deterministic equivalents of functionals of Gram matrices, the techniques can be carried over, provided similar results hold for Gram matrices in the feature space setting.
>
> Regarding the ridge penalty, we believe that similar analyses can be conducted in the presence of other convex penalties (e.g., $\ell_1$ norm), borrowing tools from Approximate Message Passing (AMP) theory. Debiasing methods based on AMP have been studied in the context of linear models [1] and doubly robust estimators such as AIPW under the regime $p < n$ of ref. [23]. These developments suggest that similar techniques could be adapted to our setting.
>
> > 2. **Gaussian assumption:**
>
> Thank you for raising this concern. We would first like to point out that for the majority of our results (Theorems 3.1 - 3.3, which show that the estimators are $\sqrt{n}$-consistent), we do not need normality of the covariates or the errors. The following two conditions are sufficient for these three estimators: (i) $X_{ij}$ are i.i.d. with mean $0$ and variance $1$ and finite $(4 + \delta)$ moment, and (ii) $(\epsilon, \mu) \overset{i.i.d.}{\sim} F$ where $F$ with mean $0$, variance $1$, correlation $\rho$, and has finite $(2 + \delta)$ moment for some $\delta > 0$.
>
> Regarding allowing dependence between the covariates, we believe that our results can be extended to the settings where $X_i$ is a subgaussian random variable in $\mathbb{R}^p$ with bounded sub-gaussian norm using more modern techniques. However, we leave this as future work, as we do not believe it will add anything substantial to the main message of the paper, and the current analysis is already quite complicated.
>
> Theorem 3.4 indeed relies on the assumption of Gaussian covariates and error distribution to derive the exact form of the limiting variance. That said, we believe these assumptions could potentially be relaxed by appealing to suitable universality results. Nonetheless, doing so would require substantial theoretical development and a careful analysis to rigorously establish the necessary universality results, which is a challenging task in random matrix theory.
>
> We added additional simulations below to show robustness against the Gaussian assumption. See our response to ``Additional simulations" below.
>
> > 3. **Minimaxity:**
>
> In this paper, we establish that ECC can be estimated at the $\sqrt{n}$ rate, which is known to be minimax optimal for estimating finite-dimensional parameters under standard assumptions. Consequently, our estimator is minimax rate optimal. Obtaining sharp efficiency bounds beyond rate minimaxity for functional estimation under proportional asymptotics remains an open direction. Therefore, we explore optimal estimation under a given strategy of estimation, i.e., ridge tuning, sample splitting, and given choice of estimators such as INT, NR, or DR.
>
> > 4. **Confidence interval coverage:**
>
> In the analyses in the current paper, the bias of the debiased estimators is negligible compared to their standard errors. We also discuss standard error estimation in the following bullet point, which we found to be within 5\% of the true standard error. Together, these suggest near-nominal confidence interval coverage. However, we would be happy to report how sensitive confidence intervals are to variance estimation in the revised paper.
>
> > 5. **Practical methods for selecting tuning parameters:**
>
> Thank you for raising this concern. While the expressions for the limiting variances may appear complex, they are in fact smooth deterministic functions of three unknown quantities: $(||\alpha_0||_2^2, ||\beta_0||_2^2, \alpha_0^\top \beta_0)$. This allows practitioners to adopt one of two practical strategies, as outlined below:
>
> (i) **consistent estimation of unknown quantities**  One may consistently estimate $(||\alpha_0||\_2^2, ||\beta_0||\_2^2, \alpha\_0^\top \beta\_0)$ and plug them into the variance expressions. To estimate $||\alpha_0||\_2^2$, one can first compute a simple plugged-in estimator $||\hat \alpha(\lambda)||\_2^2$, ($\hat \alpha(\lambda)$ is a ridge estimator of $\alpha\_0$) with tuning parameter $\lambda$, and then apply bias correction to obtain a consistent estimator. Similar procedures apply to $ ||\beta_0||\_2^2$ and $\alpha\_0^\top \beta\_0$.
>
> (ii) **Bootstrapping** Alternatively, a parametric bootstrap approach can be employed: we first simulate noise terms $(\varepsilon, \mu)$ from a bivariate Gaussian distribution with mean $0$, variance $1$, and correlation $\hat \theta$ and add them to signals (with $\hat \alpha(\lambda_1)$ and $\hat \beta(\lambda_2)$) to generate synthetic responses, from which a bootstrap estimate $\hat \theta^{\rm Boot}$ is computed. Repeating this procedure multiple times (say $B$ times) yields a collection of bootstrap estimates, say $\hat \theta\_j^{\rm Boot}$ for $1 \le j \le B$, whose empirical variance can then be used as an estimate of the variance of $\hat \theta$. Finally, to ensure consistency of this variance estimator, a bias-correction step is needed, for which we need to estimate $(||\alpha_0||_2^2,  ||\beta_0||_2^2, \alpha_0^\top \beta_0)$ consistently as before. This method circumvents the need to evaluate the complex analytical form of the variance. We would be happy to provide further details in the final version.
>
> Finally, we evaluate the estimated variance over a grid of $(\lambda_1, \lambda_2)$ values and select the pair minimizing estimated variance. This would give us an estimator of $\hat \theta$, which is $\sqrt{n}$-consistent and has \emph{minimum asymptotic variance} among all estimators in our proposed class. This aids in inference via giving a confidence interval with a small width. Due to space constraints, we omit the technical details here, but we would be happy to elaborate upon request and add a discussion in the final version of the paper.
>
> > 6. **Additional simulations**
>
> **Parametric Bootstrap:** We run simulations for $\hat{\theta}^{{\rm INT, db}}_{{\rm 3sp}}$ using bootstrap. In particular, we compute the ratio of the bootstrap estimator of the limiting standard deviation (s.d.) to that of the gold-standard oracle Monte-Carlo s.d. (obtained from repeated draws from the true data-generating process). We set $p = 250, n = 500, \rho = 0.5, B = 10000$ and and a grid of 100 $\lambda$ values from $0.05$ to $10$.  Below is the summary of this ratio (mean = $1.024$) over $\lambda$:
>
> |Min|$Q_1$|Median|$Q_3$|MAX|
> |--------|--------|---|---|--------|
> |1.014|1.016|1.023|1.029|1.051|
>
> This shows that the bootstrap estimator closely matches the true asymptotic variance across different $\lambda$, and we have already shown (see Figure 2,4, and 6 in the appendix) that bias is negligible. Thus, our data-driven procedure yields valid coverage.
>
> **Non-gaussian error:** We perform simulations with non-Gaussian errors for $\hat{\theta}^{{\rm INT, db}}_{{\rm 3sp}}$, where we now generate errors from a bivariate Laplace distribution with mean $0$, marginal variances $1$, and correlation $\rho=0.5$, same as the Gaussian setting in the paper. The de-biased estimators had negligible bias in this case, and the following table provides a summary of the ratio (Laplace to Gaussian) of s.d. over 10000 Monte Carlo iterations:
>
> |Min|$Q_1$|Median|$Q_3$|MAX|
> |--------|--------|---|---|--------|
> |1.039|1.041|1.048|1.053|1.06|
>
> This implies our results continue to hold beyond Gaussianity. We would be happy to elaborate upon request and add this to the final version of the paper.
>
> > 7. **Variance blowup:**
>
> The variance blow-up is primarily driven by the bias correction factor. For instance, consider the definition of $\hat{\theta}\_{2sp}^{\rm INT, db}$ in Equation (3.4). It is immediate from the definition that the variance of this estimator diverges when $$ \left(1 - \frac{g\_{2, 2sp}^{\rm INT}(\lambda_1, \lambda_2)}{g\_{1, 2sp}^{\rm INT}(\lambda_1, \lambda_2)}\right) \approx 0 \,.$$ This scenario arises in our simulations when $\lambda_1 = \lambda_2 = \lambda$ is around 1.5 (Top left picture of Figure 1), where $g\_{1, 2sp}^{\rm INT}(\lambda, \lambda) \approx g\_{2, 2sp}^{\rm INT}(\lambda, \lambda)$, causing the variance to explode. However, this phenomenon does not affect our method. Since our goal is to select $(\lambda_1, \lambda_2)$ that minimizes the asymptotic variance, such values where the variance is unstable are naturally excluded from consideration.
>
> [1] Hastie, T., Montanari, A., Rosset, S., & Tibshirani, R. J. (2022). Surprises in high-dimensional ridgeless least squares interpolation. Annals of statistics, 50(2), 949.

---

> > ### Comment · Reviewer_AoFP · 2025-08-05
> > **Thank you!**
> >
> > Thank you for the detailed and thoughtful rebuttal. The clarifications on practical tuning, confidence interval coverage, and the explanation of the variance blow-up were clear and convincing. These responses address my concerns well, and I’m happy to raise my score to 5.

---

> > > ### Author Response · Authors · 2025-08-07
> > >
> > > Thank you for your thoughtful engagement and for considering our responses. We truly appreciate your precious time. We're glad our clarifications were helpful and are grateful for your raised score.

---

### Official Review · Reviewer_Wn4b · 2025-07-03

**Clarity:** 3
**Significance:** 3
**Originality:** 2
**Rating:** 5
**Confidence:** 3

**Summary:**

The paper studies the optimal tuning parameter choice in ridge regression to estimate the Expected Conditional Covariance in the setting of linear data-generating model and the proportional asymptotic regime (where $p>n$ is allowed). It studies three estimators and two data-splitting methods, and proves the $\sqrt{n}-$consistency and derives the asymptotic variances of the estimators.

**Questions:**

The proofs rely heavily on results from Random Matrix Theory. To what extent can the techniques be extended beyond the restrictive assumptions of a linear model and the ridge estimator?

The paper provides analytic expressions for the asymptotic variances. Do these expressions offer any intuitive understanding of the influence of the parameters involved? Can they help explain the variance blowup observed in the experiment?

**Ethical Concerns:**

["NO or VERY MINOR ethics concerns only"]

**Final Justification:**

Thank you for addressing the concerns. I will raise the score.

**Limitations:**

The paper discusses the limitation on Gaussian assumption. The limitations of the proving technique applied to linear model and ridge estimator merits consideration.

**Quality:**

3

**Strengths And Weaknesses:**

The paper clearly explained the novelties compared with existing work and proves novel results for proportional asymptotic regime. It reveals that prediction-optimal tuning parameters may not lead to the lowest asymptotic variance of the ECC estimator.

The proposed sample-splitting approach appears to sacrifice the sample efficiency of DML. In addition, there is a considerable body of work on high-dimensional linear regression, where RMT is widely applied. The paper can benefit from discussing connections to this literature.

---

> ### Author Rebuttal · Authors · 2025-07-30
>
> We greatly appreciate the reviewer’s feedback and suggestions, and would like to address them as follows.
>
> > 1. **The sample-splitting approach appears to sacrifice the sample efficiency of DML:**
>
> Thank you for raising this important point. You are absolutely right that sample splitting generally sacrifices statistical efficiency, and cross-fitting is a widely adopted strategy to address this issue.
> However, our analysis is conducted under the proportional asymptotic regime where $p/n \to c \in (1, \infty)$. In this setting, a rigorous analysis of cross-fitting becomes substantially more challenging. To the best of our knowledge, recent work (ref. [23] of our paper) analyzed cross-fitting in the context of AIPW estimators, but only in the regime $p/n \to c \in (0, 1)$, where the dimensionality is smaller than the sample size. The main technical difficulty is that, in classical fixed-$p$ settings (or in high-dimensional settings with sparsity constraints), the estimators obtained from different sample splits are only weakly correlated and the correlation is asymptotically negligible. However, in the proportional regime, this correlation is often non-negligible, rendering the analysis of cross-fitting significantly more challenging. Moreover, whether cross-fitting effectively reduces variance, i.e., whether the non-negligible correlation between estimators from different splits is negative, can depend on several factors, including the sample splitting scheme, the type of estimator used, and the specific parameter settings. In fact, recent work (ref. [23] of our paper) has observed a striking phenomenon: the correlation between estimators from different splits can be positive and non-vanishing, leading to increased variance of the cross-fitted estimator. This highlights the nuanced and context-dependent behavior of cross-fitting in the proportional asymptotic regime. While we believe it may be possible to adapt the techniques from [23] and extend them carefully to this setting, doing so is non-trivial and would likely require substantial new insights. As such, we leave this as an important direction for future work. We expect that a complete theoretical treatment of cross-fitting under the regime $p/n \to c \in (1, \infty)$ would constitute a contribution in its own right.
>
>
> > 2. **Extension beyond linear model and ridge:**
>
> Thank you for raising this interesting point! While, as a first step, we focus on a linear data-generating model, we agree that extending the analysis to more expressive non-linear models—such as kernel ridge regression (KRR) or random feature (RF) models is an important direction for future work.
> Recent advances (e.g., [1]) have analyzed KRR and RF estimators under the proportional asymptotic regime, leveraging tools from random matrix theory. Therefore, we believe that it is possible to construct a consistent estimator of $\theta_0$ when the conditional mean function is a polynomial. However, extending these results to enable $\sqrt{n}$-consistent estimation and drawing a valid inference remains challenging. In particular, debiasing non-linear estimators in the proportional regime is a highly non-trivial task, and to the best of our knowledge, no prior work has addressed this issue, even in the context of simple regression. We believe this represents a promising but technically demanding research avenue for future work.
>
> If we replace the squared error loss with another loss function (e.g., Huber loss), it is still possible to leverage similar tools from random matrix theory (RMT) to arrive at the desired conclusions. This is supported by existing work on general M-estimation problems in high dimensions using RMT techniques (e.g., [2]). However, if the penalty function is changed to the $\ell_1$ norm or other convex penalties, the analysis becomes more involved. In such settings, the tools from Approximate Message Passing (AMP), Convex Gaussian minmax theorem (CGMT), or Stein's identity become useful to obtain a $\sqrt{n}$-consistent and asymptotically normal (CAN) estimator of $\theta_0$. Debiasing methods based on these techniques have been studied in the context of linear models (ref. [25] of our paper) and doubly robust estimators such as AIPW under the regime $p < n$ (ref. [23] of our paper). These developments suggest that similar techniques could be adapted to our setting. We leave this direction as a promising avenue for future investigation.
>
>
> > 3. **Asymptotic variance:**
>
> The analytic expression of the variances indicates that given any $(\lambda_1, \lambda_2)$, the limiting variance is a deterministic function of three unknowns: $(||\alpha_0||_2^2, ||\beta_0||_2^2, \alpha_0^\top \beta_0)$. The final goal is to choose $(\lambda_1, \lambda_2)$ which minimizes these variance expressions. This also echoes the main point of our paper: the prediction-optimal tuning parameters, i.e., the value of $\lambda_1$ (resp. $\lambda_2$) that yields the minimum prediction risk of $\alpha_0$ (resp. $\beta_0$), may not yield an estimator of ECC with minimum asymptotic variance. This contrasts with the conventional wisdom in double machine learning (DML), where nuisance parameters are typically estimated by minimizing prediction error, a point which is further highlighted in our simulation section. We next present two practical strategies for consistently estimating the limiting variance and selecting optimal tuning parameters:
>
> (i) **consistent estimation of unknown quantities**  One may consistently estimate $(||\alpha_0||\_2^2, ||\beta_0||\_2^2, \alpha\_0^\top \beta\_0)$ and plug them into the variance expressions. To estimate $||\alpha_0||\_2^2$, one can first compute a simple plugged-in estimator $||\hat \alpha(\lambda)||\_2^2$, ($\hat \alpha(\lambda)$ is a ridge estimator of $\alpha\_0$) with tuning parameter $\lambda$, and then apply bias correction to obtain a consistent estimator. Similar procedures apply to $ ||\beta_0||\_2^2$ and $\alpha\_0^\top \beta\_0$.
>
> (ii) **Bootstrapping** Alternatively, a parametric bootstrap approach can be employed: we first simulate noise terms $(\varepsilon, \mu)$ from a bivariate Gaussian distribution with mean $0$, variance $1$, and correlation $\hat \theta$ and add them to signals (with $\hat \alpha(\lambda_1)$ and $\hat \beta(\lambda_2)$) to generate synthetic responses, from which a bootstrap estimate $\hat \theta^{\rm Boot}$ is computed. Repeating this procedure multiple times (say $B$ times) yields a collection of bootstrap estimates, say $\hat \theta\_j^{\rm Boot}$ for $1 \le j \le B$, whose empirical variance can then be used as an estimate of the variance of $\hat \theta$. Finally, to ensure consistency of this variance estimator, a bias-correction step is needed, for which we need to estimate $(||\alpha_0||_2^2,  ||\beta_0||_2^2, \alpha_0^\top \beta_0)$ consistently as before. This method circumvents the need to evaluate the complex analytical form of the variance. We would be happy to provide further details in final version.
>
> Finally, we evaluate the estimated variance over a grid of $(\lambda_1, \lambda_2)$ values and select the pair minimizing estimated variance. This would give us an estimator of $\hat \theta$, which is $\sqrt{n}$-consistent and has \emph{minimum asymptotic variance} among all estimators in our proposed class. This aids in inference via giving a confidence interval with a small width.
>
> Our simulation suggests that the bootstrap estimator works well, but we cannot add it due to space constraints. Kindly see our response to Comment 3 of Reviewer FNJX and Comment 6 of AoFP for the details.
>
> > 4. **Variance blowup:**
>
> The variance blow-up is primarily driven by the bias correction factor. For instance, consider the definition of $\hat{\theta}\_{2sp}^{\rm INT, db}$ in Equation (3.4). It is immediate from the definition that the variance of this estimator diverges when $$ \left(1 - \frac{g\_{2, 2sp}^{\rm INT}(\lambda_1, \lambda_2)}{g\_{1, 2sp}^{\rm INT}(\lambda_1, \lambda_2)}\right) \approx 0 \,.$$
> This scenario arises in our simulations when $\lambda_1 = \lambda_2 = \lambda$ is around 1.5 (Top left picture of Figure 1), where $g\_{1, 2sp}^{\rm INT}(\lambda, \lambda) \approx g\_{2, 2sp}^{\rm INT}(\lambda, \lambda)$, causing the variance to explode. However, this phenomenon does not affect our method. Since our goal is to select $(\lambda_1, \lambda_2)$ that minimizes the asymptotic variance, such values where the variance is unstable are naturally excluded from consideration.
>
> [1] *Spectrum of inner-product kernel matrices in the polynomial regime and multiple descent phenomenon in kernel ridge regression*. Theodor Misiakiewicz.
>
> [2] *Asymptotics For High Dimensional Regression M-Estimates: Fixed Design Results*. Lihua Lei, Peter J. Bickel, Noureddine El Karoui.

---

> > ### Comment · Reviewer_Wn4b · 2025-08-05
> >
> > The author has addressed most of the concerns. It would be helpful to discuss how this work connects to the broader high-dimensional regression literature. In particular, do the proof techniques and the distinction drawn between estimation and prediction have parallels in that setting?

---

> > > ### Author Response · Authors · 2025-08-07
> > > **Response to the comment from authors**
> > >
> > > Thank you once again for your thoughtful feedback and for recognizing our efforts. As you correctly noted, our work is closely related to the literature on high-dimensional regression under proportional asymptotics. Below, we elaborate on the key connections and differences.
> > >
> > > We aim to estimate a univariate functional of high-dimensional parameters, specifically, the expected conditional covariance $\theta_0$ between $(A, Y)$ given $X$, which, in our model, corresponds to the correlation $\rho$ between the unobserved noise terms $\epsilon$ and $\mu$. Since $(\epsilon, \mu)$ are unobserved, $\theta_0 = \rho$ cannot be estimated directly. However, under our model, $\theta_0$ admits several equivalent expressions:
> > > $$\theta_0 = \rho = E[AY] - \alpha^\top \beta = E[A(Y - X^\top \beta)] = E[(Y - X^\top \beta)(A - X^\top \alpha)]. $$ These representations motivate three estimators we have studied in our paper. Focusing on the first, we need to estimate $E[AY]$ and $\alpha^\top \beta$ to estimate $\theta_0$. The former is easy as it is an expectation of a univariate random variable $AY$ and can be estimated efficiently via the corresponding sample average. However, estimating later poses the main challenge, as both $\alpha$ and $\beta$ are high-dimensional.
> > >
> > > **Similarity** To estimate the inner product, we first estimate the high-dimensional coefficients $(\alpha, \beta)$ via ridge regression, which has been studied in [DW18, HMRT22]. This step relies on similar techniques, such as approximating random matrices and their functionals by deterministic equivalents."
> > >
> > > **Distinction** However, this is where our approach diverges from prior work. A naive plug-in estimator using ridge estimates is biased—both due to the inherent bias of ridge regression and the potential correlation between $\hat{\alpha}$ and $\hat{\beta}$ when estimated from the same data, as in the two-sample split setting. To achieve $\sqrt{n}$-consistency of $\rho$, careful bias correction is essential. Our key contribution, distinct from previous high-dimensional regression studies [DW18, HMRT22], is the construction of a $\sqrt{n}$-consistent estimator of a univariate functional in the proportional asymptotic regime ($p/n \to c > 1$). While we do not formally prove asymptotic normality, we derive the limiting variance of $\sqrt{n}(\hat{\theta} - \theta_0)$, which essentially captures the main difficulty. We believe standard tools from random matrix theory (e.g., martingale CLTs and Lindeberg’s method) can extend our analysis to full normality. To our knowledge, ours is among the first works to achieve $\sqrt{n}$-consistent estimation of a functional of $p$-dimensional parameters in this regime. While [26] offers one such instance, it also emphasizes the need to understand debiased regularization schemes for more efficient estimation. Our work advances this direction by quantifying the optimal choice of nuisance functions through a detailed analysis of how nuisance tuning interacts with sample splitting and estimation strategy to yield the most efficient debiased estimator, providing the first systematic theory of its kind in the proportional asymptotic setting, complementing earlier developments in fixed-$p$ nonparametric regimes [7,10,14,15].
> > >
> > > **Different choices of estimators** Our choice of estimators is grounded in the rich literature on doubly robust functional estimation[1]. While much of this work focuses on fixed-dimensional nonparametric settings [7,10,12,14,15] or high-dimensional regimes under sparsity constraints [19], our aim is to extend these ideas to the proportional asymptotic regime. In prior works, estimating the expected conditional covariance (ECC) has often served as a foundational case for studying more general doubly robust functionals (e.g., [7,10,14,15]). Notably, phenomena such as bias propagation and sensitivity to nuisance estimation in ECC estimation also arise in problems like average treatment effect estimation and inference under missing data. For ECC, we consider three estimators corresponding to three representations mentioned above, which can be classified as: (i) plug-in estimators, (ii) bias-corrected estimators via influence function, and (iii) doubly robust estimators, and have been well studied in fixed-dimensional settings. Motivated by this framework, and with the broader goal of understanding doubly robust functionals under proportional asymptotics, we begin with ECC and analyze all three estimator classes. Thus, while our estimator design is informed by the classical literature, our contribution lies in characterizing their behavior in the high-dimensional proportional regime.
> > >
> > > Note: References are numbered as they appear in the paper.
> > >
> > > [DW18]: Dobriban, E., & Wager, S. (2018). High-dimensional asymptotics of prediction: Ridge regression and classification.
> > >
> > > [HMRT22]: Hastie, T., Montanari, A., Rosset, S., & Tibshirani, R. J. (2022). Surprises in high-dimensional ridgeless least squares interpolation.

---

### Official Review · Reviewer_FNJX · 2025-07-03

**Clarity:** 3
**Significance:** 2
**Originality:** 3
**Rating:** 3
**Confidence:** 3

**Summary:**

The authors study the expected conditional covariance (ECC) under proportional asymptotics. The regressions are assumed to be linear with independent Gaussian noise. For some results the covariates are assumed to be independent and sub-Gaussian. For the main results they are assume to be independent and Gaussian.
The first result proves that bias corrected ridge estimators are n^{1/2} consistent. There are six variations of the estimator.
The second results gives the asymptotic variances of the six estimators.

**Questions:**

What is the step by step method that a researcher should use to tune the nuisance functions in the inconsistency regime?

How does this step by step method perform in coverage simulations?

**Ethical Concerns:**

["NO or VERY MINOR ethics concerns only"]

**Final Justification:**

In my assessment, the main results do not provide a complete enough answer to the main question posed in the paper. I will keep my score.

**Limitations:**

Yes.

**Paper Formatting Concerns:**

No.

**Quality:**

2

**Strengths And Weaknesses:**

Quality

The main weakness of the paper is that the results do not line up to the title and seem incomplete. First, the paper is about ECC not general functional estimation. Second, the assumptions in (2.1) are already very strong. Third, asymptotic normality is not proved, instead normality of all variables is assumed with independent covariates. Fourth, the way to operationalize (3.11) is not described. The variance seems to contain ECC=rho and also two-norm limits that are unknown, so it does not seem to given a feasible estimate of lamba1 and lambda2. Fifth, there is no analysis of a data-driven choice of lambda1 and lambda2. Sixth, the debiasing functions called g are not given in the main text so it is unclear what they depend on and how to implement them.

If I understand correctly, optimality is never established, even though it appears in the title. For a given ridge estimator on Gaussian data, the best ridge regularization is described as the minimizer of some objective. But this ridge estimator is not optimal, and so terminology  like “sharp asymptotic analyses” seems misleading.

Clarity

The main strength is that the paper summarizes the literature and ideas of the inconsistent regime well. The ideas are clearly presented.

Significance

I am concerned about how significant the result is because it ultimately does not derive the asymptotic variance without assuming all the data are Gaussian. The result also does not give practical guidance on choosing lambda1 and lambda2, since they seem to depend on unknown limits and maybe a preliminary estimate of ECC, which is not discussed.

Originality

The originality depends on how much the derivations of the debiasing g functions departs from the literature on the inconsistent regime. I have not studied the appendix closely. If it is using new techniques, then there are originality points. If it is using the same techniques, the less so.

---

> ### Author Rebuttal · Authors · 2025-07-31
>
> We greatly appreciate the reviewer’s feedback and suggestions, and would like to address them as follows:
>
> > 1. **ECC, not general functional:**
>
> We chose ECC as the main parameter, as in the literature it is often considered a starting point for analyzing more general doubly robust functionals (see refs. [7,10,14,15,19,25] of our paper). ECC is studied extensively in both fixed-dimensional nonparametric and ultra-high-dimensional setup under sparsity constraints (e.g., ref [19]). Moreover, as noted in those references, deriving necessary and sufficient ways of tuning nuisance functions for optimal doubly robust functional estimation has been carried out in the nonparametric fixed $p$ regime -- often using the ECC functional as a template. In contrast, we work in a more challenging proportional asymptotic regime ($p/n \to c \in (0, \infty)$), where dimension can exceed sample size and no sparsity is assumed. Here, there are no results for the necessary and sufficient ways of tuning nuisance functions. We view ECC as a foundational step for extending this theory to more general functionals, as has been achieved in other asymptotic regimes. That said, we appreciate your feedback and will revise the paper title to avoid confusion.
>
> > 2. **Assumption (2.1) and asymptotic normality:**
>
> We would first like to point out, for the majority of our results (Theorem 3.1 - 3.3, which shows that the estimators are $\sqrt{n}$-consistent), we do not need asymptotic normality of the covariates or errors; all we need is: (i) $X_{ij}$ are i.i.d. with mean $0$ and variance $1$ and finite $(4 + \delta)$ moment, and (ii) $(\epsilon, \mu) \overset{i.i.d.}{\sim} F$ where $F$ with mean $0$, variance $1$, correlation $\rho$, and finite $(2 + \delta)$ moment for some $\delta > 0$.
>
> We believe the results can be extended to settings where $X_i$ is a subgaussian random variable in $\mathbb{R}^p$ with bounded sub-gaussian norm using longer Martingale arguments, but we leave it as future work, as it doesn't change the main message of the paper and allows us to keep our proofs relatively compact.
>
> For exact asymptotic variance, we use the normality of the covariates to shorten the length of the proof significantly.
> One may once again use Martingale arguments along with deterministic equivalents for suitable functionals of reolvents of large covariance matrices, but it would yield a much longer proof without changing the message.
>
> As for the linearity assumption in regression, one could extend the analysis to feature space linear models or consider kernel ridge regression for nuisance functions, leveraging tools from [1]. While the overall framework would remain similar, the extension would require handling more involved deterministic equivalents of the associated Gram matrices, leading to significantly more complex computations. Given the current length and scope of our paper, we leave this promising direction for future work.
>
> Lastly, while we do not provide asymptotic normality, we note that the main crux of the proof shares the same complexity as that of obtaining precise asymptotic variance as we describe above.
>
> > 3. **Practical implementation of (3.11) and a data-driven optimal estimator:**
>
> The analytic expression of the variances indicates that given any $(\lambda_1, \lambda_2)$, the limiting variance is a deterministic function of three unknowns: $(||\alpha_0||_2^2, ||\beta_0||_2^2, \alpha_0^\top \beta_0)$. The final goal is to choose $(\lambda_1, \lambda_2)$ which minimizes these variance expressions. This also echoes the main point of our paper: the prediction-optimal tuning parameters, i.e., the value of $\lambda_1$ (resp. $\lambda_2$) that yields the minimum prediction risk of $\alpha_0$ (resp. $\beta_0$), may not yield an estimator of ECC with minimum asymptotic variance. This contrasts with the conventional wisdom in double machine learning (DML), where nuisance parameters are typically estimated by minimizing prediction error, a point which is further highlighted in our simulation section. We next present two practical strategies for consistently estimating the limiting variance and selecting optimal tuning parameters:
>
> (i) **consistent estimation of unknown quantities**  One may consistently estimate $(||\alpha_0||\_2^2, ||\beta_0||\_2^2, \alpha\_0^\top \beta\_0)$ and plug them into the variance expressions. To estimate $||\alpha_0||\_2^2$, one can first compute a simple plugged-in estimator $||\hat \alpha(\lambda)||\_2^2$, ($\hat \alpha(\lambda)$ is a ridge estimator of $\alpha\_0$) with tuning parameter $\lambda$, and then apply bias correction to obtain a consistent estimator. Similar procedures apply to $ ||\beta_0||\_2^2$ and $\alpha\_0^\top \beta\_0$.
>
> (ii) **Bootstrapping** Alternatively, a parametric bootstrap approach can be employed: we first simulate noise terms $(\varepsilon, \mu)$ from a bivariate Gaussian distribution with mean $0$, variance $1$, and correlation $\hat \theta$ and add them to signals (with $\hat \alpha(\lambda_1)$ and $\hat \beta(\lambda_2)$) to generate synthetic responses, from which a bootstrap estimate $\hat \theta^{\rm Boot}$ is computed. Repeating this procedure multiple times (say $B$ times) yields a collection of bootstrap estimates, say $\hat \theta\_j^{\rm Boot}$ for $1 \le j \le B$, whose empirical variance can then be used as an estimate of the variance of $\hat \theta$. Finally, to ensure consistency of this variance estimator, a bias-correction step is needed, for which we need to estimate $(||\alpha_0||_2^2,  ||\beta_0||_2^2, \alpha_0^\top \beta_0)$ consistently as before. This method circumvents the need to evaluate the complex analytical form of the variance. We would be happy to provide further details in final version.
>
> Finally, we evaluate the estimated variance over a grid of $(\lambda_1, \lambda_2)$ values and select the pair minimizing estimated variance. This would give us an estimator of $\hat \theta$, which is $\sqrt{n}$-consistent and has \emph{minimum asymptotic variance} among all estimators in our proposed class. This aids in inference via giving a confidence interval with a small width.
>
> **Simulation** To validate, we run simulations for $\hat{\theta}^{{db, INT}}_{{3sp}}$ using bootstrap. In particular, we compute the ratio of the bootstrap estimator of the limiting standard deviation (s.d.) to that of the gold-standard oracle Monte-Carlo s.d. (obtained from repeated draws from the true data-generating process). We set $p = 250, n = 500, \rho = 0.5, B = 10000$ and a grid of 100 $\lambda$ values from $0.05$ to $10$. We have the summary of this ratio (mean = $1.024$):
>
> |Min|$Q_1$|Median|$Q_3$|MAX|
> |--------|--------|---|---|--------|
> |1.014|1.016|1.023|1.029|1.051|
>
> This shows that the bootstrap estimator closely matches the true asymptotic variance, and we have already shown (see Figure 2,4, and 6 in the appendix) that the bias is negligible. Thus, the data-driven procedure yields valid coverage.
>
> > 4. **Debiasing function g:**
>
> The debiasing functions $g$ are defined using $\triangleq$, e.g., in line 184 $$g^{\rm INT}\_{\rm 3sp}(\lambda_1, \lambda_2)= \left( \int \frac{x \, dF_{\rm MP}(x) }{x+\lambda_1}\right) \left( \int \frac{x \, dF_{\rm MP}(x) }{x + \lambda_2}\right),$$ and in line 198 $$g_{1, \rm 2sp}^{\rm INT}(\lambda_1, \lambda_2) = \int \frac{x^2 \ dF_{\rm MP}(x)}{(x + \lambda_1)(x + \lambda_2)} \,,$$ where $F_{\rm MP}$ is Marchenko-Pasteur (MP)distribution, the limiting spectral distribution of eigenvalues of sample covariance matrices. In practice, we calculated by 10000 Monte Carlo samples of MP distribution.
>
> > 5. **Optimality of ridge estimator and sharp asymptotic analysis:**
>
> In parameter estimation, optimality typically refers to either i) rate optimality in the minimax sense, and ii) semi-parametric efficiency. As our estimator is proven to be $\sqrt{n}$-consistent, it achieves the former, as it attains the gold-standard parametric estimation rate.
>
> In classical fixed-dimensional settings, semiparametric efficiency typically refers to the smallest achievable asymptotic variance among all regular estimators. In contrast, our analysis operates under the proportional asymptotic regime (e.g., see ref. [2] in our paper), where the covariate dimension $p$ grows proportionally larger than the sample size $n$, i.e., $p/n \to c \in (1, \infty)$, without assuming sparsity in $E[Y|X]$ or $E[A|X]$.
>
> In this high-dimensional context, foundational concepts like regularity of estimators and Hajek–LeCam local minimax theory are not yet well developed. Thus, rather than pursuing classical efficiency bounds, we focus on identifying the best estimator within a given estimation strategy. Specifically, we ask: given a debiasing approach with regularized nuisance estimation and a fixed sample-splitting scheme (e.g., 1/2 or 1/3 splits), which configuration yields the lowest asymptotic variance?
>
> Can that be achieved by merely a priori choosing the level of regularization from a prediction optimality perspective -- as is the case for DML estimators under fixed $p$ or ultra-high dimensional sparse regimes? Our notion of sharp optimality is grounded in this perspective. We argue that when ridge regularization is used to estimate the nuisance functions, the regularization parameter should be chosen by minimizing the asymptotic variance of $\theta_0$ directly, rather than the prediction optimal one (as advocated by DML). That said, it remains possible that alternative regularization schemes, such as LASSO, may yield even lower variance. However, identifying the minimum possible achievable variance in this regime is highly nontrivial and is unknown even for simple linear regression. Therefore, this remains an important open problem for future investigation. We would be happy to include this discussion in the final version and are open to incorporating any other suggestions you may have.

---

> > ### Comment · Reviewer_FNJX · 2025-08-05
> >
> > I thank the authors for the rebuttal. In my assessment, the main results do not provide a complete enough answer to the main question posed in the paper. I will keep my score.

---

### Decision · Program_Chairs · 2025-09-17

**Decision:**

Accept (spotlight)

**Comment:**

**Summary**
Double Machine Learning (DML) relies on estimating nuisance functions accurately. However, in high-dimensional settings, estimators such as ridge, LASSO, and other become inconsistent. Besides, the proportional regime ($p/n = c \longrightarrow (0, \infty)$), the product of two inconsistent nuisance estimators can introduce substantial bias. Furthermore, cross-fitting of DML becomes problematic since estimates from different folds can correlated.

This paper studies how to optimally tune nuisance function estimators in high-dimensional settings of the ECC. They derive bias-corrected version of three ECC estimators under two different sample-splitting strategies: (plug-in, Newey-Robins, and doubly-robust) for the ECC estimators, and (2-splitting and 3-splitting) for sample-splitting. All bias-corrected ECC estimators achieve $\sqrt{n}$-consistency. Next, the authors derive exact asymptotic variance formulas for these estimators, showing that the tuning parameters that are optimal for prediction do not necessarily minimize the variance of the final ECC estimator.

Simulations validate the theoretical findings and provide practical insights into selecting tuning parameters for reliable inference in high-dimensional settings.

**Strength**

Except Rev.1 (FNjX), all reviewers agree on the following (best summarized by Rev. 3 AoFP) :

1. The paper is technically strong; it addresses a the challenge of inference for doubly-robust robust functionals when nuisance functions cannot be consistently estimated due to high-dimensionality.

2. The authors provide rigorous bias correction techniques for three ECC estimators, and prove $\sqrt{n}$-consistency and asymptotic normality.

3. The asymptotic variance analysis is detailed, revealing a nuanced dependence of estimator performance on tuning parameters and sample splitting schemes.

4. Simulation studies illustrate theoretical findings.

5. The insights regarding inference-optimal and prediction-optimal can be useful for the causal and robust statistical inference applications.

6. The paper builds on concepts like sample splitting and debiasing from classical DML literature, the methodological contributions, especially the derivation of explicit variance formulas under ridge regularization, are **novel**.

The concerns raised by Rev.1 (FNjX) are marginal and do not reflect a deep comprehension for the topic and contribution discussed in this submission. The authors responded with a thorough rebuttal to the reviewer, yet I find the reviewer's comment to the authors (after the rebuttal) to be disconnected from the author's rebuttal. I believe Rev.1 (FNjX) did not read, or did not carefully read the authors rebuttal, and/or even try to review the submission in order to have a better understanding.

Accordingly, the AC recommends acceptance for this nice and strong submission.